

# A new phylogenetic hypothesis of Tanystropheidae (Diapsida, Archosauromorpha) and other "protorosaurs", and its implications for the early evolution of stem archosaurs

Stephan N.F. Spiekman[1], Nicholas C. Fraser[2] and Torsten M. Scheyer[1]

[1] University of Zurich, Palaeontological Institute and Museum, Zurich, Switzerland
[2] National Museums Scotland, Edinburgh, UK

## ABSTRACT

The historical clade "Protorosauria" represents an important group of archosauromorph reptiles that had a wide geographic distribution between the Late Permian and Late Triassic. "Protorosaurs" are characterized by their long necks, which are epitomized in the genus *Tanystropheus* and in *Dinocephalosaurus orientalis*. Recent phylogenetic analyses have indicated that "Protorosauria" is a polyphyletic clade, but the exact relationships of the various "protorosaur" taxa within the archosauromorph lineage is currently uncertain. Several taxa, although represented by relatively complete material, have previously not been assessed phylogenetically. We present a new phylogenetic hypothesis that comprises a wide range of archosauromorphs, including the most exhaustive sample of "protorosaurs" to date and several "protorosaur" taxa from the eastern Tethys margin that have not been included in any previous analysis. The polyphyly of "Protorosauria" is confirmed and therefore we suggest the usage of this term should be abandoned. Tanystropheidae is recovered as a monophyletic group and the Chinese taxa *Dinocephalosaurus orientalis* and *Pectodens zhenyuensis* form a new archosauromorph clade, Dinocephalosauridae, which is closely related to Tanystropheidae. The well-known crocopod and former "protorosaur" *Prolacerta broomi* is considerably less closely related to Archosauriformes than was previously considered.

## INTRODUCTION

Non-archosauriform archosauromorphs lived during the late Permian and Triassic and belong to the archosaurian stem-lineage, the ancestral lineage of crocodylians and birds. Historically, many members of this group were placed within either "Protorosauria" or "Prolacertiformes". These two groups generally encompassed the same taxa and the usage of one term over the other depended on the inclusion within the clade of either *Protorosaurus speneri* or *Prolacerta broomi*, or both. Since both names generally apply to

Corresponding author
Stephan N.F. Spiekman,
stephanspiekman@gmail.com

the same taxa and are often used interchangeably, and because "Protorosauria" *Huxley, 1871* predates "Prolacertiformes" *Camp, 1945*, we refer to the members of these groups here as "Protorosauria" (sensu *Chatterjee, 1986*). Apart from the two above mentioned genera, the terrestrial and aquatic long-necked tanystropheids (e.g., *Tanystropheus, Macrocnemus, Langobardisaurus,* and *Tanytrachelos*) represent the most morphologically diverse and best-known members of "Protorosauria". Formerly, the enigmatic arboreal drepanosaurids were also referred to the clade, but they have recently been revealed to represent a separate clade of non-saurian diapsids (*Pritchard & Nesbitt, 2017*; *Pritchard et al., 2016*). As Permo-Triassic non-archosauriform archosauromorphs, "protorosaurs" represent some of the earliest members of the lineage that gave rise to Archosauria and as such are important both for our understanding of early archosauromorph evolution and the acquisition of traits within the archosaur character complex. For instance, the Chinese *Dinocephalosaurus orientalis* and an unnamed closely related taxon represent the only known viviparous archosauromorphs (*Li, Rieppel & Fraser, 2017*; *Liu et al., 2017*).

Recent cladistic studies have extensively dealt with early archosauromorph phylogeny (early Archosauria, *Nesbitt, 2011*; early Archosauromorpha with a focus on proterosuchians, *Ezcurra, 2016*; Allokotosauria, *Nesbitt et al., 2015*; Rhynchosauria, *Butler et al., 2015* and *Ezcurra, Montefeltro & Butler, 2016*; and Tanystropheidae, *Pritchard et al., 2015*). These, and some earlier analyses, indicate that "Protorosauria" does not form a monophyletic clade as historically considered, but rather represents a paraphyletic or polyphyletic grouping of non-archosauriform archosauromorphs (Fig. 1, but for an exception see *Simões et al., 2018*, who recovered Protorosauria excluding *Prolacerta* as a monophyletic clade outside Archosauromorpha). However, none of these analyses were constructed to specifically address the interrelationships of "Protorosauria" and many recently described taxa (e.g., the genera *Pectodens, Fuyuansaurus, Dinocephalosaurus, Raibliania, Elessaurus,* and *Sclerostropheus*) attributed to the group or an equivalent grade in the archosauromorph tree have not been included (*Dalla Vecchia, 2020*; *De Oliveira et al., 2020*; *Fraser, Rieppel & Li, 2013*; *Li et al., 2017*; *Rieppel, Li & Fraser, 2008*; *Spiekman & Scheyer, 2019*). Moreover, the two best-known tanystropheid genera, *Tanystropheus* and *Macrocnemus*, were recently revised extensively, revealing much additional morphological information, particularly with regards to the skull, which has not been incorporated in the abovementioned analyses (*Miedema et al., 2020*; *Spiekman et al., 2020a, 2020b*).

Here we present an extensive phylogenetic analysis, focusing on "protorosaur" and other non-archosauriform archosauromorph interrelationships. The new dataset includes 42 operational taxonomic units (OTUs), of which 23 are "protorosaurs", and employs 307 morphological characters, many of which are new or distinctly revised from previous analyses. Since the definition of "Protorosauria" in the literature is inconsistent, with many taxa having been placed alternately within and outside the group, we first provide a historical overview of "protorosaur" systematics and discuss the taxa that have formerly been included in the group. Several of these are represented by very fragmentary material or have since been identified as belonging to an entirely separate lineage to that of the archosauromorph "protorosaurs", and they were therefore not included in our phylogenetic analysis.

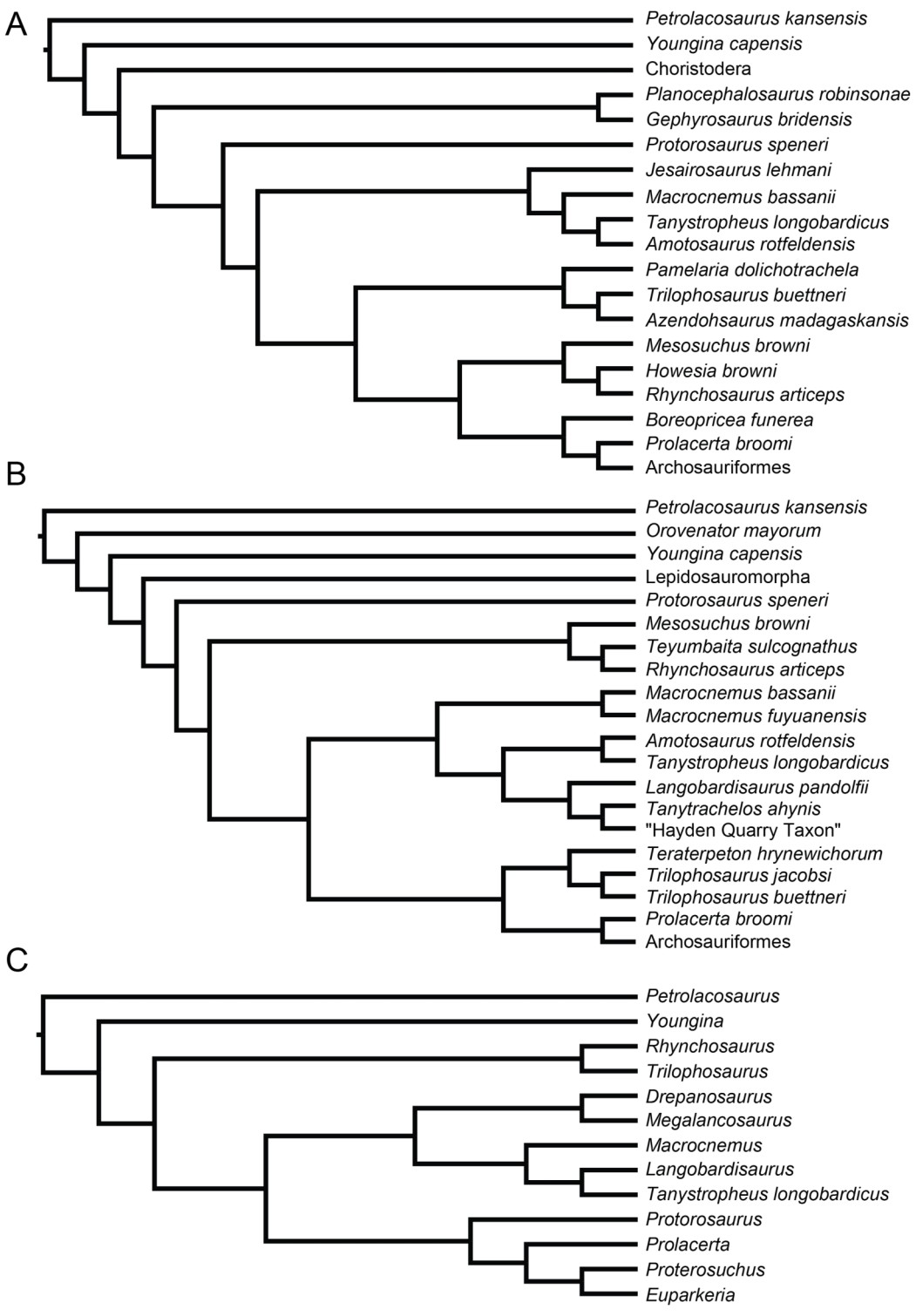

**Figure 1 Selected phylogenetic hypotheses for "protorosaur" relationships.** (A) *Ezcurra (2016)*. (B) *Pritchard et al. (2015)*. (C) *Rieppel, Fraser & Nosotti (2003)*, which represents a compilation of the matrices of *Benton & Allen (1997)*, *Jalil (1997)*, and *Dilkes (1998)*.

## Historical background of "Protorosauria"

*Protorosaurus speneri* is one of the earliest known fossil reptiles, first described in Latin by *Spener (1710)*. He considered *Protorosaurus* to be a crocodile, with many similarities specifically to the Nile crocodile, *Crocodylus niloticus* (*Gottmann-Quesada & Sander, 2009*). More than a century later, *Protorosaurus* was recognized as an extinct reptile (*Meyer, 1830*), and subsequently assigned a species definition (*Meyer, 1832*) and covered in an extensive monograph (*Meyer, 1856*). The clade "Protorosauria", with *Protorosaurus* as the only representative, was erected by *Huxley (1871)*, as part of Sauropsida, then as now considered to be the clade that encompasses all modern birds and reptiles and their direct ancestors. In his classification of the reptiles, *Osborn (1903)* provided the first definition of "Protorosauria" and assigned *Palaeohatteria*, a synapsid (*Fröbisch et al., 2011*), and *Kadaliosaurus*, an araeoscelid diapsid (*DeBraga & Reisz, 1995*), to the clade. Therein, the group was closely related to dinosaurs. *Williston (1925)* placed "protorosaurs" within "Parapsida" alongside squamates, ichthyosaurs, and mesosaurs. Other genera that were included within "Protorosauria" were *Sapheosaurus* and *Pleurosaurus*, now firmly established rhynchocephalians (*Hsiou et al., 2019*; *Rauhut et al., 2012*), and *Araeoscelis* and *Aphelosaurus*, now considered to be non-neodiapsid diapsids (*Ezcurra, Scheyer & Butler, 2014*; *Reisz, Modesto & Scott, 2011*).

After extensive excavations at the Anisian-Ladinian deposits of Monte San Giorgio on the border between Switzerland and Italy, newly discovered specimens allowed for the first comprehensive description of both *Tanystropheus longobardicus* and *Macrocnemus bassanii* (*Peyer, 1931*; *1937*). Initially *Tanystropheus longobardicus* was placed within a newly erected suborder "Tanysitrachelia", which apart from *Tanystropheus* also included *Trachelosaurus fischeri*, a small, long-necked reptile from the Buntsandstein (Early to Middle Triassic) of Germany (*Broili & Fischer, 1918*). "Tanysitrachelia" was placed within Sauropterygia (*Peyer, 1931*). *Trachelosaurus* is only known from limited disarticulated remains and its phylogenetic position is uncertain, although it is currently still considered a "protorosaur" (*Benton & Allen, 1997*; *Jalil, 1997*; *Rieppel, Fraser & Nosotti, 2003*). However, in the later report on *Macrocnemus bassanii*, *Peyer (1937)* found many similarities between *Protorosaurus* and both *Macrocnemus* and *Tanystropheus*, and therefore both taxa were reassigned to "Protorosauria", which was considered closely related to squamates and rhynchocephalians rather than archosaurs therein.

Around the same time *Prolacerta broomi* was described and assigned to the newly erected family "Prolacertidae" (*Parrington, 1935*). "Prolacertidae" was classified as part of "Thecodontia", a group that was at the time considered either as a "primitive" lineage within Archosauria (*Watson, 1917*), or ancestral to both archosaurs and lepidosaurs (*Broom, 1914*). "Thecodontia" is now unequivocally a paraphyletic grouping and has been abandoned as a clade (*Benton, 2005*). However, based on its incomplete infratemporal bar, *Prolacerta* was considered to be intermediate between "lacertilians" (i.e., squamates) and more "primitive thecodonts" such as *Youngina capensis* (*Parrington, 1935*). The description of a new specimen of *Prolacerta* led to the consideration that it was more

closely related to *Protorosaurus* and resulted in the first inclusion of *Prolacerta* into "Protorosauria" (*Camp, 1945*). *Camp (1945)* favored "Protorosauria" over "Eosuchia" based on seniority, and included taxa placed in "Eosuchia", "Trachelosauria", and "Protorosauria" by *Williston (1925)* within this clade and established it within Lepidosauria. This superorder "Protorosauria" was further subdivided in the orders "Prolacertiformes", which he synonymized with "Eosuchia" (sensu *Broom (1914)*, meaning it also included "Younginiformes"), "Trachelosauria" or Tanystropheidae, and, more tentatively, Thalattosauria and "Acrosauria" (the latter containing the rhynchocephalian pleurosaurids).

*Kuhn-Schnyder (1954)* defined the Middle Triassic *Macrocnemus* and *Tanystropheus* as squamates (German: Eidechsen, which literally translates to lacertids) that were morphologically intermediate between the Jurassic squamates and the supposed "squamate ancestor" *Prolacerta*. *Protorosaurus* was not considered, since this interpretation was based mainly on skull anatomy, which was insufficiently understood in *Protorosaurus* at this point. This hypothesis differed from that of *Colbert (1945*, *1965)* and *Romer (1956*, *1966*, *1968)*, who considered "protorosaurs" as "Euryapsida" (sometimes also called "Synaptosauria"; *Cope, 1900*), a clade which consisted of "protorosaurs" and sauropterygians, thus being similar to Sauropterygia as defined previously by *Peyer (1931)*. This classification was largely based on the temporal fenestration of the skull and represented an important systematic paradigm for amniotes. "Euryapsids" were considered as a group that was separate from "anapsids", synapsids, and diapsids, based on the presence of an upper temporal fenestra surrounded by the postorbital, squamosal, and parietal, and the absence of a lower temporal fenestra. The inclusion of "protorosaurs" within "euryapsids" was mainly based on *Araeoscelis*, which shows this fenestration type, in contrast to other "protorosaurs" that show the typical diapsid condition. Among others, the "Protorosauria" of *Romer (1966)* included *Protorosaurus*, *Tanystropheus*, *Trachelosaurus*, and *Trilophosaurus* (the last taxon is currently considered an allokotosaur within non-archosauriform Archosauromorpha; *Ezcurra, 2016*; *Nesbitt et al., 2015*; *Sengupta, Ezcurra & Bandyopadhyay, 2017*). On the other hand, *Prolacerta* and *Macrocnemus* were assigned to "Prolacertiformes" within "Eosuchia", interpreted as the "basalmost" order of Lepidosauria ("Eosuchia" was maintained *contra Camp, 1945*). "Euryapsida" has generally not been used as a grouping in recent years, and its former members are now distributed within Diapsida (*Benton, 2005*; *Merck, 1997*). Furthermore, an extensive redescription of *Araeoscelis* has shown various differences with taxa such as *Protorosaurus* and *Prolacerta* (*Vaughn, 1955*), and it is now considered an early diapsid that it is not closely related to "protorosaurs" (*Ford & Benson, 2020*). The hypothesis of "Euryapsida" comprised of Sauropterygia and "Protorosauria" was criticized by *Kuhn-Schnyder (1963*, *1967*, *1974)*. *Kuhn-Schnyder (1967)* and *Wild (1973)* argued that because of the ventrally opened lower temporal bar of *Macrocnemus* and *Tanystropheus*, "Protorosauria", including *Protorosaurus*, belonged to "Prolacertidae" within Lepidosauria.

*Prolacerta* was extensively redescribed by *Gow (1975)* based on new material, which included the first detailed description of postcranial remains. This study was the first to

conclude that *Prolacerta*, together with *Macrocnemus* and *Tanystropheus*, was not part of the lepidosaurian lineage, but instead was archosaurian in many of its features. These taxa were grouped in the newly erected order "Parathecodontia", with *Prolacerta* and *Macrocnemus* being further classified together within "Prolacertidae" and *Tanystropheus* within Tanystropheidae. Nevertheless, the need for a detailed phylogenetic re-examination of these taxa was stressed and this revision did not consider *Protorosaurus*.

In the 1970s and the subsequent two decades, a considerable number of taxa were included within "Protorosauria", further indicating the significance of the group: *Tanytrachelos ahynis* (*Olsen, 1979*), *Langobardisaurus pandolfii* (*Bizzarini & Muscio, 1995*), *Cosesaurus aviceps* (originally considered an avian ancestor; *Ellenberger & De Villalta, 1974*, but later designated as a "protorosaur" by *Olsen, 1979*), *Malerisaurus robinsonae* (*Chatterjee, 1980*), *Kadimakara australiensis* (*Bartholomai, 1979*), *Prolacertoides jimusarensis* (*Young, 1973*), *Malutinisuchus gratus* (*Ochev, 1986*), and *Boreopricea funerea* (*Tatarinov, 1978*). In addition, "Protorosauria" as designated by *Evans (1988)* included *Megalancosaurus preonensis*, a member of the Drepanosauridae, a family of highly specialized, arboreal diapsids (*Calzavara, Muscio & Wild, 1980*; *Pritchard et al., 2016*; *Renesto et al., 2010*). *Chatterjee (1980)* also included the Carboniferous *Petrolacosaurus kansensis* within "Prolacertiformes", although this view was swiftly disputed (*Evans, 1988*; *Reisz, Berman & Scott, 1984*), and *Petrolacosaurus* is now widely considered an araeoscelid diapsid instead (*Ezcurra, Scheyer & Butler, 2014*; *Ford & Benson, 2020*; *Reisz, Modesto & Scott, 2011*).

Cladistics became widespread as a method for establishing phylogenetic relationships between taxa during the 1980s and its implementation on diapsid phylogeny quickly led to a relatively clear-cut division between Lepidosauromorpha and Archosauromorpha, with "Protorosauria" firmly established within the latter group (*Bennett, 1996*; *Benton, 1984, 1985*; *Evans, 1988*; *Gauthier, 1984, 1994*; *Gauthier, Kluge & Rowe, 1988*). *Chatterjee (1986)* pointed out the priority of "Protorosauria" over "Prolacertiformes" based on seniority, but since "Protorosauria" had previously often included *Araeoscelis* and was therefore shown to be polyphyletic, many authors preferred "Prolacertiformes" (see *Evans, 1988*, pages 226–227 for an overview of the use of both terms within the literature between 1945 and 1988). However, although the place of "protorosaurs" among Archosauromorpha became firmly established, the interrelationships of the various "protorosaurs" was not evaluated cladistically except by *Chatterjee (1986)* and *Evans (1988)*. *Olsen (1979)* and *Wild (1980a)* also provided a hypothesis of "protorosaur" interrelationships on a non-cladistic basis.

This issue would soon be addressed in more detail in several papers. One study included 11 "protorosaurs" (excluding the poorly known *Prolacertoides*) and three outgroups and 48 morphological characters (*Benton & Allen, 1997*). In the same year, the description of a new "protorosaur", *Jesairosaurus lehmani*, was accompanied by an analysis including ten "protorosaurs" and eight outgroup taxa, employing 71 characters (*Jalil, 1997*; the initial analysis also included *Trachelosaurus*, *Prolacertoides*, *Malutinisuchus*, and *Kadimakara*, but these poorly known taxa were excluded from the final analysis, as the inclusion of these taxa left "protorosaurs" interrelationships unresolved). Another study addressing early archosauromorph phylogeny also included several "protorosaurs" (*Dilkes, 1998*). This

analysis included 144 characters and 23 taxa, out of which seven were traditionally considered as "protorosaurs", including two drepanosaurid taxa, which were not included in *Benton & Allen (1997)* and *Jalil (1997)*. It recovered a monophyletic "Protorosauria" in which *Protorosaurus* formed a sister taxon to two lineages, Drepanosauridae and Tanystropheidae, whereas *Prolacerta* was placed outside the clade as the sister taxon of Archosauriformes. *Peters (2000)* used the matrices of *Evans (1988)*, *Jalil (1997)*, and *Bennett (1996)* and reran each of them after adding a number of characters and rescoring some characters for certain taxa, for a total taxon sample that included 11 "protorosaurs", other non-archosauriform archosauromorphs, the pterosaur *Eudimorphodon*, and two enigmatic and possibly gliding diapsids, *Longisquama insignis* (*Sharov, 1970*) and *Sharovipteryx mirabilis* (*Cowen, 1981*; *Sharov, 1971*). *Sharovipteryx* is an enigmatic gliding reptile with a membrane stretched between the hindlimbs, which represents an entirely unique morphology among gliding reptiles. It has been tentatively ascribed to "protorosaurs" or tanystropheids by some authors (*Gans, Darevski & Tatarinov, 1987*; *Pritchard & Sues, 2019*; *Tatarinov, 1989*, *1994*; *Unwin, Alifanov & Benton, 2000*), but its phylogenetic position is highly uncertain due to its highly specialized, yet very poorly known morphology. *Peters (2000)* found "protorosaurs", and *Longisquama* and *Sharovipteryx*, to be very closely associated with *Eudimorphodon*, from which a "protorosaurian" ancestry for pterosaurs was concluded. However, the exact topologies varied strongly between the different analyses, and this hypothesis of pterosaur ancestry has widely been rejected by other phylogenetic studies on pterosaurs and early archosaurs (e.g., *Ezcurra, 2016*; *Ezcurra et al., 2020*; *Hone & Benton, 2007*; *Nesbitt, 2011*; *Padian, 1997*). The datasets of *Benton & Allen (1997)*, *Dilkes (1998)*, and *Jalil (1997)* were combined into one larger character list of 239 characters by *Rieppel, Fraser & Nosotti (2003)*, which was used specifically to address "protorosaur" phylogeny, and in particular the question of "protorosaur" monophyly, which had now been put in doubt (*Dilkes, 1998*). This approach included seven "protorosaur" taxa (*Protorosaurus*, *Drepanosaurus*, *Megalancosaurus*, *Prolacerta*, *Macrocnemus*, *Langobardisaurus*, and *Tanystropheus longobardicus*), and four outgroup taxa (*Petrolacosaurus*, *Youngina*, *Rhynchosaurus*, and *Trilophosaurus*). Additional analyses were performed after subsequently including *Euparkeria* and *Proterosuchus*, and the lesser known "protorosaurs" *Boreopricea* and *Jesairosaurus*. Although the first analysis found a monophyletic "Protorosauria", the other two resulted in paraphyly for the group. Although *Rieppel, Fraser & Nosotti (2003)* concluded that the monophyly of "Protorosauria" as previously regarded (e.g., *Benton & Allen, 1997*; *Jalil, 1997*) could not be maintained, they argued the need for an extensive phylogenetic investigation into "protorosaurs". *Senter (2004)* investigated the phylogenetic position of drepanosaurids in an analysis that comprised "protorosaurs" (*Prolacerta*, *Macrocnemus*, and *Langobardisaurus*), *Longisquama*, non-archosaurian Archosauriformes, birds, a non-avian dinosaur, and a number of early diapsids. This study found drepanosaurids to form a clade with *Longisquama* and *Coelurosauravus*, which was termed "Avicephala", as the sister group to Neodiapsida, which in his analysis encompassed *Youngina*, the rhynchocephalian *Gephyrosaurus*, and several archosauromorphs. The included "protorosaurs" formed a monophyletic clade within Archosauromorpha. However, an analysis using the same character list by *Renesto & Binelli*

(2006) could not reproduce the same topology. *Renesto et al. (2010)* reaffirmed the position of drepanosaurids among "protorosaurs", whereas *Pritchard & Nesbitt (2017)* recovered Drepanosauromorpha as a separate clade of non-saurian diapsids. *Müller (2004)* included four different "protorosaur" taxa in his broad-scale analysis of diapsid relationships, which consisted of 184 characters compiled mainly from *Rieppel, Mazin & Tchernov (1999)* and *DeBraga & Rieppel (1997)*. This study also inferred a polyphyletic "Protorosauria", with *Tanystropheus*, *Macrocnemus*, and *Prolacerta* being successive sister taxa to rhynchosaurs and *Trilophosaurus*, whereas drepanosaurids were only quite distantly related to these taxa.

"Protorosaurs" were virtually unknown from China until about 15 years ago, with the exception of the poorly known and tentative "protorosaur" *Prolacertoides jimusarensis* (*Young, 1973*). However, a number of new finds have been referred to "Protorosauria", including *Tanystropheus* cf. *longobardicus* (*Rieppel et al., 2010*; now *Tanystropheus* cf. *hydroides*, see *Spiekman et al., 2020a*), *Tanystropheus* sp. (*Li, 2007*), and *Macrocnemus fuyuanensis* (*Jiang et al., 2011*; *Li, Zhao & Wang, 2007*), forms very similar to European counterparts, as well as completely new taxa, such as *Dinocephalosaurus orientalis* (*Li, 2003*; *Li, Rieppel & LaBarbera, 2004*; *Liu et al., 2017*; *Rieppel, Li & Fraser, 2008*), *Fuyuansaurus acutirostris* (*Fraser, Rieppel & Li, 2013*), an unnamed taxon closely related to *Dinocephalosaurus* (*Li, Rieppel & Fraser, 2017*), and potentially *Pectodens zhenyuensis* (*Li et al., 2017*). This has revealed that "protorosaurs" had a Tethys-wide distribution and are considerably more morphologically diverse than previously appreciated. Except for *Dinocephalosaurus orientalis*, which has been included in phylogenetic analyses of *Rieppel, Li & Fraser (2008)*, *Liu et al. (2017)*, and *De Oliveira et al. (2020)*, none of these Chinese taxa have been phylogenetically assessed so far except by the matrix of *Ezcurra & Butler (2018)*. However, the aim of this analysis was not to investigate the phylogenetic relationships between the included taxa, but rather to serve as a discrete character matrix to investigate their morphological disparity.

But new "protorosaur" findings have also been reported from outside of China. *Fraser & Rieppel (2006)* re-examined the *"Tanystropheus antiquus"* material from the Upper Buntsandstein of Baden-Württemberg, Germany, and assigned it to a new taxon, *Amotosaurus rotfeldensis*. Furthermore, *Gottmann-Quesada & Sander (2009)* provided a monograph on the German *Protorosaurus speneri* material, including the first detailed description and reconstruction of the skull, based on the discovery of a well-preserved skull in 1972, which previously had only been briefly documented (see *Haubold & Schaumberg, 1985* p. 223; *Fichter, 1995* and references therein). *Gottmann-Quesada & Sander (2009)* also provided a phylogenetic analysis, which employed the matrix of *Dilkes (1998)*, with several modifications to the character scorings of *Mesosuchus*, *Prolacerta*, and *Protorosaurus*. This resulted in a tree with a polyphyletic "Protorosauria" that recovered *Protorosaurus* as the sister taxon to *Megalancosaurus*. A new species of *Macrocnemus*, *Macrocnemus obristi*, has been described from Alpine Europe (*Fraser & Furrer, 2013*), and a specimen from Monte San Giorgio on the border of Switzerland and Italy was recently assigned to *Macrocnemus fuyuanensis*, a species that was previously only known from China (*Jaquier et al., 2017*; *Scheyer et al., 2020b*). A new species of *Tanystropheus*, *Tanystropheus hydroides*, has also been described from Monte San Giorgio (*Spiekman*

 

*et al., 2020a*). This new species was previously considered to represent the adult stage of *Tanystropheus longobardicus* (*Wild, 1973*), but long bone histology revealed that the small-sized specimens of *Tanystropheus longobardicus* were skeletally mature, thus representing a separate species from the newly recognized *Tanystropheus hydroides*. Two new "protorosaurs" have been reported from Russia based on limited, isolated remains: the large-sized *Vritramimosaurus dzerzhinskii*, considered to be closely related to *Prolacerta* (*Sennikov, 2005*), and *Augustaburiania vatagini*, a medium-sized tanystropheid (*Sennikov, 2011*). From Poland two new, possibly "protorosaur", archosauromorphs have been described. *Czatkowiella harae* has been interpreted as being closely related to *Protorosaurus* (*Borsuk-Białynicka & Evans, 2009b*), whereas the highly gracile, and putative glider, *Ozimek volans* is similar to *Sharovipteryx* (*Dzik & Sulej, 2016*). *Ezcurra, Scheyer & Butler (2014)* re-examined material consisting of five vertebrae, three fragmented forelimb elements, and some indeterminable fragments from the late Permian of Tanzania previously described by *Parrington (1956)* and assigned it to the new taxon *Aenigmastropheus*. Following an analysis modified from *Reisz, Laurin & Marjanović (2010)*, used to address both synapsid and diapsid relationships, it was recovered among "protorosaurs" as the sister taxon to *Protorosaurus*. In addition, they found *Eorasaurus*, previously assigned as a "protorosaur" by *Sennikov (1997)*, to likely be an archosauriform, which would make *Aenigmastropheus* the second known "protorosaur" and non-archosauriform archosauromorph from the Permian, the other being *Protorosaurus*. Two more tanystropheid genera, *Sclerostropheus fossai* and *Raibliania calligarisi* were recently identified, based on partial postcranial remains (*Dalla Vecchia, 2020*; *Spiekman & Scheyer, 2019*). Finally, recent findings have shone light on the occurrence and distribution of tanystropheids in the Americas. Isolated material from the Middle and Late Triassic of North America, largely consisting of cervical vertebrae, as well as some other postcranial remains, indicate that tanystropheids were more widespread than previously thought (*Formoso et al., 2019*; *Lessner et al., 2018*; *Pritchard et al., 2015*; *Sues & Olsen, 2015*). From South America, archosauromorph remains from the Induan to early Olenekian of Brazil have been described, and a new species, *Elessaurus gondwanoccidens*, was recovered as the sister taxon to Tanystropheidae (*De Oliveira et al., 2018*; *De Oliveira et al., 2020*). If this South American material is referrable to Tanystropheidae, it would represent some of the earliest records of the clade, and would indicate a wide, if not nearly cosmopolitan distribution for Tanystropheidae during the Early Triassic. In addition, material with possibly "protorosaur" affinities are also known from the Permo-Triassic Buena Vista Formation of northeastern Uruguay (*Ezcurra et al., 2015*).

The original phylogenetic matrices by *Pritchard et al. (2015)* and *Ezcurra (2016)*, and their subsequently modified iterations (e.g., *Butler et al., 2019*; *Ezcurra & Butler, 2018*; *Ezcurra et al., 2017*; *2019*; *Maidment et al., 2020*; *Nesbitt et al., 2017a*, *2015*; *Pritchard et al., 2018*; *Pritchard & Nesbitt, 2017*; *Pritchard & Sues, 2019*; *Scheyer et al., 2020a*; *Sengupta, Ezcurra & Bandyopadhyay, 2017*; *Spiekman, 2018*; *Spiekman et al., 2020a*; *Stocker et al., 2017*), represent the two separate datasets that most comprehensively addressed "protorosaur" relationships. The former focused specifically on tanystropheid relationships. It found *Protorosaurus* as the sister taxon to all other archosauromorphs,

whereas *Prolacerta* formed the sister taxon to Archosauriformes. Tanystropheidae was recovered as a monophyletic clade and consisted of *Macrocnemus*, *Amotosaurus*, *Tanystropheus*, *Langobardisaurus*, *Tanytrachelos*, and the new Hayden Quarry material that was presented therein. The character list consisted of 200 characters, including novel characters and characters derived from many previous analyses (*Benton, 1985*; *Benton & Allen, 1997*; *Conrad, 2008*; *DeBraga & Rieppel, 1997*; *Dilkes, 1998*; *Gauthier, 1984*; *Gauthier, Estes & De Queiroz, 1988*; *Gauthier, Kluge & Rowe, 1988*; *Hutchinson, Skinner & Lee, 2012*; *Jalil, 1997*; *Merck, 1997*; *Modesto & Sues, 2004*; *Müller, 2004*; *Nesbitt, 2011*; *Rieppel, 1994*). *Ezcurra (2016)* presented a very extensive analysis of early archosauromorph interrelationships that used 600 characters to analyze 96 taxa. Out of these characters, 96 were new. The remaining characters were compiled from the literature (mainly *Desojo, Ezcurra & Schultz, 2011*; *Dilkes, 1998*; *Dilkes & Arcucci, 2012*; *Ezcurra, Desojo & Rauhut, 2015*; *Ezcurra, Lecuona & Martinelli, 2010*; *Ezcurra, Scheyer & Butler, 2014*; *Gower & Sennikov, 1996*; *Gower & Sennikov, 1997*; *Jalil, 1997*; *Nesbitt, 2011*; *Nesbitt et al., 2015*; *Parrish, 1992*; *Pritchard et al., 2015*; *Senter, 2004*; *Trotteyn & Ezcurra, 2014*). Like *Pritchard et al. (2015)*, it found *Protorosaurus* to be the sister taxon to all other archosauromorphs. *Prolacerta* was recovered as the sister taxon to *Kadimakara australiensis*, and the clade formed by these two taxa was the sister clade to Archosauriformes + *Tasmaniosaurus triassicus*. *Boreopricea* was found as the sister taxon to Prolacertidae + *Tasmaniosaurus* + Archosauriformes, whereas *Jesairosaurus* formed the sister taxon to a monophyletic Tanystropheidae, composed of *Macrocnemus*, *Amotosaurus*, and *Tanystropheus*.

## Overview of "protorosaur" taxa

In the following, an overview is provided of taxa that have previously been assigned to "Protorosauria", but which have not been included in the present analysis, since they are either represented by insufficient material for inclusion or because it is now widely considered that they are not closely related to *Protorosaurus speneri*, *Prolacerta broomi*, or Tanystropheidae.

### *Aenigmastropheus parringtoni* Ezcurra, Scheyer & Butler, 2014.

*Aenigmastropheus parringtoni* is known from a single specimen, UMZC T836, from the Wuchiapingian (middle late Permian) of Tanzania. It comprises five cervical and dorsal vertebrae, the distal part of a right humerus, the proximal part of the right ulna, and several small fragments. The specimen was first described by *Parrington (1956)* and was considered to be insufficiently preserved for a confident taxonomic diagnosis. However, it was noted that its morphology contained both "primitive" diapsid traits as well as archosaurian characteristics. The specimen was recently revised and assigned to a new taxon, which was recovered as the sister taxon to *Protorosaurus speneri* in a cladistic analysis (*Ezcurra, Scheyer & Butler, 2014*). A later analysis found *Aenigmastropheus parringtoni* as the sister taxon to all other archosauromorphs (*Ezcurra, 2016*).

### *Kadimakara australiensis* Bartholomai, 1979.

*Kadimakara australiensis* is known from two partial skulls first described by *Bartholomai (1979)*. The holotype is represented by the postorbital region, whereas the other specimen comprises a partial snout. Although both specimens do not have any shared preserved regions, they were attributed to the same taxon based on their similar size and shared close similarity to *Prolacerta broomi*. The validity of *Kadimakara australiensis* has been questioned and *Borsuk-Białynicka & Evans (2009b)* and *Evans & Jones (2010)* considered the specimens to be congeneric with *Prolacerta broomi*. *Ezcurra (2016)* corroborated the close affinity of *Kadimakara australiensis* to *Prolacerta broomi*, but only considered the holotype in the revised diagnosis of the taxon therein, since the lack of overlapping morphology precludes the direct comparison between the holotype and referred specimen. *Ezcurra (2016)* argued in favour of the validity of *Kadimakara australiensis*, pointing out a medial fossa on the posterior half of the parietals as a distinguishing feature between this species and *Prolacerta broomi*. However, other distinguishing features indicated by *Bartholomai (1979)* were revealed to result from an erroneous interpretation of the morphology of the postorbital bar. *Kadimakara australiensis* comes from the lower beds of the upper part of the Arcadia Formation, central Queensland, Australia, which are of Induan (earliest Triassic) age.

### *Megacnemus grandis* Huene, 1954.

*Megacnemus grandis* was described based on one isolated long bone exceeding 20 cm in length, which was identified as a femur (*Huene, 1954*). Although the provenance of the specimen is unknown, *Huene (1954)* considered the specimen to most likely derive from the Gogolin Formation (Lower Muschelkalk) of southwest Poland, which is lower Anisian (Middle Triassic) in age, based on a similarity in preservation to fossils known from this formation. *Skawiński, Talanda & Sachs (2015)* re-examined the specimen and corroborated its "protorosaurian" affinities. However, they also considered the possibility that the specimen represents a humerus rather than a femur, and therefore only identified the bone as a propodial. It has not been included in any phylogenetic analyses due to its extremely poorly known morphology.

### *Trachelosaurus fischeri* Broili & Fischer, 1918.

*Trachelosaurus fischeri* is represented by three associated slabs originating from the Thüringischer Chirotheriensandstein (Buntsandstein) of Bernburg (late Olenekian to early Anisian, late Early to early Middle Triassic), north of Halle, Germany (*Schoch, 2019*). The remains preserved on these slabs comprise a premaxilla, two additional incomplete skull elements, associated gastralia, a likely coracoid, an ilium, a femur, and many, mostly disassociated, vertebrae and ribs, representing all sections of the vertebral column. The cervical ribs are remarkable in that they are short and bifurcated posteriorly. Many of the dorsal vertebrae bear conspicuously wide transverse processes. Alongside these elements referred to *Trachelosaurus fischeri* pedal or manual imprints and several fish scales are also preserved on the slabs. This material was first mentioned and partially figured by *Huene (1902)* when it was only partially prepared, and it was considered to

represent a taxon closely related to *Protorosaurus speneri*. After further preparation, the species *Trachelosaurus fischeri* was erected and fully described based on this material by *Broili & Fischer (1918)*. Therein, it was concluded that the specimen showed similarities to *Protorosaurus speneri* in the shape of the cervical vertebrae, but also differed distinctly from this taxon in the number of cervical vertebrae, which was interpreted to be 20 or 21, the height of the neural spines of the dorsal vertebrae, and the shape of the femur, which is relatively shorter and stockier compared to *Protorosaurus speneri* (*Broili & Fischer, 1918*). *Trachelosaurus fischeri* was assigned to Sauropterygia and was suggested to have been closely related to nothosaurs. *Huene (1944)* later revised the material of *Trachelosaurus fischeri* based on the detailed drawings presented by *Broili & Fischer (1918)* and only identified 15 cervical vertebrae, nine of which were preserved in articulation. Four additional vertebrae, which were interpreted as cervical vertebrae by *Broili & Fischer (1918)*, were reinterpreted as caudal vertebrae. *Huene (1944)* disagreed with the high vertebral number of the cervical column suggested by *Broili & Fischer (1918)* and therefore concluded that the 15 cervical vertebrae belonged to more than one individual of *Trachelosaurus fischeri*. Furthermore, several disassociated vertebrae that are considerably smaller than the rest of the preserved vertebrae, were considered to belong to an unidentified taxon other than *Trachelosaurus fischeri*. Considering the reduced cervical count of *Trachelosaurus fischeri*, *Huene (1944)* found that its morphology was in clear correspondence with that of a "protorosaur", with the notable deviation in the morphology of the cervical ribs. *Trachelosaurus fischeri* has been incorporated in several previous phylogenetic analyses. It formed a large polytomy at the base of Prolacertiformes in the analysis of *Evans (1988)*. In one of the analyses presented by *Benton & Allen (1997)* *Trachelosaurus fischeri* formed a polytomy with *Tanystropheus meridensis* (now considered a junior synonym of *Tanystropheus longobardicus*, *Spiekman & Scheyer, 2019*) and a clade comprising *Tanystropheus longobardicus* and *Tanytrachelos ahynis*. However, *Trachelosaurus fischeri* was excluded from the final analysis in this study to increase tree resolution. Similarly, *Trachelosaurus fischeri* was part of a large polytomy within Prolacertiformes in one analysis and subsequently excluded from the final analysis to increase tree resolution in *Jalil (1997)*. *Trachelosaurus fischeri* was also omitted from the analyses to optimize tree resolution by *Rieppel, Fraser & Nosotti (2003)*. A revision of the holotype of *Trachelosaurus fischeri*, including detailed figures, is desirable and should carefully consider how many individuals and taxa are represented on the slabs.

### *Prolacertoides jimusarensis* Young, 1973.

*Prolacertoides jimusarensis* represents the first described "protorosaur" from China. It is known from a single, poorly preserved skull (IVPP V3233) and was considered to be closely related to *Prolacerta broomi* in its initial description in Chinese (*Young, 1973*). *Ezcurra (2016)* provided a more detailed osteological description of the holotype in English, but *Prolacertoides jimusarensis* was omitted from the final analysis therein, due to a lack of preserved, phylogenetically informative morphological characters. In analyses 1 and 2 of *Ezcurra (2016)*, which included *Prolacertoides* and other poorly represented taxa, *Prolacertoides jimusarensis* was positioned in a large polytomy including most early

archosauromorphs with *Protorosaurus speneri* as the sister taxon of that clade. *Prolacertoides jimusarensis* has previously been included in the phylogenetic analyses of *Benton & Allen (1997)*, *Evans (1988)*, *Jalil (1997)*, and *Rieppel, Fraser & Nosotti (2003)*. Notably, *Benton & Allen (1997)* retrieved *Prolacertoides jimusarensis* as the sister taxon to *Trilophosaurus buettneri* and to thus fall outside of the traditional "Protorosauria". However, its exact phylogenetic affiliations were questioned by all authors due to a lack of morphological information for the taxon.

### *Rhombopholis scutulata* Owen, 1842.

*Rhombopholis scutulata* was originally described as an amphibian (*Owen, 1842*). It is represented by a single block that contains a number of postcranial bones, including four vertebrae, several ribs, and five limb elements that belong to at least two individuals (*Benton & Walker, 1996*). In a revision of the reptile material from the Keuper Sandstone Group of England (Anisian, Middle Triassic), this specimen together with some other specimens were considered to be closely related to *Macrocnemus bassanii* (*Walker, 1969*). *Benton & Walker (1996)* provided a redescription of *Rhombopholis scutulata*, and identified it as a "prolacertiform" metataxon, meaning that no autapomorphies could be identified to distinguish it from other "prolacertiforms" and that the different specimens possibly belong to more than one taxon.

### *Sharovipteryx mirabilis* Sharov, 1971.

A virtually complete but poorly preserved specimen with long and gracile hindlimbs and an apparent skin membrane present between the legs was initially described as *Podopteryx mirabilis* and interpreted as a gliding reptile (*Sharov, 1971*). However, because the name *Podopteryx* was already occupied by a genus of damselflies, the taxon was renamed *Sharovipteryx mirabilis* by *Cowen (1981)*. *Sharovipteryx mirabilis* was described in detail by *Gans, Darevski & Tatarinov (1987)*. Although the phylogenetic position of *Sharovipteryx mirabilis* is exceedingly hard to assess due to the lack of visible morphological details, it has been identified as a "protorosaur" by various authors (e.g., *Ivakhnenko & Kurochkin, 2008*; *Peters, 2000*; *Unwin, Alifanov & Benton, 2000*).

### *Cosesaurus aviceps* Ellenberger & De Villalta, 1974.

*Cosesaurus aviceps* is known from a single specimen, which represents an impression of a complete skeleton. As such, the outline of the specimen is well-preserved, but the detailed morphology of the taxon is very poorly known. The specimen was found at the Montral-Alcover outcrop (Ladinian, Middle Triassic), Sierra de Prades, Tarragona Province, Spain. Due to the lack of morphological information, the phylogenetic affinities of *Cosesaurus aviceps* are unclear. It was initially thought to represent an ancestor to birds (*Ellenberger, 1977*, *1978*; *Ellenberger & De Villalta, 1974*). However, the now widely accepted view that birds represent a derived clade of theropod dinosaurs refutes this hypothesis. *Cosesaurus aviceps* was redescribed by *Sanz & López-Martínez (1984)* and considered to bear many similarities to various "protorosaurs". *Cosesaurus aviceps* has also been found among "protorosaurs" in subsequent phylogenetic analyses (*Benton & Allen, 1997*; *Evans, 1988*; *Jalil, 1997*; *Rieppel, Fraser & Nosotti, 2003*). In a reinterpretation of

previous analyses by *Peters (2000)*, which has been widely criticized (e.g., *Hone & Benton, 2007*), it was concluded that Pterosauria are a derived lineage within "Prolacertiformes". This was largely based on several morphological characters observed in *Cosesaurus aviceps*, as well as the poorly known, gracile reptiles *Sharovipteryx mirabilis* and *Longisquama insignis* (*Peters, 2000*). Although *Cosesaurus aviceps* might represent a "protorosaur", the lack of morphological information does not allow this taxon to be reliably incorporated in phylogenetic analyses, and recent phylogenetic investigations into archosauromorph or "protorosaurian" affinities did not consider this taxon.

### *Vritramimosaurus dzerzhinskii* Sennikov, 2005.

The holotype of *Vritramimosaurus dzerzhinskii* is a single cervical vertebra, and referred material comprises another cervical vertebra, a caudal vertebra, and two additional, fragmentary vertebrae. They were originally discovered in 1953 and 1954 by B.P. Vjuschkov. *Vritramimosaurus dzerzhinskii* has been described as a "large, specialized prolacertilian" (*Sennikov, 2005*). The material originates from the Rassypnaya locality of the Petropavlovka Formation, Orenburg Region, Russia, which is of uppermost Olenekian (Early Triassic) age. Its estimated overall body size is at least three meters, making *Vritramimosaurus dzerzhinskii* one of the larger early archosauromorphs and considerably larger than *Prolacerta broomi*, to which it is considered to be closely related (*Sennikov, 2005*). However, the limited and fragmentary material allows for only a very limited comparison to other taxa and the taxon has therefore not been included in phylogenetic analyses.

### *Microcnemus efremovi* Huene, 1940.

The referred material of *Microcnemus efremovi* consists of several isolated elements, comprising vertebrae, long bones, a partial scapulacoracoid, a largely complete ischium, and possibly a few skull elements including teeth (*Huene, 1940*). These specimens were recovered from Early Triassic sites (*Benthosuchus* assemblage zone, Olenekian) in Vologda Oblast, Russia, from which the proterosuchid *Chasmatosuchus rossicus* has also been described (*Gower & Sennikov, 2000*; *Huene, 1940*). The cervical vertebrae of *Microcnemus efremovi* are elongate and amphicoelous, with all vertebrae having been described as thin-walled (*Huene, 1940*). The partial scapulacoracoid exhibits a strongly posteriorly curved scapular blade as is typical of tanystropheids (e.g., *Macrocnemus bassanii*, PIMUZ T 4355; *Tanystropheus longobardicus*, PIMUZ T 1277). Additionally, *Huene (1940)* considered a fragment of a maxilla and a partial mandible including teeth to be possibly referrable to *Microcnemus efremovi*. The teeth were described as robust with an acrodont implantation, which deviates strongly from all known archosauromorphs. *Huene (1940)* considered the cervical vertebrae and scapulacoracoid to be most similar to those of *Macrocnemus* and *Tanystropheus*, whereas the dorsal and caudal vertebrae, as well as the tentatively included skull elements, were considered most similar to *Araeoscelis*. From the same localities *Huene (1940)* additionally referred a partial cervical vertebra to ? Protorosauridae gen. indet. that was considerably larger than the material of *Microcnemus efremovi*, and erected a new taxon, *"Chasmatosuchus parvus"*, based on another partial

vertebra. The latter element was found to be morphologically indistinguishable from *Microcnemus efremovi* by *Sennikov (1995)* and was thus reassigned to this species. Postcranial material that was initially identified as *Microcnemus* sp. (*Garyainov & Rykov, 1973*; *Rykov & Ochev, 1966*) was later reassigned to the tanystropheid taxon *Augustaburiania vatagini* by *Sennikov (2011)*. *Microcnemus efremovi* has not been included in any phylogenetic analyses but was suggested to be closely related to *Macrocnemus bassanii* by *Sennikov (2011)*.

### *"Exilisuchus tubercularis"* Ochev, 1979.

*"Exilisuchus tubercularis"* is known from a single partial ilium from the Sludkian Gorizont of the Orenburg Region, Russia (early Olenekian, Early Triassic; *Gower & Sennikov, 2000*). It has most commonly been interpreted as a possible proterosuchian, although its taxonomic assignment is uncertain due to the highly limited available morphological information (*Ezcurra, 2016*; *Gower & Sennikov, 2000*; *Ochev, 1979*). *Ezcurra (2016)* briefly described and figured the holotype and recovered *"Exilisuchus tubercularis"* within Tanystropheidae based on the presence of a dorsally rimmed caudifemoralis brevis muscle origin on the lateroventral surface of the postacetabular process. *"Exilisuchus tubercularis"* has been considered a nomen dubium by *Gower (1994)* and *Ezcurra (2016)*.

### *Malerisaurus robinsonae* Chatterjee, 1980 *and "Malerisaurus langstoni"* Chatterjee, 1986.

*Malerisaurus robinsonae* is known from two individuals that are part of the stomach content of two specimens of the phytosaur *Parasuchus hislopi* from the Maleri Formation (late Carnian, early Late Triassic) of central India (*Chatterjee, 1980*). Another specimen from the Tecovas Member, lower Dockum Formation of western Texas, US, (Carnian to early Norian, early Late Triassic) was recognized as representing a taxon that was very closely related to *Malerisaurus robinsonae* and assigned to the new species *Malerisaurus langstoni* (*Chatterjee, 1986*). However, this holotype and only known specimen is actually composed of elements belonging to several diapsid taxa, most notably *Trilophosaurus buettneri* (*Spielmann et al., 2006*). Therefore, *"Malerisaurus langstoni"* is no longer considered a valid taxon. Furthermore, the validity of the Indian *Malerisaurus robinsonae* was questioned, as this taxon also showed distinct similarities to *Trilophosaurus buettneri* (*Spielmann et al., 2006*). Following the original interpretation by *Chatterjee (1980*; *1986)* of *Malerisaurus robinsonae* as a "protorosaur" closely related to *Protorosaurus speneri*, it has been incorporated in several phylogenetic analyses (*Benton, 1985*; *Benton & Allen, 1997*; *Evans, 1988*; *Jalil, 1997*; *Rieppel, Fraser & Nosotti, 2003*). *Malerisaurus robinsonae* was removed from the final analyses due to insufficient character preservation in *Benton (1985)* and *Rieppel, Fraser & Nosotti (2003)*, whereas it was retrieved as part of a polytomy within Archosauromorpha by *Evans (1988)*. *Benton & Allen (1997)* included only *"Malerisaurus langstoni"* in the final analysis and found it as the sister taxon to all included "protorosaurs" except *Protorosaurus speneri*, *Prolacerta broomi*, and *Boreopricea funerea*. Finally, *Jalil (1997)* included both *Malerisaurus* species as a single OTU and found it to be the sister taxon to *Jesairosaurus lehmani*. *Malerisaurus* spp. have not been included

in any of the recently published phylogenetic analyses of early archosauromorphs. Recently, *Nesbitt et al. (2017b)* identified both *Malerisaurus* species as separate from *Trilophosaurus buettneri*, and considered them to belong to Allokotosauria, more specifically as members of Azendohsauridae.

### *Malutinisuchus gratus* Ochev, 1986.

*Malutinisuchus gratus* is a very poorly known taxon that has been considered a "protorosaur". It is known from Belyaevsky I, Bukobay Svita, Ladinian, Orenburg region, Russia (*Ochev, 1986*; *Tverdokhlebov et al., 2003*). The known material comprises several fragmentary remains, including an elongated cervical vertebra, two partial limb bones, and likely pectoral girdle elements. *Malutinisuchus gratus* was incorporated into phylogenetic analyses by *Jalil (1997)* and *Rieppel, Fraser & Nosotti (2003)*, but in both cases omitted from the final analysis due to a lack of morphological information. In one of the trees recovered by *Jalil (1997)*, *Malutinisuchus gratus* formed a polytomy with all other taxa forming the clade "Prolacertiformes" therein.

### *Boreopricea funerea* Tatarinov, 1978.

*Boreopricea funerea* is known from a nearly complete specimen and an anterior end of a snout, collected from a borehole, number 141, at 1,112.3 m deep at Kolguyev Island in the Barents Sea. This borehole is part of the Vetluzhian Series (Induan, earliest Triassic; *Benton & Allen, 1997*). The specimen comprising the anterior end of a snout is likely lost (*Benton & Allen, 1997*). *Boreopricea funerea* was originally considered to represent an intermediate form between *Prolacerta broomi* and *Pricea longiceps* (now considered a junior synonym of *Prolacerta broomi*), and *Macrocnemus bassanii* (*Tatarinov, 1978*). The taxon was later redescribed in more detail by *Benton & Allen (1997)*, who commented on the poor state of the specimen and the absence of certain elements described by *Tatarinov (1978)* as a consequence of damage that the holotype had sustained after this description, such as the crushing of the skull and the displacement and in some cases disappearance of certain postcranial elements. Among these are the interclavicle and ossified sternum, which contained characters that were important in distinguishing *Boreopricea funerea* from other "protorosaurs". Furthermore, because these elements were removed and later placed back on the card on which the specimen is kept, the identification of the tarsal bones is difficult and ambiguous (*Rieppel, Fraser & Nosotti, 2003*). *Boreopricea funerea* has been included in several phylogenetic analyses (*Benton & Allen, 1997*; *Evans, 1988*; *Ezcurra, 2016*; *Jalil, 1997*; *Rieppel, Fraser & Nosotti, 2003*) and an emended diagnosis was provided by *Ezcurra (2016)*. *Boreopricea funerea* was found as the sister taxon to a clade comprising *Macrocnemus*, *Cosesaurus*, *Tanystropheus*, and *Tanytrachelos* within a monophyletic Protorosauria (Prolacertiformes therein) that also included *Prolacerta* (*Evans, 1988*). In the phylogenetic analysis accompanying the redescription of the taxon, *Boreopricea funerea* represented the sister taxon to *Prolacerta* (*Benton & Allen, 1997*). In the final tree of *Jalil (1997)* *Boreopricea funerea* was recovered within a tanystropheid clade together with *Cosesaurus*, *Tanystropheus*, and *Tanytrachelos*. In the various trees produced by *Rieppel, Fraser & Nosotti (2003)* the placement of *Boreopricea funerea* varied. In some cases it was

positioned as closely related to *Protorosaurus* and in others as being more closely related to *Prolacerta*. *Ezcurra (2016)* recovered *Boreopricea funerea* as the sister taxon to the clade composed of Prolacertidae, *Tasmaniosaurus triassicus*, and all Archosauriformes. Because of the badly damaged nature of the holotype, certain previously observed cranial characters by *Tatarinov (1978)* and *Benton & Allen (1997)* could not be scored based on personal observation by *Ezcurra (2016)*. Characters in which *Tatarinov (1978)* and *Benton & Allen (1997)* were in disagreement were scored as missing data by *Ezcurra (2016)*.

### *Eorasaurus olsoni* Sennikov, 1997.

*Eorasaurus olsoni*, one of the very few known Permian archosauromorphs, is known from several vertebrae. The taxon was originally considered to be most closely related to *Protorosaurus* and was therefore placed within "Protorosauridae" (*Sennikov, 1997*). *Ezcurra, Scheyer & Butler (2014)* provided additional observations and an emended diagnosis for *Eorasaurus olsoni*, and it was retrieved as an archosauriform that formed a trichotomy with *Euparkeria capensis* and *Erythrosuchus africanus* in the phylogenetic analysis of that study. *Eorasaurus olsoni* was also included by *Ezcurra (2016)* and formed a massive polytomy at the base of Archosauriformes in analyses 1 and 2 therein, but it was pruned from the final analysis.

### *Hayden Quarry tanystropheid.*

Recently many postcranial elements with clearly tanystropheid affinities were described, encompassing vertebrae, femora, and a calcaneum (*Pritchard et al., 2015*). Because the material is represented by isolated elements, it is unclear whether they all belong to the same species, and it was therefore not referred to any specific taxon. The calcaneum was shown to share apomorphies with the calcanea of *Tanytrachelos ahynis* and this element was therefore assigned to this species. This material was collected from the Hayden Quarry in western North America, which is of approximately middle Norian age (Late Triassic; *Irmis et al., 2011*), making it among the youngest known material referrable to Tanystropheidae. Although it was not concluded that the Hayden Quarry material represents a single taxon, a hypothetical Hayden Quarry taxon was included in the phylogenetic analysis of *Pritchard et al. (2015)*, in which it was recovered as the sister taxon to the North American, Late Triassic *Tanytrachelos ahynis*. Since the material represents only limited postcranial material that cannot unequivocally be assigned to a single taxon, it is not considered for our phylogenetic analysis.

### *Gwynneddosaurus erici* Bock, 1945.

*Gwynneddosaurus erici* is known from a single specimen from the Lockatong Formation near the town of Gwynedd, close to Philadelphia, Pennsylvania. The Lockatong Formation is of late Carnian age (Late Triassic), and is approximately contemporaneous to the Cow Branch Formation from which the better known *Tanytrachelos ahynis* is known (*Colbert & Olsen, 2001*). The only known specimen was originally described by *Bock (1945)* and considered to be closely related to *Podokesaurus holyokensis*, a poorly known early theropod dinosaur (*Carrano & Sampson, 2004*). The holotype and only known specimen represents a disarticulated skeleton preserving several vertebrae, ribs, parts of the pectoral

girdle, and limb elements. *Huene (1948)* revised the specimen and identified it as a "protorosaur" similar to *Macrocnemus*. *Pritchard et al. (2015)* suggested that this material might be referrable to the same taxon as *Tanytrachelos ahynis* and credited this hypothesis to *Olsen (1979)*. However, although he considered some tanystropheid material from the lower Lockatong Formation to be referrable to *Tanytrachelos*, *Olsen (1979)* suggested that *Gwyneddosaurus erici* was quite distantly related to *Tanytrachelos ahynis*.

### *Drepanosaurus unguicaudatus* Pinna, 1980.

*Drepanosaurus unguicaudatus* was first descibed based on the holotype, which consists of a largely complete, articulated skeleton, missing the skull and anterior cervical vertebrae, and several juvenile specimens (*Pinna, 1980*). This was followed by a more extensive description in Italian (*Pinna, 1984*), and these findings were later summarized in English (*Pinna, 1986*). *Renesto (1994c)* revised the morphology of *Drepanosaurus unguicaudatus*, especially regarding the highly specialized forelimbs, and the juvenile specimens were reassigned to a different species, *Megalancosaurus preonensis*. This identification was corroborated by *Renesto (2000)*, who considered the holotype as the only known specimen of *Drepanosaurus unguicaudatus*, whilst a juvenile specimen (MCSNB 4783), previously described by *Renesto & Paganoni (1995)*, was attributed to *Drepanosaurus* sp. A revised diagnosis and overview of the provenance of drepanosaurid species was provided in *Renesto et al. (2010)*. *Pritchard et al. (2016)* described new remains from North America, which were assigned to *Drepanosaurus* sp. and provided new insight into the unique configuration of the grasping forelimb of the taxon. *Drepanosaurus unguicaudatus* is among the best-known drepanosaurids and has been included in several phylogenetic analyses (*Dilkes, 1998*; *Evans, 1988*; *Pritchard & Nesbitt, 2017*; *Pritchard et al., 2016*; *Renesto et al., 2010*; *Senter, 2004*). *Drepanosaurus unguicaudatus* was omitted from the final analysis of *Evans (1988)* due to a lack of observable, phylogenetically informative characters. In the same analysis another drepanosaurid, *Megalancosaurus preonensis*, was recovered as the sister taxon to a large "Prolacertiformes" clade. *Benton & Allen (1997)* also included *Megalancosaurus preonsensis* as the only drepanosaurid in their analysis, in which it was recovered as the sister taxon to *Protorosaurus speneri*. *Dilkes (1998)* included both *Drepanosaurus unguicaudatus* and *Megalancosaurus preonensis*, which formed the sister clade to Tanystropheidae in that analysis. *Senter (2004)* included five drepanosaurid OTUs, which were found in a monophyletic clade that, together with the sister clade formed by *Coelurosauravus* sp. and *Longisquama insignis*, were referred to Avicephala. This clade was recovered outside Neodiapsida and thus, this was the first cladistic analysis to indicate that drepanosaurids were quite distantly related to tanystropheids. Based on modifications of the data matrices of *Dilkes (1998)* and *Laurin (1991)*, as well as a newly formed character matrix, *Renesto et al. (2010)* concluded that the drepanosaurids, including *Drepanosaurus unguicaudatus*, formed the sister clade to Tanystropheidae within a monophyletic Protorosauria. Finally, data matrices employed by *Pritchard et al. (2016)* and *Pritchard & Nesbitt (2017)*, which greatly enhanced both character and taxon sampling, recovered a monophyletic Drepanosauromorpha including *Drepanosaurus unguicaudatus* outside Sauria. Due to their highly derived morphology and because they

likely represent a lineage outside Archosauromorpha, *Drepanosaurus unguicaudatus* and other drepanosauromorphs are not included in the present analysis.

### *Vallesaurus cenensis* Wild, 1991.

*Vallesaurus cenensis* is known from a single, well-preserved and complete specimen that was discovered in the Cene quarry, which represents the upper part of the Zorzino Limestone (Revueltian, early-middle Norian, Late Triassic), in Lombardy, Italy (*Renesto & Binelli, 2006*). *Wild (1991)* mentioned the specimen and assigned it to the genus *Vallesaurus* but did not formally describe it. The specimen (*Renesto, 2000*) and species (*Pinna, 1993*) were subsequently referenced to, but a formal description was only provided later by *Renesto & Binelli (2006)*. *Vallesaurus cenensis* has additionally been compared to other drepanosaurids by *Renesto et al. (2010)*. Therein, the new species *Vallesaurus zorzinensis* was included in the genus. This species differs from *Vallesaurus cenensis* in having an opposable hallux with two phalanges, of which the first one is straight. *Vallesaurus cenensis* has been included in phylogenetic analyses focusing on drepanosaurid interrelationships (*Pritchard & Nesbitt, 2017*; *Pritchard et al., 2016*; *Renesto et al., 2010*; *Senter, 2004*).

### *Megalancosaurus preonensis* Calzavara, Muscio & Wild, 1980.

*Megalancosaurus preonensis* is known from the middle Norian Forni Dolostone of Friuli and Zorzino Limestone of Lombardy, Italy (*Renesto et al., 2010*). The holotype of *Megalancosaurus preonensis*, which comprises a complete skull and cervical series, the expanded neural spines of the anterior dorsal vertebrae, several fragments of dorsal ribs, and a right forelimb, was described by *Calzavara, Muscio & Wild (1980)* and interpreted as an arboreal archosaur. *Feduccia & Wild (1993)* and *Feduccia (1996)* suggested that *Megalancosaurus preonensis* was possibly a glider and considered it to be closely related to birds, thus arguing that a lineage of Triassic non-dinosaurian archosauromorphs, rather than theropod dinosaurs, are the sister group to birds. An additional specimen of *Megalancosaurus preonensis* was described, which provided new information on the postcranium of the taxon (*Renesto, 1994a*). Additionally, three specimens that were previously identified as juvenile specimens of *Drepanosaurus unguicaudatus* (*Pinna, 1980*) were re-assigned to the taxon as well (*Renesto, 1994a*). The arboreal lifestyle suggested for *Megalancosaurus preonensis* was questioned by *Padian & Chiappe (1998)* and they instead considered an aquatic lifestyle for the taxon. The hypothesis that drepanosaurids are the sister group to birds was refuted in a study that also assigned two additional specimens to the species (*Renesto, 2000*). The skull of *Megalancosaurus preonensis* was redescribed in detail by *Renesto & Dalla Vecchia (2005)*. A second species, *Megalancosaurus endennae*, was erected and two specimens that were previously identified as *Megalancosaurus preonensis* were re-assigned to this species (*Renesto et al., 2010*). *Megalancosaurus endennae* mainly differs from *Megalancosaurus preonensis* in the presence of an opposable hallux in the pes. Another specimen lacking the hindlimb, MFSN 18443a, was reassigned to *Megalancosaurus* sp. A functional interpretation of the forelimbs of *Megalancosaurus* spp. was provided by *Castiello, Renesto & Bennett (2016)*.

*Megalancosaurus preonensis* has been included in several phylogenetic analyses, the results of which are outlined above in the description of *Drepanosaurus unguicaudatus* (*Benton & Allen, 1997*; *Dilkes, 1998*; *Evans, 1988*; *Pritchard & Nesbitt, 2017*; *Pritchard et al., 2016*; *Renesto et al., 2010*; *Senter, 2004*).

The following taxa are included as OTUs for the phylogenetic analysis:

### *Petrolacosaurus kansensis* Lane, 1945

**Age.** Late Missourian, late Pennsylvanian, Late Carboniferous.

**Occurrence.** Garnett Quarry, Rock Lake Member of the Stanton Formation, Kansas, USA (*Reisz, 1981*; *Reisz, Heaton & Pynn, 1982*).

**Holotype.** KUVP 1424, largely complete right hindlimb.

**Referred specimens.** The referred specimens are listed in *Reisz (1981*, p. 4–5*)*.

**Diagnosis.** The most recent emended diagnosis is provided by *Reisz (1981)*.

**Remarks.** *Petrolacosaurus kansensis* was first described based on a largely complete hindlimb (the holotype KUVP 1424) and pelvis, and identified as a pelycosaur (*Lane, 1945*). Additional postcranial elements from the same locality were assigned to *Podargosaurus hibbardi* in the same study. Additional specimens, including skull material, revealed that *Podargosaurus* was indistinguishable from *Petrolacosaurus kansensis* and therefore the former is now considered a junior synonym of the latter (*Peabody, 1952*). The systematic position of *Petrolacosaurus kansensis* was disputed, but an additional specimen preserving the skull in more detail revealed it as an early diapsid reptile (*Reisz, 1977*) and it has been described in detail by *Reisz (1981)*. *Petrolacosaurus kansensis* represents one of the best-known Carboniferous diapsids and as such has been widely used as an outgroup or important early taxon in studies on saurian or diapsid phylogeny (e.g., *Dilkes, 1998*; *Evans, 1988*; *Ezcurra, 2016*; *Ezcurra, Scheyer & Butler, 2014*; *Jalil, 1997*; *Pritchard et al., 2015*; *Simões et al., 2018*). A recent phylogenetic hypothesis of early amniotes suggests that the diapsid temporal configuration of *Petrolacosaurus kansensis* was likely independently acquired from that of neodiapsids, including saurians (*Ford & Benson, 2020*).

### *Orovenator mayorum* Reisz, Modesto & Scott, 2011

**Age.** Earliest Artinskian, early Permian (*Cohen et al., 2013*; *Woodhead et al., 2010*).

**Occurrence.** Claystone fissure fills in the Ordovician Arbuckle Limestone, Dolese Brothers Limestone Quarry, Richards Spur, Comanche County, Oklahoma, USA.

**Holotype.** OMNH 74606, a crushed partial skull missing several elements, including large parts of the skull roof and occipital region.

**Referred specimen.** OMNH 74607, a crushed partial skull preserving most of the skull roof, as well as an axis, two postaxial cervical vertebrae, and a single caudal vertebra.

**Diagnosis.** The most recent diagnosis is provided by *Ford & Benson (2018)*.

**Remarks.** *Orovenator mayorum* is an early Permian diapsid known from the Richards Spur locality, which represents a unique upland fissure fill deposit (*Ford & Benson, 2020*; *MacDougall et al., 2017*; *Sullivan, Reisz & May, 2000*). It was hypothesized that early and middle Permian diapsids were mostly restricted to upland environments, and that this would explain their rare occurrence in the fossil record during this period (*Reisz, Modesto & Scott, 2011*). Following the initial description and phylogenetic analysis including *Orovenator mayorum* by *Reisz, Modesto & Scott (2011)*, in which it was recovered as the sister-taxon to all other known neodiapsids, the taxon was redescribed by *Ford & Benson (2018)* based on µCT scans. An extensive recent phylogenetic analysis recovered *Orovenator mayorum* as a member of Varanopidae, a clade historically considered to belong to Synapsida, but which was there found on the reptilian lineage outside Neodiapsida (*Ford & Benson, 2020*). The cranial morphology suggests nocturnality and burrowing behavior in *Orovenator mayorum* (*Ford & Benson, 2018*).

### *Acerosodontosaurus piveteaui* Currie, 1980

**Age.** Lopingian (roughly equivalent to the *Dicynodon* AZ of the Karoo Basin), late Permian (*Smith, Rubidge & Van der Walt, 2012*).

**Occurrence.** Sakamena River Valley, Lower Sakamena Formation, southern Madagascar (*Currie, 1980*).

**Holotype.** MNHN 1908-32-57, a skeleton preserved partially as imprints in two slabs. The posterior half of the skull and mandible is preserved. Of the postcranium, most of the dorsal vertebral series is preserved, as well as part of the forelimbs, the pelvis and sacral region, and hindlimbs.

**Diagnosis.** The most recent emended diagnosis is provided by *Ezcurra (2016)*.

**Remarks.** *Acerosodontosaurus piveteaui* was first described by *Currie (1980)* and indicated to be closely related to *Youngina capensis*. A redescription of the only known specimen revealed that the infratemporal bar is incomplete, in contrast to *Youngina capensis* (*Bickelmann, Müller & Reisz, 2009*). An aquatic lifestyle has been suggested for *Acerosodontosaurus piveteaui*, which is supported by observed skeletal paedomorphosis of the carpal bones. Recent phylogenetic analyses recovered *Acerosodontosaurus piveteaui* as a diapsid that is closely related to Sauria (e.g., *Bickelmann, Müller & Reisz, 2009*; *Ezcurra, Scheyer & Butler, 2014*; *Ford & Benson, 2020*; *Pritchard & Nesbitt, 2017*; *Pritchard & Sues, 2019*). One recent analysis addressing the relationships of stem-turtles recovered *Acerosodontosaurus piveteaui* in a clade with *Claudiosaurus germaini* within Pantestudines as the sister group to all other members of this clade (*Li et al., 2018*).

### *Claudiosaurus germaini* Carroll, 1981

**Age.** Lopingian (roughly equivalent to the *Dicynodon* AZ of the Karoo Basin), late Permian (*Smith, Rubidge & Van der Walt, 2012*).

**Occurrence.** Lower Sakamena Formation near the village of Leoposa, southern Madagascar (*Caldwell, 1995*; *Carroll, 1981*).

**Holotype.** MNHN 1978-6-1, a largely complete skeleton, including a poorly preserved skull but missing the posterior tail section.

**Referred specimens.** A list of referred specimens can be found in *Carroll (1981*; p. 337-338*)*. Several specimens are located in private collections. Additional undescribed specimens are housed in the SAM (*Simões et al., 2018*).

**Diagnosis.** The diagnosis was provided by *Carroll (1981)*.

**Remarks.** *Claudiosaurus germaini* is a non-saurian diapsid known from various specimens from the late Permian of southern Madagascar. Its depositional environment, as well as its enlarged hindlimbs and pedes, and skeletal paedomorphosis, suggest it had an aquatic lifestyle (*Caldwell, 1995*; *Carroll, 1981*).

### *Youngina capensis* Broom, 1914

**Age.** Capitanian to Changhsingian, middle to late Permian (*Rubidge et al., 2013*; *Smith & Evans, 1996*)

**Occurrence.** *Tropidostema*, *Cistecephalus*, and *Dicynodon* AZs (Assemblage Zones), Balfour and Middelton Formations of the Beaufort Group, part of the Karoo Supergroup, South Africa (*Broom, 1914*; *Smith & Evans, 1996*).

**Holotype.** AMNH 5561, a complete skull and mandible and a partial articulated vertebral column.

**Referred specimens.** The most inclusive hypodigm has been provided by *Ezcurra (2016)*, who supplemented a previous hypodigm of *Gow (1975)* with specimens found since then.

**Diagnosis.** The most recent emended diagnosis of *Youngina capensis* was provided by *Ezcurra (2016)*.

**Remarks.** *Youngina capensis* is a middle to late Permian non-saurian neodiapsid with a generalized morphology known from an array of well-preserved specimens. Its morphology has been investigated frequently (e.g., *Broom, 1914*; *Broom, 1922*; *Currie, 1981*; *Evans, 1987*; *Gardner, Holiday & O'Keefe, 2010*; *Goodrich, 1942*; *Gow, 1975*; *Smith & Evans, 1996*; *Watson, 1957*). *Youngina capensis* represents an important taxon for phylogenetic analyses that investigate early diapsid and saurian relationships (e.g., *Ezcurra, 2016*; *Ezcurra, Scheyer & Butler, 2014*; *Ford & Benson, 2020*; *Simões et al., 2018*). Specimens that were previously assigned to *Youngoides romeri* (*Olson & Broom, 1937*), *Youngoides minor* (*Broom & Robinson, 1948*), *Youngopsis kitchingi* (*Broom, 1937*), *Youngopsis rubidgei* (*Broom & Robinson, 1948*), and *Acanthotoposaurus bremneri* (*Evans & Van Den Heever, 1987*), have all been shown to be conspecific with *Youngina capensis* (*Evans, 1987*; *Gow, 1975*; *Reisz, Modesto & Scott, 2000*).

### Gephyrosaurus bridensis *Evans, 1980*

**Age.** Hettangian, and possibly Sinemurian, Early Jurassic (*Whiteside et al., 2016*).

**Occurrence.** Fissure fill deposits of Pontalun quarry southern Wales (*Evans & Kermack, 1994*; *Whiteside & Duffin, 2017*; *Whiteside et al., 2016*).

**Holotype.** UCL T.1503, a right dentary.

**Referred specimens.** The material of *Gephyrosaurus bridensis* comprises an extensive amount of isolated remains (over 1,000 specimens according to *Evans, 1980*). No complete list of referred specimens is currently available in the literature.

**Diagnosis.** The diagnosis was provided by *Evans (1980*, p. 204–205*)*

**Remarks.** *Gephyrosaurus bridensis* is exclusively known from extensive isolated remains that have been assigned to a single taxon based on the complementary articulation surfaces between the various elements, as well as their morphological similarity and relative size. *Gephyrosaurus bridensis* has been described in detail in one study addressing the skull (*Evans, 1980*), and another addressing the postcranium (*Evans, 1981*). Although known from younger, Early Jurassic, deposits, *Gephyrosaurus bridensis* is considered the sister taxon to the clade encompassing the Triassic rhynchocephalians *Planocephalosaurus robinsonae*, *Clevosaurus* spp., and other more crownward rhynchocephalians (e.g., *Hsiou, De França & Ferigolo, 2015*; *Scheyer et al., 2020a*; *Simões et al., 2018*). An additional species of this genus, *Gephyrosaurus evansae*, was recently described from the Rhaetian (Late Triassic) '*Microlestes*' quarry at Holwell near Bristol, UK (*Whiteside & Duffin, 2017*).

### Planocephalosaurus robinsonae *Fraser, 1982*

**Age.** Early Rhaetian, Late Triassic (*Whiteside et al., 2016*).

**Occurrence.** Fissure fill deposits of Cromhall and Tytherington quarries, Bristol and Gloucestershire, England (*Fraser, 1982*; *Whiteside & Marshall, 2008*).

**Holotype.** AUP No. 11061, an isolated left maxilla.

**Referred specimens.** As for *Gephyrosaurus bridensis*, *Planocephalosaurus robinsonae* is represented by a large amount of isolated elements (at least 750 specimens from Cromhall quarry according to *Fraser, 1982*), and no complete list of referred specimens is currently available in the literature.

**Diagnosis.** The diagnosis of *Planocephalosaurus robinsonae* is provided by *Fraser (1982*, p. 710*)*.

**Remarks.** Like *Gephyrosaurus bridensis*, *Planocephalosaurus robinsonae* is known from the Late Triassic to Early Jurassic fissure fills of southwestern England and southern Wales. However, whereas *Gephyrosaurus bridensis* is known from an Early Jurassic Welsh locality, *Planocephalosaurus robinsonae* is known from Late Triassic English localities. Its material is also composed of a large amount of three-dimensionally preserved, isolated

remains that can be assigned to a single taxon based on their relative connectivity, morphological similarity, and size (although *Simões et al. (2018)* considered the assignment of postcranial elements to this taxon only tentative). The skull was described by *Fraser (1982)*, and the postcranium has subsequently been described by *Fraser & Walkden (1984)*. *Planocephalosaurus robinsonae* represents one of the best-known early rhynchocephalians and bears several plesiomorphic features compared to *Clevosaurus* spp. and other more derived rhynhocephalians. Specimens of *Planocephalosaurus* have also been identified from the Ruthin quarry, southern Wales, but were not assigned to the species level (*Whiteside et al., 2016*). Small tooth bearing fragments from the lower Tecovas Formation, Chinle Group (late Carnian) in Texas, USA, were assigned to a new species, *Planocephalosaurus lucasi* (*Heckert, 2004*).

### *Protorosaurus speneri* Meyer, 1832

**Age.** Traditionally Tatarian, although conodont data points towards a more specific Wuchiapingian age, late Permian (*Ezcurra, Scheyer & Butler, 2014*; *Legler & Schneider, 2008*).

**Occurrence.** The Middridge and Quarrington quarries near Durham, Marl Slate, England (*Evans & King, 1993*) and various localities of the Kupferschiefer Formation of central Germany (all localities are listed in Table 1 of *Gottmann-Quesada & Sander, 2009*).

Lectotype. Since no formal holotype had previously been assigned, NHMW 1943I4, known as the Swedenborg specimen, was assigned the lectotype by *Gottmann-Quesada & Sander (2009)*.

**Referred specimens.** Table 1 of *Gottmann-Quesada & Sander (2009)* listed 28 specimens that were included in that study. More specimens that can tentatively be assigned to the species are known, which are distributed among various institutions and private collections across Europe, and a complete hypodigm is missing. Most specimens consist of postcranial material, whereas skull material is comparatively rare and only known from five different specimens: RCSHC/Fossil Reptiles 308, WMsN P 47361, TWCMS S1348(.1 and .2), IGWuG 463016, and NMK S 180. Only NMK S 180 represents a complete and well-preserved skull.

**Diagnosis.** *Ezcurra (2016)* provided the most recent diagnosis for the species.

**Remarks.** *Protorosaurus speneri* currently represents by far the best-known Permian archosauromorph. The first specimen to be discovered, RCSHC/Fossil Reptiles 308 or the Spener specimen, was described by *Spener (1710)* and interpreted as a fossil of a Nile crocodile (*Crocodylus niloticus*). *Protorosaurus speneri* was erected and described in detail based on additional material (*Meyer, 1830*, *1832*, *1856*). Other specimens were described more recently (e.g., *Evans & King, 1993*; *Fichter, 1995*; *Haubold & Schaumberg, 1985*), and the species was extensively revised by *Gottmann-Quesada & Sander (2009)*. Most of the approximately 40 known specimens derive from the Kupferschiefer Formation of Germany, whereas two come from the contemporary Marl Slate of England (*Evans & King,*

*1993*). Most recent phylogenetic analyses recovered *Protorosaurus speneri* as one of the earliest diverging archosauromorphs (e.g., *Ezcurra, 2016*; *Pritchard et al., 2015*; for an alternative placement of *Protorosaurus speneri* and tanystropheids outside Archosauromorpha, see *Simões et al., 2018*).

### *Czatkowiella harae* Borsuk-Białynicka & Evans, 2009b

**Age.** Earliest late Olenekian, Early Triassic (*Shishkin & Sulej, 2009*).

**Occurrence.** Czatkowice 1, a fissure or cave infill of the Czatkowice quarry near Kraków, Poland.

**Holotype.** ZPAL R.V/100, an isolated, nearly complete right maxilla bearing teeth.

**Referred specimens.** A large number of isolated cranial and postcranial elements that could confidently be distinguished from other tetrapod remains of the Czatkowice 1 locality. A large number of these bones are presented and described by *Borsuk-Białynicka & Evans (2009b)*.

**Diagnosis.** The diagnosis was provided by *Borsuk-Białynicka & Evans (2009b)*.

**Remarks.** The material now referred to *Czatkowiella harae* was originally discovered in 1978 at Czatkowice 1. It is represented by many isolated and fragmented specimens, which were found among similar remains belonging to other small diapsids, such as the euparkeriid *Osmolskina czatkowicensis* (*Borsuk-Bialynicka & Evans, 2003*; *2009a*; *Borsuk-Białynicka & Sennikov, 2009*), the lepidosauromorph *Sophineta cracoviensis* (*Evans & Borsuk-Białynicka, 2009*), the kuehneosaurid *Pamelina polonica* (*Evans, 2009*), and three distinct procolophonids (*Borsuk-Białynicka & Lubka, 2009*). Apart from the most diagnostic elements, bones were assigned to *Czatkowiella harae* largely based on size and fitting individual elements together. The most distinguishing feature of *Czatkowiella harae* is the presence of three-headed anterior dorsal ribs. It has only been considered phylogenetically by *Borsuk-Białynicka & Evans (2009b)*, who recovered *Czatkowiella harae* as the sister taxon to *Protorosaurus speneri*. The disarticulated and fragmented remains of *Czatkowiella harae* were found in a bonebed comprising a diverse fauna and it is possible that multiple taxa are represented among its referred material (*Ezcurra, Scheyer & Butler, 2014*), making this taxon somewhat problematic for inclusion in phylogenetic analyses. Here, we follow the identification by *Borsuk-Białynicka & Evans (2009b)* in all but the most tentatively assigned bones (e.g., the squamosal) and as for all other taxa scored, the specimen(s) referred to for the scoring of each character are provided in the Supplementary Material. This allows subsequent workers to critically evaluate scorings and exclude certain specimens that further investigation might find to belong to a different taxon. Because the inclusion of potentially composite taxa can negatively influence the accuracy of phylogenetic analyses, *Czatkowiella harae* is omitted from the analyses 2 and 4 here.

### *Tanystropheus longobardicus* *Bassani, 1886*

**Age.** Latest Anisian-Ladinian, Middle Triassic (*Spiekman et al., 2020a*; *Spiekman & Scheyer, 2019*; *Stockar, 2010*).

**Occurrence.** Besano Formation and the Cassina beds, Meride Limestone, of Monte San Giorgio, Switzerland and Italy.

**Neotype.** The holotype specimen was destroyed in Milan during World War II (*Nosotti, 2007*; *Spiekman & Scheyer, 2019*; *Wild, 1973*). A neotype was established by *Wild (1973)*: PIMUZ T 2791 – An almost complete and largely articulated, bituminous specimen, lacking the posterior half of the tail.

**Referred specimens.** PIMUZ T 2779, PIMUZ T 2781, PIMUZ T 2795, PIMUZ T 2485, PIMUZ T 2482, PIMUZ T 2484, PIMUZ T 3901, PIMUZ T 1277, MSNM BES SC 265, and MSNM BES SC 1018.

**Diagnosis.** The most recent diagnosis for the taxon was provided by *Spiekman et al. (2020a*, Methods S1*).

**Remarks.** *Tanystropheus longobardicus* was first described based on a single, partially articulated specimen from the Besano Formation of Monte San Giorgio. It was interpreted as a pterosaur and assigned to *Tribelesodon longobardicus*, with the generic name referring to the tricuspid marginal teeth (*Arthaber, 1922*; *Bassani, 1886*; *Nopcsa, 1923*). This specimen has unfortunately been lost, but it is figured in *Arthaber (1922)*. The discovery of additional specimens from the Besano Formation revealed that the elements that were interpreted as elongated phalanges represented elongated cervical vertebrae that were similar to those known from the Upper Muschelkalk of the Germanic Basin, which had been assigned to *Tanystropheus conspicuus* (*Peyer, 1930*, *1931*). Therefore, the species was re-assigned to *Tanystropheus longobardicus*. *Wild (1973)* described the species in detail and assigned PIMUZ T 2791 as the neotype. Additional specimens were described in *Wild (1980a)*, including a specimen from the slightly younger Meride Limestone, which was assigned to the separate species *Tanystropheus meridensis*. However, this specimen, as well as an additional specimen that was found from the Meride Limestone (*Renesto, 2005*), were shown to be morphologically indistinguishable from the specimens from the Besano Formation, and therefore *Tanystropheus meridensis* is considered a junior synonym of *Tanystropheus longobardicus* (*Nosotti, 2007*; *Spiekman & Scheyer, 2019*). A small-sized *Tanystropheus* skeleton lacking the skull from the Zhuganpo Formation of China was identified as *Tanystropheus* sp. and could represent the only known occurrence of *Tanystropheus longobardicus* from China, indicating a Tethys-wide distribution of the species (*Li, 2007*). However, since no diagnostic cranial material is known for this specimen, it cannot be assigned to the species level and therefore the occurrence of *Tanystropheus longobardicus* in China is currently unclear (*Spiekman et al., 2020a*, Methods S1; *Spiekman & Scheyer, 2019*). *Nosotti (2007)* described in detail specimens from the Italian side of the Besano Formation. Recently, a combined morphological and palaeohistological study revealed that the small-sized specimens of *Tanystropheus* from

Monte San Giorgio, which bear the distinct tricuspid marginal teeth, are skeletally mature (*Spiekman et al., 2020a*). This indicates that the small-sized specimens represent a separate species from the large-sized specimens, and the latter were re-assigned to a new species, *Tanystropheus hydroides*. *Tanystropheus longobardicus* was therefore a relatively small-sized *Tanystropheus* species, likely not exceeding 2 m in total length, that fed on small prey, including soft-shelled invertebrates.

### *Tanystropheus hydroides* Spiekman, Neenan, Fraser, Fernandez, Rieppel, Nosotti & Scheyer, 2020

**Age.** Latest Anisian-earliest Ladinian, Middle Triassic (*Spiekman et al., 2020a*; *Stockar, 2010*).

**Occurrence.** Besano Formation of Monte San Giorgio, Switzerland and Italy.

**Holotype.** PIMUZ T 2790, a compressed skull and anterior eight cervical vertebrae in semi-articulation.

**Referred specimens.** MSNM BES 351, MSNM V 3663, PIMUZ T 1270, PIMUZ T 1307, PIMUZ T 2480, PIMUZ T 2483, PIMUZ T 2497, PIMUZ T 2787, PIMUZ T 2788, PIMUZ T 2793, PIMUZ T 2818, PIMUZ T 2819, PIMUZ T 183, PIMUZ T 2817, SNSB-BSPG 1953 XV 2. For additional information, see *Spiekman et al. (2020b)*.

**Diagnosis.** The diagnosis for *Tanystropheus hydroides* has been provided in *Spiekman et al. (2020a*, p. 3890*)*.

**Remarks.** Specimens of *Tanystropheus hydroides* were previously considered as the adult morphotype of *Tanystropheus longobardicus*, but it was recently shown that they represent a separate, large-sized species (*Spiekman et al., 2020a*). Specimens of *Tanystropheus hydroides* were described as *Tanystropheus longobardicus* in *Peyer (1931)*, *Kuhn-Schnyder (1947*, *1959)*, and *Wild (1973)*. A *Tanystropheus* specimen, GMPKU P 1527, has been described from China that attained a size similar to *Tanystropheus hydroides* (*Rieppel et al., 2010*). Although the postcranial skeleton of GMPKU P 1527 cannot be distinguished from *Tanystropheus hydroides*, the absence of a skull has not allowed the specimen to be assigned to this species with certainty, and it is therefore currently considered as *Tanystropheus* cf. *T. hydroides* (*Spiekman et al., 2020b*). As such, it has been included as a separate OTU in the present analysis. However, it is clear that the genus had a Tethys-wide distribution (*Spiekman & Scheyer, 2019*). *Tanystropheus hydroides* has recently been interpreted as an aquatic ambush predator that employed its long-neck and a snapping bite to catch its prey, which consisted of fish and cephalopods (*Spiekman et al., 2020a*, *2020b*). A detailed description of the cranial morphology and the anterior cervical column of *Tanystropheus hydroides* was provided by *Spiekman et al. (2020b)*.

### GMPKU P 1527

**Age.** Ladinian, Middle Triassic (*Sun et al., 2016*).

**Occurrence.** The upper part of the Zhuganpo Formation of Nimaigu near Xingyi City, Wusha District, Guizhou Province, southwestern China.

**Remarks.** A large-sized *Tanystropheus* skeleton from China that is largely complete but lacks the skull, anterior segment of the neck, the posterior end of the tail, and most of the pedes, was identified as *Tanystropheus* cf. *T. longobardicus* (*Rieppel et al., 2010*). Recently, it was shown that the large-sized specimens of *Tanystropheus* from Monte San Giorgio represent a separate species from the small-sized specimens and they were assigned to *Tanystropheus hydroides* (*Spiekman et al., 2020a*). Therefore, the assignment of GMPKU P 1527 has consequently been altered to *Tanystropheus* cf. *T. hydroides* (*Spiekman et al., 2020b*). The recent distinction between *Tanystropheus longobardicus* and *Tanystropheus hydroides* has indicated that the skull appears to be considerably more variable than the postcranium between species within this genus. Therefore, since GMPKU P 1527 lacks any cranial material and because it is known from the eastern side of the Tethys Ocean, whereas the referred specimens of *Tanystropheus hydroides* derive from its western margin, this specimen cannot be unequivocally assigned to the same species, even though its preserved postcranial skeleton is considered to be virtually morphologically indistinguishable from the known postcranium of *Tanystropheus hydroides* (*Rieppel et al., 2010*; *Tanystropheus hydroides* is referred to as the adult type of *Tanystropheus longobardicus* therein). GMPKU P 1527 was incorporated as a separate OTU here to test its phylogenetic position relative to the known *Tanystropheus* species.

### *Tanystropheus "conspicuus"* Meyer, 1852 (part of Meyer, 1847–1855)

**Age.** Late Anisian to Ladinian, Middle Triassic (*Menning & Hendrich, 2016*; *Spiekman & Scheyer, 2019*).

**Occurrence.** Various localities of the Upper Muschelkalk and Lettenkeuper of Central Europe. An overview of all known localities can be found in Supplementary Table 1 of *Spiekman & Scheyer (2019)*.

**Lectotype.** U-MO BT 740, an isolated, three-dimensionally preserved cervical vertebra.

**Referred specimens.** The referred specimens of *Tanystropheus "conspicuus"* are listed in Supplementary Table 1 of *Spiekman & Scheyer (2019)*.

**Remarks.** Several elongate bones from the Upper Muschelkalk of Bayreuth, Germany, were identified as reptilian vertebrae and assigned to *Tanystropheus "conspicuus"* by *Meyer (1847–1855)*. These bones had previously also been described by Count Georg zu Münster, who had interpreted these elements as limb bones of a saurian reptile, which he had named "*Macroscelosaurus*". However, since this work has been lost and this generic name has fallen into disuse (*nomen oblitum*), the generic name *Tanystropheus* has received precedence (*Melville, 1981*; *Wild, 1973*, p. 148). Following the description of the semi-articulated specimens of *Tanystropheus longobardicus* (*Peyer, 1930*, *1931*), *Huene (1931)* considered material previously identified as "*Thecodontosaurus latespinatus*", "*Thecodontosaurus primus*", and "*Procerosaurus cruralis*" from the Upper Muschelkalk of

Europe to very likely belong to *Tanystropheus "conspicuus"*. *Wild (1973)* provided a systematic palaeontology section in which these taxa were synonymized with *Tanystropheus conspicuus*. Fragmentary and isolated remains of *Tanystropheus "conspicuus"* are known from the Upper Muschelkalk and Lettenkeuper throughout Central Europe (late Anisian to Ladinian; *Menning & Hendrich, 2016*). This material comprises isolated cervical, dorsal, sacral, and caudal vertebrae, two femora, and an ischium. *Peyer (1931)* refrained from providing a detailed comparison of *Tanystropheus longobardicus* with *Tanystropheus "conspicuus"* and *"Tanystropheus antiquus"* from the Germanic Basin. *Wild (1973)* distinguished *Tanystropheus "conspicuus"* from *Tanystropheus longobardicus* on the basis of comparatively wider rib attachment sites and a concavity on the anterior end of the neural spine of the cervical vertebrae. Although he considered these minor differences to be insufficient for a species definition, the distinction between the two taxa was maintained in expectation of additional specimens that would allow for a more complete comparison. A recent revision of *Tanystropheus* spp. revealed that no distinct morphological differences could be identified between *Tanystropheus "conspicuus"*, *Tanystropheus hydroides* (therein the large morphotype of *Tanystropheus longobardicus*), and *Tanystropheus "haasi"* (*Spiekman & Scheyer, 2019*). However, since the hypodigms of both *Tanystropheus "conspicuus"* and *Tanystropheus "haasi"* are insufficient for a detailed comparison and both are only known from fragmentary and isolated postcranial elements, these taxa were considered as nomina dubia.

### *"Tanystropheus antiquus"* Huene, 1905

**Age.** Latest Olenekian to middle Anisian, latest Early Triassic to Middle Triassic (*Menning & Hendrich, 2016*; *Spiekman & Scheyer, 2019*).

**Occurrence.** Lower Muschelkalk of Silesia, Poland (Gogolin Formation), Germany (Schaumkalk Formation), and the Netherlands (Vossenveld Formation) (see also Supplementary Table 1 of *Spiekman & Scheyer, 2019*).

**Syntype.** SMNS 16687, SMNS 10110, MGUWr 3872s, MGUWr 3888s, MGUWr 3895s, MGUWr 3902s and some uncatalogued MGUWr specimens, all consisting of isolated cervical vertebrae. *Wild (1973)* assigned SMNS 10110 as the lectotype.

**Referred specimens.** All specimens assigned to "*Tanystropheus* (c.f.) *antiquus*" are listed in Supplementary Table 1 of *Spiekman & Scheyer (2019)*.

**Diagnosis.** Recent diagnoses were provided for this taxon by *Sennikov (2011)* for "*Protanystropheus antiquus*" and *Fraser & Rieppel (2006)* for "*Tanystropheus antiquus*".

**Remarks.** Following the description of the syntype of "*Tanystropheus antiquus*" from the Lower Muschelkalk of Gogolin and Krapkowice, Silesia, Poland (*Huene, 1905*), other isolated *Tanystropheus*-like remains from the Lower Muschelkalk were attributed to the species (e.g., *Huene, 1931*; *Kuhn, 1971*; *Schmidt, 1928*, *1938*; *Spiekman et al., 2019*; *Wild & Oosterink, 1984*). *Ortlam (1966)* referred material of the uppermost Buntsandstein (Anisian) to *Tanystropheus longobardicus* and *Macrocnemus bassanii*, but this material

was later assigned to "*Tanystropheus antiquus*" by *Wild (1980b)*. The Buntsandstein precedes the Muschelkalk and in contrast to the latter represents largely fluvial sediments (*Feist-Burkhardt et al., 2008*), and *Wild (1980b)* concluded that the discovery of "*Tanystropheus antiquus*" from the Buntsandstein indicated that at least the juvenile individuals of this species had a terrestrial lifestyle. Both *Wild (1987)* and *Evans (1988)* later suggested that "*Tanystropheus antiquus*" might belong to a separate genus, based on the large morphological discrepancy between this taxon and other *Tanystropheus* species. *Fraser & Rieppel (2006)* revised the Buntsandstein specimens and concluded that it represented a separate taxon from the Lower Muschelkalk specimens of "*Tanystropheus antiquus*" and assigned it to the new species *Amotosaurus rotfeldensis*. Despite a lack of diagnostic characters in the material, *Fraser & Rieppel (2006)* tentatively maintained the assignment of the Lower Muschelkalk specimens to "*Tanystropheus antiquus*". *Sennikov (2011)* compared "*Tanystropheus antiquus*" to *Augustaburiania vatagini* and concluded that the former was sufficiently distinct from *Tanystropheus* spp. to assign it to a new genus, resulting in the combination "*Protanystropheus antiquus*".

The relative length of the cervical vertebrae might indeed indicate that "*Tanystropheus antiquus*" is more closely related to *Augustaburiania vatagini* or *Amotosaurus rotfeldensis* than to other *Tanystropheus* species. However, the taxonomic status of "*Tanystropheus antiquus*" is currently unclear since many specimens of the syntype material (*Huene, 1902*, *1905*) were long considered to have been lost (*Fraser & Rieppel, 2006*; *Sennikov, 2011*; *Wild, 1973*, *1980b*). These specimens have recently resurfaced and were briefly discussed by *Skawiński et al. (2017)*. Any taxonomic evaluation of this taxon would first require a detailed revision of this type material to assess whether subsequently referred specimens of "*Tanystropheus antiquus*" from other localities represent the same species (*Spiekman et al., 2019*; *Spiekman & Scheyer, 2019*). Such a revision is currently underway (T. Szczygielski, 2019, personal communication), and therefore the taxonomic status of "*Tanystropheus antiquus*" is not addressed here. However, we include a preliminary "*Tanystropheus antiquus*" OTU in analyses 1 and 3 based on the strong morphological similarity of the tanystropheid cervical vertebrae from the Lower Muschelkalk of Central Europe. Our scoring of this OTU is based on two complete cervical vertebrae, SMNS 16687 and Coll. Oosterink A638. The former specimen comes from the Lower Muschelkalk of Krapkowice, Poland, and constitutes part of the syntype of "*Tanystropheus antiquus*", and the latter derives from the Lower Muschelkalk of Winterswijk, the Netherlands (*Spiekman et al., 2019*; *Wild & Oosterink, 1984*).

### *Sclerostropheus fossai Wild, 1980a*

**Age.** Late Norian, Late Triassic (*Rigo, Galli & Jadoul, 2009*; *Tackett & Tintori, 2019*).

**Occurrence.** N-slope of Canto Alto, near Poscante in Val Brembana, Bergamo Province, Italy (*Wild, 1980a*).

**Holotype.** MCSNB 4035, a partial, articulated cervical column.

**Diagnosis.** The most recent diagnosis was provided by *Spiekman & Scheyer (2019)*.

**Remarks.** *Sclerostropheus fossai* is known from a single specimen, which constitutes a partial, semi-articulated cervical column, and was previously considered within the genus *Tanystropheus* (*Wild, 1980a*). However, the morphology of the cervical vertebrae and ribs differs distinctly from that of other *Tanystropheus* species, as was briefly indicated by *Renesto (2005)*, and it was recently assigned to the new genus *Sclerostropheus* (*Spiekman & Scheyer, 2019*). After *Langobardisaurus pandolfii*, *Sclerostropheus fossai* represents a second tanystropheid taxon known from the Norian of northern Italy.

### *Macrocnemus bassanii* Nopcsa, 1930

**Age.** latest Anisian-Ladinian, Middle Triassic.

**Occurrence.** Besano Formation and Meride Limestone of Monte San Giorgio, Switzerland and Italy (*Jaquier et al., 2017*; *Peyer, 1937*; *Renesto & Avanzini, 2002*; *Rieppel, 1989*; *Stockar, 2010*).

**Holotype.** MSNM 14624, a cast of MSNM specimen Besano I, a poorly preserved specimen that was destroyed in Milan during WWII (*Fraser & Furrer, 2013*).

**Referred specimens.** *Rieppel (1989*, p. 374*)* provided a referred specimen list for the *Macrocnemus bassanii* material housed in the PIMUZ. Among those specimens, the specimen listed there as A 111/208 is now catalogued as PIMUZ T 4822. Additionally, two specimens of *Macrocnemus bassanii* are housed in the MSNM: MSNM BES SC 111; a complete and fully articulated juvenile including skin remains; and MSNM V 457, a disarticulated adult specimen, in which a number of skull and jaw bones are preserved, as well as several cervical, dorsal, and caudal vertebrae, gastralia, ribs, and pelvic girdles and both hindlimbs, excluding the feet.

**Diagnosis.** The most recent diagnosis was provided by *Jaquier et al. (2017)*.

**Remarks.** *Macrocnemus bassanii* is the type species of the genus and is known from the Middle Triassic of Switzerland and Italy. It was first described by *Nopcsa (1930)* based on the poorly preserved holotype. This specimen was lost during World War II, but a cast has been preserved. The taxon was described in more detail following the discovery of multiple well-preserved specimens (*Peyer, 1937*). Further details of the skull were provided by *Kuhn-Schnyder (1962)* and *Rieppel & Gronowski (1981)*. The postcranium and its functional considerations were discussed by *Rieppel (1989)*, which indicated that *Macrocnemus bassanii* was facultatively bipedal. An excellently preserved juvenile specimen preserving soft tissue was described by *Premru (1991)* and *Renesto & Avanzini (2002)*. The skull and atlas-axis complex of *Macrocnemus bassanii* were recently redescribed in detail with the use of a synchrotron microtomographic scan, revealing several previously obscured anatomical regions, such as the braincase (*Miedema et al., 2020*). *Macrocnemus bassanii* is currently firmly established as a tanystropheid (e.g., *Ezcurra, 2016*; *Pritchard et al., 2015*).

### *Macrocnemus fuyuanensis* Li, Zhao & Wang, 2007

**Age.** Late Anisian-Ladinian, Middle Triassic (*Stockar, 2010*; *Sun et al., 2016*).

**Occurrence.** Besano Formation of Monte San Giorgio, Switzerland and Zhuganpo Formation of Huabi, Yun-nan Province, China (*Jaquier et al., 2017*; *Jiang et al., 2011*; *Li, Zhao & Wang, 2007*; *Scheyer et al., 2020b*).

**Holotype.** IVPP V15001, a mostly complete and largely articulated skeleton missing most of the skull.

**Referred specimens.** GMPKU-P-3001, almost complete and fully articulated specimen, missing most of the tail; PIMUZ T 1559, virtually complete and disarticulated specimen, missing parts of the skull, almost the complete tail, and the hindlimbs.

**Diagnosis.** The most recent diagnosis for the species was provided by *Scheyer et al. (2020b)*.

**Remarks.** The holotype specimen of *Macrocnemus fuyuanensis* was first described by *Li, Zhao & Wang, 2007* and distinguished from *Macrocnemus bassanii* based on the relative proportions of the limbs as well as the number of dorsal vertebrae. *Jiang et al. (2011)* described another specimen of *Macrocnemus fuyuanensis* with a completely preserved skull, GMPKU-P-3001, and concluded that it differed from *Macrocnemus bassanii* in several cranial characters. A specimen from the upper Besano Formation of Switzerland, PIMUZ T 1559, was described by *Jaquier et al. (2017)*. This specimen was more similar in limb proportions to *Macrocnemus fuyuanensis* than to *Macrocnemus bassanii*, and also differed from the latter in the morphology of the interclavicle, and it was therefore identified as *Macrocnemus* aff. *M. fuyuanensis*. Furthermore, the cranial morphology of *Macrocnemus fuyuanensis* specimen GMPKU-P-3001 was revised in this study, indicating that it did not substantially differ from that of *Macrocnemus bassanii*. The holotype IVPP V15001 was recently redescribed, which revealed new anatomical details for the taxon, particularly with regards to the palate and pectoral girdle (*Scheyer et al., 2020b*).
The morphology of the interclavicle of the holotype was in correspondence with that of the Swiss specimen PIMUZ T 1559, and distinctly differed from that of specimens assigned to *Macrocnemus bassanii*. Therefore, PIMUZ T 1559 was re-assigned to *Macrocnemus fuyuanensis* and the species thus occurred on both the eastern and western margins of the Tethys Ocean.

### *Macrocnemus obristi* Fraser & Furrer, 2013

**Age.** Early Ladinian, Middle Triassic.

**Occurrence.** Prosanto Formation of Ducanfurgga near Davos, canton Graubünden, Switzerland (*Fraser & Furrer, 2013*).

**Holotype.** PIMUZ A/III 1467 (housed in the Bündner Naturmuseum, Chur, Switzerland), an articulated partial skeleton, which consists of the posterior dorsal vertebrae, pelvic girdle and hindlimbs, and most of the tail.

**Referred specimens.** PIMUZ A/III 722, a right pes preserved in dorsal view.

**Diagnosis.** The diagnosis was provided by *Fraser & Furrer (2013*, p. 200*)*.

Remarks. *Macrocnemus obristi* is known from two specimens from the Prosanto Formation (*Fraser & Furrer, 2013*). It differs from *Macrocnemus bassanii* and *Macrocnemus fuyuanensis* based on the length proportions of the femur and tibia (*Fraser & Furrer, 2013*; *Jaquier et al., 2017*). Due to its recent description and only partially known morphology, *Macrocnemus obristi* has previously only been included in the phylogenetic analysis of *Ezcurra & Butler (2018)*, in which it was scored for the purpose of a disparity analysis. Therein, *Macrocnemus obristi* was recovered within a monophyletic *Macronemus* clade as the sister taxon to *Macrocnemus fuyuanensis*.

### *Tanytrachelos ahynis* Olsen, 1979

**Age.** Late Carnian, early Late Triassic.

**Occurrence.** Virginia Solite Quarry B, Upper member of the Cow Branch Formation, part of the Dan River Group (Newark Supergroup), USA (*Casey, Fraser & Kowalewski, 2007*; *Liutkus-Pierce, Fraser & Heckert, 2014*; *Olsen, 1979*).

**Holotype.** YPM 7496, a largely complete, articulated specimen.

**Referred specimens.** A hypodigm was listed by *Olsen (1979*, p. 4-5, and note on p. 13*)*. Most specimens are housed in the VMNH.

**Diagnosis.** The diagnosis for *Tanytrachelos ahynis* was provided by *Olsen (1979)*.

**Remarks.** *Tanytrachelos ahynis* was described by *Olsen (1979)* and is known from hundreds of specimens from Solite Quarry B in Virginia (*Casey, Fraser & Kowalewski, 2007*). However, many detailed morphological features are unknown for *Tanytrachelos ahynis*, due to the poor preservation of the specimens. Recently, the authors of the current study subjected a relatively well-preserved specimen (NMS G.2017.11.1) to synchrotron radiation micro-computed tomography. This revealed the inner anatomy of the cervical vertebrae in some detail, highlighting that as in *Tanystropheus* spp. and *Macrocnemus bassanii* the neural canal passes through the vertebral centrum in *Tanytrachelos ahynis*. However, most morphological details could not be observed due to the poor preservation of the specimen, which is likely attributable to diagenetic factors (*Liutkus-Pierce, Fraser & Heckert, 2014*). The Solite Quarry B is represented by lacustrine deposits (*Fraser et al., 1996*) and *Tanytrachelos ahynis* is considered to have had an aquatic lifestyle (*Casey, Fraser & Kowalewski, 2007*; *Olsen, 1979*). As in *Tanystropheus longobardicus* and *Tanystropheus hydroides*, paired heterotopic bones parallel to the anterior caudal vertebrae occur in approximately half of the articulated specimens preserving this region, which indicates that the presence of these elements is likely related to sexual dimorphism.

### *AMNH FARB 7206*

**Age.** Carnian, early Late Triassic (*Colbert & Olsen, 2001*).

**Occurrence.** Lockatong or Stockton Formation of Hudson County, New Jersey, USA.

**Remarks.** Small reptilian specimens have been recovered from the Lockatong Formation of New Jersey, which are approximately contemporaneous to the Cow Branch Formation in Virginia from which *Tanytrachelos ahynis* is known. Although it could not be excluded that some of these specimens might represent *Tanytrachelos ahynis*, not enough diagnostic features were preserved to positively identify any of these specimens to this taxon (*Olsen, 1979*). However, one of the best-preserved specimens, AMNH FARB 7206, was recently referred to *Tanytrachelos ahynis* by *Pritchard et al. (2015)*. This specimen was here scored as a separate OTU to test this hypothesis. Additionally, *Pritchard et al. (2015)* referred a single calcaneum from the middle Norian Hayden Quarry locality of New Mexico to *Tanytrachelos ahynis* based on the striking similarities in morphology between this element and the calcaneum of AMNH FARB 7206. Since it is currently uncertain whether the latter can be referred to *Tanytrachelos ahynis*, we also consider the assignment of the Hayden Quarry calcaneum to *Tanytrachelos ahynis* as equivocal. Therefore, only specimens from the Solite Quarry B can currently be confidently assigned to *Tanytrachelos ahynis*.

### *Amotosaurus rotfeldensis* Fraser & Rieppel, 2006

**Age.** Early Anisian, early Middle Triassic.

**Occurrence.** Quarry Kossig (Upper Buntsandstein) of Baden-Württemberg, Germany (*Fraser & Rieppel, 2006*; *Ortlam, 1966*).

**Holotype.** SMNS 50830, a largely disarticulated skeleton, including an articulated cervical series, maxilla, parabasisphenoid, scapulacoracoids and pelvic girdles, and scattered dorsal vertebrae.

**Referred specimens.** Many specimens housed in the SMNS, some of which are unprepared or unaccessioned, including: SMNS 54783 a and b, a slab and counterslab preserving two mostly disarticulated skeletons, including a poorly preserved skull roof, cervical vertebrae, an articulated dorsal vertebral series, three articulated hindlimbs including pedes, and a partial forelimb including manus; SMNS 50691, three slabs, preserving a partial pes, a coracoid and maxilla, and a partial skull in ventral view, ilium, and dorsal vertebrae, respectively; SMNS 54784a and b, a slab and counterslab preserving the palatal region of a skull and a partial cervical series; SMNS 54810, disarticulated skeletons, including both cranial and extensive postcranial remains; SMNS 90600, posterior part of the vertebral column, including sacral and anterior caudal vertebrae; SMNS 90601, articulated maxilla and jugal; SMNS 90540, two skulls in palatal view; SMNS unnumbered (1), partial mandible and cervical vertebrae and ribs; SMNS unnumbered (2), disarticulated cranial elements and a partial cervical series; SMNS unnumbered (3), skull in palatal view and three anterior cervical vertebrae; SMNS unnumbered (4), sacral region.

**Diagnosis.** The diagnosis of *Amotosaurus rotfeldensis* was provided by *Fraser & Rieppel (2006*, p. 867*)*.

**Remarks.** Several specimens of associated skeletons from the Buntsandstein of Baden-Württemberg were assigned to *Macrocnemus bassanii* and *Tanystropheus longobardicus* by *Ortlam (1966)*. However, *Wild (1980b)* considered this material to represent juvenile specimens of "*Tanystropheus antiquus*", which is known from several isolated remains, mostly cervical vertebrae, from the Lower Muschelkalk of the Germanic Basin (*Spiekman & Scheyer, 2019*). *Fraser & Rieppel (2006)* re-examined the specimens from the Buntsandstein and assigned it to the new taxon *Amotosaurus rotfeldensis*. *Ezcurra (2016)* and *Pritchard et al. (2015)* incorporated *Amotosaurus rotfeldensis* in their phylogenetic analyses and provided several new morphological observations for the taxon.

### *Augustaburiania vatagini* Sennikov, 2011

**Age.** Latest Olenekian, latest Early Triasssic.

**Occurrence.** Donskaya Luka locality, right slope of the Don River valley, Lipovskaya Formation, Ilovlyanskii District, Volgograd Region, Russia (*Sennikov, 2011*).

**Holotype.** PIN 1043/587, an isolated mid-cervical vertebra.

**Referred specimens.** The referred specimens are listed in *Sennikov (2011*, p. 98*)*.

**Diagnosis.** The diagnosis was provided by *Sennikov (2011)*.

**Remarks.** *Augustaburiania vatagini* is known from the latest Olenekian (Early Triassic) of Donskaya Luka of the Don River valley, Russia, and thus represents one of the earliest known tanystropheids together with likely tanystropheid material from the Sanga do Cabral Formation (Induan–early Olenekian) of Brazil (*De Oliveira et al., 2018*, *2020*). Like other tanystropheids, such as "*Tanystropheus antiquus*" and *Tanystropheus* "*conspicuus*", *Augustaburiania vatagini* is solely known from isolated postcranial remains largely represented by cervical vertebrae. The relative length of the mid-cervical vertebrae of *Augustaburiania vatagini* is longer than that of "*Tanystropheus antiquus*" and *Amotosaurus rotfeldensis*, but shorter than that of other *Tanystropheus* species. Furthermore, the cervical vertebrae of *Augustaburiania vatagini* can be distinguished by a distinct concave ventral margin of the centrum of the cervical vertebrae, although the expression of this character in the referred material is subject to much intraspecific variation. The number of cervical vertebrae of *Augustaburiania vatagini* was considered to be eight or nine by *Sennikov (2011)* but cannot be unambiguously established, since no articulated cervical vertebrae have been preserved.

### *Raibliania calligarisi* Dalla Vecchia, 2020

**Age.** Julian, early Carnian, early Late Triassic.

**Occurrence.** Predil Limestone near Prasnig Brook, Tarvisio, Udine Province, Italy (*Dalla Vecchia, 2020*).

**Holotype.** MFSN 27532, a partial skeleton comprising the thoracic part of the vertebral column, a single partial cervical vertebra, sacral vertebrae, part of the pelvic girdle and left femur, and a purported tooth.

**Diagnosis.** The diagnosis was provided by *Dalla Vecchia (2020)*.

**Remarks.** *Raibliania calligarisi* was recently described from a single specimen from the early Carnian of northern Italy. It is morphologically very similar to *Tanystropheus longobardicus* and is slightly younger than the known occurrence of this species (*Spiekman et al., 2020a*). *Raibliania calligarisi* is distinguished from *Tanystropheus longobardicus* based on differences in the shape of the neural spines of the dorsal vertebrae, the pleurapophyses of the posterior dorsal vertebrae, the iliac blade, the anterior portion of the pubis, and a single, disarticulated tooth (*Dalla Vecchia, 2020*). The identification of the single, isolated tooth to the same individual as the rest of the specimen is somewhat equivocal, as it is located far from where the head would have been preserved. *Raibliania calligarisi* is here considered in a phylogenetic context for the first time. Another specimen discovered in the vicinity of the holotype of *Raibliania calligarisi*, MFSN 13228, consisting of three articulated caudal vertebrae, represents the only other described tetrapod remains from the locality. This specimen has not been referred to *Raibliania calligarisi* due to the lack of overlapping morphology between it and the holotype (*Dalla Vecchia, 2020*). Although the relative size of the vertebrae corresponds to that of the holotype of *Raibliania calligarisi*, the morphology of the neural spine differs distinctly from that of *Tanystropheus longobardicus*.

### *Prolacerta broomi* Parrington, 1935

**Age.** Induan, Early Triassic.

**Occurrence.** Middle Beaufort beds, *Lystrosaurus* AZ, Katberg Formation, South Africa, and Fremouw Formation, Transantarctic Mountains, Antarctica (*Groenewald & Kitching, 1995*; *Peecook, Smith & Sidor, 2019*).

**Holotype.** UMZC 2003.40—A partial skull and mandible.

**Referred specimens.** The referred specimens are listed in *Spiekman (2018*, p. 4–5*)*.

**Diagnosis.** The latest diagnosis of *Prolacerta broomi* was provided by *Spiekman (2018)*.

Remarks. *Prolacerta broomi* was first described by *Parrington (1935)* based on a crushed partial skull found in the Katberg Formation, *Lystrosaurus* Zone, near Harrismith, South Africa. *Prolacerta broomi* has played an important role in discussions on the evolutionary origin of modern reptile groups and has been considered both as an ancestral lepidosaur (e. g., *Camp, 1945*; *Parrington, 1935*) and archosaur (e.g., *Romer, 1956*). *Prolacerta broomi* was first identified as a "protorosaur" by *Camp (1945)*. Following the discovery of more specimens, the complete morphology of *Prolacerta broomi*, including the postcranium, was described (*Gow, 1975*). This revealed that *Pricea longiceps* Broom & Robinson, 1948 represented a junior synonym of *Prolacerta broomi*. The braincase of *Prolacerta broomi*

was described by *Evans (1986)*. Based on new specimens as well as a reappraisal of previously described South African material, *Modesto & Sues (2004)* provided a redescription of the skull of *Prolacerta broomi*. Specimens of *Prolacerta broomi* have also been described from Antarctica, consisting of several smaller, likely juvenile specimens, and a single, large-sized specimen, which is slightly larger than the specimens known from South Africa (*Colbert, 1987*; *Spiekman, 2018*). Although previously considered a member of the "protorosaurs", recent phylogenetic analyses indicate that *Prolacerta broomi* is more closely related to Archosauriformes than *Protorosaurus speneri* and tanystropheids (e.g., *Dilkes, 1998*; *Ezcurra, 2016*; *Modesto & Sues, 2004*; *Pritchard et al., 2015*; *Rieppel, Fraser & Nosotti, 2003*). *Prolacerta broomi* has been used widely as an outgroup in phylogenetic analyses focused on Archosauriformes and early crown-archosaurs (e.g., *Butler et al., 2015*; *Desojo, Ezcurra & Schultz, 2011*; *Dilkes & Sues, 2009*; *Nesbitt, 2011*; *Sookias, 2016*).

### *Ozimek volans* Dzik & Sulej, 2016

**Age.** Late Carnian or early Norian, Late Triassic (*Dzik & Sulej, 2016*; *Szulc, Racki & Bodzioch, 2017*).

**Occurrence.** Grabowa Formation (Silesian Keuper) of Krasiejów, Upper Silesia, Poland.

**Holotype.** ZPAL AbIII/2512, partial skeleton missing the skull.

**Referred specimens.** A complete hypodigm can be found in *Dzik & Sulej (2016)*.

**Diagnosis.** The diagnosis was provided by *Dzik & Sulej (2016)*.

**Remarks.** *Ozimek volans* was recently described based on several partial and disarticulated skeletons (*Dzik & Sulej, 2016*). An elongate vertebra now referred to this taxon was previously linked to either pterosaurs or *Tanystropheus* due to its extreme elongation (*Dzik & Sulej, 2007*). It is considered a close relative of the gliding reptile *Sharovipteryx mirabilis* and was possibly also a glider, although a comparison is limited due to the poorly known morphology of *Sharovipteryx mirabilis*. The morphology of *Ozimek volans* is highly derived and differs distinctly from other "protorosaurs" in the relative length and gracile construction of the limb bones and the configuration of the pectoral girdle, which includes an enlarged coracoid and possibly an ossified sternum. Formally assigned to the family Sharovipterygidae, *Ozimek volans* was considered a "protorosaur" based on the presence of elongate cervical vertebrae, the posterior curvature of the scapula, and the procoelous articulation surfaces of the cervical vertebrae (which occur in *Tanytrachelos ahynis* and *Langobardisaurus pandolfii* among tanystropheids, but which is widespread among some other diapsids, e.g., drepanosaurids; *Dzik & Sulej, 2016*). Unfortunately, the skull morphology of *Ozimek volans* is only partially known and identification of many cranial bones is uncertain due to their disassociation and peculiar morphology. *Ozimek volans* has been included in the phylogenetic analysis of *Pritchard & Sues (2019)*, in which it was recovered within Tanystropheidae as the sister taxon to a clade comprising *Langobardisaurus pandolfii* and *Tanytrachelos ahynis*.

### *Elessaurus gondwanoccidens* De Oliveira et al., 2020

**Age.** Induan-Olenekian, Early Triassic (*De Oliveira et al., 2020*; *Dias-da-Silva et al., 2017*).

**Occurrence.** Bica São Tomé, Sanga do Cabral Formation, São Francisco de Assis, Rio Grande do Sul, southern Brazil.

**Holotype.** UFSM 11471, a left hindlimb, partial pelvis, a single sacral vertebra and three caudal vertebrae.

**Diagnosis.** The diagnosis was provided by *De Oliveira et al. (2020)*.

**Remarks.** *Elessaurus gondwanoccidens* is known from a single hind limb, partial pelvis, and a few sacral and caudal vertebrae, and has been identified as the sister taxon to Tanystropheidae (*De Oliveira et al., 2020*). In addition to *Elessaurus gondwanoccidens*, several isolated cervical vertebrae with a typical tanystropheid morphology (*De Oliveira et al., 2018*), as well as the non-archosauriform crocopod *Teyujagua paradoxa* (*Pinheiro, Simão-Oliveira & Butler, 2019*), are also known from the Sanga do Cabral Formation. Due to the limited morphological information currently available for *Elessaurus gondwanoccidens*, its phylogenetic placement is somewhat uncertain.

### *Jesairosaurus lehmani* Jalil, 1997

**Age.** Late Olenekian-early Anisian, late Early Triassic to early Middle Triassic (*Jalil, 1999*).

**Occurrence.** Site 5003 of Busson, at the base of the Zarzaitine Formation, Algeria.

**Holotype.** ZAR 06, a nearly complete skull and mandible, the neural arches of the five posteriormost cervical vertebrae, the complete left and partial right pectoral girdle, and the proximal end of the left humerus.

**Referred specimens.** The hypodigm was provided by *Jalil (1997)* and *Ezcurra (2016)*.

**Diagnosis.** The most recent diagnosis was provided by *Ezcurra (2016)*.

**Remarks.** The specimens now assigned to *Jesairosaurus lehmani* were originally interpreted as procolophonid remains (*Lehman, 1971*). However, detailed observation was hampered by a hard hematite layer that covered the specimens. Additional preparation revealed the diapsid affinity of the material (*Jalil, 1990*), and it was later described in detail and assigned to *Jesairosaurus lehmani* (*Jalil, 1997*). More recently a morphological redescription of *Jesairosaurus lehmani* was provided by *Ezcurra (2016)*. *Jalil (1997)* identified *Jesairosaurus lehmani* as a "protorosaur" and in a phylogenetic analysis found it to be the sister taxon to "*Malerisaurus langstoni*", whereas the clade they formed was recovered as the sister group to a tanystropheid clade that included *Boreopricea funerea* and *Cosesaurus aviceps*. In a re-analysis of this matrix by *Rieppel, Fraser & Nosotti (2003)*, *Jesairosaurus lehmani* formed a polytomy with other "protorosaurs" (including drepanosaurids) and a clade composed of *Prolacerta broomi* and the archosauriforms *Proterosuchus* and *Euparkeria*. In the recent analysis by *Ezcurra (2016)*, *Jesairosaurus lehmani* was recovered as the sister taxon to Tanystropheidae. The unstable position of

*Jesairosaurus lehmani* might be related to the poorly resolved relationships of former "protorosaurs" and the difficulties to confidently assess morphological details for this taxon because of the preservation of the specimens within the hard hematite layer (*Jalil, 1997*).

### *Langobardisaurus pandolfii* Renesto, 1994b

**Age.** Alaunian to Revueltian, middle Norian, Late Triassic.

**Occurrence.** The uppermost section of the Zorzino Limestone Cene quarry, Lombardy, Italy (*Renesto, 1994b*), lower member of the Forni Dolostone of Friuli, Italy (*Renesto & Dalla Vecchia, 2000*), and the Seefeld Formation, near Innsbruck, Austria (*Saller, Renesto & Dalla Vecchia, 2013*).

**Holotype.** MCSNB 2883, an articulated partial skeleton, missing both forelimbs completely, as well as parts of the skull, feet, and tail.

**Referred specimens.** MCSNB 4860, complete and articulated juvenile specimen preserved in ventral view, with the skull covered by the neck and trunk; MFSN 1921, a virtually complete and articulated adult specimen, including a well-preserved skull. Only the posterior part of the tail and part of the left forelimb are missing; MFSN 26829, a partial articulated adult specimen, preserving a nearly complete right hindlimb, a partial left hindlimb, and some poorly visible parts of the vertebral column and possibly the pelvic girdle; P 10121, a nearly complete impression of an articulated adult, only missing part of the tail, with some fragments of the limb bones and teeth preserved.

**Diagnosis.** The latest emended diagnosis for *Langobardisaurus pandolfii* was provided by *Saller, Renesto & Dalla Vecchia (2013)*.

**Remarks.** *Renesto (1994b)* was the first to describe the genus *Langobardisaurus* based on two specimens originally found in 1974, which were assigned to *Langobardisaurus pandolfii* and considered to be closely related to known tanystropheids, specifically *Macrocnemus bassanii*. It was interpreted as a terrestrial insectivore based on the presence of tricuspid teeth. *Bizzarini & Muscio (1995)* proposed a new species, *Langobardisaurus rossii*, based on a new but poorly preserved specimen from the Forni Dolostone of Friuli, Italy. This specimen was later considered as a probable rhynchocephalian lepidosauromorph, mainly inferred from its body proportions, particularly the relative size of the skull and cervical and trunk regions (*Renesto & Dalla Vecchia, 2007*). However, the poor preservation of this specimen prevents an unequivocal taxonomic determination. Another species, *Langobardisaurus? tonelloi*, was tentatively ascribed to the genus based on a complete specimen (MFSN 1921) by *Muscio (1996)*. The species was considered to differ from *Langobardisaurus pandolfii* in its phalangeal formula and dentition. However, *Renesto & Dalla Vecchia (2000)* could not find any differences between the phalangeal formula in these two taxa and considered the minor differences in dentition to be attributable to ontogenetic variation, and thus considered *Langobardisaurus tonelloi* to likely represent a junior synonym of *Langobardisaurus pandolfii*, which was later

supported by *Saller, Renesto & Dalla Vecchia (2013)*. MFSN 1921 allowed the first detailed description of the skull of *Langobardisaurus pandolfii* and revealed a unique dentition among archosauromorphs, consisting of an edentulous premaxilla and anterior margin of the maxilla, followed by tricuspid teeth more posteriorly on the maxilla and dentary, and terminating in a very large, molar-like crushing tooth on both the maxilla and dentary. *Renesto & Dalla Vecchia (2000)* hypothesized that *Langobardisaurus pandolfii* used this highly specialized dentition to feed on large insects, crustaceans, and small scaly fishes. *Renesto, Dalla Vecchia & Peters (2002)* described another specimen, MFSN 26829, and considered facultative bipedal locomotion for *Langobardisaurus pandolfii*, which has also been proposed for *Macrocnemus bassanii* (*Rieppel, 1989*). P 10121, a poorly preserved specimen of *Langobardisaurus pandolfii*, consists of the impression, as well as bone fragments, of a nearly complete skeleton. This specimen was found in the Seefeld Formation of Austria, extending the biogeographic range of the taxon outside Italy (*Saller, Renesto & Dalla Vecchia, 2013*).

### *Dinocephalosaurus orientalis* Li, 2003

**Age.** Anisian, Middle Triassic (*Sun et al., 2016*).

**Occurrence.** Member II of the Guanling Formation of Xinmin close to Panxian, and Luoping, Guizhou Province, China (*Li, 2003*; *Liu et al., 2017*; *Rieppel, Li & Fraser, 2008*).

**Holotype.** IVPP V13767, an almost complete skull and the three anteriormost cervical vertebrae and associated ribs.

**Referred specimens.** ZMNH M8752, an undescribed specimen of which the pelvic morphology was briefly mentioned in comparison to that of *Fuyuansaurus* by *Fraser, Rieppel & Li (2013)*; LPV 30280, a partial, articulated skeleton including some disarticulated skull bones, most of the cervical column, parts of the thorax, hindlimbs, and anterior part of the tail. Within the thorax of this specimen some elements belonging to an embryo are preserved (*Liu et al., 2017*); IVPP V13898, a relatively complete skeleton including a skull preserved in ventral view, a complete cervical series, and parts of the thorax including an articulated fore and hindlimb (*Rieppel, Li & Fraser, 2008*).

**Diagnosis.** The most recent diagnosis of *Dinocephalosaurus orientalis* was provided by *Rieppel, Li & Fraser (2008)*.

**Remarks.** *Dinocephalosaurus orientalis* was first described based exclusively on the holotype, which only preserves the skull and three anteriormost cervical vertebrae (*Li, 2003*). The discovery of a specimen preserving much of the postcranial skeleton subsequently revealed a striking convergence between *Dinocephalosaurus orientalis* and *Tanystropheus* spp. (*Li, Rieppel & LaBarbera, 2004*). *Dinocephalosaurus orientalis* shares the extreme elongation of the neck with *Tanystropheus* spp., but achieved this elongation through different means, since its neck is composed of at least 33 vertebrae that are comparatively short, whereas that of *Tanystropheus hydroides* and *Tanystropheus longobardicus* is composed of 13 hyperelongate cervical vertebrae (*Li, Rieppel & Fraser,*

*2017*; *Rieppel et al., 2010*; *Rieppel, Li & Fraser, 2008*). Additionally, the postcranial morphology of *Dinocephalosaurus orientalis* shows clear adaptations to a fully aquatic lifestyle, most notably in the paddle-like limbs. The unique morphology of *Dinocephalosaurus orientalis* has led to various hypotheses regarding its lifestyle and feeding method. *Li, Rieppel & LaBarbera (2004)* tentatively suggested suction feeding for *Dinocephalosaurus orientalis*, which was refuted by *Peters (2005)* and *Demes & Krause (2005)*. The former argued that *Dinocephalosaurus orientalis* was a benthic ambush predator and a very poor swimmer. This suggestion was in turn repudiated by *LaBarbera & Rieppel (2005)*. A general anatomical description of *Dinocephalosaurus orientalis* was provided by *Rieppel, Li & Fraser (2008)*. In this study *Dinocephalosaurus orientalis* was incorporated into the combined dataset provided by *Rieppel, Fraser & Nosotti (2003)* for a phylogenetic analysis, which found a polytomy formed by *Jesairosaurus lehmani*, *Dinocephalosaurus orientalis*, drepanosaurids, and tanystropheids. *Liu et al. (2017)* described a new *Dinocephalosaurus* specimen, which preserves articulated remains of a much smaller *Dinocephalosaurus* specimen within the thorax of the adult, indicating the first example of vivipary in an archosauromorph reptile. This study also provided an updated version of the phylogenetic analysis of *Rieppel, Li & Fraser (2008)*, in which they recovered *Dinocephalosaurus orientalis* as the sister taxon to Tanystropheidae. Another, isolated embryo bearing close similarities to *Dinocephalosaurus orientalis* was described by *Li, Rieppel & Fraser (2017)*. It represents a separate taxon, since it differs distinctly from *Dinocephalosaurus orientalis* in its relative limb proportions and in having 24 rather than at least 33 cervical vertebrae. However, it has not been assigned to a separate species due to the very early ontogenetic stage of the specimen and it was instead referred to as a "dinocephalosaur", indicating the presence of multiple closely related *Dinocephalosaurus*-like taxa.

### *Fuyuansaurus acutirostris* Fraser, Rieppel & Li, 2013

**Age.** Ladinian, Middle Triassic (*Sun et al., 2016*).

**Occurrence.** Zhunganpo Formation of Guizhou Province, China (*Fraser, Rieppel & Li, 2013*).

**Holotype.** IVPP V17983, a partial skeleton preserving a skull, cervical vertebral column, a few dorsal vertebrae, and a partial pectoral and pelvic girdle.

**Diagnosis.** The diagnosis of *Fuyuansaurus acutirostris* was provided by *Fraser, Rieppel & Li (2013)*.

**Remarks.** *Fuyuansaurus acutirostris* has a long neck and an elongate rostrum, and is known from a single, possibly juvenile, specimen. It has been interpreted as an aquatic taxon and bears typical tanystropheid features in the presence of a long neck composed of elongated cervical vertebrae and corresponding ribs, and a scapular blade that is semi-lunar in shape (*Fraser, Rieppel & Li, 2013*). However, *Fuyuansaurus acutirostris* differs from other tanystropheids in its elongate and tapered rostrum and a pelvic girdle that lacks a thyroid fenestra between the pubis and ischium. *Fuyuansaurus acutirostris* has previously only been included in the phylogenetic analysis of *Ezcurra & Butler (2018)*, in

which it was scored for the purpose of a disparity analysis. Therein it was recovered in a large polytomy within Tanystropheidae.

### *Pectodens zhenyuensis* Li et al., 2017

**Age.** Anisian, Middle Triassic (*Sun et al., 2016*).

**Occurrence.** Member II of the Guanling Formation, Luoping, Yunnan Province, China (*Li et al., 2017*).

**Holotype.** IVPP V18578, a nearly complete and articulated skeleton including the skull.

**Diagnosis.** The diagnosis was provided by *Li et al. (2017)*.

**Remarks.** *Pectodens zhenyuensis* is a small, highly gracile archosauromorph with an elongate neck, tail, and limbs. It bears certain characteristics typical of "protorosaurs", most notably in having a long neck with elongate cervical vertebrae and ribs. *Li et al. (2017)* addressed the similarities of *Pectodens zhenyuensis* to "Protorosauria" but considered its inclusion in this group only tentative since it lacks several diagnostic features, such as the presence of a hooked fifth metatarsal and a thyroid fenestra between the pubis and ischium, and because it differed from other "protorosaurs" in the shape of its skull and marginal dentition. Most carpal bones are missing in the only known specimen, even though the manus is fully articulated, possibly because these bones had not yet ossified due to the early ontogenetic stage of the specimen. The phylogenetic position of *Pectodens zhenyuensis* has previously only been tested in the phylogenetic analysis of *Ezcurra & Butler (2018)*. Like *Fuyuansaurus acutirostris*, it was recovered in a large polytomy within Tanystropheidae.

### *Mesosuchus browni* Watson, 1912

**Age.** Early Anisian, early Middle Triassic.

**Occurrence.** Burgersdorp Formation of the Beaufort Group near Aliwal North, Subzone B of the *Cynognathus* AZ, Eastern Cape Province, South Africa (*Dilkes, 1998*; *Hancox, 2000*).

**Holotype.** SAM-PK-5882, a partial skull consisting of the rostrum, braincase, and palatal regions, mandible, a partial vertebral column, an incomplete scapula and pelvic girdle, and partial fore and hindlimbs.

**Referred specimens.** The hypodigm of *Mesosuchus browni* was provided by *Dilkes (1998)* and was also listed by *Ezcurra (2016)*.

**Diagnosis.** The most recent diagnosis was provided by *Dilkes (1998)*.

**Remarks.** *Mesosuchus browni* is considered the best-known non-rhynchosaurid rhynchosaur (*Butler et al., 2015*; *Hone & Benton, 2008*). Like the other early rhynchosaurs *Howesia browni* and *Eohyosaurus wolvaardti*, as well as the archosauriforms *Euparkeria capensis* and *Erythrosuchus africanus*, it is known from the Burgersdorp Formation near Aliwal North in the Eastern Cape, South Africa (*Butler et al., 2015*; *Ezcurra, Montefeltro &*

*Butler, 2016*; *Rubidge, 2005*). The morphology of *Mesosuchus browni* has been studied several times (*Broom, 1913a*, *1913b*, *1925*; *Haughton, 1922*, *1924a*; *Watson, 1912*) and has been most comprehensively described by *Dilkes (1998)*. In many ways *Mesosuchus browni* shows a morphology that is intermediate between that of rhynchosaurids and other early archosauromorphs. SAM-PK-6536 represents a particularly informative specimen, as it includes a complete, virtually undistorted skull. The braincase of this specimen was recently described in detail by *Sobral & Müller (2019)*, which revealed the presence of a pneumatic sinus between the basal tubera. Pneumatization of the braincase was previously considered a derived archosaur trait, but its presence in *Mesosuchus browni* indicates it had evolved much earlier in the archosauromorph lineage.

### *Howesia browni* Broom, 1905a

**Age.** Early Anisian, early Middle Triassic.

**Occurrence.** Burgersdorp Formation of the Beaufort Group near Aliwal North, Subzone B of the *Cynognathus* AZ, Eastern Cape Province, South Africa (*Dilkes, 1995*; *Hancox, 2000*).

**Holotype.** SAM-PK-5884, a flattened partial skull missing the rostrum and occipital regions, and a mandible.

**Referred specimens.** SAM-PK-5885, a flattened partial skull, mandible, and atlas-axis complex; SAM-PK-5886, a postcranial skeleton consisting of a partial vertebral column, an incomplete pelvic girdle and left hindlimb, and a complete right tarsus.

**Diagnosis.** The most recent diagnosis was provided by *Dilkes (1995)*.

**Remarks.** *Howesia browni* is a non-rhynchosaurid rhynchosaur closely related to *Mesosuchus browni* that was first described by *Broom (1905a)*. Following additional preparation of the three known specimens, *Howesia browni* was extensively described by *Dilkes (1995)*.

### *Eohyosaurus wolvaardti* Butler et al., 2015

**Age.** Early Anisian, early Middle Triassic.

**Occurrence.** Burgersdorp Formation of the Beaufort Group near Aliwal North, Cynognathus AZ, Subzone B of the Cynognathus AZ, Eastern Cape Province, South Africa (*Butler et al., 2015*).

**Holotype.** SAM-PK-K10159, skull and mandible missing the anterior part of the rostrum.

**Diagnosis.** The diagnosis was provided by *Butler et al. (2015)*.

**Remarks.** *Butler et al. (2015)* described *Eohyosaurus wolvaardti* and included it in a phylogenetic analysis. It was recovered as the sister taxon to rhynchosaurids and was thus found to be more closely related to this clade than *Mesosuchus browni* and *Howesia browni*. In another phylogenetic analysis focusing on rhynchosaurs, it was found in a polytomy with rhynchosaurids, *Mesosuchus browni*, *Howesia browni*, and *Noteosuchus*

*colletti*, the last being a poorly known early rhynchosaur from the Induan of South Africa (*Ezcurra, Montefeltro & Butler, 2016*). However, more recent iterations of this data matrix corroborated the phylogenetic placement of *Eohyosaurus wolvaardti* by *Butler et al. (2015)* as the sister taxon to rhynchosaurids (e.g., *Ezcurra, 2016*).

### *Pamelaria dolichotrachela* Sen, 2003

**Age.** Anisian, Middle Triassic (*Lucas, 2010*).

**Occurrence.** Yerrapalli Formation, Gondwana Supergroup, Pranhita–Godavari Basin, southern India.

**Holotype.** ISIR 316, a partial skeleton including a largely complete skull.

**Referred specimens.** The referred specimens are listed in *Sen (2003)* and *Ezcurra (2016)*.

**Remarks.** *Pamelaria dolichotrachela* is known from three specimens originating from the Yerrapalli Formation (Middle Triassic) of India and was originally identified as a "protorosaur" (*Sen, 2003*). However, recent phylogenetic analyses have revealed that *Pamelaria dolichotrachela* is an allokotosaur that is closely related to *Azendohsaurus* spp. and *Shringasaurus indicus* (*Ezcurra, 2016*; *Nesbitt et al., 2015*; *Sengupta, Ezcurra & Bandyopadhyay, 2017*; and subsequent modifications of these matrices).

### *Azendohsaurus madagaskarensis* Flynn et al., 2010

**Age.** Late Ladinian to early Carnian, late Middle Triassic to early Late Triassic.

**Occurrence.** Locality M-28 close to the eastern bank of the Malio River, west of Isalo National Park, southern Madagascar, Isalo II or the Makay Formation (*Nesbitt et al., 2015*).

**Holotype.** UA 7-20-99-653, partial skull and five anterior cervical vertebrae.

**Referred specimens.** A list of referred specimens can be found in appendix 1 of *Nesbitt et al. (2015)*.

**Diagnosis.** The most recent diagnosis was provided in *Nesbitt et al. (2015)*.

**Remarks.** *Azendohsaurus laaroussii* is the type species, but *Azendohsaurus madagaskarensis* represents the best-known member of the genus. The former species was originally described from a few teeth and dental fragments and interpreted as an ornithischian dinosaur (*Dutuit, 1972*) and later as a sauropodomorph dinosaur (e.g., *Gauffre, 1993*). Postcranial remains from the type locality of *Azendohsaurus laaroussii* can likely also be referred to this species, and indicate that the taxon did not belong to Dinosauria (*Cubo & Jalil, 2019*; *Jalil & Knoll, 2002*). Extensive three-dimensionally preserved remains of various individuals from the late Middle Triassic to early Late Triassic of southern Madagascar, which closely resembled the known material of *Azendohsaurus laaroussii*, were assigned to *Azendohsaurus madagaskarensis*. The skull and mandible were initially described by *Flynn et al. (2010)*. A description of the postcranium and a phylogenetic hypothesis for *Azendohsaurus madagaskarensis* was provided by *Nesbitt et al. (2015)*.

This revealed a new clade of non-archosauriform archosauromorphs, Allokotosauria, that includes *Azendohsaurus* spp., *Trilophosaurus* spp., *Pamelaria dolichotrachela*, *Spinosuchus caseanus*, and *Teraterpeton hrynewichorum*. *Shringasaurus indicus* was later also referred to this clade (e.g., *Sengupta, Ezcurra & Bandyopadhyay, 2017*), which has also been recovered in subsequent phylogenetic analyses (e.g., *Ezcurra, 2016*; *Pritchard & Sues, 2019*), confirming it as one of the three major lineages of non-archosauriform archosauromorphs previously recognized, together with Rhynchosauria and Tanystropheidae. *Azendohsaurus madagaskarensis* represents one of the best-known non-archosauriform archosauromorphs. It was herbivorous and has a relatively large body size among early archosauromorphs, being approximately 2 to 3 m in length.

### *Trilophosaurus buettneri* Case, 1928

**Age.** Early Norian, Late Triassic (*Kligman et al., 2020*).

**Occurrence.** *Trilophosaurus* site 1 and *Trilophosaurus* quarries 1-3, Colorado City Formation; Walker's Tank and lower Kalgary site, Tecovas Formation, western Texas, USA (*Spielmann et al., 2008*).

**Holotype.** UMMP 2338, an incomplete right dentary bearing teeth.

**Referred specimens.** A list of referred specimens can be found in appendix 1 of *Spielmann et al. (2008)*.

**Diagnosis.** The most recent diagnosis is provided by *Spielmann et al. (2008*, p. 11*)*

**Remarks.** *Trilophosaurus buettneri* was first described based on a dentary fragment bearing teeth (*Case, 1928*) and interpreted to be closely related to procolophonids. Additional specimens that gave a much more complete account of the taxon were described by *Gregory (1945)*. *Gregory (1945)* referred *Trilophosaurus buettneri* to "Protorosauria". The skull of *Trilophosaurus buettneri* was later redescribed by *Parks (1969)*. Another species referred to the genus, "*Trilophosaurus jacobsi*", was proposed to represent a junior synonym of *Spinosuchus caseanus* by *Nesbitt et al. (2015)*.
Two additional species of the genus, *Trilophosaurus phasmalophus* and *Trilophosaurus dornorum*, are exclusively known from limited and isolated cranial or dental remains (*Kligman et al., 2020*; *Mueller & Parker, 2006*). *Trilophosaurus buettneri* was redescribed and reinterpreted as a non-archosauriform archosauromorph outside "Protorosauria" by *Spielmann et al. (2008)*. *Trilophosaurus buettneri* was later found within the newly erected clade Allokotosauria (e.g., *Ezcurra, 2016*; *Nesbitt et al., 2015*). The manus of *Trilophosaurus buettneri* was redescribed by *Nesbitt et al. (2015)*. *Trilophosaurus buettneri* was herbivorous and has a remarkable dentition characterized by an edentulous, beak-like rostrum and labiolingually very wide tricuspid teeth further posterior in the jaws.
An arboreal lifestyle has been suggested for *Trilophosaurus buettneri* by *Spielmann, Heckert & Lucas (2005)*, based on the relative proportions of the appendicular skeleton and the presence of large curved claws. Although similar claws occur in other allokotosaurs (e.

g., *Azendohsaurus madagaskarensis*), a similar interpretation has not been made for these taxa (*Nesbitt et al., 2015*).

### *Teyujagua paradoxa* Pinheiro et al., 2016

**Age.** Induan-Olenekian, Early Triassic (*Dias-da-Silva et al., 2017*; *Pinheiro, Simão-Oliveira & Butler, 2019*).

**Occurrence.** Bica São Tomé, Sanga do Cabral Formation, São Francisco de Assis, Rio Grande do Sul, southern Brazil.

**Holotype.** UNIPAMPA 653, a nearly complete skull and mandible, as well as parts of the first five cervical vertebrae.

**Diagnosis.** The most recent diagnosis was provided by *Pinheiro, Simão-Oliveira & Butler (2019)*.

**Remarks.** *Teyujagua paradoxa* was first described by *Pinheiro et al. (2016)* and subsequently described in further detail with the aid of µCT data by *Pinheiro, Simão-Oliveira & Butler (2019)*. The skull and mandible of *Teyujagua paradoxa* exhibit a remarkable combination of plesiomorphic features, typical of non-archosauriform archosauromorphs (e.g., absence of the antorbital fenestra), and derived features that had previously been considered to represent synapomorphies for Archosauriformes (e.g., the presence of an external mandibular fenestra). This is also reflected in the position of *Teyujagua paradoxa* in previous phylogenetic analyses including this species, in which it was recovered as being very closely related to Archosauriformes (*Pinheiro et al., 2016*; *Pinheiro, Simão-Oliveira & Butler, 2019*).

### *Proterosuchus fergusi* Broom, 1903

**Age.** Induan, Early Triassic (*Lucas, 2010*).

**Occurrence.** Several localities of the upper Balfour Formation and/or lower Katberg Formation, *Lystrosaurus* AZ, Karoo Supergroup in the Eastern Cape and Free State provinces of South Africa (a list of the exact localities is provided in table 3 of *Ezcurra & Butler, 2015b*).

**Neotype.** RC 846 was assigned as the neotype of *Proterosuchus fergusi* by *Ezcurra & Butler (2015b)*, since the holotype SAM-PK-591, a partially preserved skull, is undiagnostic.

**Hypodigm.** A list of the referred specimens is presented in *Ezcurra & Butler (2015b*, p. 164–165) and *Ezcurra (2016*, p. 47).

**Diagnosis.** The most recent emended diagnosis was provided by *Ezcurra & Butler (2015b)*.

**Remarks.** *Proterosuchus fergusi* is a medium-sized, predatory archosauriform characterized by a distinctly downturned premaxilla. The genus *Proterosuchus*, of which *Proterosuchus fergusi* is the type species, has a complicated taxonomic history and specimens have been referred to various species within the genus and the now synonymous

genera "*Chasmatosaurus*" and "*Elaphrosuchus*". An extensive revision of the genus was provided by *Ezcurra & Butler (2015b)*, in which the species were distinguished based on a combination of cranial proportions and discrete characters. Quantitative investigation of cranial proportions was also used to investigate the ontogeny of *Proterosuchus fergusi* (*Ezcurra & Butler, 2015a*). A detailed morphological description of *Proterosuchus fergusi* has been provided by *Cruickshank (1972*; *Proterosuchus vanhoepeni* therein). Recently, the endocast of *Proterosuchus fergusi* was described by *Brown et al. (2019)*, which supported a semi-aquatic lifestyle for the taxon, although a histological and sedimentological study suggests a more terrestrial habit (*Botha-Brink & Smith, 2011*). Although the skull of *Proterosuchus fergusi* is represented by several, well-preserved specimens, the postcranium is comparatively much less well-known.

### *Proterosuchus alexanderi* Hoffman, 1965

**Age.** Induan, Early Triassic (*Lucas, 2010*).

**Occurrence.** Farm Zeekoegat close to Venterstad, upper Balfour Formation and/or lower Katberg Formation, *Lystrosaurus* AZ, Karoo Supergroup, Eastern Cape Province, South Africa (*Ezcurra & Butler, 2015b*; *Hoffman, 1965*).

**Holotype.** NMQR 1484, a well-preserved specimen comprising a largely complete skull missing most of the premaxillae, and a postcranial skeleton missing part of the tail and the appendicular skeleton.

**Diagnosis.** The most recent diagnosis was provided by *Ezcurra & Butler (2015b)*.

**Remarks.** NMQR 1484 includes the best-preserved postcranial skeleton of the genus *Proterosuchus*. This specimen was first described by *Hoffman (1965)* and assigned to "*Chasmatosaurus*" *alexanderi*. This taxon was later synonymized with "*Chasmatosaurus vanhoepeni*" (e.g., *Cruickshank, 1972*), which in turn was later synonymized with *Proterosuchus fergusi* (*Welman, 1998*). However, the most recent revision of proterosuchid taxonomy (*Ezcurra & Butler, 2015b*) found NMQR 1484 to be taxonomically distinct from *Proterosuchus fergusi*, yet referrable to the same genus, and it was therefore reassigned as the only known specimen of the new combination *Proterosuchus alexanderi*. *Proterosuchus alexanderi* is included here as a separate OTU because its postcranial morphology is better known to that of the type species *Proterosuchus fergusi*.

### *Euparkeria capensis* Broom, 1913b

**Age.** Early Anisian, early Middle Triassic.

**Occurrence.** Burgersdorp Formation of the Beaufort Group near Aliwal North, Subzone B of the *Cynognathus* AZ, Eastern Cape Province, South Africa (*Ewer, 1965*; *Hancox, 2000*; *Sookias, 2016*).

**Holotype.** SAM-PK-5867, a complete skull and mandible and a largely complete and articulated postcranial skeleton, only missing most of the hands and feet and the majority of the tail.

**Referred specimens.** A complete hypodigm is listed in *Sookias (2016)* and an updated specimen list of specimens preserving cranial elements is provided in *Sookias et al. (2020)*.

**Diagnosis.** An emended diagnosis was recently provided in *Sookias et al. (2020)*.

**Remarks.** *Euparkeria capensis* is a small, carnivorous archosauriform that has received a lot of interest because it represents a well-known early archosauriform that is closely related to Archosauria. The most extensive morphological description of *Euparkeria capensis* was provided by *Ewer (1965)*. Recently, a detailed revision of the Euparkeriidae, a clade comprising *Euparkeria capensis* and its closest relatives, provided a morphological reevaluation of this taxon (*Sookias, 2016*; *Sookias & Butler, 2013*; *Sookias et al., 2014*), including a detailed cranial description (*Sookias et al., 2020*). In addition, the braincase has been described in detail by *Sobral et al. (2016)*, and the posture of *Euparkeria capensis* has been investigated through a biomechanical study by *Demuth, Rayfield & Hutchinson (2020)*.

### *Erythrosuchus africanus* Broom, 1905b

**Age.** Early Anisian, early Middle Triassic (*Abdala, Hancox & Neveling, 2005*).

**Occurrence.** Various localities in South Africa, most notably near Aliwal North and Burgersdorp, Eastern Cape Province, and Rouxville, Free State Province, Burgersdorp Formation of the Beaufort Group, *Cynognathus* AZ subzone B, Karoo Supergroup (*Abdala, Hancox & Neveling, 2005*; *Gower, 2003*).

**Holotype.** SAM-PK-905, an incomplete postcranial skeleton, mainly consisting of the pectoral and pelvic girdles, a partial forelimb, and vertebrae.

**Referred specimens.** A list of the referred specimens is provided in Appendix 1 of *Gower (2003)*.

**Diagnosis.** The most recent diagnosis for *Erythrosuchus africanus* is provided by *Ezcurra (2016)*.

**Remarks.** *Erythrosuchus africanus* is a large-sized carnivorous archosauriform with a skull that is particularly large compared to the postcranium. It was first described based on a partial postcranial skeleton by *Broom (1905b)*. A new and more completely preserved specimen, including a partial skull, was described by *Huene (1911)*. *Erythrosuchus africanus* was more recently described extensively by *Gower (2003)*, and separate studies addressed the morphology of the pes (*Gower, 1996*), and braincase (*Gower, 1997*).

## MATERIALS AND METHODS

To resolve the phylogenetic relationships of tanystropheids and other "protorosaurs", a new comprehensive character matrix was constructed consisting of 307 characters. These include 40 new characters, with the remaining characters having been compiled and modified from the literature (see "Character sampling and formulation" below). The matrix contains 25 ratio characters, and 55 multistate characters are ordered as they

are considered to form a transformational series in which at least one state represents a clear intermediate between two other states. Currently there is ongoing debate whether discrete characters should be ordered and whether to discretize continuous ratio data (e.g., *Grand et al., 2013*; *Simões et al., 2016*), but their application has been considered phylogenetically informative in previous studies of archosauromorph relationships (*Ezcurra, 2016*; *Nesbitt, 2011*). To test for the influence of ordered and ratio characters for our dataset, one round of analyses was performed including the ratio characters and with characters indicated as ordered treated as such, and another round without ratio characters and with all remaining characters treated as unordered. A total of 42 OTUs are included, the large majority of which were scored based on personal observations of relevant specimens. *Petrolacosaurus kansensis* was assigned as the outgroup. The continuous values calculated for the ratio characters are listed in the Supplementary Information for the sampled specimens of each OTU. These values were quantitatively partitioned into discrete character states using a classical cluster analysis in PAST 2.17c (*Hammer, Harper & Ryan, 2001*). Following the methodology proposed by *Ezcurra (2016)*, in the tree derived from the cluster analysis a set of values were considered as a separate character state when the internal distance of that set was higher than the distance that separated it from (an) other value(s). In order to account for taphonomic deformation and other measurement uncertainties, character states that were separated by less than 5% of the total range present in the raw ratio values were merged. A full list of all character scorings and the specimens and literature that have been employed are provided in the Supplementary Information. The specimens that were scored are specified for each character individually for future assessment and comparison.

Of the 42 included OTUs, *Czatkowiella harae*, *Tanystropheus "conspicuus"*, and *"Tanystropheus antiquus"* can be considered problematic and their inclusion could therefore result in more ambiguous tree topologies. *Czatkowiella harae* is exclusively known from fragmented and isolated remains from the fissure deposits of Czatkowice, Poland (*Borsuk-Białynicka & Evans, 2009b*), and it is possible that multiple taxa are represented in the material (*Ezcurra, Scheyer & Butler, 2014*). The material currently known for *Tanystropheus "conspicuus"* is undiagnostic at the species level and this taxon is therefore currently considered a nomen dubium (*Spiekman & Scheyer, 2019*). *"Tanystropheus antiquus"* is currently insufficiently defined, since much of the type material that was considered to have been lost was recently rediscovered and is in need of revision (*Skawiński et al., 2017*; *Spiekman & Scheyer, 2019*). Therefore, both analyses outlined above (one round excluding ratio characters and treating all characters as unordered, and one round including ratio and ordered characters) were performed once including all 42 OTUs, and once excluding *Czatkowiella harae*, *Tanystropheus "conspicuus"*, and *"Tanystropheus antiquus"* a priori. Thus, in total four different analyses were performed. For the fourth analysis, which is the analysis including both ratio and ordered characters and which excluded the three problematic OTUs, we performed several heuristic searches in which specific constraints were defined (e.g., "Protorosauria" was enforced as a monophyletic group) to explore how many additional steps were required to obtain these alternative topologies.

The analyses were performed in TNT 1.5 (maximum parsimony criterion; *Goloboff & Catalano, 2016*), using several rounds of equally weighted "New Technology Search" algorithms to adequately explore tree space and maximize the likelihood of finding the global optimum. Initial trees were calculated with "Sectorial Search", "Ratchet", "Drift" and "Tree Fusing" algorithms using 100 iterations each and 1,000 random addition sequences (RAS). Relative fit difference was set at 0.1 and up to 10 suboptimal trees were retained. The saved trees were subsequently put through two separate analyses of three rounds each. One analysis applying "Sectorial search", "Ratchet", and "Ratchet", in that order; and the other "Ratchet", "Sectorial search", and "Ratchet". All rounds ran 1,000 iterations and additionally included 1,000 iterations of "Tree fusing". At this stage suboptimal trees were discarded, and the strict consensus tree was calculated from the remaining trees. Branch support and stability were assessed with Bremer and Bootstrap support values, respectively. Bremer support values were calculated with the Bremer support function in TNT, using the absolute supports setting and TBR branch swapping on existing trees. Bootstrap support values were calculated using a "Traditional search" at 1,000 iterations.

We employed the iter PCR function (*Pol & Escapa, 2009*) implemented in TNT to identify the presence of topologically unstable OTUs in the four different analyses we performed. The incorporation of these so called "wild card" OTUs results in polytomies in the strict consensus tree (SCT) derived from the most parsimonious trees (MPTs). The iter PCR function iteratively removes the unstable OTUs from the analyses *a posteriori* to calculate a number of reduced strict consensus trees (RSCTs) that are sequentially more resolved. This way we were able to assess the phylogenetic relationships between more stable OTUs that were obscured by the incorporation of the unstable OTUs.

## Character sampling and formulation

Following detailed investigations of early archosaur phylogenetic relationships (e.g., *Nesbitt, 2011*), the phylogeny of non-archosaurian archosauromorphs has received much attention in recent years and several detailed character lists for this group exist, with one analysis focusing on tanystropheids (*Pritchard et al., 2015*) and another on allokotosaurs (*Nesbitt et al., 2015*). However, the most comprehensive analysis, consisting of 600 characters, was provided by *Ezcurra (2016)*, in which characters of these previous analyses were included, as well as those of many other studies. Furthermore, for many of the characters included at least one of the character states was figured, limiting subjective interpretation of the characters by the reader. Several subsequent studies have used and slightly modified the matrix provided by *Ezcurra (2016)* depending on the clade that was focused on in each respective study (e.g., *Butler et al., 2019*; *Ezcurra et al., 2019*; *Maidment et al., 2020*; *Sengupta, Ezcurra & Bandyopadhyay, 2017*; *Stocker et al., 2017*).

Due to its comprehensiveness and well-explained characters, the character list of *Ezcurra (2016)* was used as the main source for our characters. However, only those characters that were relevant to the sampled taxa were included, as many characters in the original list were used to differentiate between taxa not included herein, such as proterochampsids, archosaurs, and choristoderans. Additional characters were
incorporated mainly from the character list of *Pritchard et al. (2015)* and supplemented by characters from *Pritchard et al. (2018)*, *Nesbitt et al. (2015)*, *Nesbitt (2011)*, *Sookias (2016)*, *Simões et al. (2018)*, *Dilkes (1998)*, *Jalil (1997)*, *Benton & Allen (1997)*, and *Senter (2004)*. Certain characters taken from the literature were modified to fit more precisely with the specific morphologies observed in the sampled taxa. Finally, new characters were constructed based on detailed morphological comparisons of the included taxa. Certain autapomorphies for individual species were also incorporated into characters as they represent important morphological information and since these characters might prove phylogenetically relevant in future studies. New characters and characters that have been distinctly modified from previous analyses are discussed and figured.

All characters were critically assessed on their logical construction and whether character states represent valid tests of similarity or primary homology. Issues regarding character construction, specifically how to optimize the construction of characters as to represent similarity tests, are a continuing source of debate (e.g., *Brazeau, 2011*; *Kearney & Rieppel, 2006*; *Rieppel & Kearney, 2007*). Criteria for character construction to minimalize "non-meaningful" character scorings have been suggested by *Simões et al. (2016)* and examples of problematic characters have also been pointed out by *Nesbitt (2011)*. We assessed all our characters in light of these suggestions, as these criteria provide important considerations for character construction. However, we find that the application of each criterion is dependent on various factors. The taxa included in our analysis all represent Permo-Triassic non-archosaur members of the archosauromorph lineage, as well as early lepidosauromorphs and non-saurian diapsids. This phylogenetically relatively narrow sample is expected to exhibit less morphological variation than larger scale analyses (e.g., both extant and extinct Lepidosauromorpha and closely related taxa as in *Simões et al., 2018*). Therefore, in certain cases, characters that might otherwise not follow the criteria proposed in *Simões et al. (2016)* (e.g., the shape of the orbit, Type I A.7, or the use of continuous characters, Type II of *Simões et al., 2016*) are maintained when, based on detailed comparisons and careful consideration, the character states therein were deemed to likely represent valid similarity tests to the taxa involved. We pose that although these criteria as formulated represent useful tools, the complexity of morphological variation entails that careful observation and logical assessments of similarity by experts on the taxonomic sample at hand should be leading in character construction (*Kearney & Rieppel, 2006*; *Rieppel & Kearney, 2007*). Therefore, a character that might be problematic as a test of homology when applied to one set of taxa, might still be valid when looking at a different taxonomic sample. Following *Brazeau (2011)*, the presence or absence of a feature was formulated as a separate character from its morphology for unordered characters. In these cases, the character describing the morphology of this feature was scored as inapplicable in taxa in which the feature is absent.

## Character list

**Character 1 (***Ezcurra, 2016***: ch. 20).** *Rostrum, antorbital length (anterior tip of the skull to anterior margin of the orbit) versus total length of the skull: 0.32–0.40 (0); 0.43–0.62 (1), RATIO (***Ezcurra, 2016***: Figs. 17 and 18).*

This character is considered to be interdependent with character 76 of *Ezcurra (2016)* for the taxonomic sample of our analysis. Since this character 20 could be applied to more taxa than character 76, the former was preferred and the latter excluded.

**Character 2 (*Ezcurra, 2016*: ch. 21).** *Rostrum, dorsoventral height at the level of the anterior tip of the maxilla versus dorsoventral height at the level of the anterior border of the orbit: 0.20–0.27 (0); 0.32–0.48 (1); 0.56–0.78 (2), ORDERED RATIO (*Ezcurra, 2016*: Figs. 17 and 19).*

**Character 3 (*Ezcurra, 2016*: ch. 22).** *Rostrum, proportions at the level of the anterior border of the orbit: transversely broader than dorsoventrally tall or subequal (0); dorsoventrally taller than transversely broad (1) (*Ezcurra, 2016*: Fig. 16).*

**Character 4 (Modified from *Ezcurra, 2016*: ch. 27).** *Premaxilla, main body size: length of the tooth bearing margin in lateral view (in edentulous taxa the ventral margin of the premaxilla contributing to the ventral margin of the upper jaw; =main body) versus the length of the rostrum (anterior tip of the skull to the anterior border of the orbit): 0.09-0.10 (0); 0.13-0.20 (1); 0.23-0.38 (2); 0.45-0.54 (3), ORDERED RATIO (*Ezcurra, 2016*: Fig. 17).*

The original distinction between the character states was not considered to be phylogenetically relevant for the sampled taxa and therefore it was decided to distinguish states based on calculated ratios.

**Character 5 (*Ezcurra, 2016*: ch. 29).** *Premaxilla, downturned main body: absent, alveolar margin sub-parallel to the main axis of the maxilla (0); slightly, in which the alveolar margin is angled at approximately 20 degrees to the alveolar margin of the maxilla (1); strongly, prenarial process obscured by the postnarial process in lateral view (if the postnarial process is long enough) and postnarial process parallel or posteroventrally orientated with respect to the main axis of the premaxillary body (2), ORDERED (*Ezcurra, 2016*: Figs. 16–19).*

**Character 6 (*Ezcurra, 2016*: ch. 30).** *Premaxilla, angle formed between the alveolar margin and the anterior margin of the premaxillary body in lateral view: acute or right-angled (0); obtuse (1) (*Ezcurra, 2016*: Figs. 20 and 21). This character is inapplicable in taxa with a hooked premaxilla.*

**Character 7 (Modified from *Ezcurra, 2016*: ch. 34).** *Premaxilla, prenarial process: absent or incipient (0); present and less than the anteroposterior length of the main body of the premaxilla (1); present and longer than the anteroposterior length of the main body of the premaxilla (2), ORDERED (*Ezcurra, 2016*: Figs. 17 and 21). This character is scored as inapplicable in taxa with confluent external nares.*

An absent or incipient state was added and the character was ordered, since the character is considered a transformational series and state 1 represents a clear intermediate between states 0 and 2. Furthermore, the inapplicability criterion was included, since confluent external nares preclude the presence of a well-developed prenarial process.

**Character 8 (Modified from *Ezcurra, 2016*: ch. 35).** *Premaxilla, base of the prenarial process: anteroposteriorly shallow, being not much wider at its base than further distally on the process (0); anteroposteriorly deep, being much wider at its base than further distally on the process (1)* (*Ezcurra, 2016*: Figs. 12, 17, 20 and 21). *This character is inapplicable in taxa that lack a prenarial process.*

This character was further clarified in its description and the inapplicability criterion has been modified.

**Character 9 (Modified from *Ezcurra, 2016*: ch. 36 and 40).** *Premaxilla, postnarial process (=posterodorsal process, =maxillary process, =subnarial process): absent (0); short, ends well anterior to the posterior margin of the external naris (1); well-developed, forms most of the ventral border of the external naris or excludes the maxilla from participation in the external naris but process does not contact prefrontal (2); well-developed, forms most of the ventral border of the external naris and postnarial process of premaxilla contacts prefrontal (3), ORDERED* (*Ezcurra, 2016*: Figs. 17 and 19).

Characters 36 and 40 of *Ezcurra (2016)* were combined here because a contact between the premaxilla and the prefrontal always requires the premaxilla to exclude the maxilla from the external nares. Therefore, these conditions can be considered as part of the same transformational series.

**Character 10 (*Ezcurra, 2016*: ch. 37).** *Premaxilla, postnarial process (=posterodorsal process, =maxillary process, =subnarial process): wide, plate-like (0); thin (1). This character is not applicable to taxa that lack a postnarial process* (Ezcurra, 2016: Fig. 20).

**Character 11 (Modified from *Ezcurra, 2016*: ch. 41, and *Nesbitt et al. (2015)*: ch. 247).** *Premaxilla, plate-like palatal shelf or process on the medial surface (contribution to secondary palate by premaxillae): absent (0); present (1)* (*Ezcurra, 2016*: Figs. 12, 20 and 21)

The character was redescribed to indicate that the "process" referred to represents a rather wide shelf-like structure. For further explanation, see character 247 of *Nesbitt et al. (2015)*.

**Character 12 (Modified from *Pritchard et al., 2015*: ch. 1).** *Premaxilla, distinct posterodorsally to anteroventrally directed grooves terminating at the ventral margin of the bone: absent (0); present (1).*

This character describes the presence of posterodorsally to anteroventrally directed grooves present in *Langobardisaurus pandolfii* MFSN 1921 (*Saller, Renesto & Dalla Vecchia, 2013*) that were previously mistakenly identified as premaxillary teeth. This character was reformulated here to describe this feature more specifically.

**Character 13 (Modified from *Ezcurra, 2016*: ch. 42).** *Premaxilla, number of tooth positions: 8 or more (0); 5 or 6 (1); 4 (2); 3 (3); 2 (4); 1 or edentulous (5) ORDERED* (*Ezcurra, 2016*: Figs. 16 and 17).

The states of this character were modified since the original distinction did not cover all observed variation and because state 1 partially covered the same number of teeth as state 0 in the original description. Characters 69 and 278 of *Ezcurra (2016)* were omitted here, because we consider them to be strongly interdependent with this character for the current taxonomic sample.

**Character 14 (*Ezcurra, 2016*: ch. 43).** *Premaxilla, orientation of the tooth series or the occlusal surface of premaxilla in ventral view: approximately parasagittal (0); strongly transverse and (in case of tooth-bearing premaxillae) anterior teeth covering each other in lateral view (1)* (*Ezcurra, 2016*: Fig. 21). *This character is inapplicable in taxa with a hooked and beak-like premaxilla.*

The inapplicability criterion of this character was slightly modified. It was previously scored as inapplicable in taxa with an edentulous premaxilla. We consider that the differentiating morphology addressed by this character can also occur in taxa that lack premaxillary teeth. However, a hooked or beak-like shape morphology does not allow for a transverse occlusal surface and therefore taxa that exhibit this morphology should be scored as inapplicable for this character.

**Character 15 (*Ezcurra, 2016*: ch. 24).** *Premaxilla-maxilla, suture: simple continuous contact (0); notched along the ventral margin (1)* (*Ezcurra, 2016*: Figs. 17 and 19).

This character was also illustrated and discussed in Supplementary Figure 10 of *Pritchard et al. (2018)*.

**Character 16 (Modified from *Ezcurra, 2016*: ch. 25).** *Premaxilla-maxilla, subnarial foramen between the elements: absent (0); present (1)* (Nesbitt, 2011: Figs. 14, 17 and 19). *This character is inapplicable in taxa that have a ventral notch on the suture of the premaxilla and maxilla.*

The inapplicability criterion was included because the presence of a ventral notch on the border between the premaxilla and maxilla precludes the presence of a subnarial foramen. Character states 1 and 2 of character 25 of *Ezcurra (2016)* were fused here because the distinction between these two states is hard to establish confidently and is most likely irrelevant to the taxon sample included in this analysis.

**Character 17 (New, similar to *Ezcurra, 2016*: ch. 33, *Pritchard et al., 2015*: ch. 6, *Dilkes, 1998*: ch. 17, and *Pritchard et al., 2018*: ch. 6).** *Premaxilla-maxilla, contact between the premaxilla and maxilla: simple abutting contact in which the premaxilla might overlap the maxilla slightly laterally (0); overlapping contact in which the maxilla considerably overlaps the premaxilla laterally (1); contact in which the premaxilla has a posteriorly directed peg on its posterolateral margin articulating with the maxilla, often accompanied by a groove (2); complex connection in which the premaxilla has posteriorly projected peg on its medial surface which locks the maxilla against the premaxilla medially (3)* (Fig. 2).

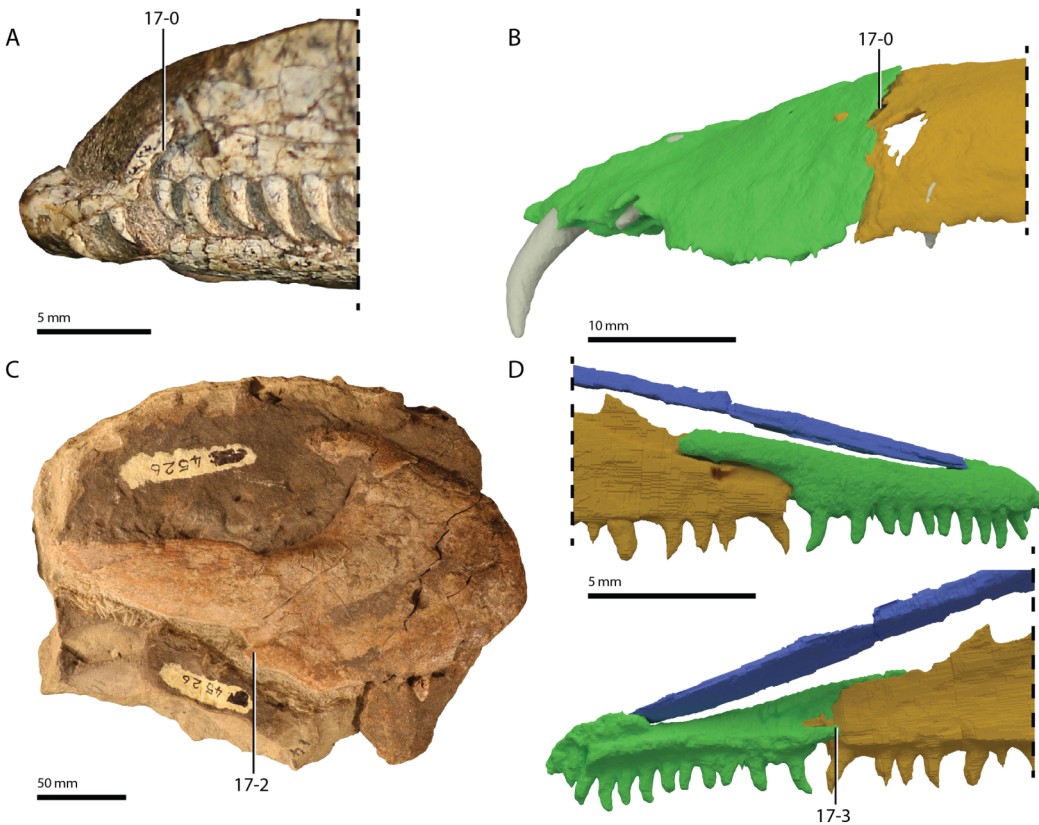

**Figure 2 Illustration of character 17.** (A) State 0 in *Prolacerta broomi* (BP/1/5880, anterior snout in left lateral view). (B) State 1 in a digital reconstruction of *Tanystropheus hydroides* (PIMUZ T 2790, anterior snout in left lateral view). (C) State 3 in *Erythrosuchus africanus* (BP/1/4526, isolated right premaxilla in lateral view). (D) State 4 in a digital reconstruction of *Macrocnemus bassanii* (PIMUZ T 2477, anterior snout in right lateral view above and medial view below).

The connection between the premaxilla and maxilla in early archosauromorphs has been discussed in depth and is considered phylogenetically informative. We have reformulated this character based on recent new findings with regards to the articulation between these elements in *Macrocnemus bassanii* (state 3; *Miedema et al., 2020*). Among the taxa sampled here, most have a simple abutting contact between premaxilla and maxilla, in which in many cases the premaxilla slightly overlaps the maxilla laterally (state 0; the sampled non-archosauromorph diapsids, *Czatkowiella harae*, *Tanystropheus hydroides*, *Prolacerta broomi*, *Trilophosaurus buettneri*, *Teyujagua paradoxa*, *Euparkeria capensis*, and *Proterosuchus fergusi*). In the sampled rhynchosaurs, the maxilla broadly overlaps the premaxilla laterally (state 1; *Mesosuchus browni*). Certain taxa bear a small peg, meaning a short pin or bolt, on the posterolateral end of their premaxilla, which connects to the lateral surface of the maxilla (state 2; *Azendohsaurus madagaskarensis* and *Erythrosuchus africanus*). The configuration of *Macrocnemus bassanii*, in which a peg is present on the medial side of the premaxilla that interlocks with an anteriorly facing peg on the medial side of the maxilla (*Miedema et al., 2020*), is considered to be morphologically distinct and almost certainly non-homologous to the pegs on the

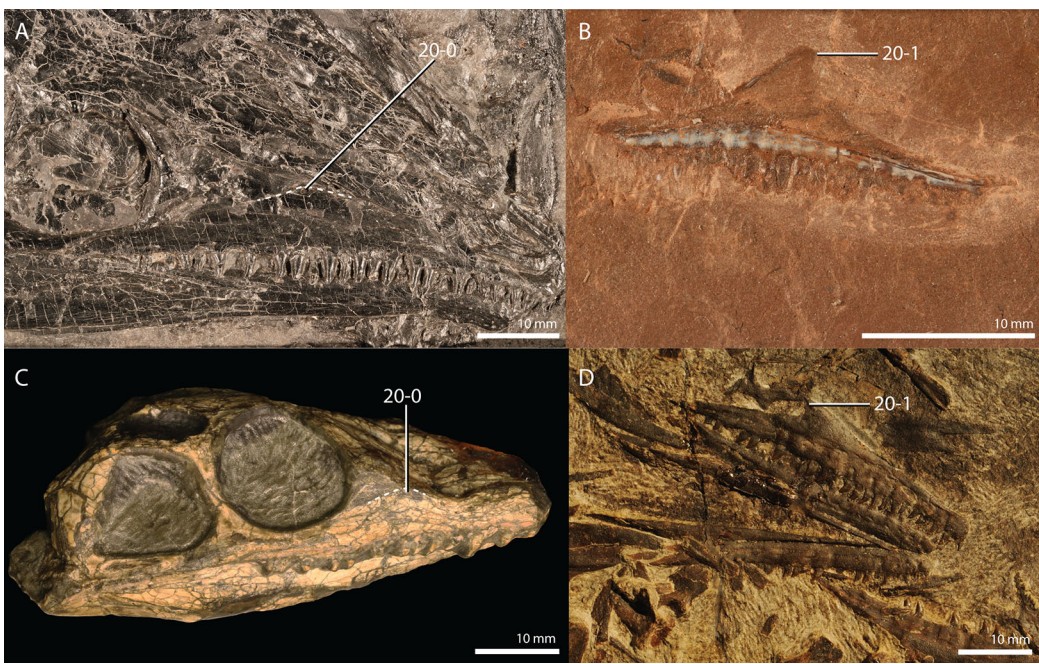

**Figure 3 Illustration of character 20.** (A) State 0 in *Protorosaurus speneri* (NMK S 180, anterior part of the skull in right laterodorsal view). (B) State 1 in *Amotosaurus rotfeldensis* (SMNS 50830, right maxilla in medial view). (C) State 0 in *Youngina capensis* (AMNH FARB 5561, skull in right lateral view). (D) State 1 in *Macrocnemus fuyuanensis* (PIMUZ T 1559, disarticulated skull with right maxilla visible in lateral view). The stippled line indicates the transition from the dorsal to the posterior margin of the maxilla in (A) and (B).

posterolateral end of the premaxilla described for state 2 and is therefore scored as a separate state here.

**Character 18 (*Ezcurra, 2016*: ch. 45).** *Septomaxilla: present (0); absent (1)* (*Ezcurra, 2016*: Fig. 16).

The presence of this small element in the anterior region of the rostrum is hard to establish, and we were not able to confidently consider it as absent for any taxon. Therefore, this character is not phylogenetically informative for the present analysis, but it is nevertheless maintained as it presents an overview of the presence of the septomaxilla among early archosauromorphs. The elements identified as the anterodorsal portion of the vomers in specimen SAM-PK-6536 of the rhynchosaur *Mesosuchus browni* by *Dilkes (1998)* can confidently be interpreted as the septomaxillae and are scored as such here.

**Character 19 (*Ezcurra, 2016*: ch. 52 and *Nesbitt et al., 2015*: ch. 203).** *Maxilla, anterior maxillary foramen: absent (0); present (1)* (*Ezcurra, 2016*: Fig. 17).

**Character 20 (Modified from *Nesbitt et al., 2015*: ch. 202.** *Maxilla, dorsal portion, shape: gradual transition between the dorsal and posterior margin of the maxilla and no distinct process is formed (0); the dorsal apex of the maxilla ends abruptly and its posterior margin is concave (1)* (Fig. 3).

The description of state 0 was modified to more precisely describe the condition observed in *Youngina capensis* and *Protorosaurus speneri*. The character as described by *Nesbitt et al. (2015)* was preferred over character 58 of *Ezcurra (2016)*, since state 2 of the latter is likely strongly interdependent with the presence of an antorbital fenestra, which is scored here already in a separate character (22). In the description of character 202 of *Nesbitt et al. (2015)* it was pointed out that in *Azendohsaurus madagaskarensis* and *Azendohsaurus laaroussii* the posterodorsal margin of the maxilla is concave, which is similar to the condition in Archosauriformes in which this margin forms the anterior margin of the antorbital fenestra. A curved posterodorsal margin of the maxilla occurrs in many non-archosauriform archosauromorphs (see also character 58 of *Ezcurra, 2016* and its scoring). The presence of a distinct ascending process of the maxilla, considered to be a saurian trait (*sensu* the scoring of character 57 in *Ezcurra, 2016*), is considered to be too ambiguous as a phylogenetic character in the sampled taxa. For instance, the maxillae of *Youngina capensis* (AMNH FARB 5561), *Protorosaurus speneri* (NMK S 180), and *Prolacerta broomi* (BP/1/5880) do not bear a clearly defined process and the maxillae are approximately equally tall relative to their respective rostra in all three taxa. Nevertheless, the latter two taxa were previously considered to bear an ascending process, in contrast to the non-saurian diapsid *Youngina capensis*. Furthermore, the presence of a process-like dorsal portion of the maxilla is strongly dependent on the relative height of the anterior portion of the rostrum at the level of the anterior margin of the orbit, and this morphology is already considered by character 10 here.

**Character 21 (Modified from *Ezcurra, 2016*: ch. 59).** *Maxilla, anterior part of the dorsal margin: convex (0); straight (1); concave (2)* (*Ezcurra, 2016*: Fig. 22).

The anterior part of the dorsal margin is convex in *Prolacerta broomi* (BP/1/5880; *Spiekman, 2018*), *Trilophosaurus buettneri* (TMM 31025-207), *Protorosaurus speneri* (NMK S 180), *Mesosuchus browni* (SAM-PK-6536), *Youngina capensis* (SAM-PK-K7578), *Orovenator mayorum* (OMNH 74606), *Petrolacosaurus kansensis* (KUVP 9951), and *Gephyrosaurus bridensis* (*Evans, 1980*). It is completely straight in *Macrocnemus bassanii* (PIMUZ T 4822) and *Proterosuchus fergusi* (RC 846; *Ezcurra & Butler, 2015b*), which is here incorporated as a separate state, as it is considered homologous to neither the concave nor convex state, but instead represents an intermediate condition.

**Character 22 (*Ezcurra, 2016*: ch. 13, and *Pritchard et al., 2015*: ch. 13).** *Antorbital fenestra: absent (0); present (1).*

**Character 23** New. *Maxilla, maxillary fossa (="antorbital fossa" sensu Rieppel, Li & Fraser, 2008): absent (0); or present (1)* (Fig. 4). *This character is inapplicable in taxa that have an antorbital fenestra.*

A large circular concavity is present on the lateral margin of the maxilla of *Dinocephalosaurus orientalis* (IVPP V13767), which was referred to as an antorbital fossa by *Rieppel, Li & Fraser (2008)*. This character is clearly different from the antorbital fossa as described for *Erythrosuchus* (*Gower, 2003*), which refers to a depression in the rostrum

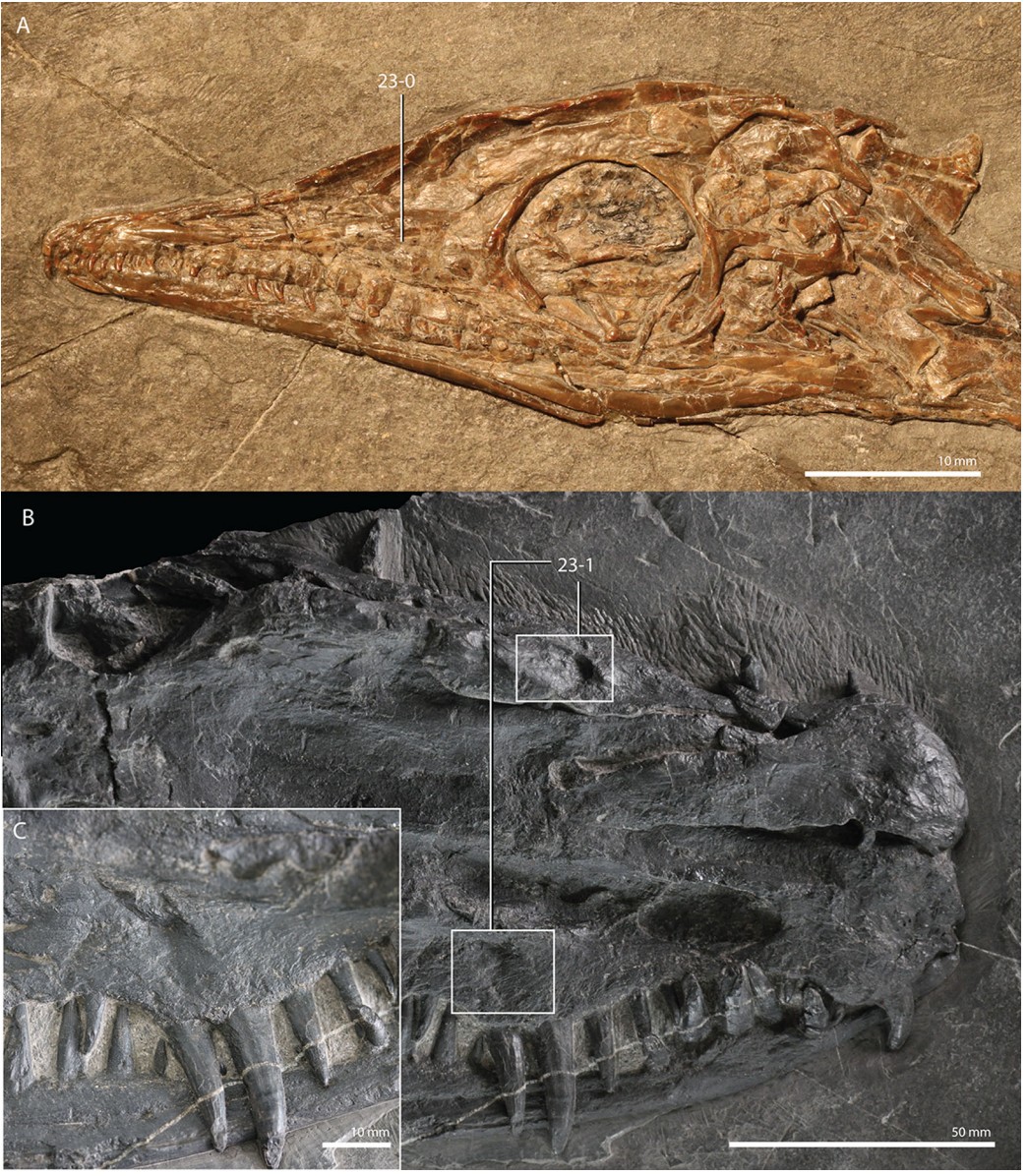

**Figure 4** **Illustration of character 23.** (A) State 0 in *Macrocnemus bassanii* (PIMUZ T 4822, skull in left lateral view). (B) State 1 in *Dinocephalosaurus orientalis* (IVPP V13767, anterior part of the skull in right laterodorsal view). (C) Close up of the maxillary fossa on the right maxilla of *Dinocephalosaurus orientalis* (IVPP V13767).

within which the antorbital fenestra is located. Because of the clear affinities with the antorbital fenestra, this antorbital fossa is considered to be non-homologous to the fossa described here, which we refer to as the maxillary fossa. It is currently only known to be present in *Dinocephalosaurus orientalis* among the sampled taxa.

**Character 24 (*Ezcurra, 2016*: ch. 64).** *Maxilla, posterior end of the horizontal process distinctly ventrally deflected from the main axis of the alveolar margin: absent (0); present (1)* (*Ezcurra, 2016*: Figs. 17 and 22).

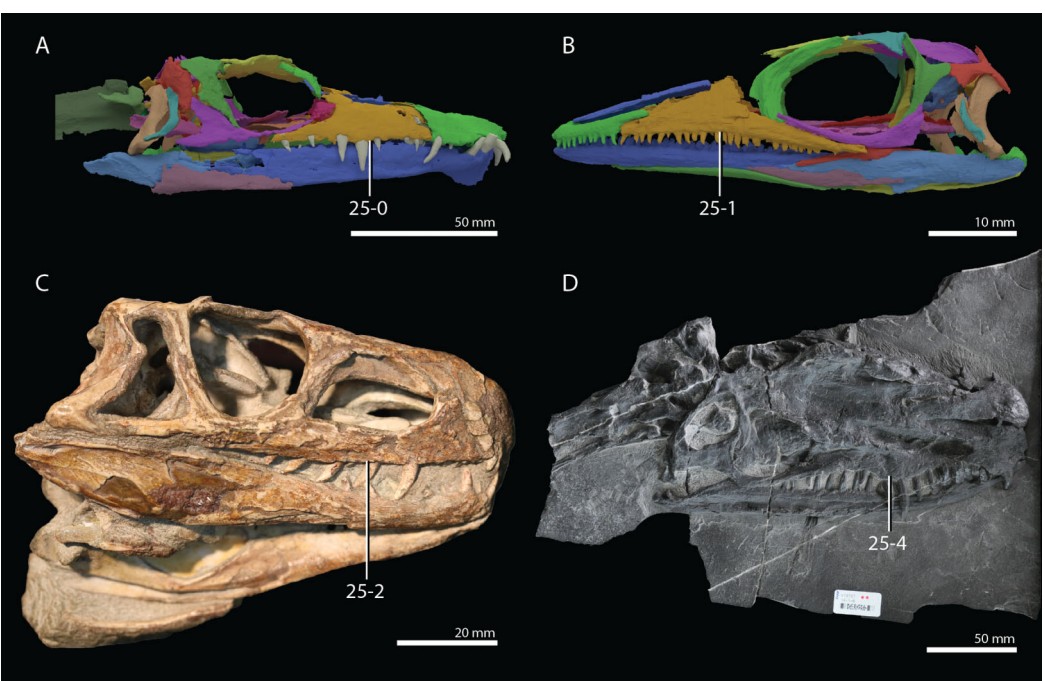

**Figure 5** **Illustration of character 25.** (A) State 0 in a digital reconstruction of *Tanystropheus hydroides* (PIMUZ T 2790, skull in right lateral view). (**B**) State 1 in a digital reconstruction of *Macrocnemus bassanii* (PIMUZ T 2477, skull in left lateral view). (C) State 2 in *Euparkeria capensis* (SAM-PK-5867, skull in right lateral view). (D) State 4 in *Dinocephalosaurus orientalis* (IVPP V13767, skull in right dorsolateral view).

**Character 25 (Modified from *Ezcurra, 2016*: ch. 68)**. *Maxilla, alveolar margin in lateral view: straight (0); concave (1); convex (2); sigmoid, anteriorly concave and posteriorly convex (3); sigmoid, anteriorly convex, starting close to mid-length, and posteriorly concave (4)* (Fig. 5; *Ezcurra, 2016*: Figs. 16 and 19).

This character was modified to distinguish between a concave, straight, or convex margin, since it was considered that these distinctions might be phylogenetically informative for this sample of taxa.

**Character 26 (Modified from *Ezcurra, 2016*: ch. 75)**. *Maxilla, number of tooth positions: 11–17 (0); 19–40 (1). This character is inapplicable in taxa with multiple tooth rows in the maxilla.*

Character states were defined after determining the tooth count in all included taxa.

**Character 27 (*Ezcurra, 2016*: ch. 47)**. *Maxilla-jugal, anguli oris crest: absent (0); present (1)* (*Ezcurra, 2016*: Fig. 16).

See the comments for character 28.

**Character 28 (New, combination of *Ezcurra, 2016*: ch. 47, and part of *Pritchard et al., 2015*: ch. 8)**. *Maxilla-jugal, anguli oris crest: both the jugal and the maxilla are distinctly*

*laterally offset (0); only the jugal is distinctly laterally offset (1) (Ezcurra, 2016*: Fig. 16*). This character is scored as inapplicable in taxa that lack an anguli oris crest.*

The term anguli oris crest is typically used to describe the very conspicuous lateral offset of the jugal seen in rhynchosaurid rhynchosaurs (e.g., *Butler et al., 2015*; *Langer & Schultz, 2000*; *Montefeltro, Langer & Schultz, 2010*). This crest might have facilitated a muscular cheek (*Benton, 1983*). A much less conspicuous anguli oris crest, which is partially formed by the posterolateral end of the maxilla, was recently described for the non-rhynchosaurid rhynchosaur *Eohyosaurus wolvaardti* (*Butler et al., 2015*). A similar lateral offset of the maxilla as seen in this taxon, which creates a substantial space between the lateral margin of the crest and the posterior portion of the maxillary tooth row, was considered for the trilophosaurid allokotosaurs *Trilophosaurus buettneri* and *Teraterpeton hrynewichorum* in character 8 of *Pritchard et al. (2015)*. We consider the description of this character to address the same morphological structure as seen in *Eohyosaurus wolvaardti* and therefore fused the characters. A similar lateral offset might also be present in *Langobardisaurus pandolfii*. However, scoring this character for *Langobardisaurus pandolfii* is currently ambiguous since the preservation of this region is poor in the only specimen in which it is visible (MFSN 1921).

**Character 29 (*Ezcurra, 2016*: ch. 9).** *External nares, confluent: absent (0); present (1)* (*Ezcurra, 2016*: Fig. 16, 17 and 20).

**Character 30 (*Ezcurra, 2016*: ch. 12).** *External naris, shape: sub-circular (0); oval (1)* (*Ezcurra, 2016*: Fig. 19). *This character is inapplicable in taxa with confluent external nares.*

An inapplicability criterion was added to this character.

**Character 31 (New, similar to *Ezcurra, 2016*: ch. 10).** *External naris: located close to the anterior end of the skull (0); a thick anterior margin of the premaxilla results in the external nares being posteriorly displaced (1)* (Fig. 6).

In most taxa scored, the anterior margin of the external naris is positioned near towards the anterior end of the rostrum. However, in the tanystropheids *Tanystropheus hydroides* (PIMUZ T 2790), *Tanystropheus longobardicus* (MSNM BES SC 1018), *Macrocnemus bassanii* (PIMUZ T 2477), and in *Pectodens zhenyuensis* (IVPP V18578) and *Dinocephalosaurus orientalis* (IVPP V13767) the anterior margin of the external naris is separated considerably from the anterior end of the rostrum by the main body of the premaxilla.

**Character 32 (Modified from *Ezcurra, 2016*: ch. 78).** *Nasal, shape of anterior margin at midline: strongly convex with anterior process, and nasal forming a partial internarial bar (0); transverse with little convexity (1)* (*Ezcurra, 2016*: Fig. 16). *This character is inapplicable in taxa in which the external nares are completely separated by an internarial bar.*

This character is scored as inapplicable in taxa with a complete internarial bar, since this is always formed, at least in part, by an anterior process of the nasal. Thus, scoring taxa with a
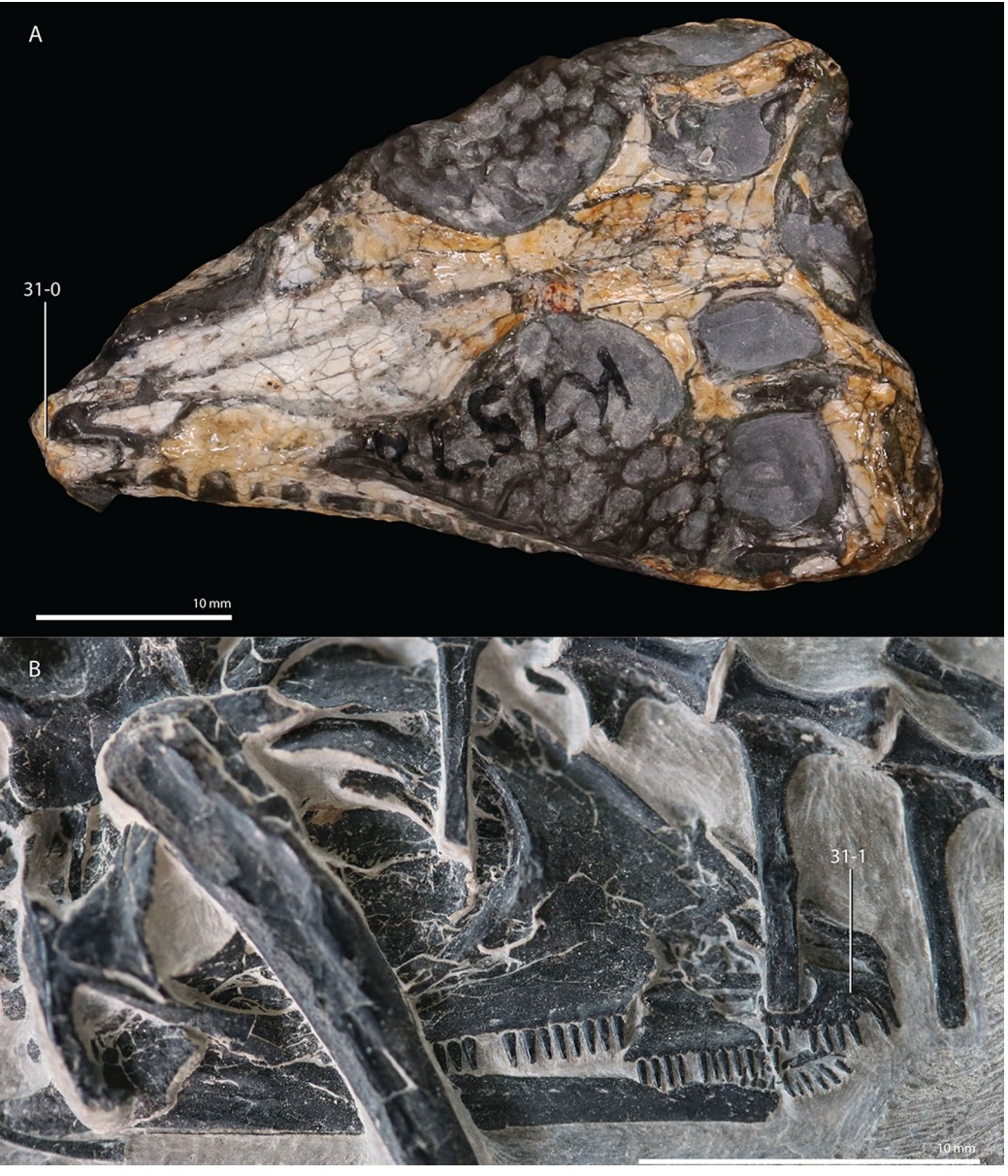

**Figure 6 Illustration of character 31.** (A) State 0 in *Youngina capensis* (SAM-PK-K7578, skull in dorsal view). (B) State 1 in *Pectodens zhenyuensis* (IVPP V18578, skull in right lateral view).

complete internarial bar for this character would result in overscoring this trait, as it is already addressed in character 5. Among the sampled taxa with confluent external nares, only the nasals of *Azendohsaurus madagaskarensis* bear clear anterior processes on the anteromedial margin of the nasal.

**Character 33 (New).** *Nasal, antorbital recess: absent (0); or present (1)* (Fig. 7).

The antorbital recess was first described for *Dinocephalosaurus orientalis* by *Rieppel, Li & Fraser (2008)*. It is a large gully posterior to the external naris that is largely formed by the

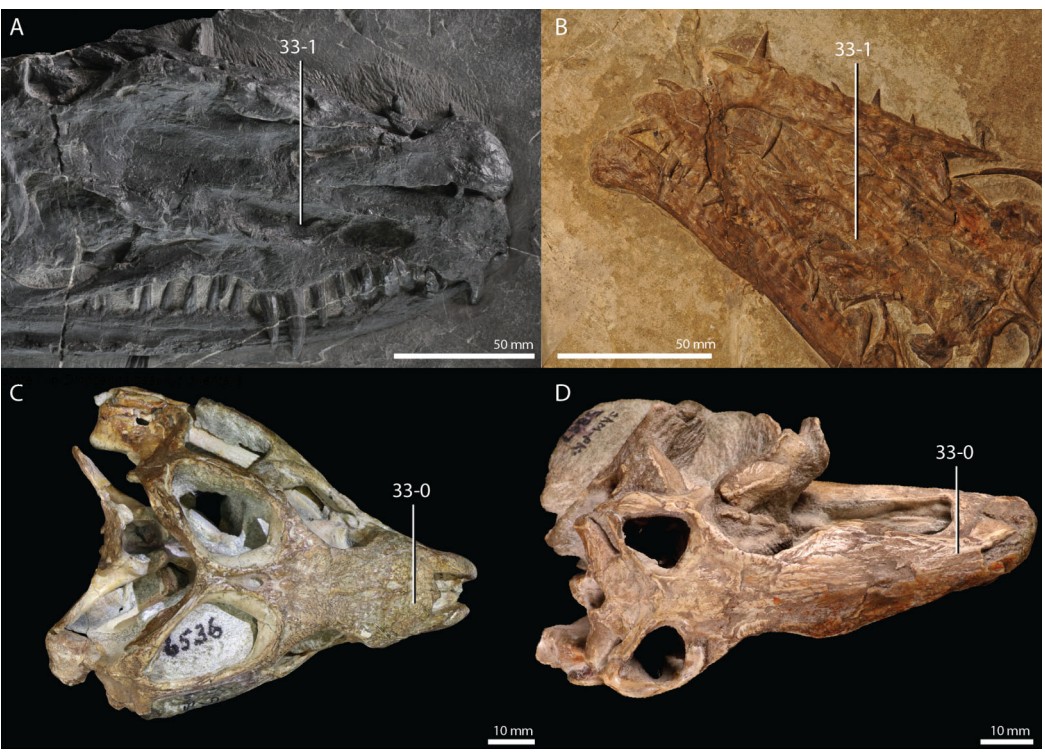

**Figure 7** **Illustration of character 33.** (A) State 1 in *Dinocephalosaurus orientalis* (IVPP V13767, anterior part of the skull in right dorsolateral view). (B) State 1 in *Tanystropheus hydroides* (PIMUZ T 2819, anterior part of the skull in dorsal view). (C) State 0 in *Mesosuchus browni* (SAM-PK-6536, skull in dorsal view). (D) State 0 in *Euparkeria capensis* (SAM-PK-5867, skull in dorsal view).

nasal bone, but the maxilla and possibly the prefrontal also contribute to it. This recess is non-homologous to the depression of the nasal described by character 80 of *Ezcurra (2016)*. The antorbital recess has recently also been identified in *Tanystropheus hydroides* (*Jiang et al., 2011*; *Spiekman et al., 2020a*, *2020b*).

**Character 34 (*Pritchard et al., 2018*: ch. 311).** *Nasal, lateral surface: meets dorsoventrally short length of medial surface of dorsal process/portion of the maxilla (0); meets entire dorsoventral height of medial surface of supra-alveolar portion of maxilla (1). This character is inapplicable in taxa with an antorbital fenestra.*

See character description of character 311 of *Pritchard et al. (2018)*. The inapplicability criterion was added because we consider that the presence of an antorbital fenestra implies that the nasal cannot have a wide articulation facet with the maxilla.

**Character 35 (*Dilkes, 1998*: ch. 15).** *Lacrimal, contacts nasal and reaches external naris (0); contacts nasal but does not reach naris (1); or does not contact nasal or reach naris (2), ORDERED. This character is inapplicable in taxa in which the premaxilla contacts the prefrontal.*
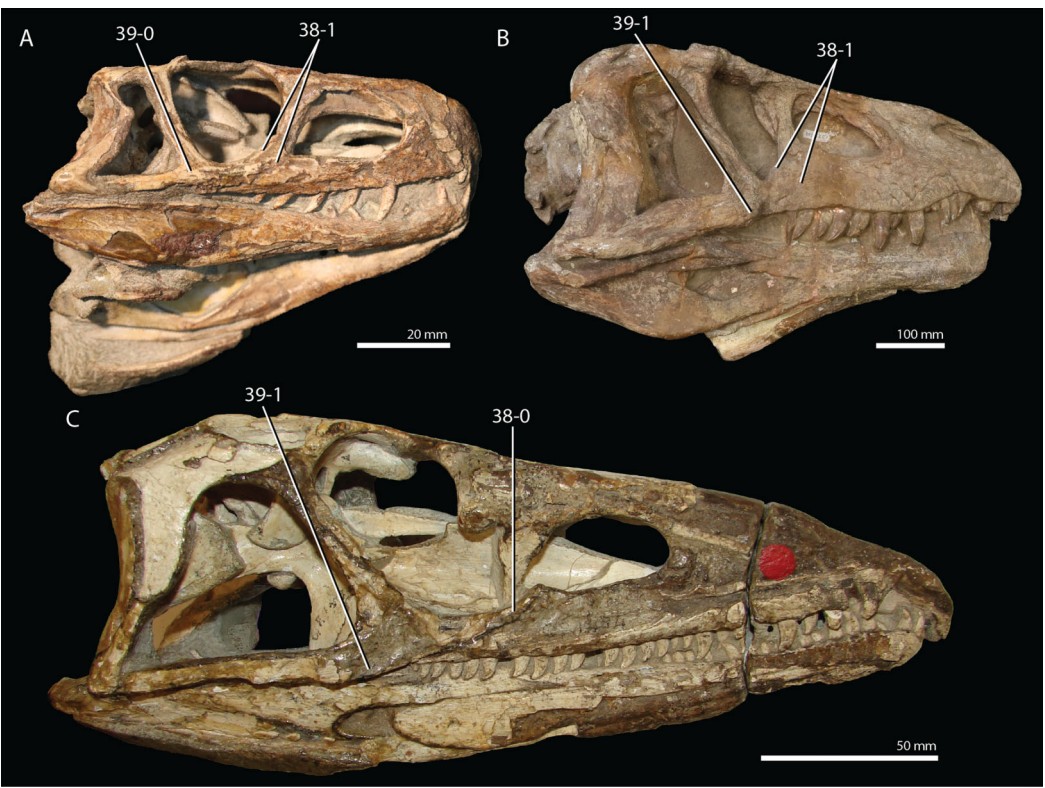

**Figure 8 Illustration of characters 38 and 39.** (A) 38-1 and 39-0 in *Euparkeria capensis* (SAM-PK-5867, skull in right lateral view). (B) 38-1 and 39-1 in *Erythrosuchus africanus* (BP/1/5207, skull in right lateral view). (C) 38-0 and 39-1 in *Proterosuchus alexanderi* (NM QR 1484, skull in right lateral view). Image of *Proterosuchus alexanderi* courtesy of Martín Ezcurra.

This character was ordered because it is considered to represent a transformational series and state 1 represents an intermediate between states 0 and 2. Furthermore, the inapplicability criterion was included, because a contact between the premaxilla and prefrontal precludes the possibility of a contact between the lacrimal and nasal.

**Character 36 (*Ezcurra, 2016*: ch. 90).** *Lacrimal, naso-lacrimal duct position: opens on the posterolateral edge of the lacrimal (0); opens on the posterior surface of the lacrimal (1)* (*Ezcurra, 2016*: Fig. 19). *This character is inapplicable if the prefrontal encloses part of the naso-lacrimal duct.*

**Character 37 (*Ezcurra, 2016*: ch. 95).** *Jugal, anterior extension of the anterior process: anterior to the level of mid-length of the orbit (0); up to or posterior to the level of mid-length of the orbit (1).*

**Character 38 (Modified from *Ezcurra, 2016*: ch. 92).** *Jugal, anterior process is dorsoventrally expanded anteriorly: absent, the anterior process tapers anteriorly and articulates with the dorsal surface of the posterior process of the maxilla (0); present, the anterior process of the jugal is expanded and partially covers the lateral surface of the posterior process of the maxilla (1)* (Fig. 8).

In the majority of the taxa sampled here, the anterior process of the jugal fits into a groove or slot on the dorsal surface of the posterior process of the maxilla and is in some cases partially covered by the maxilla in lateral view. In the archosauriforms *Euparkeria capensis* (SAM-PK-5867) and *Erythrosuchus africanus* (BP/I/5207) the anterior end of the jugal is distinctly dorsoventrally taller and partially overlaps the maxilla in lateral view.

**Character 39 (Modified from *Sookias, 2016*: ch. 81).** *Jugal, bulges ventrolaterally at the point where its three processes meet: absent (0); present (1)* (Fig. 8). *This character is scored as inapplicable in taxa that lack a posterior process of the jugal.*

Character state 1 describes the condition in the archosauriforms *Proterosuchus fergusi* (SAM-PK-11208), *Erythrosuchus africanus* (BP/I/5207), and *Teyujagua paradoxa* (*Pinheiro, Simão-Oliveira & Butler, 2019*). In these taxa the posterior process of the jugal is positioned further laterally than its anterior process. This is caused by a lateral bulging of the jugal at the point where the anterior, posterior, and ascending/dorsal processes of the bone meet. This morphology is clearly distinct from the anteromedially to posterolaterally directed crest described as the anguli oris crest and therefore coded as a separate character.

**Character 40 (*Ezcurra, 2016*: ch. 98).** *Jugal, multiple pits on the lateral surface of the main body: absent (0); present (1)* (*Ezcurra, 2016*: Fig. 17).

**Character 41 (*Ezcurra, 2016*: ch. 99).** *Jugal, ascending process forming the entire anterior border of the infratemporal fenestra: absent (0); present, postorbital excluded from the anterior border of the infratemporal fenestra (1)* (*Ezcurra, 2016*: Fig. 17). *This character is inapplicable in taxa in which the anterior process of the squamosal possesses an extensive contact with the postorbital and contacts the jugal, and in taxa that lack an infratemporal fenestra or an ascending process on the jugal.*

**Character 42 (New).** *Jugal, posterior process: present (0); absent (1)* (Fig. 9).

A posterior process, typically present in archosauromorphs, is completely absent in *Claudiosaurus germaini* (*Carroll, 1981*), *Pectodens zhenyuensis* (IVPP V18578), *Dinocephalosaurus orientalis* (IVPP V13767), and *Trilophosaurus buettneri* (*Spielmann et al., 2008*).

**Character 43 (*Ezcurra, 2016*: ch. 100).** *Jugal, length of the posterior process versus the height of its base: 0.62-2.28 (0); 2.64-3.64 (1); 4.48-4.74 (2); 5.29-5.84 (3), ORDERED RATIO* (Fig. 9; *Ezcurra, 2016*: Figs. 17 and 19).

*This character is inapplicable in taxa that lack a posterior process of the jugal.* This character and character 42 (presence or absence of posterior process of the jugal) could also be treated as a single character. However, these characters were separated here because we omitted the ratio-based characters from some of our analyses. If characters 42 and 43 were combined the presence or absence of the posterior process of the jugal would be omitted from these analyses, despite representing a discrete rather than a continuous distinction.

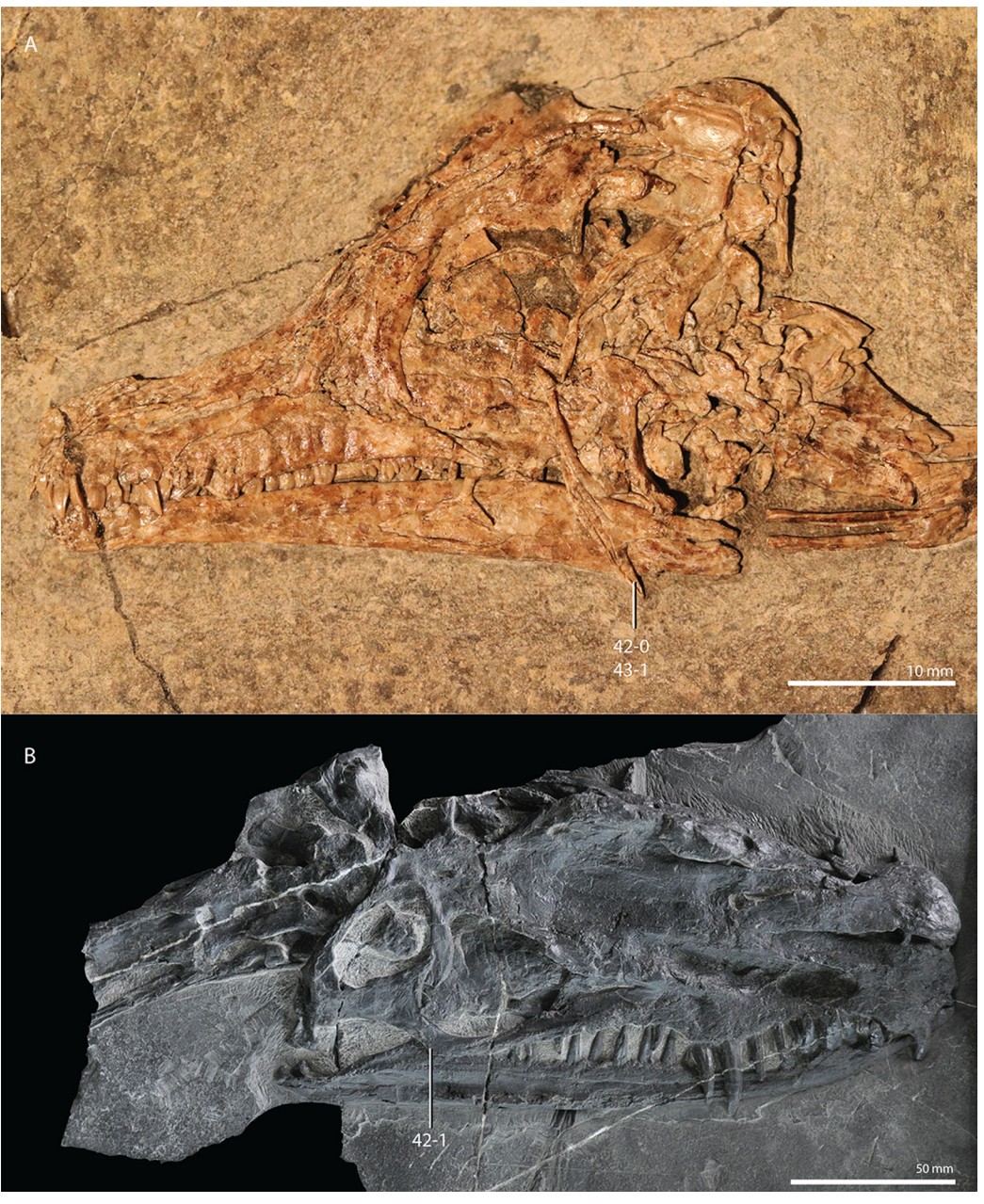

**Figure 9 Illustration of characters 42 and 43.** (A) 42-0 and 43-1 in *Tanystropheus longobardicus* (PIMUZ T 3901, skull in left lateral view). (B) 42-1 in *Dinocephalosaurus orientalis* (IVPP V13767, skull in right laterodorsal view).

**Character 44 (Modified from *Ezcurra, 2016*: ch. 5).** *Skull, dermal sculpturing on the dorsal surface of the frontals, parietals, and nasals: absent (0); shallow or deep pits scattered across surface and/or low ridges (1)* (*Ezcurra, 2016*: Fig. 16).

State 2 of character 5 in *Ezcurra (2016)* was not included here, because it was not applicable to the taxa sampled.

**Character 45 (***Ezcurra, 2016***: ch. 109).** *Prefrontal, subtriangular medial process: absent, nasal-frontal suture transversely broad (0); present, nasal-frontal suture strongly transversely reduced (1) (*Ezcurra, 2016*: Fig. 7).*

**Character 46 (***Ezcurra, 2016***: ch. 111 and** *Nesbitt et al., 2015***: ch. 237).** *Prefrontal, lateral surface of the orbital margin: smooth or slight grooves present (0); rugose sculpturing present (1) (*Ezcurra, 2016*: Fig. 17).*

**Character 47 (Modified from** *Ezcurra, 2016***: ch. 16).** *Orbit, shape: subcircular (0); distinctly dorsoventrally taller than long (1).*

Within the current taxonomic sample states 0 and 1 of character 16 in *Ezcurra (2016)* represent a relatively minor morphological difference of which the accurate observation is easily hampered by compression of specimens. Therefore, this character was modified to distinguish between the roughly subcircular orbits present in most of the sampled taxa, and the very dorsoventrally tall orbits of *Proterosuchus fergusi* (SAM-PK-11208), *Proterosuchus alexanderi* (NM QR 1484), and *Erythrosuchus africanus* (BP/1/5207).

**Character 48 (Modified from** *Ezcurra, 2016***: ch. 17).** *Orbit, elevated rim: absent or incipient (0); present, orbital margin of the jugal and/or postorbital slightly elevated to form a rim (1) (*Ezcurra, 2016*: Figs. 16 and 17).*

State 2 of character 17 in *Ezcurra (2016)* was excluded because it was not applicable to the sampled taxa.

**Character 49 (Modified from** *Ezcurra, 2016***: ch. 113).** *Frontal, suture with the nasal: transverse (0); oblique, forming an angle of at least 60 degrees with the long axis of the skull and frontals entering between both nasals (1); oblique and nasals entering considerably between frontals in a non-interdigitate suture (2); frontals enter nasals medially and nasals enter frontals laterally creating a W-shaped suture (3); frontals possess a three-pronged anteromedial process that articulates with the nasals (4) (*Fig. 10*; *Ezcurra, 2016*: Fig. 23 and* Nesbitt, 2011*: Fig. 18). This character is inapplicable if the nasal is received by a slot in the frontal or the nasal does not contact the frontal.*

Character states 3 and 4 are new. State 3 is exhibited by *Youngina capensis* (BP/1/3859) and *Tanystropheus hydroides* (PIMUZ T 2787) and state 4 by *Tanystropheus longobardicus* (PIMUZ T 2484).

**Character 50 (***Ezcurra, 2016***: ch. 114)** (the character formulation has been slightly modified). *Frontal, orbital border in skeletally mature individuals: absent or anteroposteriorly short and forms less than half of the dorsal edge of the orbit (0); anteroposteriorly long and forms at least more than half of the dorsal edge of the orbit (1) (*Ezcurra, 2016*: Fig. 23).*

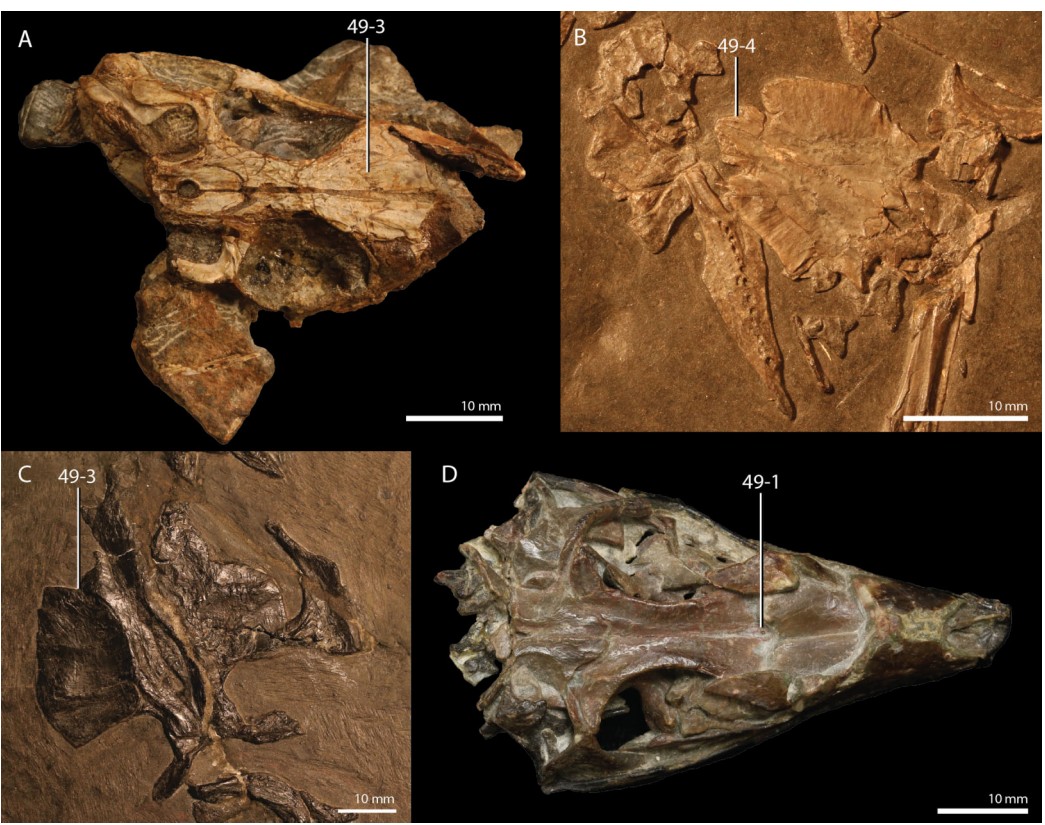

**Figure 10 Illustration of character 49.** (A) State 3 in *Youngina capensis* (BP/1/3859, partial skull in dorsal view). (B) State 4 in *Tanystropheus longobardicus* (PIMUZ T 2484, frontal, parietal, and postfrontal in dorsal view). (C) State 3 in *Tanystropheus hydroides* (PIMUZ T 2787, frontal and parietal in ventral view). (D) State 1 in *Prolacera broomi* (UCMP 37151, skull in dorsal view).

**Character 51 (*Ezcurra, 2016*: ch. 118).** *Frontal, dorsal surface adjacent to sutures with the postfrontal (if present) and parietal: flat to slightly concave (0); possesses a longitudinal and deep depression (1)* (*Ezcurra, 2016*: Fig. 16).

**Character 52 (*Ezcurra, 2016*: ch. 119).** *Frontal, longitudinal groove: longitudinally extended along most of the surface of the frontal (0); anterolaterally to posteromedially extended along the posterior half of the frontal (1)* (*Ezcurra, 2016*: Fig. 16). *This character is inapplicable in taxa that lack a longitudinal depression on the frontal.*

The inapplicability criterion has been slightly modified.

**Character 53 (*Ezcurra, 2016*: ch. 121).** *Frontal, olfactory tract on the ventral surface of the frontal: maximum transverse constriction point well posterior to the moulds of the olfactory bulbs and posterolateral margin of the bulbs delimited by a low ridge (0); maximum transverse constriction of the olfactory tract immediately posterior to the moulds of the olfactory bulbs and posterolateral margin of the bulbs well-delimited by a thick, tall ridge (1)* (*Ezcurra, 2016*: Fig. 23). *This character is inapplicable in taxa that lack olfactory bulb moulds and constriction of the olfactory tract canal.*

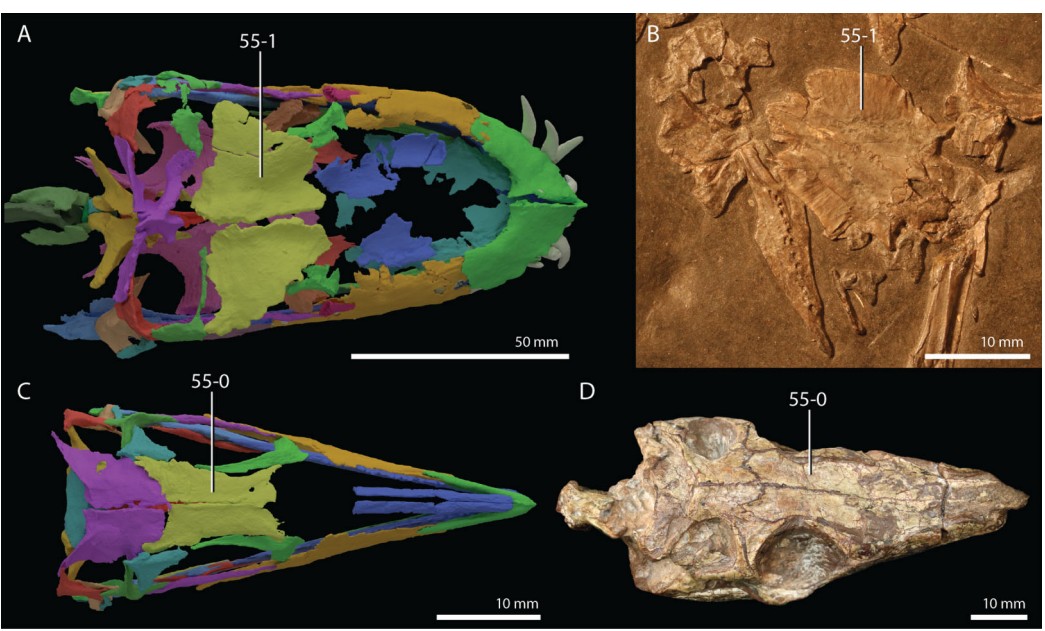

**Figure 11** **Illustration of character 55.** (A) State 1 in a digital reconstruction of *Tanystropheus hydroides* (PIMUZ T 2790, skull in dorsal view). (B) State 1 in *Tanystropheus longobardicus* (PIMUZ T 2484, frontal, parietal, and postfrontal in dorsal view). (C) State 0 in a digital reconstruction of *Macrocnemus bassanii* (PIMUZ T 2477, skull in dorsal view). (D) State 0 in *Prolacerta broomi* (BP/1/471, skull in dorsal view).

**Character 54 (*Ezcurra, 2016*: ch. 112 and *Pritchard et al., 2015*: ch. 14).** *Frontal, frontals fused to one another: absent (0); present (1)* (*Ezcurra, 2016*: Fig. 23).

State 0 can only be scored based on skeletally mature specimens.

**Character 55 (New).** *Frontal, width (or width of half of a fused frontal in taxa with fused frontals): narrow, frontal is considerably longer than wide (0); very wide and plate-like, frontal is almost as wide as long (1)* (Fig. 11).

This character describes the very wide frontals seen in *Tanystropheus hydroides* (PIMUZ T 2790) and *Tanystropheus longobardicus* (MSNM BES SC 1018).

**Character 56 (*Pritchard et al., 2015*: ch. 16).** *Frontal, shape of contact with parietal in dorsal view: roughly transverse in orientation (0); frontals exhibit posterolateral processes, forming anteriorly curved U-shaped contact with parietals (1)* (*Ezcurra, 2016*: Figs. 8 and 23).

This character is similar to character 116 in *Ezcurra (2016)*. However, the version of this character formulated by *Pritchard et al. (2015)* is preferred because it is more specific to the taxon sample studied here.

**Character 57 (New, combining information from *Ezcurra, 2016*: ch. 122, *Pritchard et al., 2015*: ch. 15, and *Pritchard et al., 2018*: ch. 313).** *Postfrontal, suture with the frontal: anteroposteriorly or sagitally orientated (0); distinctly posteromedially inclined by a*

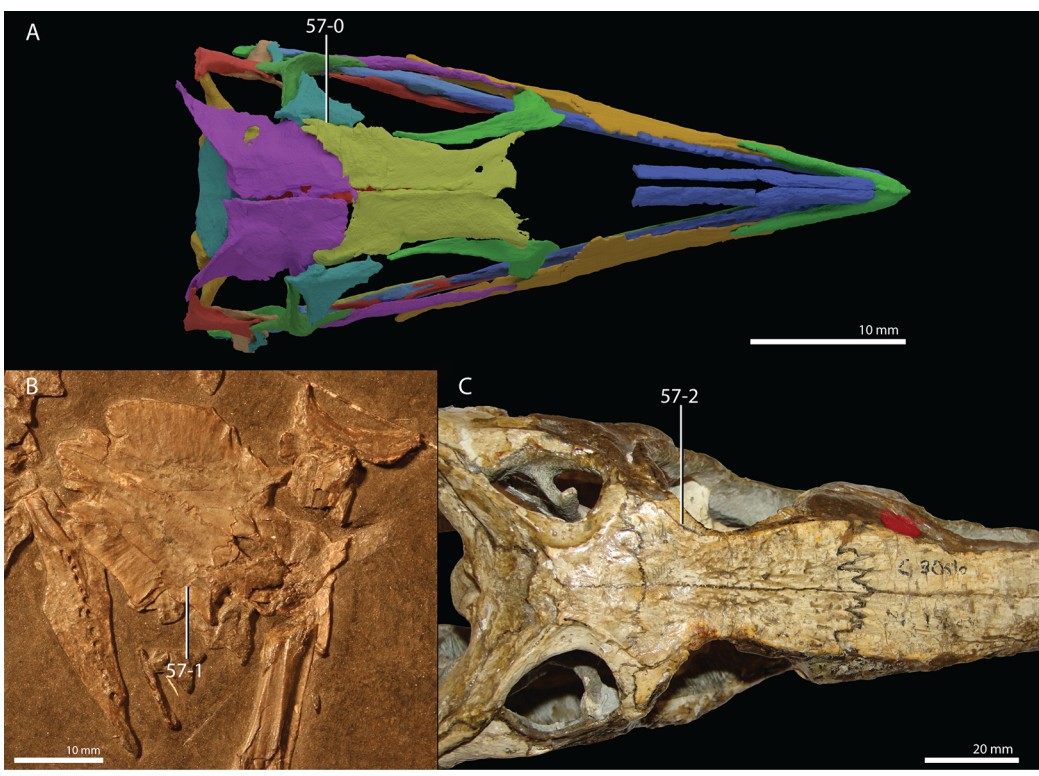

**Figure 12 Illustration of character 57.** (A) State 0 in a digital reconstruction of *Macrocnemus bassanii* (PIMUZ T 2477, skull in dorsal view). (B) State 1 in *Tanystropheus longobardicus* (PIMUZ T 2484, frontal, parietal, and postfrontal in dorsal view). (C) State 2 in *Proterosuchus alexanderi* (NM QR 1484, part of the skull in dorsal view). Image of *Proterosuchus alexanderi* courtesy of Martín Ezcurra.

*medial process of the postfrontal, resulting in a posteriorly strongly narrowed frontal (1); distinctly posterolaterally inclined, resulting in a posteriorly expanded frontal and reduced postfrontal (2)* (Fig. 12).

This character, describing the contact between the frontal and postfrontal, combines the description of several characters of previous analyses. In all non-archosauromorph diapsids included here, as well as in most non-archosauriform archosauromorphs (tanystropheids, *Prolacerta broomi*, *Czatkowiella harae*, and *Protorosaurus speneri*), the articulation between the postfrontal and frontal is sagitally orientated. However, in the rhynchosaurs *Howesia browni* and *Mesosuchus browni*, as well as the allokotosaurs *Trilophosaurus buettneri* and *Azendohsaurus madagaskarensis*, the postfrontal bears a distinct medial process, resulting in a posteriorly narrow frontal and a posteromedially orientated suture between the postfrontal and frontal. This morphology was described by character 313 of *Pritchard et al. (2018)*. However, in the archosauriforms, a posteriorly wider frontal reduces the size of the postfrontal, as seen in *Proterosuchus fergusi*, *Proterosuchus alexanderi*, and *Erythrosuchus africanus* included here. This morphology was described by character 122 of *Ezcurra (2016)* as well as character 15 of *Pritchard et al. (2015)*. In most archosaur groups, as well as Proterochampsia, the postfrontal has been

lost completely (see character 44 of *Nesbitt, 2011*). However, since no taxa belonging to these clades are included here, a separate character state referring to this condition has not been included.

**Character 58 (New, combination of *Pritchard et al., 2015*: ch. 18 and *Ezcurra, 2016*: ch. 123 [= *Pritchard et al., 2015*: ch. 27]).** *Postfrontal, lacks a posterior process and does not participate in the border of the supratemporal fenestra (0); has a posterior process and participates in the border of the supratemporal fenestra (1)* (*Ezcurra, 2016*: Fig. 16).

In all scored taxa, the presence of a posterior process of the postfrontal, or a roughly T-shaped postfrontal, implies that the postfrontal contributes to the margin of the supratemporal fenestra, and thereby prevents a contact between the postorbital and parietal (*Youngina capensis, Gephyrosaurus bridensis*, and *Planocephalosaurus robinsonae*). Therefore, we consider the contribution of the postfrontal to the supratemporal fenestra dependent on the presence of a T-shaped postfrontal, and we have combined the characters describing this morphology here.

**Character 59 *Ezcurra, 2016*: ch. 124).** *Postfrontal, shape of dorsal surface: flat or slightly concave towards raised orbital rim (0); depression with deep pits (1)* (*Ezcurra, 2016*: Fig. 16). *This character is inapplicable in taxa that lack a postfrontal.*

**Character 60 (*Ezcurra, 2016*: ch. 130).** *Postorbital, posterior process extends close to or beyond the level of the posterior margin of the supratemporal fenestrae: absent (0); present (1)* (*Ezcurra, 2016*: Fig. 17).

**Character 61 (*Ezcurra, 2016*: ch. 131).** *Postorbital, extension of the ventral process: ends much higher than the ventral border of the orbit (0); ends close to or at the ventral border of the orbit (1)* (*Ezcurra, 2016*: Fig. 17).

**Character 62 (Modified from *Dilkes, 1998*: ch. 23).** *Postorbital, length of the ventral process versus the length of the posterior process of the postorbital: 0.47-0.59 (0); 0.76-0.88 (1); 0.99-1.17 (2); 1.33-1.62 (3); 1.78-1.95 (4); 2.08-2.20 (5); 2.44-2.54 (6), ORDERED RATIO.*

The identification of the processes was slightly modified for them to be congruent with other character descriptions listed here.

**Character 63 (*Ezcurra, 2016*: ch. 126).** *Postorbital-squamosal, upper temporal bar: located approximately at level of mid-height of the orbit (0); located approximately aligned to the dorsal border of the orbit (1)* (*Ezcurra, 2016*: Figs. 17 and 19). *This character is inapplicable in taxa without an infratemporal fenestra and in taxa in which the upper temporal bar is very tall, reaching from the dorsal margin of the orbit to or beyond mid-height of the orbit.*

An inapplicability criterion was added to this character, because in *Trilophosaurus buettneri* (*Spielmann et al., 2008*) the infratemporal fenestra is absent, and therefore an

upper temporal bar is not present, and because in *Tanystropheus hydroides* (PIMUZ T 2790) the upper temporal bar is dorsoventrally tall and therefore covers the lateral side of the skull from the dorsal border of the orbit to about mid-height of the orbit, which covers both states of this character.

**Character 64 (Modified from *Ezcurra, 2016*: ch. 127).** *Postorbital-squamosal, contact: restricted to the dorsal margin of the elements (0); the anterior process of the squamosal continues along the posterior margin of the ventral process of the postorbital and contacts the jugal (1)* (*Nesbitt, 2011*: Figs. 17 and 19). *This character is inapplicable in taxa that lack an infratemporal fenestra.*

State 1 of character 127 in *Ezcurra (2016)* was not included here, because it is not applicable to any of the included taxa. An inapplicability criterion was included because it was considered that the morphology of *Trilophosaurus buettneri* (*Spielmann et al., 2008*), in which an infratemporal fenestra is absent, represents a distinctly separate morphology from state 0, even though the squamosal and jugal likely did not meet in this taxon.

**Character 65 (*Ezcurra, 2016*: ch. 18).** *Infratemporal fenestra: present (0); absent (1).*

**Character 66 (*Ezcurra, 2016*: ch. 137).** *Squamosal, anterior process forms more than half of the lateral border of the supratemporal fenestra: absent (0); present (1)* (*Ezcurra, 2016*: Fig. 16). *This character is inapplicable in taxa lacking a supratemporal fenestra.*

**Character 67 (Modified from *Ezcurra, 2016*: ch. 143 and *Pritchard et al., 2015*: ch. 33).** *Squamosal, ventral process: present (0); absent or completely confluent with anterior process (1)* (Fig. 13).

This character was modified based on the observed morphologies in the sampled taxa. In *Tanystropheus hydroides* (PIMUZ T 2790) no clear ventral process can be distinguished, but instead the anterior process of the squamosal is dorsoventrally tall and plate-like. This is possibly the result of a confluence of the anterior and ventral processes (*Spiekman et al., 2020b*). In *Trilophosaurus buettneri* (*Spielmann et al., 2008*) the ventral process is also absent.

**Character 68 (*Ezcurra, 2016*: ch. 139) (slightly reformulated).** *Squamosal, ventral process: angle between the ventral and anterior processes of the squamosal 90 degrees or less, forming a roughly square outline (0); angle between the ventral and anterior processes of the squamosal more than 90 degrees, forming a gentle, widely rounded posterodorsal border of the infratemporal fenestra (1)* (*Ezcurra, 2016*: Figs. 8, 17, 18 and 24). *This character is scored as inapplicable in taxa that lack a ventral process of the squamosal.*

**Character 69 (*Pritchard et al., 2015*: ch. 34).** *Squamosal, ventral process: forming a massive flange that covers the quadrate entirely in lateral view (0); anteroposteriorly slender (1). This character is scored as inapplicable in taxa that lack a ventral process on the squamosal.*

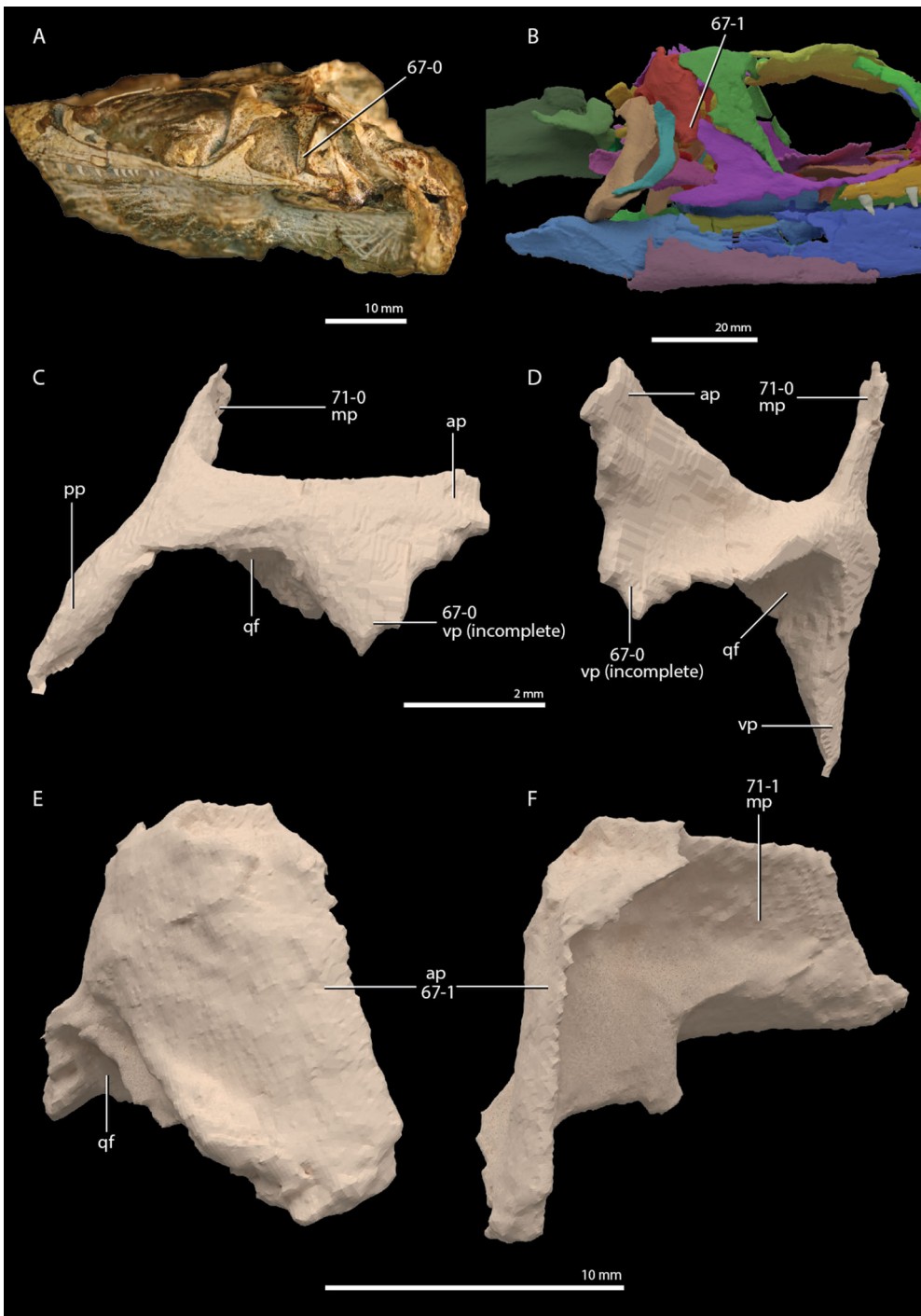

**Figure 13 Illustration of characters 67 and 71.** (A) 67-0 in *Prolacerta broomi* (BP/1/5375, partial skull in left lateral view). (B) 67-1 in a digital reconstruction of *Tanystropheus hydroides* (PIMUZ T 2790, posterior part of the skull in right lateral view). (C and D) 67-0 and 71-0 in a digital reconstruction of *Macrocnemus bassanii* (PIMUZ T 2477, right squamosal in (C) lateral and (D) angled ante-roventromedial view). (E and F) 67-1 and 71-1 in a digital reconstruction of *Tanystropheus hydroides* (PIMUZ T 2790, right squamosal in (E) lateral and (F) anterior view). Abbreviations: ap, anterior process; mp, medial process; pp, posterior process; qf, quadrate facet; vp, ventral process.

See character description of character 135 of *Ezcurra (2016)*, which covers the same distinction. The character description of *Pritchard et al. (2015)* was preferred here, because it is considered to be more informative. The inapplicability criterion has been added.

**Character 70 (*Ezcurra, 2016*: ch. 140).** *Squamosal, medial process: short, forming up to half or less of the posterior border of the supratemporal fenestra (0); long, forming entirely or almost entirely the posterior border of the supratemporal fenestra (1) (*Ezcurra, 2016*: Fig. 16). This character is scored as inapplicable in taxa that lack a medial process of the squamosal.*

The inapplicability criterion was added.

**Character 71 (New).** *Squamosal, medial process, dorsoventrally short (0); dorsoventrally tall and plate-like, forming a tall surface of the posterior margin of the supratemporal fenestra (1) (*Fig. 13*). This character is scored as inapplicable in taxa that lack a medial process of the squamosal.*

**Character 72 (Modified from *Ezcurra, 2016*: ch. 141).** *Squamosal, posterior process is distinct and extends posterior to the dorsal head of the quadrate: absent (0); present (1) (*Ezcurra, 2016*: Figs. 18, 19, and 24). This character is inapplicable in taxa where the quadrate is completely covered by the squamosal in lateral view.*

The description of this character was modified to more clearly describe the morphology observed in the taxon sample studied here.

**Character 73 (*Ezcurra, 2016*: ch. 157).** *Supratemporal: broad element (0); slender, in parietal and squamosal trough (1); absent (2) ORDERED (*Ezcurra, 2016*: Fig. 17).*

The definitive absence of the supratemporal is hard to establish because it is often a small element that is easily obscured by specimen disarticulation or compression. Therefore, following *Ezcurra (2016)*, this bone is only scored as absent when it can be confidently established as such from well-preserved specimens.

**Character 74 (*Ezcurra, 2016*: ch. 159 and *Pritchard et al., 2015*: ch. 19).** *Parietal, median contact between both parietals: suture present (0); fused with loss of suture (1) (*Ezcurra, 2016*: Fig. 16).*

State 0 can only be scored based on skeletally mature specimens.

**Character 75 (*Ezcurra, 2016*: ch. 160).** *Parietal, extension over interorbital region: absent or slight (0); present (1) (*Ezcurra, 2016*: Figs. 6 and 23).*

**Character 76 (*Ezcurra, 2016*: ch. 162).** *Parietal, pineal fossa on the median line of the dorsal surface: absent (0); present (1). This character should not be scored for early juveniles (*Fig. 14*; *Ezcurra, 2016*: Fig. 8).*

This character was considered to be present in *Kadimakara australiensis* and several archosauriforms (see scorings of *Ezcurra, 2016*, character 162). However, a similar fossa as

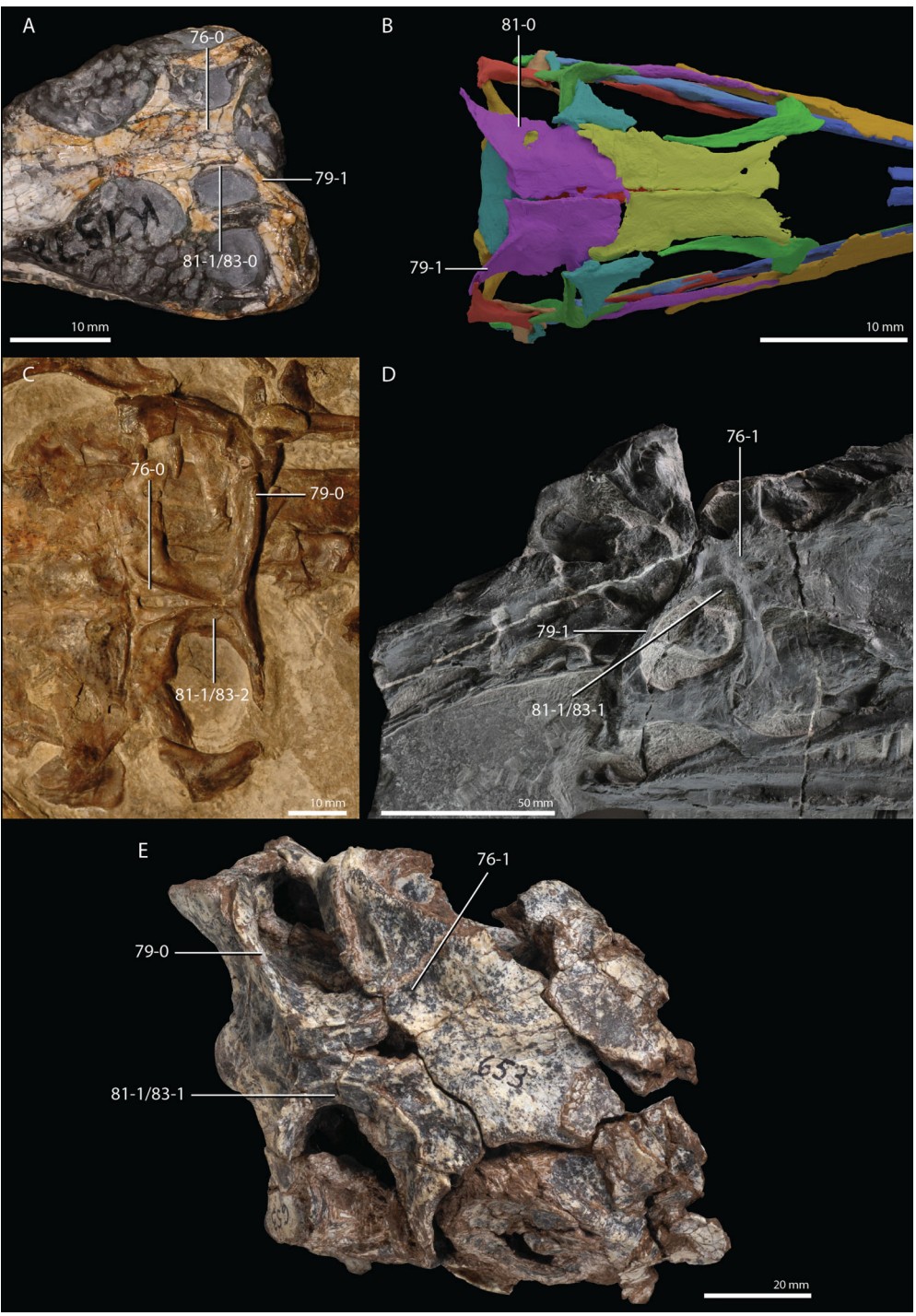

**Figure 14 Illustration of characters 76, 79, 81, and 83.** (A) 76-0, 79-1, 81-1, 83-0 in *Youngina capensis* (SAM-PK-K7578, posterior part of the skull in dorsal view). (B) 79-1, 81-0 in a digital reconstruction of *Macrocnemus bassanii* (PIMUZ T 2477, posterior part of the skull in dorsal view). (C) 76-0, 79-0, 81-1, 83-2 in *Tanystropheus hydroides* (PIMUZ T 2819, posterior part of the skull in dorsal view). (D) 76-1, 79-1, 81-1, 83-1 in *Dinocephalosaurus orientalis* (IVPP V13767, posterior part of the skull in right laterodorsal view). (E) 76-1, 79-0, 81-1, 83-1 in *Azendohsaurus madagaskarensis* (UA-7-20-99-653, partial skull in dorsal view). Image of *Azendohsaurus madagaskarensis* courtesy of Sterling Nesbitt.

present in these taxa can also be identified in *Azendohsaurus madagaskarensis* (*Flynn et al., 2010*), *Trilophosaurus buettneri* (*Flynn et al., 2010*), and *Dinocephalosaurus orientalis* (IVPP V13767).

**Character 77 (Modified from *Ezcurra, 2016*: ch. 164).** *Parietal, pineal foramen in dorsal view: large (0); reduced to a small, circular pit or concavity (1); absent (2)* (*Ezcurra, 2016*: Figs. 6 and 8), *ORDERED.*

The character was ordered, since it is considered a transformational series in which state 1 represents a clear intermediate between states 0 and 2. Furthermore the concavity statement was added to state 1, because in some taxa this depression is not pit-like.

**Character 78 (Modified from *Ezcurra, 2016*: ch. 165).** *Parietal, position of the pineal foramen in dorsal view: enclosed by parietals and clearly on the posterior part of the bones (0); enclosed by parietals at roughly mid-length of the bones (1); enclosed by parietals on the anterior part of the bones close to the frontals (2); enclosed by both frontals and parietals (3), ORDERED* (*Ezcurra, 2016*: Figs. 6 and 8). *This character is scored as inapplicable in taxa that lack a pineal foramen.*

The pineal foramen is displaced distinctly posteriorly on the parietals in the non-archosauromorph diapsids *Planocephalosaurus robinsonae* (*Fraser, 1982*), *Orovenator mayorum* (*Ford & Benson, 2018*), and *Youngina capensis* (AMNH FARB 5561), and this was therefore considered as a separate character state. This morphology was extensively discussed in *Ford & Benson (2018*, p. 208).

**Character 79 (Modified from *Pritchard et al., 2015*: ch. 21).** *Parietal, orientation of the posterolateral process: roughly transverse (0); strongly angled posterolaterally (1)* (Fig. 14).

In most of the sampled taxa, the posterolateral processes have a posterolateral orientation. However, in *Tanystropheus hydroides* (PIMUZ T 2819), *Tanystropheus longobardicus* (PIMUZ T 2484), *Protorosaurus speneri* (NMK S 180), and *Azendohsaurus madagaskarensis* (*Flynn et al., 2010*), the posterolateral processes have a completely transverse or lateral orientation.

**Character 80 (*Ezcurra, 2016*: ch 168).** *Parietal, posterolateral process height: dorsoventrally low, usually considerably lower than the supraoccipital (0); dorsoventrally deep, being plate-like in occipital view and subequal to the height of the supraoccipital (1)* (*Ezcurra, 2016*: Fig. 27).

**Character 81 (Modified from *Ezcurra, 2016*: ch. 8).** *Parietal, supratemporal fossa medial to the supratemporal fenestra: absent (0); present (1)* (Fig. 14).

**Character 82 (*Ezcurra, 2016*: ch. 161).** (inapplicability criterion slightly reformulated). *Parietal, supratemporal fossa medial to the supratemporal fenestra: well-exposed in dorsal view and mainly dorsally or dorsolaterally facing (0); poorly exposed in dorsal view and*

*mainly laterally facing (1)* (*Ezcurra, 2016*: Fig. 16). *This character is scored as inapplicable in taxa that lack a supratemporal fossa on the parietal.*

**Character 83 (Modified from *Pritchard et al., 2015*: ch. 20 and *Ezcurra, 2016*: ch. 8).** *Parietal, medial extent of the supratemporal fossa: restricted to the lateral edge of the parietal, resulting in a broad, flat parietal table (0); expanded distinctly medially, resulting in a mediolaterally narrow parietal table (1); supratemporal fossae strongly expanded medially and only separated by a ridge running along the midline of the parietal, the sagittal crest (2), ORDERED* (Fig. 14). *This character is scored as inapplicable in taxa that lack a supratemporal fossa on the parietal.*

This character was modified to clarify the distinction between the different states. The supratemporal fossa can either be restricted to the lateral portion of the parietal, expressed more widely on the parietal, or cover most of the dorsal surface of the parietal between the supratemporal fenestrae, only leaving a thin sagittal crest between the two fossae. This character is very variable in *Prolacerta broomi* with all three states observed in different specimens (state 0: BP/1/471, state 1: BP/1/5375 and UCMP 37151, state 2: BP/1/5066 and BP/1/5880).

**Character 84 (*Ezcurra, 2016*: ch. 171).** *Postparietal, size (pair of postparietals if they are not fused to each other): sheet-like, not much narrower than the supraoccipital (0); small, splint-like (1); absent as a separate ossification (2) ORDERED* (*Ezcurra, 2016*: Fig. 23).

**Character 85 (*Ezcurra, 2016*: ch. 172 and *Pritchard et al., 2015*: ch. 25).** *Postparietal, fusion between counterparts: absent (0); present, forming an interparietal (1). This character is inapplicable in taxa that lack postparietals.*

**Character 86 (*Ezcurra, 2016*: ch. 173 and *Pritchard et al., 2015*: ch. 37).** *Tabular: present (0); absent (1).*

**Character 87 (*Ezcurra, 2016*: ch. 150 and *Pritchard et al., 2015*: ch. 38).** *Quadratojugal: absent or fused to the quadrate (0); present (1)* (*Ezcurra, 2016*: Fig. 24).

**Character 88 (Modified from *Ezcurra, 2016*: ch. 153).** *Quadratojugal, anterior process: absent, anteroventral margin of the bone rounded and the quadratojugal and jugal do not connect and therefore the lower temporal bar is incomplete (0); incipient, short anterior prong on the anteroventral margin of the bone and the quadratojugal and jugal connect and therefore the lower temporal bar is complete (1); distinctly present, in which the lower temporal bar is complete, but process terminates well posterior to the base of the posterior process of the jugal (2); distinctly present, in which the lower temporal bar is complete and participates in the posteroventral border of the infratemporal fenestra, and process terminates close to the base of the posterior process of the jugal (3), ORDERED* (*Ezcurra, 2016*: Figs. 17 and 19). *This character is inapplicable in taxa that lack an infratemporal fenestra or quadratojugal.*

The description of this character was modified to more clearly describe the morphology observed in the taxon sample studied here.

**Character 89 (*Ezcurra, 2016*: ch. 156).** *Quadratojugal, posterior extension of the ventral end: absent, without a posteriorly arched quadratojugal (0); limited, ventral condyles of the quadrate broadly visible in lateral view (1); strongly developed, overlapping completely or almost completely the ventral condyles of the quadrate in lateral view (2), ORDERED* (*Ezcurra, 2016*: Fig. 18). *This character is inapplicable in taxa lacking a quadratojugal.*

**Character 90 (New, combination of ch. 176 and ch. 182 in *Ezcurra, 2016*).** *Quadrate, posterior margin in lateral view: straight along entire shaft (0); continuously concave (1); sigmoidal, with a concave dorsal portion and convex ventral portion (2)* (*Ezcurra, 2016*: Fig. 24).

Characters 176 and 182 of *Ezcurra (2016)* were combined because they both relate to the shape of the quadrate shaft. The presence of a quadrate conch is omitted because its presence is likely closely related to a fusion of the quadratojugal to the quadrate in lepidosauromorphs. This fusion is already coded for by character 87 and its inclusion here would result in the overscoring of this morphology.

**Character 91 (*Ezcurra, 2016*: ch. 180 and *Nesbitt et al., 2015*: ch. 207).** *Quadrate, dorsal end hooked posteriorly in lateral view: absent (0); present (1)* (*Ezcurra, 2016*: Figs. 17 and 24).

**Character 92 (New).** *Quadrate, ventral condyles: lateral and medial condyles not distinctly separated and therefore the ventral surface of the quadrate is rounded, flat, or slightly concave (0); condyles separated by a deep concavity on the ventral surface of the quadrate (1)* (Fig. 15).

**Character 93 (*Ezcurra, 2016*: ch. 183).** *Quadrate, ventral condyles: subequally distally extended (0); medial condyle distinctly more distally projected than the lateral one (1)* (Fig. 15).

**Character 94 (New).** *Quadrate, pterygoid flange: anteriormost extension at about mid-height of the quadrate shaft (0); dorsally located, the anteriormost extension of the flange is at close to the dorsoventral level of the dorsal head of the quadrate (1)* (Fig. 15).

This character describes the difference seen in the morphology of the pterygoid flange, as can be clearly observed between for instance *Tanystropheus hydroides* (PIMUZ T 2790) and *Macrocnemus bassanii* (PIMUZ T 2477).

**Character 95 (*Pritchard et al., 2015*: ch. 45).** *Vomer, teeth: absent (0); present (1).*

**Character 96 (Modified from *Ezcurra, 2016*: ch. 187).** *Vomer, teeth distribution: shagreen tooth distribution with no clear rows distinguishable (0); teeth distributed in multiple clearly defined rows (1); teeth distributed mainly in a single row, but multiple teeth present*

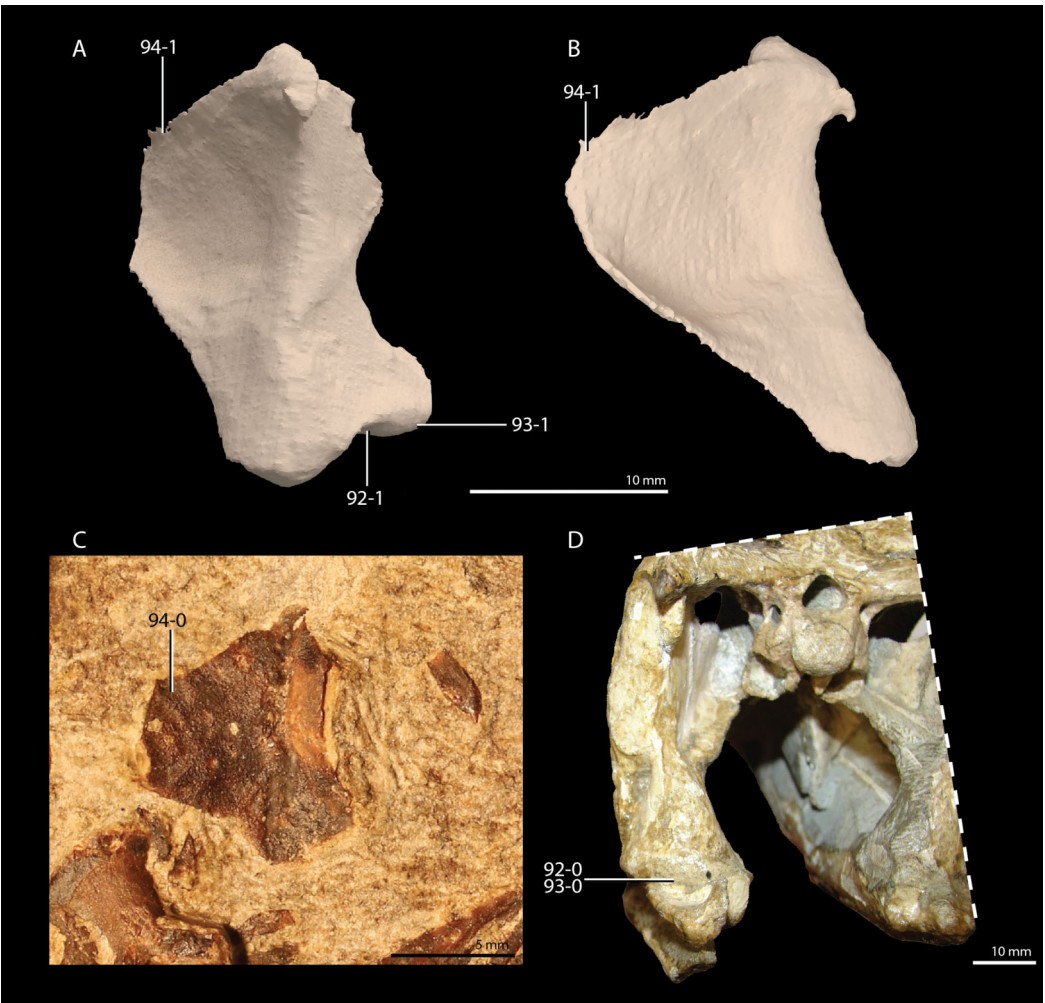

**Figure 15 Illustration of characters 92, 93, and 94.** (A and B) 92-1, 93-1, 94-1 in a digital reconstruction of *Tanystropheus hydroides* (PIMUZ T 2790, right quadrate in (A) posterior and (B) medial view). (C) 94-0 in *Macrocnemus fuyuanensis* (PIMUZ T 1559, right quadrate in anterior view). (D) 92-0 and 93-0 in *Proterosuchus alexanderi* (NM QR 1484, left side of the skull in posterior/occipital view). Image of *Proterosuchus alexanderi* courtesy of Martín Ezcurra.

*immediately anterior to the contact with the pterygoid (2); teeth distributed in a single row along entire extension (3). This character is inapplicable in taxa that lack vomerine teeth.*

The presence of vomerine teeth and their distribution were considered in one ordered character in character 187 in *Ezcurra (2016)*. However, we do not consider any of the various tooth distributions to represent an intermediate stage between any of the others. Therefore, we treat the presence of vomerine teeth as a separate character, and the distribution of these teeth, if they are present, as a separate, unordered character.

**Character 97 (New, related to *Ezcurra, 2016*: ch. 189).** *Palatal dentition, size (height and diameter) of teeth on the vomer: small, considerably smaller than those of the marginal*

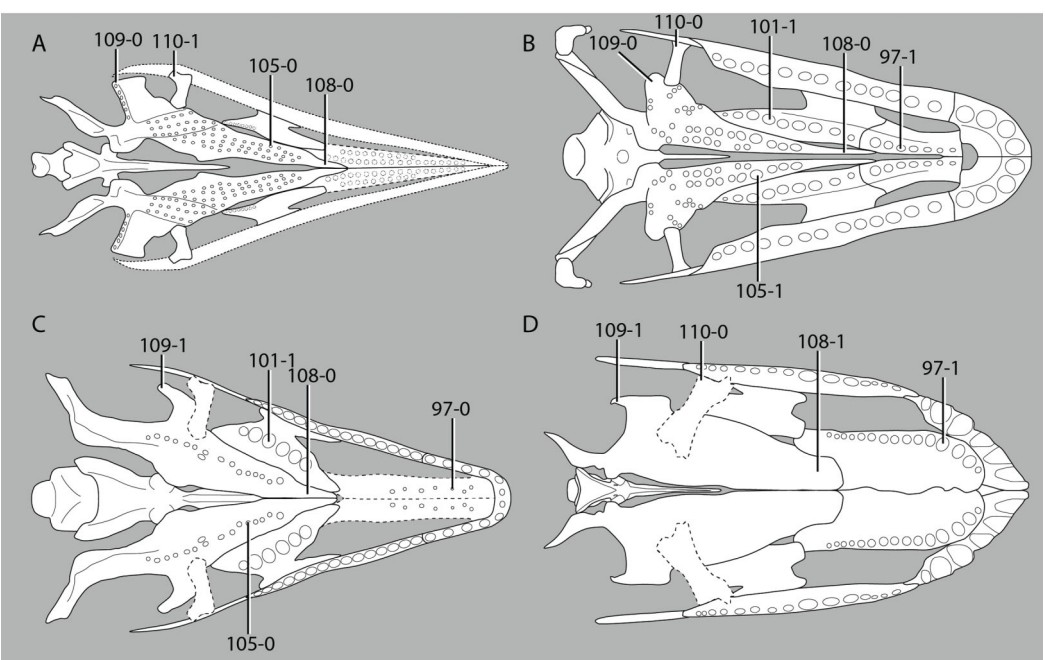

**Figure 16 Illustration of characters 97, 101, 105, 108, 109, and 110.** (A) 105-0, 108-0, 109-0, and 110-1 in *Macrocnemus fuyuanensis* (palatal reconstruction, modified from *Scheyer et al., 2020b*). (B) 97-1, 101-1, 105-1, 108-0, 109-0, and 110-0 in *Azendohsaurus madagaskarensis* (palatal reconstruction, modified from *Flynn et al., 2010*). (C) 97-0, 101-1, 105-0, 108-0, and 109-1 in *Tanystropheus longobardicus* (palatal reconstruction, modified from *Spiekman et al., 2020a*). (D) 97-1, 108-1, 109-1, and 110-0 in *Tanystropheus hydroides* (palatal reconstruction, modified from *Spiekman et al., 2020a*).

*dentition (0); relatively large, similar to those of the marginal dentition (1)* (Fig. 16). *This character is inapplicable in taxa lacking vomerine teeth.*

Character 189 in *Ezcurra (2016)* describes the relative size of the teeth on the palatine and pterygoid. However, in our sampled taxa, a distinct difference in the size of the dentition could also be observed in the vomer, and this was therefore formulated into a separate character, since the size of the vomerine teeth does not appear to be consistently dependent on the size of the palatine or pterygoid teeth in the sampled taxa.

**Character 98 (*Ezcurra, 2016*: ch. 190) (description of state 1 slightly reformulated).** *Palatine, transverse extension: narrow, subequal contribution of the palatine and pterygoid to or pterygoid main component of the palate posterior to the choanae (0); broad, the palatine is the main component of the palate posterior to the choanae (1)* (*Ezcurra, 2016*: Fig. 26).

**Character 99 (Modified from *Ezcurra, 2016*: ch. 191).** *Palatine, anterior processes forming the posterior border of the choana: subequal in anterior extension or anterolateral process longer (0); anteromedial process longer (1)* (*Ezcurra, 2016*: Fig. 26).

Character state 2 of character 191 in *Ezcurra (2016)* was not included here, because it is not applicable to any of the sampled taxa.

**Character 100 (*Ezcurra, 2016*: ch. 188).** *Palatine-pterygoid, teeth on the palatine and ventral surface of the anterior ramus of the pterygoid: present (0); absent (1)* (*Ezcurra, 2016*: Figs. 13, 24 and 26).

**Character 101 (Part of *Ezcurra, 2016*: ch. 189).** *Palatal dentition, size (height and diameter) of teeth on the palatine: small, considerably smaller than those of the marginal dentition (0); relatively large, similar to those of the marginal dentition (1)* (Fig. 16; *Ezcurra, 2016*: Figs. 25 and 26). *This character is inapplicable in taxa lacking palatine teeth.*

Character 189 in *Ezcurra (2016)* treats the size of the dentition on the palatine and pterygoid as a single character. Because the relative size of the teeth on the palatine and pterygoid differs in *Tanystropheus longobardicus* (PIMUZ T 2484) it was decided here to treat the size of the teeth on both elements as separate characters.

**Character 102 (*Ezcurra, 2016*: ch. 195).** *Pterygoid, teeth on the ventral surface of the anterior ramus (=palatal process), excluding tiny palatal teeth if present: present in two distinct fields (=T2 and T3 of* Welman, 1998*) (0); present in three distinct fields (=T2, T3a and T3b) (1); present in three distinct fields (=T2a, T2b and T3) (2); present in one field that occupies most of the transverse width of the ramus (=T2 + T3) (3); present in only one posteromedially to anterolaterally orientated field (=T2) (4); present in only one field adjacent to the medial margin of the ramus (=T3) (5); present in no definable fields but the entire pterygoid is covered by a shagreen of teeth (6)* (*Ezcurra, 2016*: Figs. 25 and 26). *This character is inapplicable in taxa that lack teeth in the palatine and the ventral surface of the anterior ramus of the pterygoid.*

**Character 103 (*Ezcurra, 2016*: ch. 196).** *Pterygoid, number of rows on palatal tooth field T2: more than two or do not dispose on distinct rows (0); two rows parallel to each other (1); single row (2)* (*Ezcurra, 2016*: Figs. 25 and 26). *This character is inapplicable if the tooth field T2 is subdivided in T2a and T2b or is absent.*

**Character 104 (*Ezcurra, 2016*: ch. 197) (state 0 slightly reformulated).** *Pterygoid, number of rows on palatal tooth field T3: more than two or teeth not placed in distinct rows (0); two parallel rows (1); single row (2)* (*Ezcurra, 2016*: Figs. 25 and 26). *This character is inapplicable if the tooth field T3 is subdivided into T3a and T3b or is absent.*

Character 199 in *Ezcurra (2016)* treats a row of teeth sticking out on the medial side of the anterior ramus of the pterygoid (=T4 of *Welman, 1998*) as a separate character. It is found here, based on observations of *Macrocnemus bassanii* (PIMUZ T 1559) and *Prolacerta broomi* (unpublished CT-scan of BP/1/5066) that tooth field T3 in these taxa bears more than two distinct rows. Furthermore, the medial margin of the anterior ramus of the pterygoid is curved, resulting in a number of these teeth facing lateroventrally, whilst others face mediolaterally. Therefore, we conclude that tooth field T4 actually represents the mediolaterally facing teeth of tooth field T3 and consequently character 199 in *Ezcurra (2016)* has not been included here.

**Character 105 (Part of** *Ezcurra, 2016***: ch. 189).** *Palatal dentition, size (height and diameter) of teeth on the ventral surface of the anterior ramus of the pterygoid: small, considerably smaller than those of the marginal dentition (0); relatively large, similar to those of the marginal dentition (1)* (Fig. 16; *Ezcurra, 2016*: Figs. 25 and 26). *This character is inapplicable in taxa lacking teeth on the anterior ramus of the pterygoid.*

See description of character 101.

**Character 106 (Part of** *Ezcurra, 2016***: ch. 202).** *Pterygoid, teeth on the lateral ramus (=transverse flange): absent (0); present (1)* (*Ezcurra, 2016*: Figs. 13, 25 and 26). *This character is inapplicable in taxa in which most of the pterygoid is covered by shagreen teeth.*

The inapplicability criterion was added because this tooth row cannot be distinguished from other pterygoid teeth when the pterygoid is covered by shagreen teeth. We separated this character from character 107 because we consider the presence of teeth on the lateral ramus of the pterygoid to represent a separate criterion from the number of tooth rows if such teeth are present. Therefore, we do not consider the presence of a single row of teeth to represent an intermediate step in a transformational series between no teeth present and two rows present.

**Character 107 (Part of** *Ezcurra, 2016***: ch. 202).** *Pterygoid, distribution of teeth on the lateral ramus (=transverse flange): teeth distributed in a single row on the posterior edge (=T1 of* Welman, 1998*) (0); teeth distributed in multiple rows (1)* (*Ezcurra, 2016*: Figs. 13, 25 and 26). *This character is inapplicable in taxa that lack teeth on the lateral ramus of the pterygoid or in taxa in which shagreen teeth cover the pterygoid.*

See description of character 107.

**Character 108 (New).** *Pterygoid, anterior end of the anterior ramus: tapers to an end (0); rounded (1)* (Fig. 16).

In most of the sampled taxa, the anterior ramus of the pterygoid gradually tapers anteriorly and thus has an anteriorly pointed end. In contrast, in *Tanystropheus hydroides* (PIMUZ T 2787) and *Dincephalosaurus orientalis* (*Rieppel, Li & Fraser, 2008*) the anterior ramus of the pterygoid is much wider anteriorly and has a rounded anterior margin.

**Character 109 (New).** *Pterygoid, lateral/distal end of the posterior margin of the lateral ramus (=transverse flange) curved posteriorly: absent (0); present (1)* (Fig. 16). *This character is scored as inapplicable in taxa with a strongly posterolaterally orientated lateral ramus of the pterygoid.*

This character is closely related to character 201 in *Ezcurra (2016)*. However, because this new description distinguishes between morphologies seen in tanystropheids, it is considered to be more informative and therefore preferred. Character 201 of *Ezcurra (2016)* was not included in order to prevent overscoring of this morphology.

**Character 110 (Modified from *Ezcurra, 2016*: ch. 207).** *Ectopterygoid, lateral process is not curved posteriorly (0); lateral process is curved posteriorly but not expanded posteriorly (1); lateral process is both curved and expanded posteriorly, giving the ectopterygoid a hook-shape in dorsal or ventral view (2)* (Fig. 16; *Ezcurra, 2016*: Figs. 7 and 26), *ORDERED*.

The lateral portion of the ectopterygoid can be separated into three different morphologies. In some taxa, it is not curved, nor expanded (e.g., *Azendohsaurus madagaskarensis*, *Flynn et al., 2010*). In other taxa, the lateral end curves posteriorly but it is not expanded anteroposteriorly (e.g., *Macrocnemus bassanii*, *Miedema et al., 2020*). Finally, in certain taxa, the lateral portion of the ectopterygoid is curved posteriorly and is expanded anteroposteriorly (e.g., *Orovenator mayorum*, *Ford & Benson, 2018*). Since state 1 is considered to represent an intermediate state between 0 and 2 in a transformational series, this character was ordered.

**Character 111 (*Ezcurra, 2016*: ch. 204) (state 0 reformulated).** *Ectopterygoid, articulation with pterygoid: ectopterygoid overlaps the pterygoid ventrally (0); interlaced articulation, complex articulation between ectopterygoid and pterygoid (1)* (*Ezcurra, 2016*: Fig. 26).

**Character 112 (*Ezcurra, 2016*: ch. 205) (the formulation of this character has been modified slightly).** *Ectopterygoid, connection with pterygoid: does not reach the posterolateral corner of the lateral ramus (=transverse flange) (0); reaches the posterolateral corner of the lateral ramus (1)* (*Ezcurra, 2016*: Fig. 26). *This character is scored as inapplicable in taxa in which the ectopterygoid simply overlaps the pterygoid.*

An inapplicability criterion is added because the ectopterygoid only reaches the posterolateral corner of the lateral ramus of the pterygoid when the ectopterygoid forms an interlacing suture with the pterygoid. In taxa with this type of articulation, the ectopterygoid wraps around the posterolateral corner of the transverse flange in some cases.

**Character 113 (*Ezcurra, 2016*: ch. 244 and *Pritchard et al., 2015*: ch. 65).** *Parasphenoid/ parabasisphenoid, dentition on cultriform process: present (0); absent (1).*

**Character 114 (New).** *Parasphenoid/parabasisphenoid, length of the cultriform process versus its height at its anteroposterior midpoint: 4.16-5.77 (0); 9.65-9.89 (1); 10.85-12.12 (2); 13.29-13.42 (3); 20.28-21.12 (4)*, ORDERED *RATIO*.

This character covers the large discrepancy in the relative length of the cultriform process. In most taxa it is a thin elongate element, whereas in allokotosaurs and rhynchosaurs it is much shorter and dorsoventrally taller.

**Character 115 (New).** *Parasphenoid/parabasisphenoid, anterior projections of the cristae trabeculares: present (0); absent (1).*

The cristae trabeculares are small bony projections on the anterolateral surface of the cultriform process of the parabasisphenoid, which occur in certain non-saurian diapsids

and lepidosaurs. These structures and their occurrence among diapsids were discussed in detail by *Ford & Benson (2018*, p. 18*).

**Character 116** *Ezcurra, 2016*: **ch. 236).** *Parasphenoid/parabasisphenoid, posterodorsal portion: incompletely ossified (0); completely ossified (1).*

**Character 117 (Modified from** *Ezcurra, 2016*: **ch. 237).** *Parasphenoid/parabasisphenoid, intertuberal plate: present (0); absent (1)* (*Ezcurra, 2016*: Figs. 10 and 28).

Character states 1 and 2 of character 237 in *Ezcurra (2016)* were fused here, because there was no clear distinction between a rounded and a straight posterior edge of the intertuberal plate in the sampled taxa, and this distinction is likely only relevant in more derived archosauriforms.

**Character 118 (Modified from** *Ezcurra, 2016*: **ch. 239).** *Parasphenoid/parabasisphenoid, recess (=median pharyngeal recess, =hemispherical sulcus, =hemispherical fontanelle): absent, the ventral floor of the parabasisphenoid posterior to the basipterygoid processes (and posterior to a potentially present intertuberal plate) is flat (0); present, the ventral floor forms a shallow depression (1); present, the ventral floor is deeply excavated (2)* (*Ezcurra, 2016*: Fig. 27), *ORDERED.*

The pharyngeal recess was originally identified in archosauriforms but has subsequently also been described for certain non-archosauriform archosauromorphs (e.g., *Azendohsaurus madagaskarensis*, *Flynn et al., 2010*; *Mesosuchus browni*, *Sobral & Müller, 2019*). Observation of this character in the sampled taxa indicates that it can occur in two states when present. The pharyngeal recess was first described as a very deep ventral cavity (e.g., the basisphenoid recess of *Witmer, 1997*). This occurs in *Tanystropheus hydroides* (PIMUZ T 2790) and *Erythrosuchus africanus* (BP/1/3893) among the sampled taxa. However, a much shallower excavation of the ventral surface of the parabasisphenoid posterior to the basipterygoid processes occurs in the majority of non-archosauriform archosauromorphs, as well as *Youngina capensis* (*Gardner, Holiday & O'Keefe, 2010*). This shallow excavation was identified as the pharyngeal recess by *Sobral et al. (2016)* and *Sobral & Müller (2019)*. We here distinguish the shallow excavation and the deeper excavation as separate character states for the first time and consider the former to likely represent an intermediate morphology between the absence of a pharyngeal recess and the deeply excavated pharyngeal recess.

**Character 119 (***Ezcurra, 2016*: **ch. 238) (formulation of the inapplicability criterion is slightly modified).** *Parasphenoid/parabasisphenoid, semilunar depression on the posterolateral surface of the bone: absent (0); present (1)* (*Ezcurra, 2016*: Fig. 28). *This character is inapplicable in taxa in which the posterodorsal portion of the parasphenoid/parabasisphenoid is not ossified, resulting in an unossified gap between this element and the prootic.*

**Character 120 (***Ezcurra, 2016*: **ch. 235 and** *Nesbitt et al., 2015*: **ch. 208) (description of state 1 slightly reformulated).** *Basisphenoid/parabasisphenoid, orientation of the body*

*between the posterior end of the bone and the basipterygoid processes: horizontal (0); oblique, main axis posterodorsally to anteroventrally orientated (1)* (*Ezcurra, 2016*: Figs. 27 and 28).

**Character 121 (*Ezcurra, 2016*: ch. 225).** *Basioccipital-parasphenoid/parabasisphenoid, contact with each other in skeletally mature individuals: loose, overlapping suture (0); tightly sutured, sometimes by an interdigitated suture, or both bones fused to each other (1)* (*Ezcurra, 2016*: Fig. 28).

**Character 122 (New).** *Basioccipital-parasphenoid/parabasisphenoid, two pneumatic foramina between the basioccipital and parabasisphenoid: absent (0); present (1)* (*Sobral & Müller, 2019*: Figs. 3 and 13).

Pneumatic foramina were described as present in several early archosauromorphs by *Sobral & Müller (2019)*. This character is now implemented in a quantitative phylogenetic analysis for non-archosauriform archosauromorphs for the first time.

**Character 123 (*Ezcurra, 2016*: ch. 226).** *Basioccipital-parasphenoid/parabasisphenoid, basal tubera: absent (0); present (1)* (*Ezcurra, 2016*: Fig. 27).

**Character 124 (Modified from *Ezcurra, 2016*: ch. 227).** *Basioccipital-parasphenoid/ parabasisphenoid, low ridge between basal tubera: absent or very strongly reduced (0); present (1)* (Fig. 17; *Ezcurra, 2016*: Fig. 27). *This character is scored as inapplicable in taxa that lack basal tubera.*

Character 227 in *Ezcurra (2016)* is applicable to a wide range of archosauromorphs. This character was modified to more specifically address the variation observed in the taxa sampled here. A clear but low, transversely orientated ridge is present between the basal tubera of the basioccipital of *Tanystropheus hydroides* (PIMUZ T 2790) and *Tanystropheus longobardicus* (PIMUZ T 2484). Such a ridge cannot be observed in any of the other sampled taxa.

**Character 125 (*Pritchard et al., 2018*: ch. 318).** *Basioccipital, ventral margin: prominent embayment or ridge between basal tubera at least as transversely broad as occipital condyle (0); transversely narrow embayment or ridge between basal tubera, narrower than occipital condyle (1). This character is scored as inapplicable in taxa that lack basal tubera.*

See the description of character 318 in *Pritchard et al. (2018)*. An inapplicability criterion has been added to this character.

**Character 126 (*Ezcurra, 2016*: ch. 229).** *Basioccipital, articular surface of the occipital condyle: concave (0); hemispherical (1)* (*Ezcurra, 2016*: Fig. 28).

**Character 127 (*Ezcurra, 2016*: ch. 211 and *Pritchard et al., 2015*: ch 62).** *Otoccipital, fusion between opisthotic and exoccipital: absent or partial (0); present (1)* (*Ezcurra, 2016*: Fig. 27).

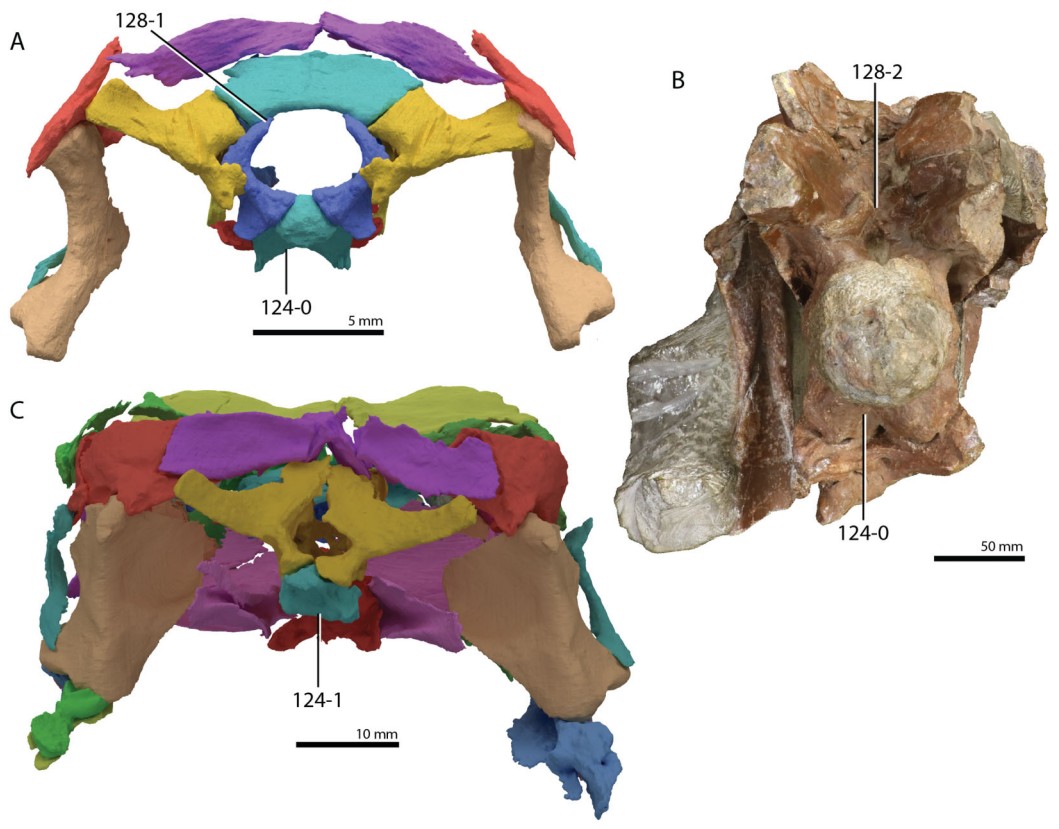

**Figure 17 Illustration of characters 124 and 128.** (A) 124-0 and 128-1 in a digital reconstruction of *Macrocnemus bassanii* (PIMUZ T 2477, skull in posterior/occipital view). (B) 124-0 and 128-2 in *Erythrosuchus africanus* (BP/1/3893, partial braincase in posterior/occipital view). (C) 124-1 in a digital reconstruction of *Tanystropheus hydroides* (PIMUZ T 2790, skull in posterior view).

**Character 128 (New, combination of *Ezcurra, 2016*: ch. 209 [= *Pritchard et al., 2015*: ch. 60] and *Ezcurra, 2016*: ch. 219 [= character 59 of *Pritchard et al., 2015*: ch. 59]).** *Exoccipital, morphology of the dorsal end: exoccipital columnar through dorsoventral height, forming transversely narrow dorsal contact with more dorsal occipital elements (0); dorsal portion of exoccipital exhibits dorsomedially inclined process that forms transversely broad contact with more dorsal occipital elements but exoccipitals do not meet on the dorsal margin of the foramen magnum (1); dorsal portion of exoccipital exhibits dorsomedially inclined process that meets the process of the opposite exoccipital on the dorsal margin of the foramen magnum, thus excluding the supraoccipital from contributing to the margin of the foramen magnum (2), ORDERED* (Fig. 17). *This character is inapplicable in taxa without a discernable suture between the supraoccipital and the exoccipital or taxa with a fused opisthotic-exoccipital.*

These two characters were fused because the exclusion of the supraoccipital from the margin of the foramen magnum implies that the exoccipitals connect to each other dorsally, which is caused by an extensive dorsomedial inclination of the dorsal portions of the exoccipitals.
**Character 129 (Modified from *Ezcurra, 2016*: ch. 221).** *Exoccipital, medial margin of their distal ends: no contact with its counterpart (0); contact with its counterpart to partially or fully exclude the basioccipital from the floor of the endocranial cavity (1)* (*Ezcurra, 2016*: Fig. 27).

States 1 and 2 of character 221 in *Ezcurra (2016)* were fused here, because it is very difficult to distinguish between them in the sampled taxa.

**Character 130 (*Ezcurra, 2016*: ch. 213) (both states slightly reformulated).** *Opisthotic, paroccipital processes orientation: extended laterally or slightly posterolaterally (0); deflected strongly posterolaterally at an angle of more than 20 degrees from the transverse plane of the skull (1)* (*Ezcurra, 2016*: Fig. 16).

**Character 131 (*Pritchard et al., 2015*: ch. 58).** *Opisthotic, paroccipital process: ends freely (0); contacts the suspensorium (1).*

**Character 132 (*Ezcurra, 2016*: ch. 216).** *Opisthotic, fossa immediately lateral to the foramen magnum: absent (0); present (1).*

**Character 133 (Modified from *Ezcurra, 2016*: ch. 217).** *Opisthotic, ventral ramus shape: pyramidal, with a tapering distal end (0); club-shaped with a large bulbous distal head (1); columnar-like shaft of the ramus and an anteroposteriorly expanded but not a bulbous distal head (2); anteroposteriorly flattened shaft of the ramus, forming a blade-like ramus in lateral view and an anteroposteriorly expanded but not bulbous distal head (3)* (Fig. 18; *Ezcurra, 2016*: Fig. 28).

This character was modified to more precisely fit the morphology of the ventral ramus of the opisthotic as we observed it for the sampled taxa.

**Character 134 (*Ezcurra, 2016*: ch. 218) (state 1 slightly reformulated).** *Opisthotic, ventral ramus: extends further laterally than the lateralmost edge of the exoccipital in posterior view (0); ventral ramus completely or almost completely covered by the lateralmost edge of the exoccipital in posterior view (1)* (*Ezcurra, 2016*: Fig. 27).

**Character 135 (*Ezcurra, 2016*: ch. 223).** *Pseudolagenar recess, opening externally between the ventral surface of the ventral ramus of the opisthotic and the basal tubera: present (0); absent (1)* (*Ezcurra, 2016*: Fig. 27).

**Character 136 (Modified from *Ezcurra, 2016*: ch. 19).** *Posttemporal fenestra, size: large, roughly similar in size to the supraoccipital (0); strongly reduced in size and much smaller than the supraoccipital (1); absent or developed as a foramen or very narrow slit (2) ORDERED* (*Ezcurra, 2016*: Fig. 27).

This character has been modified according to observations of the sampled taxa. In most taxa, the posttemporal fenestra is large with little variation in its construction. However, in *Azendohsaurus madagkarensis* the parietal encloses the fenestra laterally, distinctly reducing it in size (*Flynn et al., 2010*). In *Erythrosuchus africanus* (BP/1/4680),

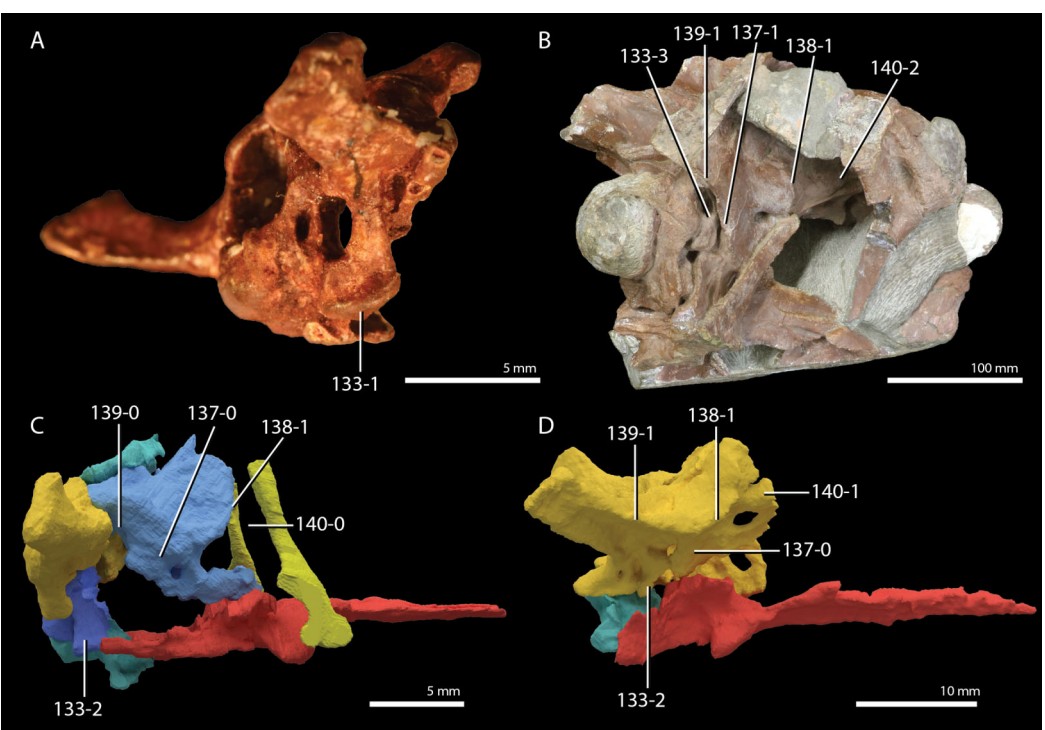

**Figure 18 Illustration of characters 133, 137, 138, 139, and 140.** (A) 133-1 in *Prolacerta broomi* (BP/1/2675, partial braincase in angled right posterolateroventral view). (B) 133-3, 137-1, 138-1, 139-1, 140-2 in *Erythrosuchus africanus* (BP/1/3893, partial braincase in angled right posterolateral view). (C) 133-2, 137-0, 138-1, 139-0, 140-0 in a digital reconstruction of *Macrocnemus bassanii* (PIMUZ T 2477, braincase right lateral view). (D) 133-2, 137-0, 138-1, 139-1, 140-1 in a digital reconstruction of *Tanystropheus hydroides* (PIMUZ T 2790, braincase right lateral view).

*Proterosuchus fergusi* (SAM-PK-K10603), and *Proterosuchus alexanderi* (NM QR 1484) the fenestra is only represented by a very narrow slit or foramen.

**Character 137 (*Ezcurra, 2016*: ch. 254) (reformulated).** *Prootic, a clear crest on the lateral surface that is roughly orientated posterodorsally to anteroventrally (=crista prootica): absent (0); present (1)* (Fig. 18; *Ezcurra, 2016*: Fig. 28).

**Character 138 (New).** *Prootic, a clear crest along the lateral surface that curves dorsally at the anterior margin of the prootic (=crista alaris): absent (0); present (1)* (Fig. 18).

Character 254 in *Ezcurra (2016)* addressed the presence of crista prootica in a phylogenetic context. However, the presence of another crest on the lateral surface of the prootic, the crista alaris, is also variable for the sampled taxa, and this is addressed with this newly formulated character for the first time.

**Character 139 (*Pritchard et al., 2015*: ch. 75) (reformulated).** *Prootic, paroccipital contribution: prootic does not contribute to the anterior surface of paroccipital process (0);*

*prootic contributes laterally tapering lamina to the anterior surface of the paroccipital process (1)* (Fig. 18).

**Character 140 (Modified from *Ezcurra, 2016*: ch. 258 and *Pritchard et al., 2015*: ch. 72).** *Laterosphenoid, ossification: absent (0); present, laterosphenoid is a narrow dorsoventrally orientated bone and lacks an anterior portion (1); present, laterosphenoid with an anterior portion located along the ventral surface of the parietals and frontals (2)* (Fig. 18; *Ezcurra, 2016*: Fig. 28), *ORDERED.*

The presence of a laterosphenoid was until recently not known for non-archosauriform archosauromorphs. However, a laterosphenoid has now been identified in *Azendohsaurus madagskarensis* (*Flynn et al., 2010*) and *Tanystropheus hydroides* (*Spiekman et al., 2020a*, *2020b*). In these taxa, the laterosphenoid is small and does not extend far anteriorly as in archosauriforms. The small, unexpanded laterosphenoid is considered to represent an intermediate step between the absence of a laterosphenoid and the larger, further anterior reaching laterosphenoid of archosauriforms, and therefore the character has been ordered.

**Character 141 (*Ezcurra, 2016*: ch. 296).** *Stapes, shape: robust, with thick shaft (0); slender, rod-like shaft (1).*

**Character 142 (*Ezcurra, 2016*: ch. 297 and *Pritchard et al., 2015*: ch. 77).** *Stapes, stapedial foramen piercing the columellar process: present (0); absent (1).*

**Character 143 (*Simões et al., 2018*: ch. 176).** *Splenial: present (0); absent (1).*

**Character 144 (Modified from *Ezcurra, 2016*: ch. 266).** *Dentary, height at the third alveolus of the bone (or directly posterior to the tapering anterior end of the dentary in taxa with an anteriorly edentulous dentary) versus length of the alveolar margin (including edentulous anterior end if present): 0.02–0.04 (0); 0.06–0.11 (1): 0.15–0.24 (2); 0.27–0.29 (3), ORDERED RATIO* (*Ezcurra, 2016*: Figs. 17 and 18).

Instead of comparing the length of the alveolar margin of the dentary to the minimum height of the dentary, it was here considered to compare it to the height of the dentary at the third alveolus, as this represents a more consistent measurement across the sampled taxa.

**Character 145 (*Ezcurra, 2016*: ch. 267) (reformulated).** *Dentary, shape of the tooth bearing portion (including edentulous anterior end if present): roughly straight (0); dorsally curved for all or most of its anteroposterior length (1); ventrally curved or deflected at its anterior end (2)* (*Ezcurra, 2016*: Figs. 17 and 29).

**Character 146 (New).** *Dentary, distinct dorsoventral expansion forming a keel at the anterior end of the dentary: absent (0); present (1)* (Fig. 19).

State 1 represents an autapomorphy for *Tanystropheus hydroides* (PIMUZ T 2790) among the sampled taxa.

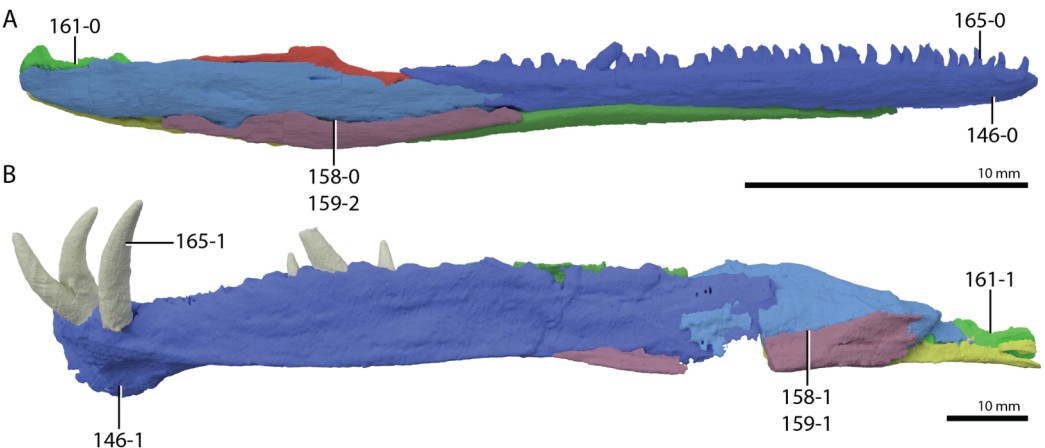

**Figure 19 Illustration of characters 146, 158, 159, 161, and 165.** (A) 146-0, 158-0, 159-2, 161-0, and 165-0 in a digital reconstruction of *Macrocnemus bassanii* (PIMUZ T 2477, right lower jaw in lateral view). (B) 146-1, 158-1, 159-1, 161-1, and 165-1 in a digital reconstruction of *Tanystropheus hydroides* (PIMUZ T 2790, left lower jaw in lateral view).

**Character 147 (*Ezcurra, 2016*: ch. 270).** *Dentary, position of the Meckelian groove on the anterior half of the bone: dorsoventral centre of the dentary (0); restricted to the ventral border (1)* (*Nesbitt, 2011*: Fig. 27).

**Character 148 (*Ezcurra, 2016*: ch. 272).** *Dentary, posterodorsal process, in which its dorsal margin is confluent with the dorsal margin of the lower jaw: absent (0); present (1)* (*Ezcurra, 2016*: Figs. 17 and 29).

**Character 149 (*Ezcurra, 2016*: ch. 273).** *Dentary, posterocentral process, in which its margins are not confluent with the dorsal or ventral margin of the lower jaw: absent (0); present (1)* (*Ezcurra, 2016*: Figs. 17 and 29).

**Character 150 (Modified from *Ezcurra, 2016*: ch. 275).** *Dentary, posteroventral process, in which its ventral margin is confluent with the ventral margin of the lower jaw: absent (0); present (1)* (*Ezcurra, 2016*: Figs. 17 and 29).

Character state 2 of character 275 in *Ezcurra (2016)* was omitted here because most of the included taxa here do not bear an external mandibular fenestra.

**Character 151 (*Ezcurra, 2016*: ch. 276).** *Dentary, posteroventral process length: extended posteriorly to the level of the posterodorsal and/or posterocentral processes (0); extended posteriorly beyond the level of the posterodorsal and/or posterocentral processes (1)* (*Ezcurra, 2016*: Fig. 29). *This character is inapplicable in taxa that lack a posteroventral process in the dentary.*

**Character 152 (*Ezcurra, 2016*: ch. 262 and *Pritchard et al., 2015*: ch. 84).** *Lower jaw, external mandibular fenestra: absent (0); present (1)* (*Ezcurra, 2016*: Figs. 17 and 29).

**Character 153 (New, combination of part of *Ezcurra, 2016*: ch. 261 and *Pritchard et al., 2018*: ch 319).** *Lower jaw, distinct dorsal process behind the alveolar margin (=coronoid*
process): absent, with only a slightly convex dorsal margin present behind the alveolar portion (0); present but low, not protruding dorsally behind the anterior process of the jugal in lateral view (1); present and tall, protruding dorsally behind the anterior process of the jugal in lateral view (2) (*Ezcurra, 2016*: Fig. 29), *ORDERED*.

Both characters 261 of *Ezcurra (2016)* and 319 of *Pritchard et al. (2018)* were considered to be informative but strongly related to each other and they were therefore combined here. It is not considered which bone forms the coronoid process because this often is hard to establish confidently in the sampled taxa. Furthermore, this information is strongly interdependent with the subsequent character (154).

**Character 154 (New).** *Separate coronoid bone: present (0); absent (1).*

Although it has been previously established that several archosauromorphs lack a separate coronoid bone, this has not been coded as a character in phylogenetic analyses until now.

**Character 155 (Modified from *Ezcurra, 2016*: ch. 286).** *Surangular, lateral shelf: absent (0); present, low ridge near dorsal margin (1); present, laterally or ventrolaterally projecting shelf with a lateral edge (2)* (*Ezcurra, 2016*: Figs. 18 and 29).

States 2 and 3 of character 286 in *Ezcurra (2016)* were combined here because this distinction was considered to be somewhat subjective and not of relevance for the sampled taxa.

**Character 156 (*Ezcurra, 2016*: ch. 288 and *Pritchard et al., 2015*: ch. 80).** *Surangular, anterior surangular foramen on the lateral surface of the bone, near surangular-dentary contact: absent (0); present (1)* (*Ezcurra, 2016*: Fig. 29).

**Character 157 (*Ezcurra, 2016*: ch. 289 and *Pritchard et al., 2015*: ch. 81).** *Surangular, posterior surangular foramen on the lateral surface of the bone, positioned directly anterolateral to the glenoid fossa: absent (0); present (1)* (*Ezcurra, 2016*: Fig. 29).

**Character 158 (Modified from *Ezcurra, 2016*: ch. 282).** *Surangular-angular, suture along the anterior half of the bones in lateral view: anteroposteriorly convex ventrally (0); roughly straight (1); anteroposteriorly concave ventrally (2)* (Fig. 19; *Ezcurra, 2016*: Fig. 29) *ORDERED*.

In *Tanystropheus hydroides* (PIMUZ T 2790), *Trilophosaurus buettneri* (*Spielmann et al., 2008*), and *Orovenator mayorum* (*Ford & Benson, 2018*) the surangular-angular suture is neither convex nor concave but straight, which was therefore included as a separate character state here. A straight suture is considered an intermediate step in a transformational series from concave to convex and the character has therefore been ordered.

**Character 159 (Modified from *Ezcurra, 2016*: ch. 290 and *Pritchard et al., 2015*: ch. 82).** *Angular, dorsoventral exposure on the lateral surface of the lower jaw: wide (0); forming*

*about half of the dorsoventral height of the mandible at its greatest width (1); narrow (2)* (Fig. 19; *Ezcurra, 2016*: Fig. 29) ORDERED.

In *Tanystropheus hydroides* (PIMUZ T 2790), *Tanystropheus longobardicus* (PIMUZ T 2484), *Azendohsaurus madagaskarensis* (*Flynn et al., 2010*), *Proterosuchus fergusi* (SAM-PK-11208), and *Proterosuchus alexanderi* (NM QR 1484) the angular covers approximately half of the lateral surface of the mandible posteriorly, which was therefore included as separate character state here. This exposure is considered an intermediate step in a transformational series from a very wide to a very narrow exposure and the character has therefore been ordered.

**Character 160 (*Pritchard et al., 2015*: ch. 83) (state 0 reformulated).** *Angular, exposure on lateral mandibular surface: terminates distinctly anterior to the glenoid (0); extends to the glenoid (1).*

**Character 161 (Modified from *Senter, 2004*: ch. 16).** *Location of glenoid fossa compared to the tooth row of the dentary: roughly at the same dorsoventral level as the tooth row (0); considerably ventrally displaced compared to the tooth row (1)* (Fig. 19).

In several archosauromorphs (*Tanystropheus hydroides*, PIMUZ T 2790; *Tanystropheus longobardicus*, PIMUZ T 2482; *Tanytrachelos ahynis*, YPM 7496a; *Pectodens zhenyuensis*, IVPP V18578; *Dinocephalosaurus orientalis*, IVPP V13767; and *Azendohsaurus madagaskarensis*, *Flynn et al., 2010*) and in *Gephyrosaurus bridensis* (*Evans, 1980*), the glenoid fossa is located distinctly ventrally compared to the dentary tooth row. This character was first employed by *Senter (2004)*.

**Character 162 (*Ezcurra, 2016*: ch. 283).** *Articular, retroarticular process: absent (0); anteroposteriorly short, being poorly developed posterior to the glenoid fossa (1); anteroposteriorly long, extending considerably posterior to the glenoid fossa (2) ORDERED* (*Ezcurra, 2016*: Figs. 17 and 29).

**Character 163 (*Ezcurra, 2016*: ch. 284).** *Articular, retroarticular process: not upturned (0); upturned (1)* (*Ezcurra, 2016*: Figs. 17 and 29). *This character is scored as inapplicable in taxa that lack a retroarticular process.*

**Character 164 (*Pritchard et al., 2015*: ch. 92).** *Marginal dentition, arrangement: single row of marginal teeth (0); multiple Zahnreihen in maxilla and dentary (1).*

Characters 73 and 279 in *Ezcurra (2016)* treat the number of tooth rows on the upper and lower jaws separately. We consider these characters to be strongly interdependent for the sampled taxa and therefore prefer to treat both jaws for this feature in one character here.

**Character 165 (New).** *Marginal dentition, anterior teeth are interlocking fangs forming a fish-trap* sensu Rieppel (2002): *absent (0); present (1)* (Fig. 19). *This character is inapplicable in taxa with an edentulous premaxilla.*

In *Tanystropheus hydroides* (PIMUZ T 2790), *Tanystropheus longobardicus* (MSNM BES SC 1018), and *Dinocephalosaurus orientalis* (IVPP V13767) the anterior marginal dentition is fang-like and elongate. These teeth interlock to form a "fish-trap" type dentition.

**Character 166 (Modified from *Ezcurra, 2016*: ch. 280).** *Marginal dentition, occlusion of marginal teeth: single-sided overlap (excluding potentially present interlocking fish-trap dentition anteriorly) (0); flat occlusion (1); teeth interlocking tightly (2)* (*Ezcurra, 2016*: Fig. 14). *This character is inapplicable in taxa in which multiple tooth rows are present on the marginal dentition.*

The character states were modified to more specifically address the morphologies observed in the sampled taxa.

**Character 167 (*Ezcurra, 2016*: ch. 298).** *Marginal dentition, posterior extent of mandibular and maxillary tooth rows: subequal (0); maxillary teeth extending further posteriorly (1).*

**Character 168 (*Ezcurra, 2016*: ch. 277).** *Marginal dentition, posteriormost dentary teeth: on the anterior half of lower jaw (0); on the posterior half of lower jaw (1)* (*Ezcurra, 2016*: Fig. 17).

**Character 169 (*Ezcurra, 2016*: ch 299).** *Marginal dentition, tooth implantation: subthecodont (=protothecodont) (0); ankylothecodont (teeth fused to the bone at the base of the crown by bony ridges and the root can be discerned; there is continuous tooth replacement) (1); pleurodont (2); acrodont (teeth fused to the bone in adults so that no root can be discerned) (3); thecodont (4)* (*Ezcurra, 2016*: Figs. 12, 14 and 22).

**Character 170 (*Ezcurra, 2016*: ch. 308).** *Marginal dentition, multiple maxillary and dentary tooth crowns distinctly mesiodistally expanded above the root: absent (0); present (1)* (*Ezcurra, 2016*: Fig. 14).

**Character 171 (Modified from *Ezcurra, 2016*: ch. 303).** *Marginal dentition, maxillary teeth: straight or very slightly recurved (0); distinctly recurved (1)* (*Ezcurra, 2016*: Fig. 14). *This character is not applicable in taxa with maxillary teeth that expand above the root or that possess multiple tooth rows in the maxilla.*

Certain taxa have very slightly recurved teeth (e.g., *Petrolacosaurus kansensis*, *Czatkowiella harae*, and *Orovenator mayorum*). However, we choose not to maintain a separate character state for this morphology as in these taxa not all teeth are recurved and many are straight, therefore forming a very minimal distinction from the straight morphology. Only taxa in which the curvature of the teeth is distinct are scored as 1.

**Character 172 (*Ezcurra, 2016*: ch. 304).** *Marginal dentition, serrations on the maxillary/dentary crowns: absent (0); distinctly present on the distal margin and usually apically*

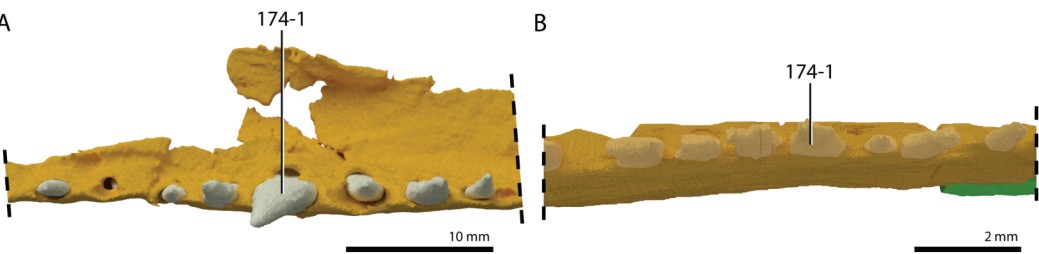

**Figure 20 Illustration of character 174.** (A) State 1 in a digital reconstruction of *Tanystropheus hydroides* (PIMUZ T 2790, part of left maxilla in ventral view). (B) State 1 in a digital reconstruction of *Macrocnemus bassanii* (PIMUZ T 2477, part of left maxilla in ventral view; the colour of the teeth in (B) has been slightly modified in Adobe Illustrator CS6 to distinguish them from the maxilla).

*restricted, low or absent on the mesial margin (1); present and distinct on both margins (2)* (*Ezcurra, 2016*: Fig. 14).

**Character 173 (*Ezcurra, 2016*: ch. 306).** *Marginal dentition, multiple maxillary or dentary tooth crowns with longitudinal labial or lingual striations or grooves: absent (0); present (1)* (*Ezcurra, 2016*: Fig. 14).

**Character 174 (Modified from *Pritchard et al., 2015*: ch. 98).** *Marginal dentition, tooth shape at crown base of the maxillary teeth: circular (0); labiolingually compressed (1); labiolingually wider than mesiodistally long (2)* (Fig. 20).

This character is only scored for the maxillary teeth because certain taxa exhibit a heterodont dentition, for instance in the form of large fang-like teeth on the premaxilla and anterior portion of the dentary (e.g., *Tanystropheus hydroides*, PIMUZ T 2790; and *Dinocephalosaurus orientalis*, IVPP V13767). The presence of marginal teeth that are oval in cross-section has widely been considered an important character that is typically diagnostic for Archosauriformes (e.g., *Dilkes, 1998*, *Jalil, 1997*, *Pritchard et al., 2015*), since it occurs widely in archosauriforms and only rarely in non-archosauriform diapsids (*Ezcurra, 2016*). However, we found that the condition in which marginal teeth are labiolingually narrower than mesiodistally wide is more widespread than previously considered, occurring, among others, in *Tanystropheus hydroides* (PIMUZ T 2790), *Tanystropheus longobardicus* (PIMUZ T 3901), *Macrocnemus bassanii* (PIMUZ T 2477), *Macrocnemus fuyuanensis* (PIMUZ T 1559), *Langobardisaurus pandolfii* (MFSN 1921), and *Dinocephalosaurus orientalis* (IVPP V13767). This morphology is distinct from the virtually circular cross-sections of teeth present in for instance *Czatkowiella harae* (ZPAL RV/100, Fig. 4 of *Borsuk-Białynicka & Evans, 2009b*).

**Character 175 (Modified from *Pritchard et al., 2015*: ch. 93).** *Marginal dentition, morphology of crown base: all tooth crowns form a single, pointed or rounded crown (0); at least some tooth crowns form a flattened platform with pointed cusps (1); at least some tooth crowns have three, mesiodistally arranged cusps (2).*

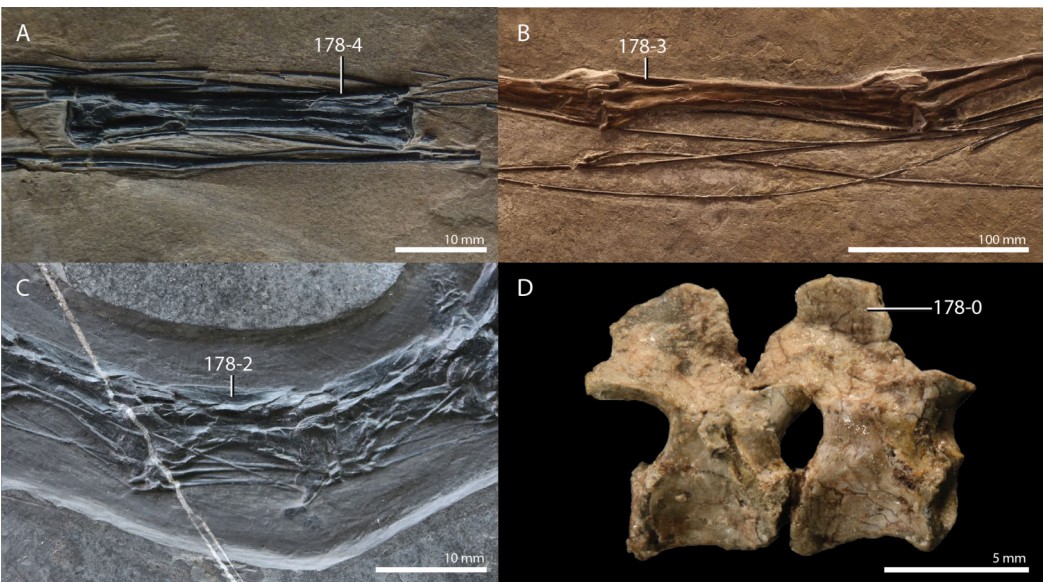

**Figure 21** **Illustration of character 178.** (A) State 4 in *Sclerostropheus fossai* (MCSNB 4035, mid-cervical vertebra in right lateral view). (B) State 3 in *Tanystropheus hydroides* (PIMUZ T 2819, mid-cervical vertebrae in left lateral view). (C) State 2 in *Fuyuansaurus acutirostris* (IVPP V17983, mid-cervical vertebrae in left lateral view). (D) State 0 in *Youngina capensis* (BP/1/3859, anterior cervical vertebrae right lateral view).

The character states were modified to more specifically address the morphologies observed in the sampled taxa.

**Character 176 (*Ezcurra, 2016*: ch. 310).** *Cervical, dorsal, sacral and caudal vertebrae, notochordal canal piercing the centrum: present throughout ontogeny (0); absent in adults (1)* (*Ezcurra, 2016*: Fig. 31).

**Character 177 (*Ezcurra, 2016*: ch. 313).** *Presacral vertebrae, at least one or more cervical or anterior dorsal with parallelogram-shaped centra in lateral view, in which the anterior articular surface is situated higher than the posterior one: absent (0); present (1)* (*Ezcurra, 2016*: Figs. 11 and 33).

**Character 178 (New, combination of ch. 342 and ch. 344 of *Ezcurra, 2016*).** *Cervical vertebrae, maximum height of postaxial anterior or mid-cervical neural spines: considerably taller than the posterior articular surface of the centrum (0); approximately equally tall as the posterior articular surface of the centrum (1); considerably shorter than the posterior articular surface of the centrum (2); low neural spines are only present at the anterior and posterior ends of the vertebrae but are completely or virtually lost at their anteroposterior midpoints (3); neural spine is completely reduced or lost (4)* (Fig. 21; *Ezcurra, 2016*: Fig. 11), *ORDERED.*

Characters 342 and 344 in *Ezcurra (2016)* addressed the height of the neural spine in the postaxial cervical vertebrae, which is a variable and phylogenetically important trait among tanystropheids. We have combined the information of these two characters, because we

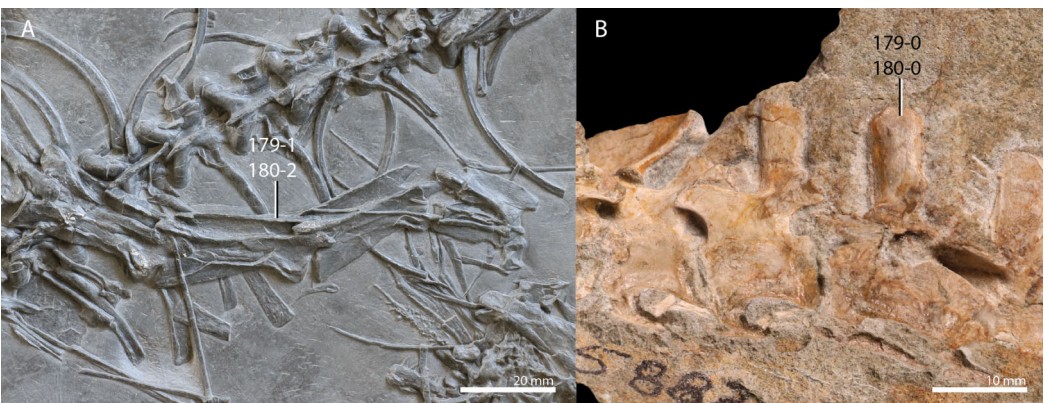

**Figure 22 Illustration of characters 179 and 180.** (A) 179-1 and 180-2 in *Macrocnemus fuyuanensis* (IVPP V15001, anterior cervical vertebrae in right lateral view). (B) 179-0 and 180-0 in *Mesosuchus browni* (SAM-PK-5882, anterior cervical vertebrae in left lateral view).

considered them to be interdependent, and modified the states distinctly to address the specific morphologies observed in the sampled taxa.

**Character 179 (New).** *Cervical vertebrae, shape of distal margin of anterior and mid-cervical postaxial neural spines in lateral view: slightly convex (0); completely straight along anteroposterior length (1); concave (2)* (Fig. 22). *This character is inapplicable in taxa that have reduced the neural spine of their anterior and mid-cervical vertebrae completely or at their anteroposterior midpoint.*

We find that in several taxa the distal margin of the neural spine of the anterior to mid-cervical vertebrae is completely straight along its entire anteroposterior length (e.g., *Macrocnemus bassanii*, PIMUZ T 4822; and *Pamelaria dolichotrachela*, ISIR 316/1). This straight margin often, but not always, occurs together with a distally expanded neural spine (=spine table). However, due to both structures also occurring without the presence of the other, they were scored here as separate characters. Furthermore, the distal margin of the neural spines of certain taxa are conspicuously concave (particularly in *Dinocephalosaurus orientalis*, *Rieppel, Li & Fraser, 2008*) and this was considered as a separate character state here.

**Character 180 (New, combination of ch. 320 and ch. 321 of *Ezcurra, 2016*).** *Cervical vertebrae, distal expansion of the anterior to mid-postaxial cervical neural spines (not mammillary process): absent (0); present, gradual transverse expansion of the distal half of the neural spine (1); present, but transverse expansion is restricted to the distal end of the neural spine (=spine table) (2)* (Fig. 22). *This character is inapplicable in taxa that have reduced the neural spine of their anterior and mid-cervical vertebrae completely.*

A distal expansion of the postaxial neural spines was previously addressed by character 117 in *Pritchard et al. (2015)* and characters 320 and 321 in *Ezcurra (2016)*. Here, we combined information from these characters to form a new character that addresses the variation

seen in this trait in the sampled taxa. We consider the gradual transverse expansion to represent a separate state from the presence of a spine table, following *Ezcurra (2016)*. However, since a gradual expansion and a distinct spine table both address a widening of the neural spine, which is separate from the presence of mammillary processes, we consider them part of the same morphological character. This character should only be scored in skeletally mature specimens since a transverse expansion of the neural spine is generally absent in early ontogenetic stages.

**Character 181 (Modified from *Simões et al., 2018*: ch. 228).** *Presacral vertebrae, type of articular surface: opisthocoelous (0); procoelous (1); amphicoelous (2); acoelous (3). This character is inapplicable in taxa that have a notochordal canal running through their centra.*

The articulation surfaces of the centra of presacral vertebrae was previously considered by characters 101 and 102 in *Pritchard et al. (2015)*, which considered the anterior and posterior surfaces separately. We follow *Simões et al. (2018)* and treat the articulation surfaces of the centra as a single character.

**Character 182 (*Nesbitt, 2011*: ch. 177).** *Presacral vertebrae, postaxial intercentra: present (0); absent (1).*

The presence of intercentra was scored separately for postaxial cervical vertebrae and dorsal vertebra in characters 346 and 366 in *Ezcurra (2016)*. However, we score the presence or absence of postaxial intercentra as a single character since in most cases the presence of intercentra often occurs in both segments of the vertebral column in the sampled taxa. Therefore, separating these segments results in overscoring of the presence of postaxial intercentra.

**Character 183 (*Ezcurra, 2016*: ch. 326 and *Nesbitt et al., 2015*: ch. 243).** *Cervical vertebrae, centrum of atlas in skeletally mature individuals: separate from axial intercentrum (0); fused to axial intercentrum (1) (Ezcurra, 2016*: Fig. 30).

**Character 184 (New).** *Cervical vertebrae, proatlas elements dorsal to atlantal neural arches: present (0); absent or fused with atlantal neural arch (1)* (Fig. 23).

No proatlases are present in *Tanystropheus hydroides* (PIMUZ T 2790), in contrast to all other sampled taxa for which this character could be scored.

**Character 185 (Modified from *Ezcurra, 2016*: ch. 328).** *Cervical vertebrae, height of the neural spine of the axis: ratio between the maximum height of the neural spine and the posterior articular surface height of the centrum of the axis: 0.46-0.59 (0); 0.76-1.21 (1): 1.33-1.66 (2); 2.06-2.23 (3), ORDERED RATIO (Ezcurra, 2016*: Fig. 30).

The distinction between the states of character 328 of *Ezcurra (2016)* is considered to be ambiguous for the taxa sampled here and we have instead modified the states into ratios.

**Character 186 (*Ezcurra, 2016*: ch. 329) (slightly reformulated).** *Cervical vertebrae, shape of the neural spine of the axis in lateral view: expanded posterodorsally or the height of the*
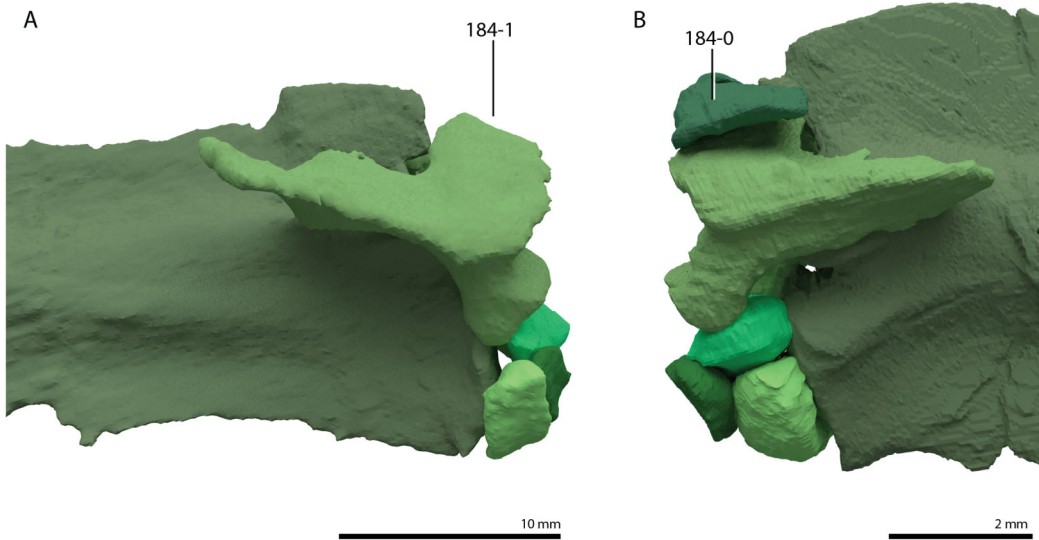

**Figure 23 Illustration of character 184.** (A) State 1 in a digital reconstruction of *Tanystropheus hydroides* (PIMUZ T 2790, atlas-axis complex in right lateral view). (B) State 0 in a digital reconstruction of *Macrocnemus bassanii* (PIMUZ T 2477, atlas-axis complex in left lateral view).

anterior portion is equivalent to the posterior height (0); expanded anterodorsally (1) (*Ezcurra, 2016*: Fig. 30).

**Character 187 (*Ezcurra, 2016*: ch. 331).** *Cervical vertebrae, lengths of the fourth or fifth cervical centra versus the heights of their anterior articular surfaces: 0.63-5.06 (0); 6.33-12.12 (1); 14.58-15.58 (2); 17.08-18.67 (3); 20.05-20.51 (4), ORDERED RATIO* (*Ezcurra, 2016*: Fig. 15).

**Character 188 (*Ezcurra, 2016*: ch. 332).** *Cervical vertebrae, diapophysis and parapophysis of anterior to mid-cervical postaxial vertebrae: single facet or both situated on the same process (0); situated on different processes and well-separated (1); situated on different processes and nearly touching (2)* (*Ezcurra, 2016*: Fig. 30).

**Character 189 (Modified from *Ezcurra, 2016*: ch. 340).** *Cervical vertebrae, laminae extending posteriorly from the base of the dia –and/or parapophysis in anterior and mid-postaxial cervical vertebrae: absent (0); present (1)* (*Ezcurra, 2016*: Fig. 30).

Laminae project from the base of the dia –and/or parapophysis in most of the sampled taxa, except for *Youngina capensis* (BP/1/3859), *Erythrosuchus africanus* (BP/1/5207), *Mesosuchus browni* (SAM-PK-5882), and *Planocephalosaurus robinsonae* (*Fraser & Walkden, 1984*).

**Character 190 (*Ezcurra, 2016*: ch. 336).** *Cervical vertebrae, epipophysis in postaxial cervical vertebrae: absent (0); present in at least the third to fifth cervical vertebrae (1)* (*Ezcurra, 2016*: Figs. 30 and 33).
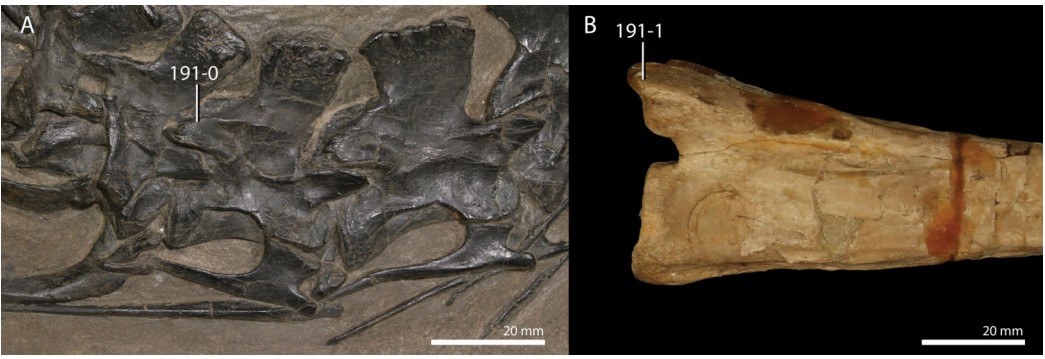

**Figure 24** **Illustration of character 191.** (A) State 0 in *Protorosaurus speneri* (WMsN P 47361, posterior cervical vertebrae in right lateral view). (B) State 1 in *Tanystropheus "conspicuus"* (U-MO BT 733, posterior part of mid-cervical vertebra in right lateral view).

**Character 191 (Modified from *Pritchard et al., 2018*: ch. 271).** *Cervical vertebrae, posterior extension of epipophysis: not extended posterior to the postzygapophysis (0); overhanging the postzygapophysis posteriorly (1)* (Fig. 24). *This character is inapplicable in taxa that lack epipophyses on their cervical vertebrae.*

In all tanystropheids except for certain specimens of *Tanystropheus "conspicuus"* (U-MO BT 740), *Sclerostropheus fossai* (MCSNB 4035), *Macrocnemus fuyuanensis* (IVPP V15001), and *Langobardisaurus pandolfii* (MCSNB 2883) the epipophyses are well-developed and extend posteriorly beyond the level of the postzygapophyses. In all other sampled taxa that bear epipophyses they are not extended as far posteriorly. The character was modified from character 271 of *Pritchard et al. (2018)* because the distinction between states 1 and 2 therein was difficult to distinguish confidently in the sampled taxa.

**Character 192 (Modified from *Ezcurra, 2016*: ch. 338 and *Nesbitt et al., 2015*: ch. 213).** *Cervical vertebrae, anterior cervical vertebrae (presacral vertebrae 3-5) postzygapophyses: postzygapophyseal trough (*sensu Rieppel, 2001*) formed by a well-developed posteriorly extending shelf (=transpostzygapophyseal lamina) that in some cases bears a notch on its posterior end: absent (0); present (1)* (Fig. 25).

This character was modified based on detailed observations of the vertebrae of *Tanystropheus* spp.

**Character 193 (*Pritchard et al., 2015*: ch. 113).** *Cervical vertebrae, neural spine base of anterior postaxial cervical vertebrae: anteroposteriorly elongate, subequal in length to the neural arch (0); anteroposteriorly shortened, spine restricted to posterior half of neural arch (1). This character is inapplicable in taxa that have completely reduced the neural spine of their anterior and mid-cervical vertebrae.*

The inapplicability criterion was added and the formulation of the character has been modified slightly.

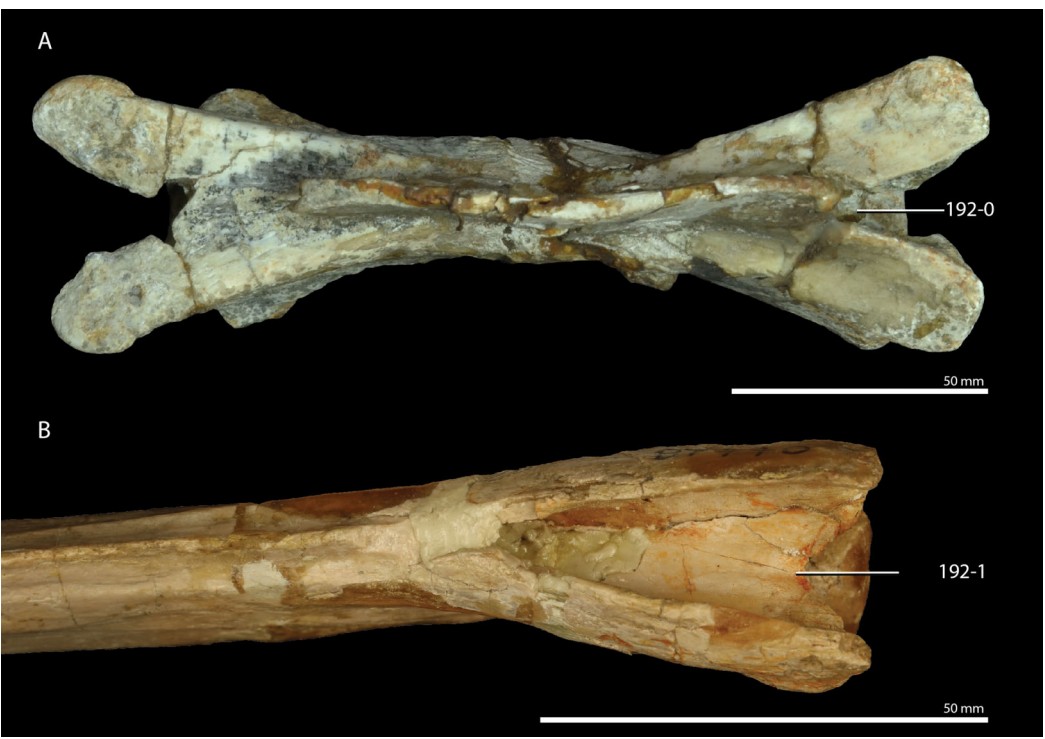

**Figure 25 Illustration of character 192.** (A) State 0 in *Pamelaria dolichotrachela* (ISIR 316, anterior cervical vertebra in dorsal view). (B) State 1 in *Tanystropheus "conspicuus"* (U-MO BT 740, posterior part of mid-cervical vertebra in dorsal view). Image of *Pamelaria dolichotrachela* courtesy of Martín Ezcurra.

**Character 194 (New, combination of *Ezcurra, 2016*: ch. 343 and *Pritchard et al., 2015*: ch. 116).** *Cervical vertebrae, orientation of the anterior margin of the neural spine of anterior and mid-postaxial cervical vertebrae: straight or posterodorsally inclined (0); anterodorsally inclined at an angle of more than 60 degrees from the horizontal plane (1); anterodorsally inclined at an angle of less than 60 degrees from the horizontal plane (2)* (*Ezcurra, 2016*: Figs. 30 and 33), *ORDERED. This character is inapplicable in taxa that have completely reduced the neural spine of their anterior and mid-cervical vertebrae.*

The notch referred to in character 115 of *Pritchard et al. (2015)* was reinterpreted as an anterior overhang or inclination in character 343 of *Ezcurra (2016)*. Here, this inclination is considered to represent a similar morphology as the inclination described by character 116 of *Pritchard et al. (2015)* and therefore these characters were fused here. The degree of an anterodorsal inclination of the anterior margin of the neural spine in the anterior to mid-postaxial cervical vertebrae is strongly variable among the sampled taxa, and therefore we distinguish between two clearly demarcated states. State 1 represents an intermediate morphology between states 0 and 2, and the character was therefore ordered. This character is scored as ? for *Tanystropheus hydroides*, *Tanystropheus longobardicus*, and *Tanystropheus "conspicuus"*. In these taxa the anterior margin of the neural spine is complex as it is bifurcated and therefore does not allow for a confident scoring of this character (see Fig. 57 of *Nosotti, 2007*).

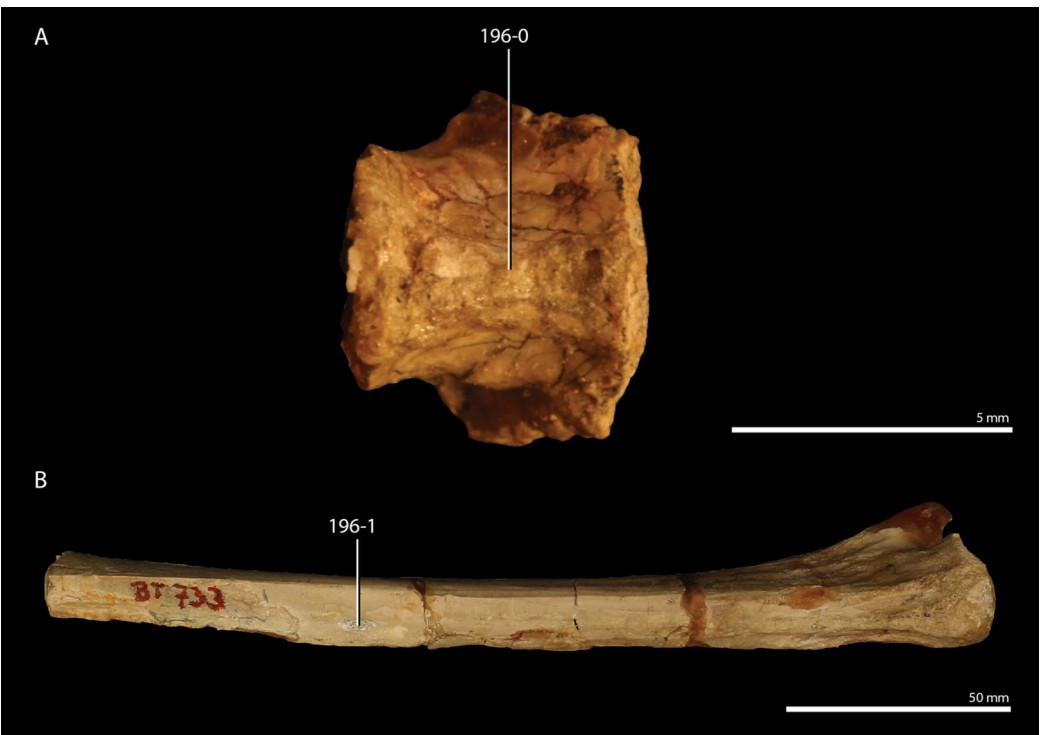

**Figure 26 Illustration of character 196.** (A) State 0 in *Youngina capensis* (BP/1/3859, anterior cervical vertebra in ventral view). (B) State 1 in *Tanystropheus "conspicuus"* (U-MO BT 733, partial cervical vertebra in ventral view).    

**Character 195 (Modified from *Ezcurra, 2016*: ch. 324).** *Cervical vertebrae, total number: six or fewer (0); between seven and 10 (1); between 11 and 13 (2); more than 13 (3), ORDERED.*

The states were modified based on the distribution of the number of cervical vertebrae in the sampled taxa.

**Character 196 (New).** *Cervical vertebrae, presence of a foramen on the ventral surface of the centrum around the anteroposterior midpoint: absent (0); present (1)* (Fig. 26).

A conspicuous nutrient foramen (foramina venae vertebralis *sensu Wild, 1973*) is present on the ventral surface of several cervical vertebrae of *Tanystropheus "conspicuus"* (*Spiekman & Scheyer, 2019*) and *Gephyrosaurus bridensis* (*Evans, 1981*). This foramen is absent in all other taxa for which this character could be assessed.

**Character 197 (*Pritchard et al., 2015*: ch. 109) (reformulated).** *Cervical vertebrae, anterior to mid-postaxial cervical vertebrae, shape of ventral surface in the coronal plane excluding keel: rounded or curved (0); ventral face flattened (1).*

**Character 198 (New).** *Cervical vertebrae, neural canal of anterior to mid-cervical vertebrae separated from vertebral centrum (0); neural canal enters into a cavity of the vertebral centrum (1)* (*Spiekman et al., 2020b*: Fig. 32).

In the tanystropheids *Macrocnemus bassanii*, *Tanytrachelos ahynis*, and *Tanystropheus* spp. the neural canal of the anterior to mid-cervical vertebrae enters the vertebral centrum. This morphology was first described for *Tanystropheus "conspicuus"* by *Edinger (1924)* and has recently been identified for several other tanystropheids through micro computed tomography. Although this character has so far not be examined for most taxa, it might represent a widespread feature among tanystropheids.

**Character 199 (*Ezcurra, 2016*: ch. 349) (state 2 reformulated).** *Cervical ribs, shape: short, being less than two times the length of its respective vertebra, and tapering at a high angle to the neck (0); short, being less than two times the length of its respective vertebra, and shaft parallel to the neck (1); very long, at least some ribs being more than two times the length of their respective vertebra, and shaft parallel to the neck (2)* (Fig. 27; *Nesbitt, 2011*: Figs. 28 and 30).

**Character 200 (Modified from *Ezcurra, 2016*: ch. 350 and *Pritchard et al., 2015*: ch. 105).** *Cervical ribs, anterior free-ending process (=accessory process) on anterior surface of anterior cervical ribs: absent (0); present and short, not reaching anterior to the prezygapophyses of the corresponding vertebra when in articulation (1); present and long, extending anterior to the prezygapophyses of the corresponding vertebra when in articulation (2)* (Fig. 27; *Ezcurra, 2016*: Fig. 30), *ORDERED*.

The anterior free-ending process of the cervical ribs in certain non-crocopodan archosauromorphs (*Czatkowiella harae*, ZPAL RV/937; *Sclerostropheus fossai*, MCSNB 4035; *Tanytrachelos ahynis*, VMNH 120346a; *Pectodens zhenyuensis*, IVPP V18578; *Dinocephalosaurus orientalis*, *Rieppel, Li & Fraser, 2008*) is particularly elongate and extends distinctly anterior to the corresponding vertebra. This represents a clearly separate morphology from the shorter processes seen in most archosauromorphs, and is therefore treated as a new, separate character state. The short processes are considered an intermediate morphology in a transformational series between the absence of the process and the elongate processes, and therefore the character has been ordered.

**Character 201 (Modified from *Ezcurra, 2016*: ch. 320).** *Presacral vertebrae, mammillary processes* (sensu *Ezcurra & Butler, 2015b*) *occurring in the posterior cervical to mid-dorsal vertebrae: absent (0); present (1)* (*Ezcurra, 2016*: Figs. 31, 32 and 34).

We follow the description of *Ezcurra & Butler (2015b)* for our identification of mammillary processes. Therefore, we differentiate mammillary processes from a transverse expansion of the neural spine (=spine table) by the presence of a longitudinal cleft between the process and the dorsal margin of the spine in the former, which results in a neural spine with three separate projections on its distal end rather than a single flattened surface, as in the latter. The presence of mammillary processes is considered to preclude the possibility of a distally expanded neural spine in the anterior to mid-dorsal vertebrae, since an expansion is already formed by the mammillary processes. This character should only be scored in skeletally mature specimens since mammillary processes are generally not yet developed in early ontogenetic stages.

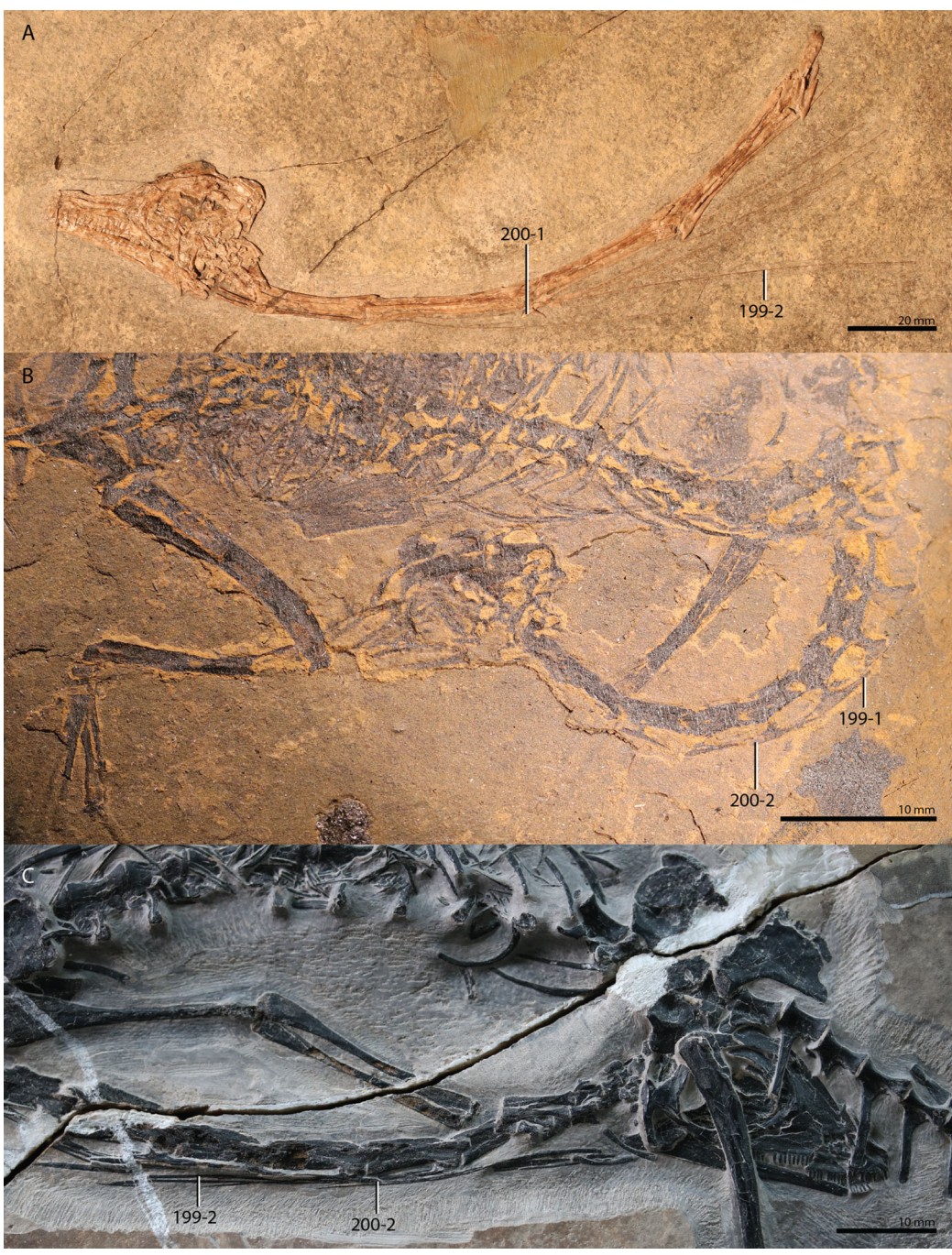

**Figure 27** **Illustration of characters 199 and 200.** (A) 199-2 and 200-1 in *Tanystropheus longobardicus* (PIMUZ T 3901, skull and partial cervical column in left lateral view). (B) 199-1 and 200-2 in *Tany-trachelos ahynis* (VMNH 120346a, partial skeleton including cervical column, cervical column in left lateral view). (C) 199-2 and 200-2 in *Pectodens zhenyuensis* (IVPP V18578, skull and cervical column in right lateral view).
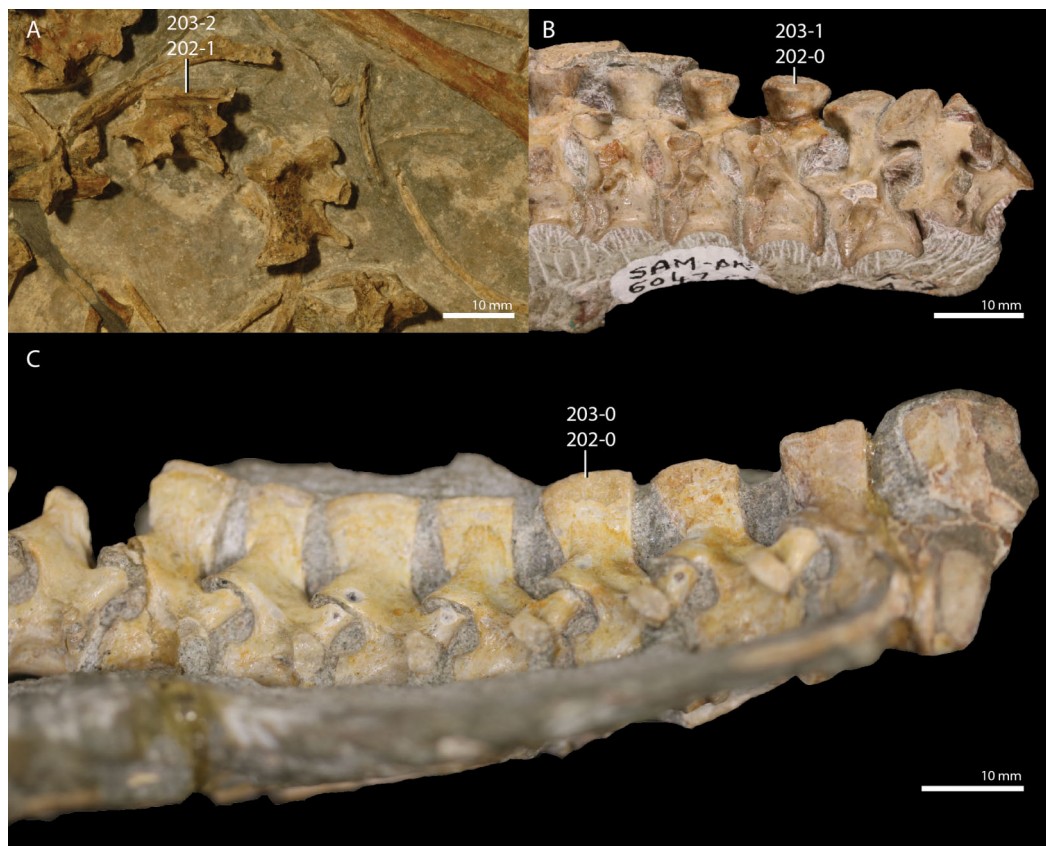

**Figure 28 Illustration of characters 202 and 203.** (A) 202-1 and 203-2 in *Tanystropheus longobardicus* (PIMUZ T 1277, disarticulated anterior dorsal vertebrae, indicated vertebra in angled left dorsolateral view). (B) 202-0 and 203-1 in *Euparkeria capensis* (SAM-PK-6047A, dorsal vertebral column in left lateral view). (C) 202-0 and 203-0 in *Mesosuchus browni* (SAM-PK-6046, dorsal vertebral column in right lateral view).

**Character 202 (New).** *Dorsal vertebrae, shape of distal margin of anterior to mid-dorsal neural spines in lateral view: slightly convex in lateral view (0); completely straight along anteroposterior length in lateral view (1)* (Fig. 28). *This character is inapplicable in taxa that possess mammillary processes.*

As for the cervical vertebrae, the anterior to mid-dorsal vertebrae of certain taxa bear a straight dorsal margin of the neural spine. This character is scored separately from character 179 because several of the sampled taxa exhibit clear variation in the shape of the distal margin of the neural spine between the dorsal and cervical vertebrae.

**Character 203 (New, combination ch. 320 and ch. 321 of *Ezcurra, 2016* and *Pritchard et al., 2015*: ch. 125).** *Dorsal vertebrae, distal expansion of the dorsal neural spines (not mammillary process) of the anterior to mid-dorsal vertebrae: absent (0); present, gradual transverse expansion of the distal half of the neural spine (1); present, but transverse expansion is restricted to the distal end of the neural spine (=spine table) (2)* (Fig. 28). *This character is inapplicable in taxa that bear mammillary processes on their dorsal vertebrae.*

This character describes the same morphology as is described for the cervical vertebrae in character 180. The occurrence of the expansion in the cervical and dorsal vertebrae is split into two different characters for the same arguments as for character 202. This character should only be scored in skeletally mature specimens since a transverse expansion of the neural spine is generally absent in early ontogenetic stages.

**Character 204 (Modified from *Senter, 2004*: ch. 42).** *Dorsal vertebrae, total number of dorsal vertebrae: ≤24 (0); ≥25 (1).*

The states of this character were modified to distinguish between the very high number of dorsal vertebrae seen in *Dinocephalosaurus orientalis* (*Rieppel, Li & Fraser, 2008*) and the 13 to 20 dorsal vertebrae seen in other taxa. No other states are incorporated because the exact number of dorsal vertebrae is hard to establish in many of the sampled taxa.

**Character 205 (*Ezcurra, 2016*: ch. 352) (reformulated).** *Dorsal vertebrae, length versus height of the centrum at the level of its posterior articular surface in posterior dorsal vertebrae: 0.83-1.25 (0); 1.36-1.88 (1); 2.16-2.20 (2); 2.31-2.40 (3); 2.53-2.76 (4), ORDERED RATIO.*

**Character 206 (Modified from *Ezcurra, 2016*: ch. 354).** *Dorsal vertebrae, lateral fossa on the centrum below the neurocentral suture: absent (0); present (1)* (*Ezcurra, 2016*: Figs. 31 and 34).

State 2 of character 254 of *Ezcurra (2016)* is excluded because it does not apply to the sampled taxa.

**Character 207 (Modified from *Nesbitt, 2011*: ch. 199).** *Dorsal vertebrae, development of the transverse process in mid-dorsal vertebrae: short, projecting only slightly beyond the lateral surface of the neural arch (0); long (1)* (*Ezcurra, 2016*: Fig. 32).

This character was modified based on the observed morphologies in the sampled taxa.

**Character 208 (*Ezcurra, 2016*: ch. 359).** *Dorsal vertebrae, hyposphene-hypantrum accessory intervertebral articulation in mid to posterior dorsal vertebrae: absent (0); present (1)* (*Ezcurra, 2016*: Figs. 31 and 32).

**Character 209 (*Ezcurra, 2016*: ch. 361).** *Dorsal vertebrae, dorsally opening pit lateral to the base of the neural spine: absent (0); shallow (fossa) (1); developed as a deep pit (2) ORDERED* (*Ezcurra, 2016*: Fig. 34).

**Character 210 (*Ezcurra, 2016*: ch. 363).** *Dorsal vertebrae, fan-shaped neural spine in lateral view: absent (0); present (1).*

**Character 211 (Modified from *Pritchard et al., 2015*: ch. 129).** *Dorsal vertebrae, height of neural spines in mid-dorsal vertebrae: tall, greater in dorsoventral height than anteroposterior length (0); long and low, approximately similar in dorsoventral height and anteroposterior length or less in height than in length (1).*

We modified the character so that it only applies to mid-dorsal vertebrae, because anterior dorsal vertebrae often have a different morphology from more posterior vertebrae, and their inclusion therefore might result in inconsistent character scoring.

**Character 212 (*Pritchard et al., 2015*: ch. 121).** *Dorsal vertebrae, position of parapophysis (or ventral margin of dorsal synapophysis) in posterior dorsal vertebrae: positioned partially on lateral margin of centrum (0); positioned entirely on neural arch (1).*

**Character 213 (*Pritchard et al., 2015*: ch. 122) (reformulated).** *Dorsal ribs, proximal end of anterior dorsal ribs: holocephalous (one facet) (0); dichocephalous (two facets) (1); tricephalous (three facets) (2).*

**Character 214 (*Ezcurra, 2016*: ch. 368).** *Dorsal ribs, proximal end of mid-dorsal ribs: dichocephalous (0); holocephalous (1). This character is inapplicable in taxa that have holocephalous anterior dorsal ribs since these imply the presence of holocephalous mid-dorsal ribs.*

The inapplicability criterion has been added.

**Character 215 (*Ezcurra, 2016*: ch. 372 and *Nesbitt et al., 2015*: ch. 216).** *Sacral ribs, anteroposterior length of the first primordial sacral rib versus the second primordial sacral rib in dorsal view: primordial sacral rib one is longer anteroposteriorly than primordial sacral rib two (0); primordial sacral rib two is about the same length or longer anteroposteriorly than primordial sacral rib one (1).*

**Character 216 (*Ezcurra, 2016*: ch. 373 and *Pritchard et al., 2015*: ch. 131).** *Sacral ribs, second rib shape: single unit (0); bifurcates distally into anterior and posterior processes (1)* (*Ezcurra, 2016*: Fig. 35).

**Character 217 (*Ezcurra, 2016*: ch. 374 and *Pritchard et al., 2015*: ch. 132).** *Sacral ribs, morphology of posterior process: pointed bluntly (0); pointed sharply (1)* (*Ezcurra, 2016*: Fig. 35). *This character is inapplicable in taxa without a bifurcated second sacral rib.*

**Character 218 (*Ezcurra, 2016*: ch. 375) (reformulated).** *Sacral and caudal vertebrae, transverse processes/ribs of sacral and anterior caudal vertebrae in skeletally mature individuals: rib/transverse process and vertebra unfused (0); rib/transverse process and vertebra fused to each other (1)* (*Ezcurra, 2016*: Fig. 35).

**Character 219 (Modified from *Ezcurra, 2016*: ch. 377).** *Caudal vertebrae, length of the transverse process + rib versus length across zygapophyses in anterior caudal vertebrae (third to fifth caudal vertebra): 0.62–1.28 (0); 1.62-1.77 (1); 1.90-2.00 (2); 2.50-2.60 (3), ORDERED RATIO* (*Ezcurra, 2016*: Fig. 35).

This character was modified slightly to specify on which caudal vertebrae this character should be scored.

**Character 220 (Modified from *Dilkes, 1998*: ch. 88).** *Caudal vertebrae, height versus maximum anteroposterior length of anterior caudal neural spine (measured in one of the first five caudal vertebrae): 0.42-0.83 (0); 1.00-1.60 (1); 2.00-2.24 (2); 2.39-2.53 (3); 2.93-3.07 (4), ORDERED RATIO.*

**Character 221 (Modified from *Pritchard et al., 2015*: ch. 134).** *Caudal vertebrae, orientation of transverse processes: base of process perpendicular to the long axis of the vertebra or slightly posterolaterally angled (0); processes distinctly angled posterolaterally from base (1).*

The states were modified to represent a clearer morphological distinction between them based on the sampled taxa.

**Character 222 (Modified from *Dilkes, 1998*: ch. 141).** *Chevrons, curvature of haemal spines in mid-caudal vertebrae: no curvature or posterior curvature (0); anterior curvature present (1).*

The states were modified to represent a clearer morphological distinction between them based on the sampled taxa.

**Character 223 (Modified from *Pritchard et al., 2015*: ch. 136).** *Chevrons, shape of haemal spine: tapers along its proximodistal length (0); maintains breadth along its proximodistal length (1); gradually broadens distally (2); broadens abruptly distally, forming an inverted T shape (3).*

This character was modified based on the observed morphologies in the sampled taxa.

**Character 224 (New).** *Chevrons, anteroposterior length of vertebral centrum versus proximodistal length of corresponding haemal spine in anterior caudal vertebrae (third to fifth caudal vertebra): 0.33-0.52 (0); 0.65-0.81 (1); 1.00-1.04 (2), ORDERED RATIO* (Fig. 29).

The relative length of the chevrons in the anterior caudal vertebrae differs among the sampled taxa and is possibly phylogenetically informative and it was therefore included as a character here.

**Character 225 (*Pritchard et al., 2015*: ch. 200).** *Heterotopic ossifications: absent in a minimum of 5 individuals (0); present (1)* (Fig. 30).

See the character description of character 200 in *Pritchard et al. (2015)*.

**Character 226 (*Ezcurra, 2016*: ch. 384).** *Scapulocoracoid, both bones fuse with each other in skeletally mature individuals: present (0); absent (1)* (*Ezcurra, 2016*: Fig. 36).

**Character 227 (New, combination of ch. 385 and ch. 388 of *Ezcurra, 2016*).** *Scapulocoracoid, anterior margin at the level of the suture between both bones: roughly continuous margin (0); distinct notch present (1); large fenestra between scapula and coracoid (scapulocoracoidal fenestra) present (2)* (Fig. 31; *Ezcurra, 2016*: Fig. 36).

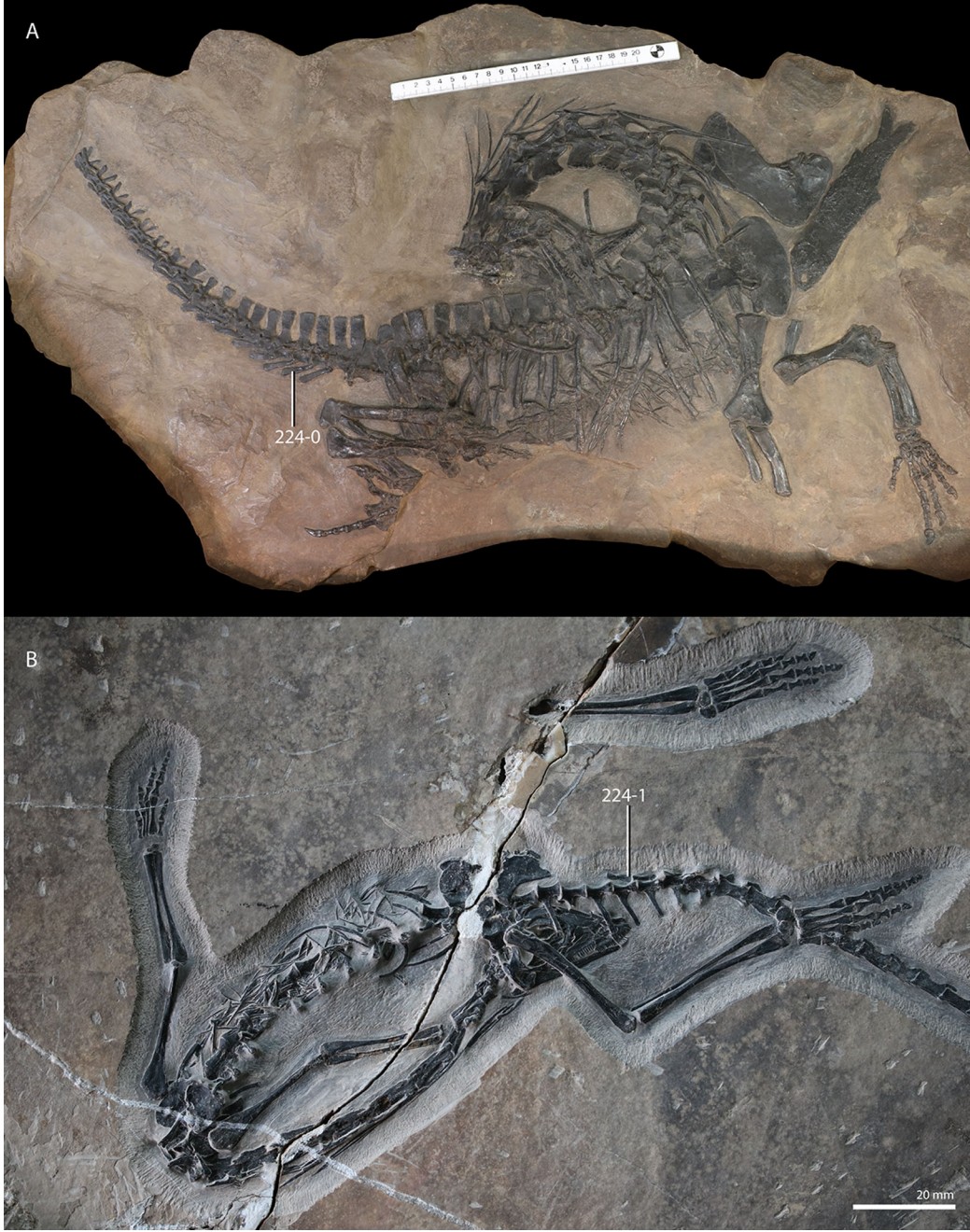

**Figure 29 Illustration of character 224.** (A) State 0 in *Protorosaurus speneri* (WMsN P 47361, largely complete skeleton in right lateral view, scale bar in cm). (B) State 1 in *Pectodens zhenyuensis* (IVPP V18578, largely complete skeleton, largely in ventral view).

Characters 385 and 388 in *Ezcurra (2016)* are fused because both refer to the anterior margin of the scapulocoracoid and the presence of state 1 precludes the possibility of state 2, and vice versa. However, because it is not clear whether the notch and the fenestra represent transitional morphologies of the same structure, the character is not ordered.

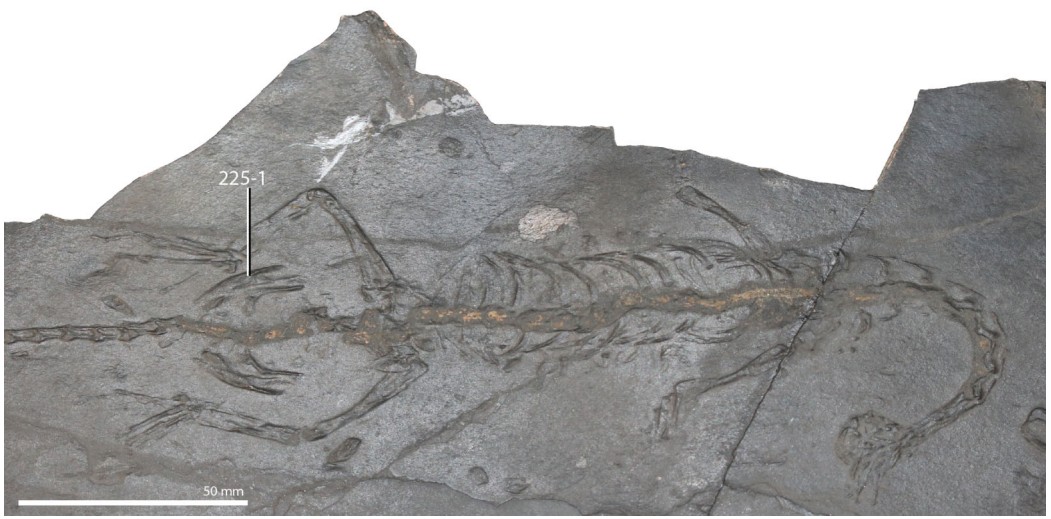

**Figure 30** Illustration of character 225 state 1 in *Tanytrachelos ahynis* (VMNH 120013, largely complete skeleton in dorsal or ventral view exhibiting heterotopic bones).

**Character 228 (Modified from *Pritchard et al., 2015*: ch. 145 and *Ezcurra, 2016*: ch. 389).** *Scapula, scapular blade dorsally or posterodorsally orientated with a rectangular outline (0); blade is largely posteriorly directed and semi-circular in outline with a continuously curved anterior/dorsal margin (1)* (Fig. 31).

The semi-circular or semi-lunar shape of the scapula in tanystropheids and *Pectodens zhenyuensis* represents a unique morphology among archosauromorphs and has been previously incorporated in phylogenetic analyses. We have redescribed this character to more specifically address this morphology as it is observed in a wide sample of taxa.

**Character 229 (Modified from *Ezcurra, 2016*: ch. 390 and *Nesbitt et al., 2015*: ch. 219).** *Scapula, anterior margin of the scapular blade in lateral view, excluding the margin of a potentially present scapulocoracoidal fenestra: straight or convex along entire length (0); distinctly concave (1)* (*Ezcurra, 2016*: Fig. 36). *This character is inapplicable in taxa that have a semi-circular scapular blade.*

This character was modified to prevent it from being interdependent with characters 227 and 228.

**Character 230 (Modified from *Ezcurra, 2016*: ch. 391 and *Nesbitt et al., 2015*: ch. 220).** *Scapula, constriction distal to the glenoid: minimum anteroposterior length greater than half the proximodistal length of the scapula (0); minimum anteroposterior length less than half but more than a quarter of the proximodistal length of the scapula (1); minimum anteroposterior length less than a quarter of the proximodistal length of the scapula (2)* (*Ezcurra, 2016*: Figs. 36 and 37), *ORDERED. This character is inapplicable in taxa that have a semi-circular scapular blade.*

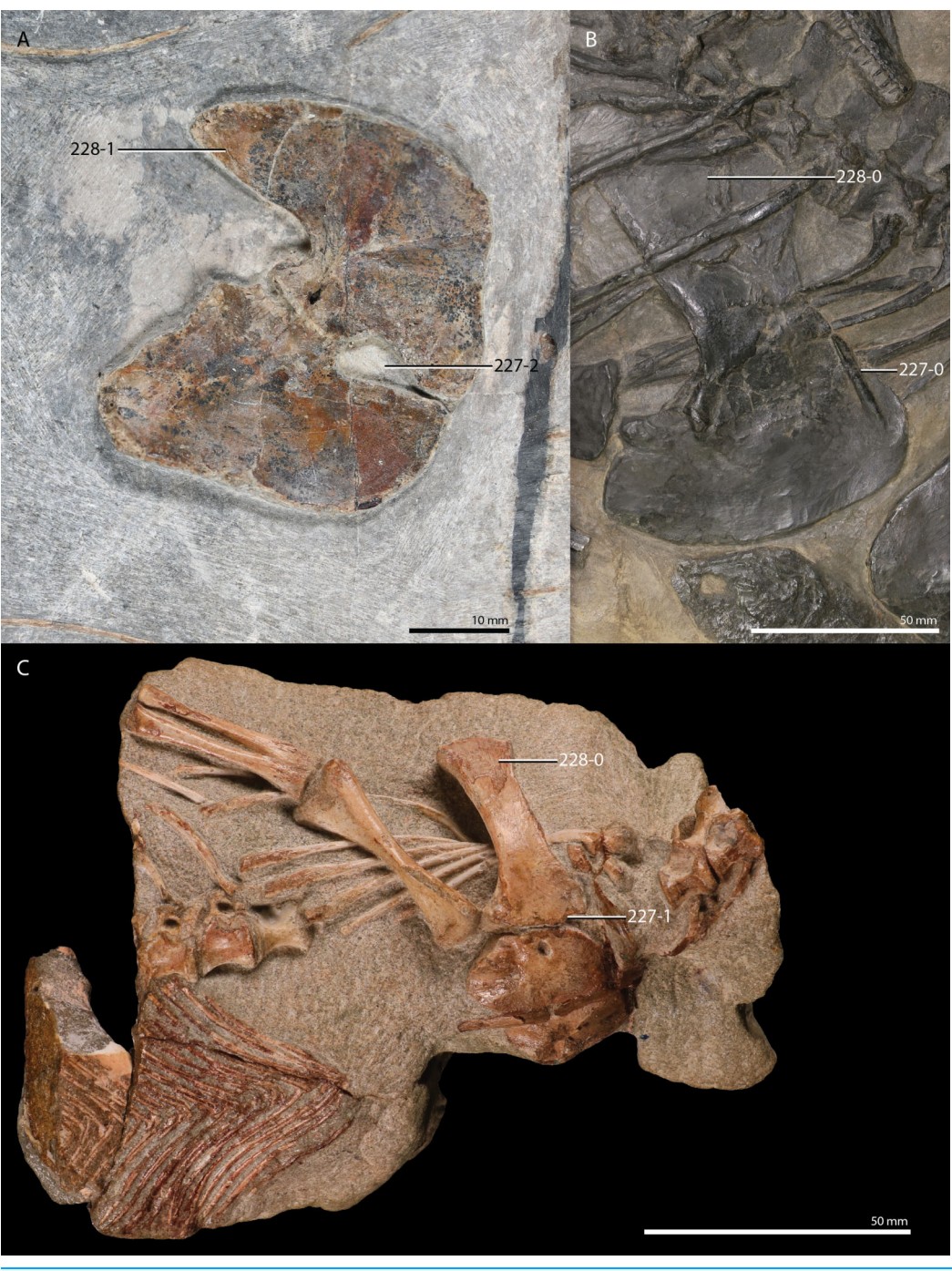

**Figure 31 Illustration of characters 227 and 228.** (A) 227-2 and 228-1 in *Tanystropheus longobardicus* (PIMUZ T 1277, left scapulocoracoid in medial view). (B) 227-0 and 228-0 in *Protorosaurus speneri* (WMsN P 47361, right scapulocoracoid in lateral view). (C) 227-1 and 228-0 in *Euparkeria capensis* (SAM-PK-5867, partial articulated skeleton including pectoral girdle in angled right lateroventral view).

This character was modified based on the observed morphologies in the sampled taxa. The inapplicability criterion was added because a semi-circular shape of the scapular blade implies that it is comparatively much wider than the other morphologies. Scoring those scapulae for this character would always result in state 0, thus representing an overscoring of the semi-circular shaped scapular blade.

**Character 231 (*Ezcurra, 2016*: ch. 392).** *Scapula, supraglenoid foramen: absent (0); present (1).*

**Character 232 (*Ezcurra, 2016*: ch. 398) (state 2 reformulated).** *Coracoid, posterior border in lateral view: unexpanded posteriorly (0); moderately expanded posteriorly (1); strongly expanded posteriorly—the entire border, not only the posteroventral region as is the case in the postglenoid process—and, as a result, the articulated scapula and coracoid are L-shaped in lateral view (in taxa in which the scapular blade is not semi-circular in shape) (2), ORDERED* (Ezcurra, 2016: Fig. 37).

**Character 233 (*Ezcurra, 2016*: ch. 404 and *Pritchard et al., 2015*: ch. 140).** *Cleithrum: present (0); absent (1).*

**Character 234 (*Ezcurra, 2016*: ch. 405).** *Interclavicle: present (0); absent (1) (Ezcurra, 2016: Fig. 15).*

**Character 235 (Modified from *Ezcurra, 2016*: ch. 406).** *Interclavicle, long anterior process, resulting in a cross-shaped interclavicle in ventral or dorsal view: present (0); absent (1) (Ezcurra, 2016: Fig. 38). This character is inapplicable in taxa that lack an ossified interclavicle.*

An inapplicability criterion was added.

**Character 236 (*Ezcurra, 2016*: ch. 407 and *Pritchard et al., 2015*: ch. 143).** *Interclavicle, anterior margin with a median notch: absent (0); present (1) (Ezcurra, 2016: Fig. 38). This character is inapplicable in taxa that lack an ossified interclavicle.*

An inapplicability criterion was added.

**Character 237 (*Ezcurra, 2016*: ch. 409).** *Interclavicle, webbed between lateral and posterior processes: present, proximal half of the bone subtriangular or diamond-shaped (0); absent, sharp angles between processes (1) (Ezcurra, 2016: Fig. 38). This character is inapplicable in taxa that lack an ossified interclavicle.*

An inapplicability criterion was added.

**Character 238 (*Ezcurra, 2016*: ch. 411 and *Pritchard et al., 2015*: ch. 144).** *Interclavicle, posterior ramus: little change in width along entire length (0); gradual transverse expansion present (1) (Ezcurra, 2016: Fig. 38). This character is inapplicable in taxa that lack an ossified interclavicle.*

An inapplicability criterion was added.

**Character 239 (New).** *Limbs, flipper-like, indicated by the presence of rod-like stylopodial and zygapodial elements, simple disc-like tarsal and carpal bones, and hyperphalangy: absent (0); present (1).*

In *Dinocephalosaurus orientalis* (*Rieppel, Li & Fraser, 2008*) the limbs have been modified to function as flippers for aquatic propulsion. The presence of at least one more *Dinocephalosaurus*-like taxon (*Li, Rieppel & Fraser, 2017*) indicates that more non-archosauriform archosauromorphs might have had flippers.

**Character 240 (New).** *Long bone histology, fibrolamellar bone tissue in the cortex: absent (0); present (1).*

Early archosauromorphs exhibit considerable variation in their bone tissue. Fibrolamellar tissue is present in the cortex of *Azendohsaurus* (*Cubo & Jalil, 2019*), *Prolacerta*, *Proterosuchus*, and *Erythrosuchus* (*Botha-Brink & Smith, 2011*) among the sampled genera, but absent in the non-saurian diapsid *Claudiosaurus* (*De Buffrénil & Mazin, 1989*) and the archosauromorphs *Macrocnemus* and *Tanystropheus* (*Jaquier & Scheyer, 2017*), *Trilophosaurus buettneri* (*Werning & Irmis, 2010*), and *Euparkeria* (*Botha-Brink & Smith, 2011*). The presence of fibrolamellar bone tissue can contain a strong phylogenetic signal, as it has important implications for growth rates and metabolism. Therefore, this character has been included in a phylogenetic context here for the first time.

**Character 241 (*Ezcurra, 2016*: ch. 415).** *Humerus, torsion between proximal and distal ends: approximately 45 degrees or more (0); 35 degrees or less (1)* (*Ezcurra, 2016*: Fig. 39).

**Character 242 (*Ezcurra, 2016*: ch. 416).** *Humerus, transverse width of the proximal end versus total length of the bone in skeletally mature individuals: 0.11-0.33 (0); 0.38-0.46 (1); 0.56-0.68 (2), ORDERED RATIO* (*Ezcurra, 2016*: Fig. 39).

**Character 243 (*Ezcurra, 2016*: ch. 420).** *Humerus, conical process on the proximal surface, placed immediately adjacent to the base of the deltopectoral crest: absent (0); present (1)* (*Ezcurra, 2016*: Fig. 39).

**Character 244 (*Ezcurra, 2016*: ch. 423).** *Humerus, ventral margin of the deltopectoral crest developed as a thick subcilindrical tuberosity that is well-differentiated from the thinner dorsal margin: present (0); absent (1)* (*Ezcurra, 2016*: Fig. 39).

**Character 245 (*Ezcurra, 2016*: ch. 425).** *Humerus, entepicondyle size in skeletally mature individuals: moderately large (0); strongly developed (1)* (*Ezcurra, 2016*: Fig. 39).

**Character 246 (*Ezcurra, 2016*: ch. 426 and *Pritchard et al., 2015*: ch. 153).** *Humerus, entepicondylar foramen: present (0); absent (1)* (*Ezcurra, 2016*: Fig. 39).

**Character 247 (*Ezcurra, 2016*: ch. 427).** *Humerus, ectepicondylar region: foramen present (0); foramen absent, supinator process and groove present (1); supinator process, groove or foramen absent (2)* (*Ezcurra, 2016*: Fig. 39).

**Character 248 (Modified from *Ezcurra, 2016*: ch. 414).** *Humerus, total length of the humerus versus the total length of the femur: 0.63–0.71 (0); 0.76–0.80 (1); 0.84–0.91 (2); 0.97–1.05 (3), ORDERED RATIO.*

This character is modified to compare the total length of the humerus to the femur rather than the entire forelimb to the entire hindlimb.

**Character 249 (*Pritchard et al., 2015*: ch. 157 and *Ezcurra, 2016*: ch. 430) (reformulated).** *Ulna, olecranon process: absent, not ossified or very low in skeletally mature individuals (0); present (1) (*Ezcurra, 2016*: Fig. 40).*

**Character 250 (*Ezcurra, 2016*: ch. 433).** *Ulna, lateral tuber (=radius tuber) on the proximal portion: absent in skeletally mature individuals (0); present (1) (*Nesbitt, 2011*: Figs. 40 and 31).*

**Character 251 (*Ezcurra, 2016*: ch. 435).** *Radius, total length versus total length of the humerus: 0.53-0.72 (0); 0.81-0.92 (1); 1.01-1.07 (2); 1.40-1.46 (3), ORDERED RATIO (*Ezcurra, 2016*: Fig. 15).*

**Character 252 (*Ezcurra, 2016*: ch. 440 and *Pritchard et al., 2015*: ch. 161).** *Carpals, perforating foramen between intermedium and ulnare: present (0); absent in skeletally mature individuals (1). This character is inapplicable in taxa that lack an intermedium.*

**Character 253 (New, combination of ch. 441 and ch. 442 of *Ezcurra, 2016*).** *Centrale of the manus of skeletally mature individuals: both the lateral and medial centrale are present (0); only the lateral centrale is present (1); only the medial centrale is present (2); both are absent (3).*

**Character 254 (*Ezcurra, 2016*: ch. 443).** *Carpals, pisiform: present (0); absent in skeletally mature individuals (1) (*Ezcurra, 2016*: Fig. 40).*

**Character 255 (*Benton & Allen, 1997*: ch. 31 and *Jalil, 1997*: ch. 46).** *First distal carpal: present (0); absent in skeletally mature individuals (1).*

**Character 256 (*Ezcurra, 2016*: ch. 444 and *Pritchard et al., 2015*: ch 159).** *Carpals, distal carpal five: absent in skeletally mature individuals (0); present (1) (*Ezcurra, 2016*: Fig. 40).*

**Character 257 (*Ezcurra, 2016*: ch 445).** *Manus, longest metacarpal + digit: longer than humeral length (0); subequal to or shorter than humeral length (1).*

**Character 258 (*Ezcurra, 2016*: ch. 446).** *Metacarpus, length of the longest metacarpal versus length of the longest metatarsal: 0.32-0.33 (0); 0.36-0.41 (1); 0.46-0.49 (2); 0.61-0.65 (3), ORDERED RATIO.*

**Character 259 (*Ezcurra, 2016*: ch. 448).** *Metacarpus, width of the distal end of the metacarpal I versus its total length: 0.25-0.33 (0); 0.38-0.46 (1); 0.49-0.50 (2); 0.56-0.61 (3); 0.65-0.67 (4), ORDERED RATIO (*Ezcurra, 2016*: Fig. 40).*

**Character 260 (*Ezcurra, 2016*: ch. 450).** *Metacarpus, metacarpal IV: longer than metacarpal III (0); equal to or shorter than metacarpal III (1)* (*Ezcurra, 2016*: Fig. 40).

**Character 261 (*Ezcurra, 2016*: ch. 453).** *Manual digits, second phalanx of manual digit II: shorter than the first phalanx of manual digit II (0); longer than the first phalanx of manual digit II (1)* (*Nesbitt, 2011*: Fig. 32).

**Character 262 (*Ezcurra, 2016*: ch. 451 and *Nesbitt et al., 2015*: ch. 222).** *Manual digits, unguals length: about the same length or shorter than the last non-ungual phalanx of the same digit (0); distinctly longer than the last non-ungual phalanx of the same digit (1). This character is inapplicable in taxa in which the terminal phalanx of each digit does not form an ungual.*

An inapplicability criterion was added because in the aquatic *Dinocephalosaurus orientalis* the terminal phalanges do not form unguals.

**Character 263 (Modified from *Ezcurra, 2016*: ch. 454).** *Manual digits, number of phalanges in digit IV: five (0); four (1)* (*Nesbitt, 2011*: Fig. 32).

State 2 of character 454 in *Ezcurra (2016)* was removed because it was irrelevant for the sampled taxa.

**Character 264 (Modified from *Ezcurra, 2016*: ch 460).** *Ilium, preacetabular process: absent or incipient (0); present, being considerably anteroposteriorly shorter than its dorsoventral height (1); present, being longer than two thirds of its height (2), ORDERED.*

States 2 and 3 of character 460 in *Ezcurra (2016)* were fused and redescribed to address the specific morphology observed in the sampled taxa.

**Character 265 (Modified from *Sookias, 2016*: ch. 268).** *Ilium, shape of preacetabular process: rounded (0); approximately straight-sided with a distinct angle between the anterior and dorsal margins (1). This character is inapplicable in taxa that lack a preacetabular process on the ilium.*

States 1 and 2 were fused and state 3 was removed compared to the original character of *Sookias (2016)* to specifically address the observed morphologies in the sampled taxa.

**Character 266 (Modified from *Pritchard et al., 2015*: ch. 170 and *Ezcurra, 2016*: ch. 461).** *Ilium, anterior process/tuber on the anterior margin of the ilium: anterior process/ tuber absent or incipient (0); clearly defined anteriorly projecting tuber present on the anterior margin of the preacetabular process (1)* (Fig. 32). *This character is inapplicable in taxa that lack a preacetabular process on the ilium.*

*Pritchard et al. (2015)* considered the presence of a small tuber on the anterior margin of the iliac blade in certain tanystropheids to represent the same structure as the preacetabular process, which was incorporated into character 170 of their dataset. This character interpreted the presence of an anteriorly expanded preacetabular process to

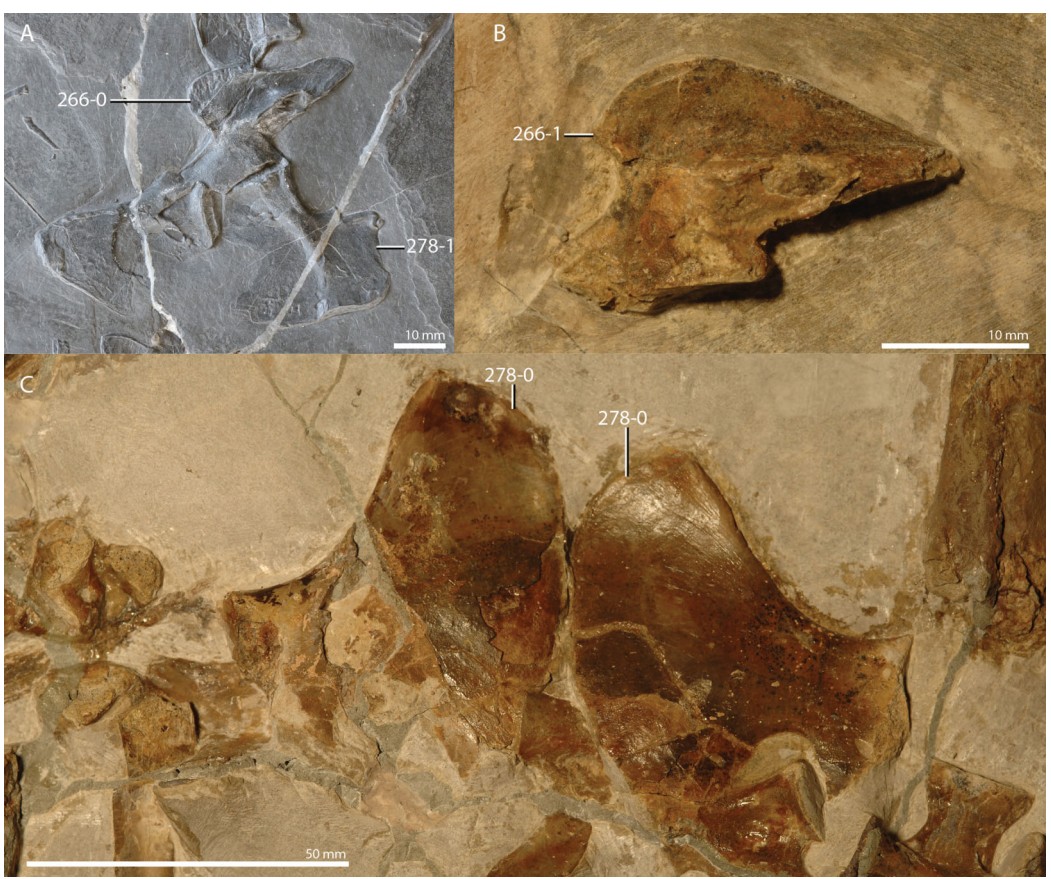

**Figure 32 Illustration of characters 266 and 278.** (A) 266-0 and 278-1 in *Macrocnemus fuyuanensis* (IVPP V15001, right half of the pelvic girdle in medial view). (B) 266-1 in *Tanystropheus longobardicus* (PIMUZ T 1277, right ilium in medial view). (C) 278-0 *Tanystropheus hydroides* (PIMUZ T 2817, ischia in lateral view).

represent a more strongly exhibited version of this tuber. However, we consider this tuber to represent a separate structure from the preacetabular process, since this tuber also occurs in certain taxa that have a smooth, anterodorsally curved preacetabular process in lateral view. The presence of a small finger-like tuber on the anterior margin of the preacetabular process is subject to intraspecific variation and occurs in *Tanystropheus longobardicus* (PIMUZ T 1277), a skeletally immature specimen of *Macrocnemus bassanii* (MSNM BES SC 111), and *Fuyuansaurus acutirostris* (IVPP V17983) among the sampled taxa and it is considered to represent an informative character independent of the size of the preacetabular process. This tuber is also present in the nomen dubium "*Exilisuchus tubercularis*" (*Ochev, 1979*; Fig. 9 of *Ezcurra, 2016*). "*Exilisuchus tubercularis*" is exclusively known from a partial ilium and was recovered as a tanystropheid in the phylogenetic analysis of *Ezcurra (2016)*.

**Character 267 (*Ezcurra, 2016*: ch. 463) (reformulated).** *Ilium, length of the postacetabular process measured from the most proximal point on the posterior/ventral margin of the*

*process versus anteroposterior length of the acetabulum: 0.50-0.71 (0); 0.88-0.91 (1); 0.98-1.14 (2); 1.22-1.57 (3); 1.72-1.78 (4), ORDERED RATIO (Ezcurra, 2016: Fig. 41).*

The character was reformulated to specify the measurement of the length of the postacetabular process.

**Character 268 (*Ezcurra, 2016*: ch. 464 and *Pritchard et al., 2015*: ch. 164).** *Ilium, main axis of the postacetabular process in lateral or medial view: posterodorsally orientated (0); mainly posteriorly orientated (1) (Ezcurra, 2016: Figs. 9 and 41).*

**Character 269 (Modified from *Ezcurra, 2016*: ch. 465).** *Ilium, caudifemoralis brevis muscle origin on the lateroventral surface of the postacetabular process: not dorsally or laterally rimmed by a brevis shelf (0); dorsally rimmed by a low brevis shelf (1) (Ezcurra, 2016: Fig. 9).*

State 0 of character 465 in *Ezcurra (2016)* has been removed because it is irrelevant to the sampled taxa.

**Character 270 (*Pritchard et al., 2015*: ch. 166).** *Ilium, supra-acetabular crest: crest absent, anterodorsal margin of acetabulum similar in development to posterodorsal margin (0); prominent anterodorsal lamina frames the anterodorsal margin of the acetabulum (1).*

**Character 271 (*Pritchard et al., 2015*: ch. 167).** *Ilium, shape of supra-acetabular margin: dorsalmost margin of acetabulum is unsculptured (0); prominent, bulbous rugosity superior to acetabulum (1). This character is inapplicable in taxa that lack a distinct supra-acetabular crest.*

**Character 272 (*Pritchard et al., 2015*: ch. 165).** *Ilium, anteroventral process extending from anterior margin of pubic peduncle: absent (0); present, process draping across anterior surface of pubis (1).*

**Character 273 (*Ezcurra, 2016*: ch. 471 and *Pritchard et al., 2015*: ch. 163).** *Pubis-ischium, thyroid fenestra: absent (0); present (1) (Ezcurra, 2016: Fig. 41).*

**Character 274 (*Pritchard et al., 2015*: ch. 175).** *Pubis, lateral surface, development of a lateral tubercle (sensu Vaughn, 1955): present (0); absent (1).*

**Character 275 (*Ezcurra, 2016*: ch. 477).** *Pubis, pubic apron: absent, symphysis extended along the ventral margin of the pelvic girdle and visible in lateral view (0); present, symphysis restricted anteriorly and obscured by the pubic shaft in lateral view (1) (Ezcurra, 2016: Fig. 41).*

This character is discussed on page 61 and figured in Fig. 58 of *Nesbitt et al. (2015)*.

**Character 276 (*Ezcurra, 2016*: ch. 482 and *Nesbitt et al., 2015*: ch. 225).** *Ischium, maximal length versus anteroposterior length of the acetabulum: 1.46–1.57 (0); 1.67–1.77 (1); 1.86–1.98 (2); 2.06–2.23 (3); 2.51–2.89 (4), ORDERED RATIO.*

**Character 277 (*Ezcurra, 2016*: ch. 486).** *Ischium, symphysis raised on a distinct low peduncle: absent (0); present (1) (Ezcurra, 2016*: Fig. 41).

This character is discussed on page 62 of *Nesbitt et al. (2015)*.

**Character 278 (Modified from *Pritchard et al., 2015*: ch. 176 and *Ezcurra, 2016*: ch. 488).** *Ischium, distinct concavity or constriction on the posterior half of the ventral margin of the ischium, thus separating a distinct posterior process from the rest of the ischium: absent (0); present (1)* (Fig. 32; *Ezcurra, 2016*: Fig. 41).

This character was discussed by *Pritchard et al. (2015)*, where it was considered homologous to the spina ischia described by *El-Toubi (1949)*. We reformulate the character based on our observations of the ischia in the sampled taxa. A posterior process of the ischium is formed by a distinct concavity or constriction of the ventral margin of the ischium in *Planocephalosaurus robinsonae*, *Pectodens zhenyuensis*, and *Langobardisaurus pandolfii*. Furthermore, this trait is also present in some, but not all, specimens of *Macrocnemus bassani*, *Macrocnemus fuyuanensis*, *Tanystropheus longobardicus*, and *Amotosaurus rotfeldensis*. Therefore, this character clearly shows a large amount of intraspecific variability.

**Character 279 (*Ezcurra, 2016*: ch. 491).** *Femur, proximal articular surface in skeletally mature individuals: well-ossified, being flat or convex (0); partially ossified, being concave and sometimes with a circular pit (1) (Ezcurra, 2016*: Fig. 42).

**Character 280 (*Ezcurra, 2016*: ch. 504).** *Femur, attachment of the caudifemoralis musculature on the posterior surface of the bone: crest-like and with intertrochanteric fossa (=internal trochanter), and convergent with proximal end (0); crest-like and with intertrochanteric fossa (=internal trochanter), and not convergent with proximal end (1); crest-like and without intertrochanteric fossa (=fourth trochanter), and not convergent with proximal end (2) (Ezcurra, 2016*: Figs. 42 and 43). *This character is inapplicable in taxa without a distinct process for the attachment of the caudifemoralis musculature on the femur.*

Character 504 in *Ezcurra (2016)* is ordered. We decided not to order this character here because we do not consider it clear that the states represent intermediate steps in a transformational series without *a priori* assumptions on phylogenetic relationships.

**Character 281 (*Ezcurra, 2016*: ch. 511).** *Femur, distal condyles: prominent, strong dorsoventral expansion (in sprawling orientation) restricted to the distal end (0); not projecting markedly beyond shaft and expand gradually if there is any expansion (1)* (*Ezcurra, 2016*: Fig. 43).

See also the description of character 318 in *Nesbitt (2011)*.

**Character 282 (*Ezcurra, 2016*: ch. 512).** *Femur, distal articular surface: uneven, lateral (=fibular) condyle projecting distally distinctly beyond medial (=tibial) condyle (0); both*

condyles prominent distally and approximately at same level (1); both condyles do not project distally (distal articular surface concave or almost flat) (2) (*Ezcurra, 2016*: Figs. 42 and 43).

**Character 283 (*Ezcurra, 2016*: ch. 513).** *Femur, anterior extensor groove: absent, anterior margin of the bone straight or convex in distal view (0); present, anterior margin of the bone concave in distal view (1) (*Ezcurra, 2016*: Fig. 42).*

**Character 284 (*Ezcurra, 2016*: ch. 515).** *Femur, shape of lateral (=fibular) condyle in distal view: lateral surface is rounded and mound-like (0); lateral surface is triangular and sharply pointed (1) (*Ezcurra, 2016*: Fig. 42).*

**Character 285 (*Benton & Allen, 1997*: ch. 39).** *Femur, length of tibia relative to length of femur: tibia shorter than, or subequal to, femur in length (0); tibia longer than femur (1).*

**Character 286 (*Pritchard et al., 2015*: ch. 177).** *Femur, shape in lateral view: femoral shaft exhibits sigmoidal curvature (0); femoral shaft linear with slight ventrodistal curvature (1).*

**Character 287 (*Ezcurra, 2016*: ch. 528).** *Fibula, transverse width at mid-length: subequal to transverse width of the tibia (0); distinctly narrower than transverse width of the tibia (1) (*Ezcurra, 2016*: Fig. 15).*

**Character 288 (*Ezcurra, 2016*: ch. 531).** *Fibula, distal end in lateral view: angled anterodorsally (asymmetrical) (0); rounded or flat (symmetrical) (1) (*Ezcurra, 2016*: Fig. 44, *Nesbitt, 2011*: Fig. 41).*

**Character 289 (Modified from *Ezcurra, 2016*: ch. 532).** *Proximal tarsals, articulation between astragalus and calcaneum: roughly flat (0); concavoconvex with concavity on the astragalus (1); fused (2) (*Ezcurra, 2016*: Fig. 45).*

State 1 of character 532 of *Ezcurra (2016)* is excluded because it does not apply to the sampled taxa. This character can best be observed in plantar view. For a detailed description of the articulation between the astragalus and calcaneum, see the extensive discussion in *Sereno (1991)*, in particular Figs. 3 and 4, 8 therein, and *Cruickshank (1979)*.

**Character 290 (*Ezcurra, 2016*: ch. 539).** *Astragalus, posterior groove: present (0); absent (1) (*Nesbitt, 2011*: Fig. 46).*

This character is extensively discussed in p. 353 of *Gower (1996)*. Due to the three-dimensional structure of the astragalus and the variation observed in its morphology in the sampled taxa, it is sometimes difficult to distinguish the groove from other curves and concavities on the bone. Taxa are scored as 0 when a clear concavity is present on the ventral/plantar surface of the astragalus that is often connected to the perforating foramen between the astragalus and calcaneum.

**Character 291 (Modified from *Pritchard et al., 2015*: ch. 184 and part of *Ezcurra, 2016*: ch. 557).** *Distal tarsals, pedal centrale: present or partially fused to the astragalus in mature*

individuals (0); absent as a separate ossification, being either unossified or fused to the astragulus in skeletally mature individuals (1) (*Ezcurra, 2016*: Figs. 45 and 46).

In character 184 of *Pritchard et al. (2015)* the absence of a pedal centrale is stated to result from the fusion of this element to the astragalus. It has indeed been documented that the loss of a separate pedal centrale in several early archosauromorphs is the result of fusion with the astragalus (*Fernández Blanco, Ezcurra & Bona, 2020*). However, in *Tanystropheus* spp. and in *Dinocephalosaurus orientalis* the absence of the pedal centrale is more likely to be attributable to skeletal paedomorphosis related to aquatic adaptations (*Rieppel, 1989*; *Rieppel, Li & Fraser, 2008*). A pedal centrale is also absent in the pedes of the only known specimen of *Pectodens zhenyuensis* (*Li et al., 2017*). However, since this specimen might represent an early ontogenetic stage, we refrain from scoring this character for this taxon. In the absence of sufficient embryological data to assess its developmental underpinnings for our taxonomic sample, our character addresses the presence or absence of this element irrespective of whether it might be attributable to paedomorphosis or fusion with the astragalus.

**Character 292 (New, combination of *Ezcurra, 2016*: ch. 558 [= *Pritchard et al., 2015*: ch. 193] and *Ezcurra, 2016*: ch. 559 [= *Pritchard et al., 2015*: ch. 194]).** *Distal tarsals of skeletally mature individuals, distal tarsal 1 and 2: both present (0); only one of the two elements is present (1); both absent (2), ORDERED.*

Characters 558 and 559 in *Ezcurra (2016)* were fused here, because in certain taxa (*Macrocnemus bassanii*, PIMUZ T 4822; *Macrocnemus fuyuanensis*, IVPP V15001; and *Amotosaurus rotfeldensis*, SMNS 54783a/b) one of the distal tarsals is present, but it cannot be established confidently whether this represents distal tarsal 1 or 2. We consider this of secondary importance, as this character treats with the degree of ossification (paedomorphosis) in the tarsus, and both distal tarsals ossify at roughly the same developmental stage (*Rieppel, 1989*). This character has been ordered because state 2, the absence of both elements, represents a larger degree of paedomorphosis than state 1, the absence of only a single element. Therefore, state 1 is considered to represent an intermediate step between states 0 and 2.

**Character 293 (*Ezcurra, 2016*: ch. 563 and *Pritchard et al., 2015*: ch. 195).** *Distal tarsals, distal tarsal 5: present (0); absent in skeletally mature individuals (1).*

**Character 294 (Modified from *Ezcurra, 2016*: ch. 564).** *Pes, foot length (articulated fourth metatarsal and digit) versus tibia-fibula length: 0.60-0.68 (0); 0.79-1.04 (1); 1.12-1.16 (2); 1.34-1.56 (3); 1.96-2.04 (4), ORDERED RATIO (*Ezcurra, 2016*: Fig. 15).*

The original distinction between the character states was not considered to be phylogenetically relevant for the sampled taxa and therefore it was decided to distinguish states based on calculated ratios.

**Character 295 (*Ezcurra, 2016*: ch. 533 and *Pritchard et al., 2015*: ch. 186).** *Proximal tarsals, foramen for the passage of the perforating artery between the astragalus and*

calcaneum (=perforating foramen): present (0); absent in skeletally mature individuals (1) (*Ezcurra, 2016*: Fig. 45).

**Character 296 (*Ezcurra, 2016*: ch. 565).** *Metatarsus, configuration: metatarsals diverging from ankle (0); compact, metatarsals I-IV tightly bunched (1)* (*Ezcurra, 2016*: Fig. 46).

**Character 297 (*Ezcurra, 2016*: ch. 569).** *Metatarsus, length of metatarsal I versus metatarsal III: 0.36–0.43 (0); 0.48–0.51 (1); 0.54–0.63 (2); 0.67–0.75 (3); 0.82–0.84 (4), ORDERED RATIO* (*Ezcurra, 2016*: Fig. 46).

**Character 298 (*Ezcurra, 2016*: ch. 571).** *Metatarsus, length of the metatarsal II versus length of the metatarsal IV: 0.55–0.67 (0); 0.70–0.76 (1); 0.80–0.86 (2); 0.89–0.91 (3); 0.94–1.02 (4), ORDERED RATIO* (*Ezcurra, 2016*: Fig. 46).

**Character 299 (*Ezcurra, 2016*: ch. 574).** *Metatarsus, length of metatarsal IV versus length of metatarsal III: 0.88–1.00 (0); 1.03–1.08 (1); 1.13–1.22 (2); 1.25–1.26 (3), ORDERED RATIO* (*Ezcurra, 2016*: Fig. 46).

Character 581 in *Ezcurra (2016)* is not included in our analysis, because the length of the entire digit is strongly dependent on the length of the metatarsal, and these characters are therefore considered interdependent. It was preferred to compare the relative lengths of the metatarsals over the lengths of the entire digits because this feature could be scored in more of the sampled taxa.

**Character 300 (*Ezcurra, 2016*: ch. 577) (reformulated).** *Metatarsus, metatarsal V with a hook-shaped proximal end: absent (0); present, with a gradually medially curved proximal process (1); present, with an abruptly medially flexed proximal process and, as a result, the metatarsal acquires a L-shape in dorsal or ventral view (2)* (*Ezcurra, 2016*: Fig. 46).

**Character 301 (*Ezcurra, 2016*: ch. 576) (reformulated).** *Metatarsus, dorsal prominence separated from the proximo-medial surface by a concave gap in metatarsal V: absent (0); present (1)* (*Ezcurra, 2016*: Fig. 46, *Nesbitt, 2011*: Fig. 47). *This character is inapplicable in taxa that lack a hook-shaped metatarsal V.*

**Character 302 (*Ezcurra, 2016*: ch. 578 and *Pritchard et al., 2015*: ch. 196).** *Metatarsus, metatarsal V outer process on the proximal lateral margin: absent, smooth curved margin (0); present, prominent pointed process (1).*

**Character 303 (*Ezcurra, 2016*: ch. 579).** *Metatarsus, metatarsal V lateral plantar tubercle: absent (0); present (1)* (*Ezcurra, 2016*: Fig. 46).

**Character 304 (*Ezcurra, 2016*: ch. 580).** *Metatarsus, metatarsal V medial plantar tubercle: absent (0); present (1)* (*Ezcurra, 2016*: Fig. 46).

**Character 305 (Modified from *Benton & Allen, 1997*: ch. 45).** *Metatarsus, length of metatarsal IV versus the proximodistal length of metatarsal V: 1.25–1.90 (0); 2.19–2.57 (1); 2.83–3.25 (2); 3.65–5.15 (3), ORDERED RATIO.*

The original distinction between the character states was not considered to be phylogenetically relevant for the sampled taxa and therefore it was decided to distinguish states based on calculated ratios.

**Character 306 (*Ezcurra, 2016*: ch. 584 and *Pritchard et al., 2015*: ch. 199).** *Pedal digits, phalanx V-1: subequal to or shorter than other non-ungual phalanges (0); metatarsal-like, considerably longer than other non-ungual phalanges (1).*

**Character 307 (*Ezcurra, 2016*: ch 587 and *Nesbitt et al., 2015*: ch. 233).** *Pedal digits, ventral tubercle in unguals: absent or small (0); well-developed and extended ventral to the articular portion of the ungual (1).*

## RESULTS

### Analyses excluding ratio characters and treating all remaining characters as unordered

Analyses 1 and 2 both exclude ratio characters and treat all characters as unordered. Analysis 1, which includes all OTUs (Fig. 33), found 1976 MPTs of 977 steps with a consistency index (CI) of 0.370 and a retention index (RI) of 0.534, whereas analysis 2, which excluded the problematic OTUs *Czatkowiella harae*, *Tanystropheus "conspicuus"*, and *"Tanystropheus antiquus"* (Fig. 34), recovered 884 MPTs that had 953 steps and a CI of 0.379 and a RI of 0.537. Support values for the branches are indicated in the SCTs (Figs. 33A and 34A). The SCTs of both analyses contain large polytomies, which are consecutively more resolved in the three sequential RSCTs calculated for each analysis (Figs. 33B–33D and 34B–34D). At the base of the SCTs of both analyses a polytomy is formed by *Claudiosaurus germaini*, *Acerosodontosaurus piveteaui*, *Youngina capensis*, and Sauria, with *Orovenator mayorum* as the sister taxon to this polytomy (Figs. 33A and 34A). Although it is generally considered that *Orovenator mayorum* is most distantly related to Sauria among our sampled OTUs except for the outgroup *Petrolacosaurus kansensis* (*Ford & Benson, 2020*; Reisz et al., 2011), there is no clear consensus in the relationships between *Youngina capensis*, *Claudiosaurus germaini*, and *Acerosodontosaurus piveteaui* (*Bickelmann, Müller & Reisz, 2009*). Since the focus of this study is not to resolve early diapsid phylogeny, only these taxa, which are among the morphologically best-known early diapsids, were included, and the lack of resolution can possibly be attributed to the low taxonomic and character samples for this part of the tree. All MPTs of both analyses recover a monophyletic Sauria, Lepidosauromorpha, Archosauromorpha, Crocopoda, Rhynchosauria, Allokotosauria, and Archosauriformes.

In the SCTs of both analyses a huge polytomy is formed at the base of Archosauromorpha that includes the monophyletic Crocopoda and all remaining "protorosaurs", some of which are recovered in less inclusive clades within the polytomy (Figs. 33A and 34A). The SCT of analysis 1 recovers *Macrocnemus bassanii* and *Macrocnemus fuyuanensis* as sister taxa but finds the poorly known *Macrocnemus obristi* outside this clade. The SCT of analysis 2 does not recover *Macrocnemus bassanii* and *Macrocnemus fuyuanensis* in a generic clade. In the SCT of analysis 1 a polytomic clade is

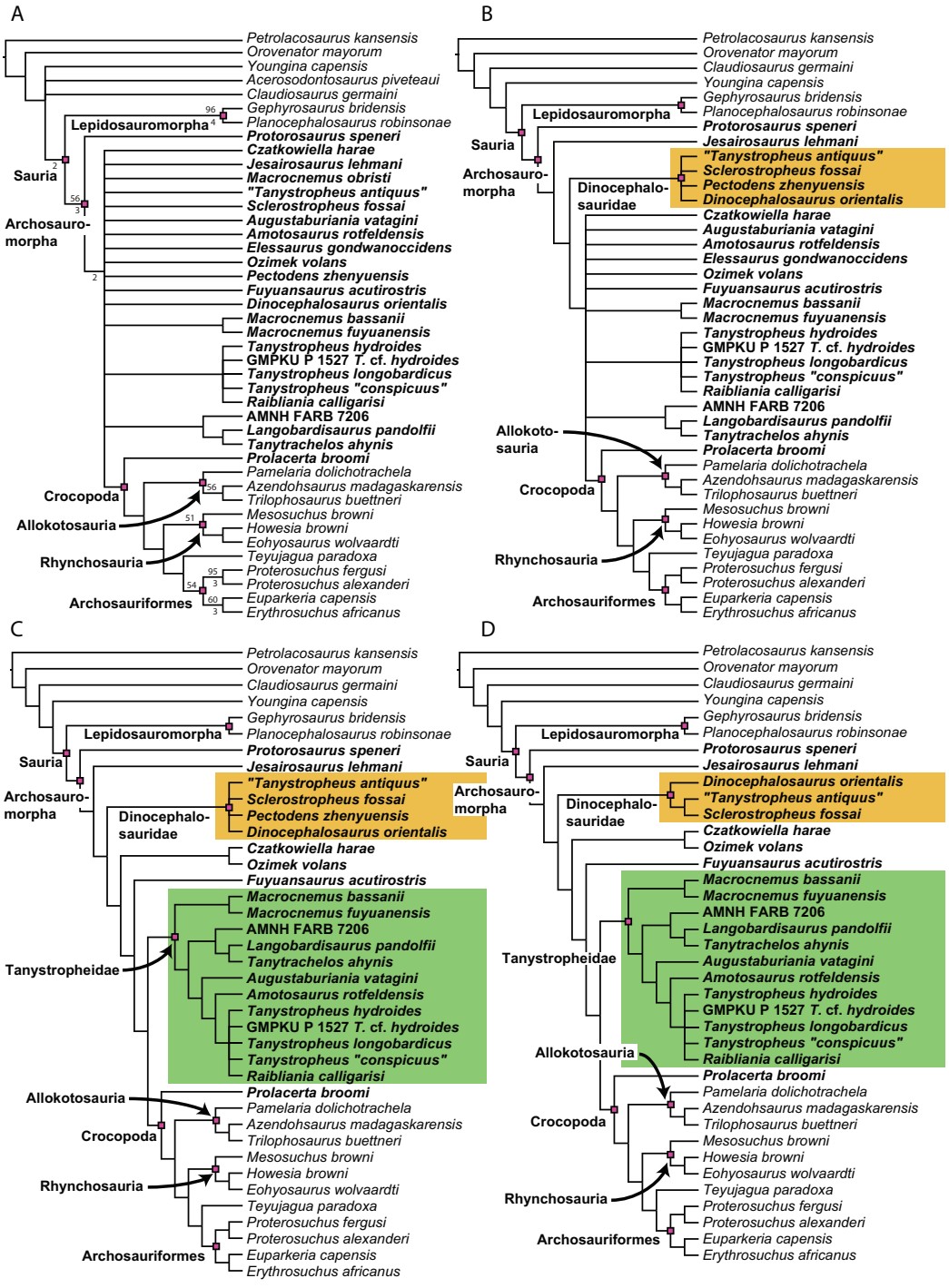

**Figure 33 Results of analysis 1 (ratio characters excluded and all characters treated as unordered; all OTUs included).** (A) Strict consensus tree out of 1976 MPTs with 977 steps. Bremer values above 1 and Bootstrap frequencies above 50% are provided above and below each node, respectively. (B) First reduced strict consensus tree after the *a posteriori* exclusion of *Acerosodontosaurus piveteaui* and *Macrocnemus obristi*. (C) Second reduced strict consensus tree after additionally excluding *Elessaurus gondwanoccidens a posteriori*. (D) Third reduced strict consensus tree after additionally excluding *Pectodens zhenyuensis a posteriori*. OTUs formerly considered to be "protorosaurs" are highlighted in bold.

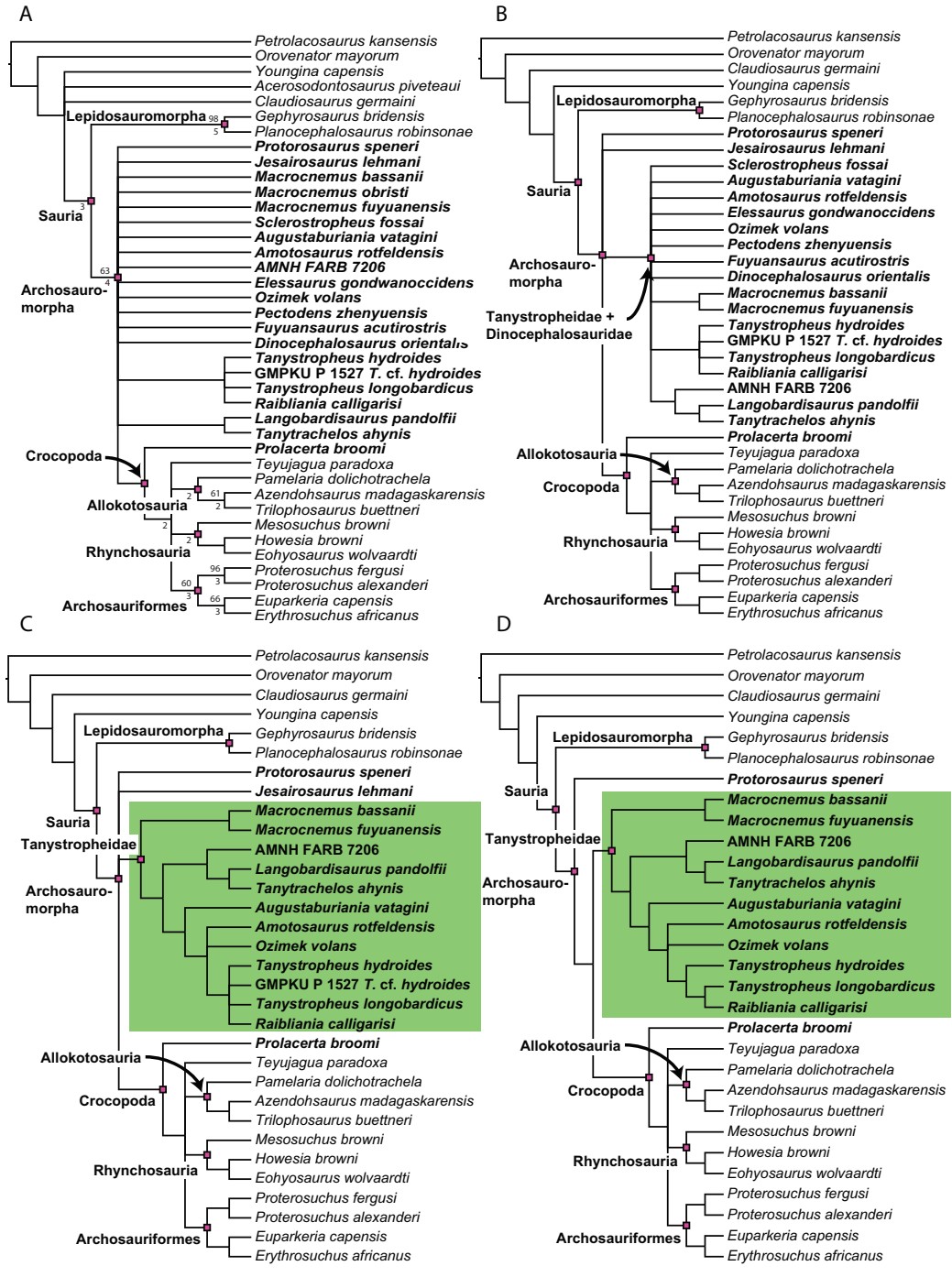

**Figure 34 Results of analysis 2 (ratio characters excluded and all characters treated as unordered;** *Czatkowiella harae, Tanystropheus "conspicuus",* **and** *"Tanystropheus antiquus"* **pruned** *a priori***).** (A) Strict consensus tree out of 884 MPTs with 953 steps. Bremer values above 1 and Bootstrap frequencies above 50% are provided above and below each node, respectively. (B) First reduced strict consensus tree after the *a posteriori* exclusion of *Acerosodontosaurus piveteaui* and *Macrocnemus obristi*. (C) Second reduced strict consensus tree after additionally excluding *Sclerostropheus fossai, Elessaurus gondwanoccidens, Pectodens zhenyuensis, Fuyuansaurus acutirostris,* and *Dinocephalosaurus orientalis a posteriori*. (D) Third reduced strict consensus tree after additionally excluding *Jesairosaurus lehmani* and GMPKU P 1527 *a posteriori*. OTUs formerly considered to be "protorosaurs" are highlighted in bold.

formed by all *Tanystropheus* species except *"Tanystropheus antiquus"* and additionally including *Raibliania calligarisi*. The SCT of analysis 2, from which both *Tanystropheus "conspicuus"* and *"Tanystropheus antiquus"* were omitted *a priori*, also recovers a polytomic clade of all remaining *Tanystropheus* OTUs and *Raibliania calligarisi*. Another clade formed by OTUs generally considered to be tanystropheid is recovered in the SCTs of both analyses. In analysis 1 this clade is formed by *Langobardisaurus pandolfii* and *Tanytrachelos ahynis* and in analysis 2 it is formed by AMNH FARB 7206, *Langobardisaurus pandolfii*, and *Tanytrachelos ahynis* as successive sister taxa. The remaining "protorosaurs" do not form more inclusive clades within the large polytomy at the base of Archosauromorpha in both SCTs.

The relationships within Crocopoda are fully resolved in the SCT of analysis 1, with successive sister groups being formed by *Prolacerta broomi*, Allokotosauria, Rhynchosauria, *Teyujagua paradoxa*, and Archosauriformes in all MPTs (Fig. 33A). In the SCT of analysis 2 *Prolacerta broomi* is the sister taxon to a polytomy formed by *Teyujagua paradoxa*, Allokotosauria, Rhynchosauria, and Archosauriformes (Fig. 34A). The position of *Prolacerta broomi* as the crocopodan most distantly related to Archosauriformes differs from other recent analyses in which *Prolacerta broomi* was found to be more closely related to Archosauriformes than both rhynchosaurs and allokotosaurs (e.g., *Ezcurra, 2016*; *Nesbitt et al., 2015*; *Pinheiro, Simão-Oliveira & Butler, 2019*; *Pritchard et al., 2018*; *Pritchard & Sues, 2019*; *Pritchard et al., 2015*; *Spiekman, 2018*). The relationships within Allokotosauria, Rhynchosauria, and Archosauriformes are the same between both analyses (Figs. 33A and 34A) and are in congruence with that of previous phylogenetic studies. Within Allokotosauria *Pamelaria dolichotrachela* is the sister taxon to the clade formed by *Trilophosaurus buettneri* and *Azendohsaurus madagaskarensis*. Rhynchosauria consists of *Mesosuchus browni*, *Eohyosaurus wolvaardti*, and *Howesia browni* as successive sister taxa. Within Archosauriformes *Euparkeria capensis* is the sister taxon to *Erythrosuchus africanus* and a generic clade is formed by *Proterosuchus fergusi* and *Proterosuchus alexanderi*.

The iter PCR option found four unstable OTUs for analysis 1. The first RSCT of analysis 1 was generated after the exclusion *a posteriori* of *Acerosodontosaurus piveteaui* and *Macrocnemus obristi* (Fig. 33B). In this RSCT the relationships of non-saurian diapsids are resolved, with *Claudiosaurus germaini* forming the sister taxon to *Youngina capensis* + Sauria. The *a posteriori* pruning of *Macrocnemus obristi* increases the resolution among non-archosauriform archosauromorphs. *Protorosaurus speneri* and *Jesairosaurus lehmani* form successive sister taxa to all other archosauromorphs. An additional clade is also recovered among non-archosauriform archosauromorphs, formed by a polytomy of *"Tanystropheus antiquus"*, *Sclerostropheus fossai*, *Pectodens zhenyuensis*, and *Dinocephalosaurus orientalis*. This clade is the sister group to the remainder of the large polytomy that is present in RSCT 1. *"Tanystropheus antiquus"* and *Sclerostropheus fossai* have generally been recognized as tanystropheids closely related to *Tanystropheus* spp. (*Spiekman & Scheyer, 2019*). The phylogenetic placement of the Chinese "protorosaurs" *Dinocephalosaurus orientalis* and *Pectodens zhenyuensis* has been more elusive (*Li et al.,*

*2017*; *Rieppel, Li & Fraser, 2008*), although they have been tentatively referred to Tanystropheidae in some studies (*Ezcurra & Butler, 2018*; *Li, 2003*; *Liu et al., 2017*).

The RSCT 2 of analysis 1 was generated after the additional *a posteriori* pruning of *Elessaurus gondwanoccidens*, which further increases the resolution among non-archosauriform archosauromorphs (Fig. 33C). In this RSCT a monophyletic Tanystropheidae clade is recovered. A generic clade of *Macrocnemus bassanii* and *Macrocnemus fuyuanensis* forms the sister group to all other taxa forming Tanystropheidae. AMNH FARB 7206, *Tanytrachelos ahynis*, and *Langobardisaurus pandolfii* are successive sister groups and together form the sister clade to all remaining members of the Tanystropheidae clade except the included *Macrocnemus* OTUs. *Augustaburiania vatagini* and *Amotosaurus rotfeldensis* form successive sister groups to the polytomic clade composed of *Raibliania calligarisi* and the *Tanystropheus* OTUs except *"Tanystropheus antiquus"*. An additional clade is formed by *Czatkowiella harae* and *Ozimek volans*. This clade is found as the sister group to a clade composed of *Fuyuansaurus acutirostris*, Tanystropheidae, and Crocopoda.

The *a posteriori* pruning of *Pectodens zhenyuensis* additionally found *"Tanystropheus antiquus"* and *Sclerostropheus fossai* as sister taxa, with *Dinocephalosaurus orientalis* forming the sister taxon to this clade in RSCT 3 of analysis 1 (Fig. 33D). The exclusion of *Pectodens zhenyuensis* is remarkable, since this OTU possesses a low amount of missing data (45.6%) compared to other OTUs in that clade (e.g., *Sclerostropheus fossai* 4.9%, and *"Tanystropheus antiquus"* 5.2%; Table 1).

For analysis 2 nine OTUs were excluded by the iter PCR function. As for analysis 1, RSCT 1 of analysis 2 was calculated after the *a posteriori* exclusion of *Acerosodontosaurus piveteaui* and *Macrocnemus obristi* and the relationships among non-saurian diapsids match those of RSCT 1 of analysis 1 (Fig. 34B). The exclusion of *Macrocnemus obristi* results in the formation of a clade that includes all remaining "protorosaurs" except *Protorosaurus speneri*, *Jesairosaurus lehmani*, and *Prolacerta broomi*. The relationships within this new clade are not further resolved relative to the SCT. This large and poorly resolved clade forms a polytomy with *Protorosaurus speneri*, *Jesairosaurus lehmani*, and Crocopoda.

*Sclerostropheus fossai*, *Elessaurus gondwanoccidens*, *Pectodens zhenyuensis*, *Fuyuansaurus acutirostris*, and *Dinocephalosaurus orientalis* are additionally excluded *a posteriori* in RSCT 2 and a monophyletic Tanystropheidae is recovered as a consequence. As for RSCT 3 of analysis 1, despite possessing a relatively low amount of missing data, *Pectodens zhenyuensis* is pruned (Table 1), in addition to *Dinocephalosaurus orientalis*, which poseses even fewer missing data (64.8%). The monophyletic Tanystropheidae recovered in RSCT 2 of analysis 2 is virtually identical to that of RSCTs 2 and 3 of analysis 1. However, in contrast to these analyses, the Tanystropheidae clade of RSCT 2 of analysis 2 includes *Ozimek volans*, which forms a trichotomy with *Amotosaurus rotfeldensis* and a clade containing all included *Tanystropheus* OTUs and *Raibliania calligarisi* (Fig. 34C). This last clade differs from that of RSCTs 2 and 3 of analysis 1 in the absence of *Tanystropheus "conspicuus"*, which was pruned *a priori* for this analysis.

**Table 1 Percentage of scored characters of each OTU for the character matrix used in this study.**

| OTU | % of characters scored |
| --- | --- |
| *Petrolacosaurus kansensis* | 87.3 |
| *Orovenator mayorum* | 47.6 |
| *Youngina capensis* | 90.6 |
| *Acerosodontosaurus piveteaui* | 31.6 |
| *Claudiosaurus germaini* | 64.2 |
| *Gephyrosaurus bridensis* | 78.2 |
| *Planocephalosaurus robinsonae* | 75.6 |
| *Czatkowiella harae* | 50.8 |
| *Protorosaurus speneri* | 73.3 |
| *Jesairosaurus lehmani* | 54.7 |
| *Macrocnemus bassanii* | 93.2 |
| *Macrocnemus obristi* | 8.1 |
| *Macrocnemus fuyuanensis* | 71.3 |
| *Tanystropheus hydroides* | 91.9 |
| GMPKU P1527 *T. cf. hydroides* | 24.4 |
| *Tanystropheus longobardicus* | 79.8 |
| *Tanystropheus conspicuus* | 4.6 |
| "*Tanystropheus antiquus*" | 5.2 |
| *Sclerostropheus fossai* | 4.9 |
| *Raibliania calligarisi* | 9.4 |
| *Augustaburiania vatagini* | 8.8 |
| *Langobardisaurus pandolfii* | 49.5 |
| *Amotosaurus rotfeldensis* | 45.9 |
| AMNH FARB 7206 | 6.2 |
| *Tanytrachelos ahynis* | 37.1 |
| *Ozimek volans* | 31.3 |
| *Elessaurus gondwanoccidens* | 8.1 |
| *Pectodens zhenyuensis* | 45.6 |
| *Fuyuansaurus acutirostris* | 25.7 |
| *Dinocephalosaurus orientalis* | 64.8 |
| *Prolacerta broomi* | 96.7 |
| *Pamelaria dolichotrachela* | 59.3 |
| *Azendohsaurus madagaskarensis* | 94.5 |
| *Trilophosaurus buettneri* | 89.3 |
| *Mesosuchus browni* | 89.3 |
| *Howesia browni* | 46.6 |
| *Eohyosaurus wolvaardti* | 25.7 |
| *Teyujagua paradoxa* | 47.9 |
| *Proterosuchus fergusi* | 56.0 |
| *Proterosuchus alexanderi* | 74.9 |
| *Euparkeria capensis* | 90.2 |
| *Erythrosuchus africanus* | 84.4 |

In RSCT 3 of analysis 2 two additional OTUs were pruned *a posteriori*, *Jesairosaurus lehmani* and GMPKU P 1527. The exclusion of the former results in a topology of *Protorosaurus speneri*, Tanystropheidae, and Crocopoda as successive sister groups among archosauromorphs (Fig. 34D). The exclusion of GMPKU P 1527, currently assigned to *Tanystropheus* cf. *T. hydroides*, resolves the relationships between *Tanystropheus hydroides*, *Tanystropheus longobardicus*, and *Raibliania calligarisi*, with the last two being sister taxa.

## Analyses including ratio characters and treating designated characters as ordered

Analysis 3, which includes all OTUs and incorporates ratio and ordered characters (Fig. 35), found 434 MPTs of 1,270 steps and has a CI of 0.350 and a RI of 0.516, whereas analysis 4, which incorporates ratio and ordered characters and excludes *Czatkowiella harae*, *Tanystropheus "conspicuus"*, and *"Tanystropheus antiquus"* (Fig. 36), recovered 154 MPTs of 1,241 steps and a CI of 0.358 and a RI of 0.518. Support values for the branches are indicated in the SCTs of both analyses (Figs. 35a and 36a). The SCTs of both analyses show a higher resolution compared to those of analyses 1 and 2 and recover a monophyletic Tanystropheidae, as well as a new clade of non-archosauriform archosauromorphs composed of at least *Pectodens zhenyuensis* and *Dinocephalosaurus orientalis*. However, both SCTs contain a large polytomy within Tanystropheidae. Three consecutively more resolved RSCTs were calculated for analysis 3 (Figs. 35B–35D), whereas only a single RSCT could be found for analysis 4 (Fig. 36B). The relationships among non-saurian diapsids are fully resolved in the SCT of analysis 3, with *Petrolacosaurus kansensis*, *Orovenator mayorum*, *Claudiosaurus germaini*, *Acerosodontosaurus piveteaui*, *Youngina capensis*, and Sauria forming successive sister groups (Fig. 35A). Even though there are no differences in character configuration with analysis 3, in this part of the SCT of analysis 4 a polytomy is formed by all these taxa except for the outgroup *Petrolacosaurus kansensis* (Fig. 36a). As in analyses 1 and 2, a monophyletic Sauria, Lepidosauromorpha, Archosauromorpha, Crocopoda, Rhynchosauria, Allokotosauria, and Archosauriformes is recovered in all MPTs of analyses 3 and 4 (Figs. 35A and 36A). However, in contrast to the previous analyses, the SCTs of analyses 3 and 4 both did not recover *Prolacerta broomi* within Crocopoda. This contrasts distinctly with recent phylogenetic analyses in which *Prolacerta broomi* has been uniformly found to be very closely related to Archosauriformes (e.g., *Ezcurra, 2016*; *Nesbitt et al., 2015*; *Pinheiro, Simão-Oliveira & Butler, 2019*; *Pritchard et al., 2018*; *Pritchard & Sues, 2019*; *Pritchard et al., 2015*; *Spiekman, 2018*). Furthermore, whereas the SCTs analyses 1 and 2 found large polytomies at the base of non-archosauriform archosauromorphs, those of analyses 3 and 4 show a higher resolution and in all MPTs of both analyses a monophyletic Tanystropheidae is recovered, as well as a previously unrecognized clade. In the SCT of analysis 3 this clade is formed by a polytomy of *Pectodens zhenyuensis*, *Dinocephalosaurus orientalis*, and *"Tanystropheus antiquus"* (Fig. 35A). In the SCT of analysis 4 it is formed only by *Pectodens zhenyuensis* and *Dinocephalosaurus orientalis*, with *Jesairosaurus lehmani* forming the sister taxon to this

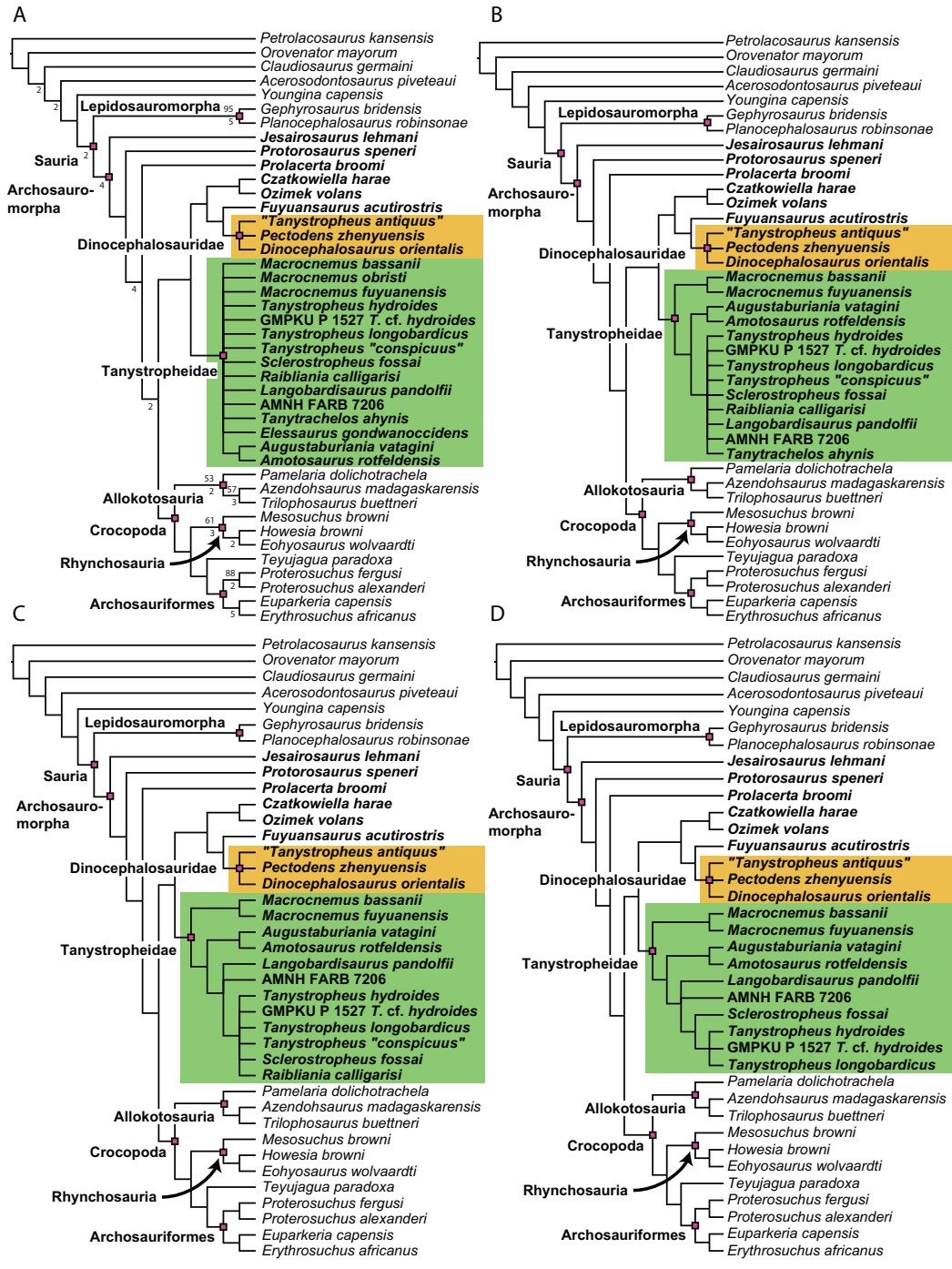

**Figure 35 Results of analysis 3 (ratio characters included and specified characters ordered; all OTUs included).** (A) Strict consensus tree out of 434 MPTs with 1,270 steps. Bremer values above 1 and Bootstrap frequencies above 50% are provided above and below each node, respectively. (B) First reduced strict consensus tree after the *a posteriori* exclusion of *Macrocnemus obristi* and *Elessaurus gondwanoccidens*. (C) Second reduced strict consensus tree after additionally excluding *Tanytrachelos ahynis a posteriori*. (D) Third reduced strict consensus tree after additionally excluding *Tanystropheus "conspicuus"* and *Raibliania calligarisi a posteriori*. OTUs formerly considered to be "protorosaurs" are highlighted in bold.

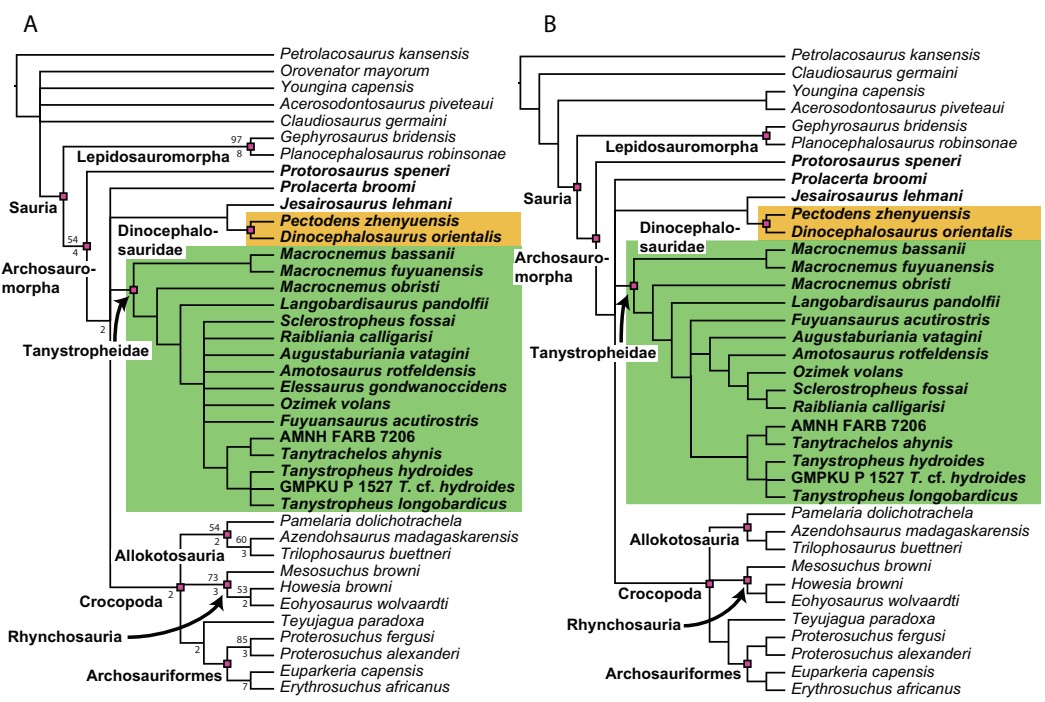

**Figure 36 Results of analysis 4 (ratio characters included and specified characters ordered;** *Czatkowiella harae, Tanystropheus "conspicuus"*, **and** *"Tanystropheus antiquus"* **pruned** *a priori*). (A) Strict consensus tree out of 154 MPTs with 1,241 steps. Bremer values above 1 and Bootstrap frequencies above 50% are provided above and below each node, respectively. (B) Reduced strict consensus tree after the *a posteriori* exclusion of *Orovenator mayorum* and *Elessaurus gondwanoccidens*. OTUs formerly considered to be "protorosaurs" are highlighted in bold.

clade (Fig. 36A). In all MPTs of analysis 3 *Jesairosaurus lehmani*, *Protorosaurus speneri*, and *Prolacerta broomi* form successive sister groups leading up to a clade composed of all remaining non-archosauriform archosauromorphs (Fig. 35A). This last clade is split into all taxa previously considered as crocopodans except *Prolacerta broomi*, and a clade encompassing Tanystropheidae, *Czatkowiella harae*, *Ozimek volans*, *Fuyuansaurus acutirostris*, and the new clade formed by *Pectodens zhenyuensis*, *Dinocephalosaurus orientalis*, and *"Tanystropheus antiquus"*. *Fuyuansaurus acutirostris* is the sister taxon to this last clade, and *Czatkowiella harae* and *Ozimek volans* form a clade that is sister to the clade formed by *Fuyuansaurus acutirostris* and the trichotomy that includes *Dinocephalosaurus orientalis*. In contrast, *Protorosaurus speneri* is found as the sister taxon to all remaining archosauromorphs in all MPTs of analysis 4 (Fig. 36A). These remaining archosauromorphs form a polytomy in the SCT of analysis 4 consisting of *Prolacerta broomi*, a monophyletic Tanystropheidae, a clade composed of all OTUs previously considered as crocopodans excluding *Prolacerta broomi*, and the clade formed by *Jesairosaurus lehmani*, *Pectodens zhenyuensis*, and *Dinocephalosaurus orientalis*. Tanystropheidae form a large polytomy in the SCT of analysis 3 and only one less inclusive clade is recovered within it, formed by *Amotosaurus rotfeldensis* and *Augustaburiania*

*vatagini* (Fig. 35A). The tanystropheid clade in the SCT of analysis 4 is considerably more resolved (Fig. 36A). *Macrocnemus bassanii* and *Macrocnemus fuyuanensis* are recovered in a clade that is the sister group to all remaining tanystropheids. *Macrocnemus obristi* does not constitute a direct sister taxon to the other two *Macrocnemus* OTUs and is instead found as the sister taxon to all other tanystropheids. *Langobardisaurus pandolfii* is recovered as the sister taxon to a large polytomy within Tanystropheidae, which is formed by all other tanystropheids except the *Macrocnemus* OTUs. Within this polytomy less inclusive clades are formed by AMNH FARB 7206 and *Tanytrachelos ahynis*, and by a trichotomy *Tanystropheus hydroides*, *Tanystropheus longobardicus*, and GMPKU P 1527. These two clades form the sister groups to each other. The relationships among allokotosaurs, rhynchosaurs, and archosauriforms are consistent among all MPTs in analysis 3 (Fig. 35A). Crocopoda is formed by Allokotosauria, Rhynchosauria, *Teyujagua paradoxa*, and Archosauriformes as successive sister groups, and the relationships within these clades are identical to those recovered in analyses 1 and 2. In the SCT of analysis 4, a polytomy is formed within Crocopoda by Allokotosauria, Rhynchosauria, and a clade composed of *Teyujagua paradoxa* and Archosauriformes (Fig. 36A). The relationships within Allokotosauria, Rhynchosauria, and Archosauriformes in both analyses is identical to that recovered in the MPTs of the other two analyses.

Five unstable OTUs were identified for analysis 3 by the iter PCR function and three RSCT were calculated. The first RSCT was generated after the exclusion *a posteriori* of *Macrocnemus obristi* and *Elessaurus gondwanoccidens*, which improved the resolution in the tanystropheid clade (Fig. 35B). *Macrocnemus bassanii* and *Macrocnemus fuyuanensis* are recovered in a generic clade that is the sister taxon to all remaining tanystropheids, and the clade formed by *Amotosaurus rotfeldensis* and *Augustaburiania vatagini* is found as the sister clade to the large polytomy formed by all remaining tanystropheids except the two *Macrocnemus* OTUs.

The second RSCT of analysis 3 additionally excludes *Tanytrachelos ahynis* and gains one additional node, which forms a polytomy consisting of *Sclerostropheus fossai*, *Raibliania calligarisi*, and all *Tanystropheus* OTUs except *"Tanystropheus antiquus"* (Fig. 35C).

RSCT 3 of analysis 3 is formed after the additional *a posteriori* pruning of *Tanystropheus "conspicuus"* and *Raibliania calligarisi*. In this RSCT, in addition to the previous improvements, *Sclerostropheus fossai* is recovered as the sister taxon to a trichotomy formed by *Tanystropheus longobardicus*, *Tanystropheus hydroides*, and GMPKU P 1527 (Fig. 35D).

Only two unstable OTUs were identified by the iter PCR function for analysis 4, resulting in a single RSCT for this analysis (Fig. 36B). The relationships among non-saurian diapsids are resolved after the *a posteriori* exclusion of *Orovenator mayorum*. In the resulting RSCT *Claudiosaurus germaini* is found as the sister taxon to Sauria and a clade composed of *Acerosodontosaurus piveteaui* and *Youngina capensis*. The *a posteriori* exclusion of *Elessaurus gondwanoccidens* results in the recovery of an additional clade within Tanystropheidae, which is composed of *Augustaburiania vatagini*, *Amotosaurus rotfeldensis*, *Ozimek volans*, *Sclerostropheus fossai*, and *Raibliania calligarisi* as successive

sister taxa. The position of *Raibliania calligarisi* is noteworthy, since this taxon is considered to be closely related to *Tanystropheus longobardicus* (*Dalla Vecchia, 2020*) and is recovered in a clade with *Tanystropheus longobardicus* and *Tanystropheus hydroides* in the SCTs of analyses 1, 2, and 3.

We tested several relevant alternative topologies with our data matrix by performing iterations of analysis 4 with specific constraints applied to the tree topology. Thus, we tested how many additional steps are needed to either recover alternative clades or force an OTU outside a clade recovered by the unconstrained iteration of analysis 4 (Fig. 36A). We found that 14 additional steps were required to achieve monophyly for "Protorosauria" (constraining a monophyletic group exclusively composed of the following clades and OTUs recovered in the unconstrained iteration of analysis 4: Tanystropheidae + Dinocephalosauridae + *Jesairosaurus lehmani* + *Protorosaurus speneri* + *Prolacerta broomi*). One additional step is required to recover a clade composed exclusively of Dinocephalosauridae and Tanystropheidae. We also tested several ingroup relationships of Tanystropheidae and found that one additional step was required to recover monophyly for *Macrocnemus* spp. (i.e., *Macrocnemus obristi* in an exclusive clade with *Macrocnemus bassanii* and *Macrocnemus fuyuanensis*) with the three *Macrocnemus* OTUs forming a trichotomy. *Langobardisaurus pandolfii* and *Tanytrachelos ahynis* were recovered as sister taxa by *Pritchard et al. (2015)*, with AMNH FARB 7206 being incorporated as part of the *Tanytrachelos* OTU therein. For analysis 4 of our data matrix two additional steps were required to recover *Langobardisaurus pandolfii* in a clade exclusively with *Tanytrachelos ahynis* and AMNH FARB 7206. It required one additional step to force *Fuyuansaurus acutirostris*, a taxon previously considered to possess several features typically not exhibited in tanystropheids (*Fraser, Rieppel & Li, 2013*), outside Tanystropheidae, and three additional steps to recover the enigmatic *Ozimek volans* outside Tanystropheidae. The extreme elongation of the neck and adaptations to an aquatic lifestyle shared by *Dinocephalosaurus orientalis* and *Tanystropheus* spp. have widely been considered a striking convergence (*Li, Rieppel & LaBarbera, 2004*) and six additional steps are required to recover *Dinocephalosaurus orientalis* and *Tanystropheus* spp. as sister taxa.

## Clade definitions and synapomorphies

Analysis 4 represents the most stable analysis as is indicated by the relatively low number of MPTs for this analysis, the comparatively high resolution of its SCT, and the identification of only two unstable OTUs. The clades recovered by all MPTs of analysis 4 are defined as follows (listed unambiguous synapomorphies were common to all MPTs of analysis 4).

**Sauria** *Gauthier, 1984*
**Definition.** The most recent common ancestor of archosaurs and lepidosaurs, and all its descendants (*Gauthier, Kluge & Rowe, 1988*).
**Temporal range.** Wuchiapingian (late Permian, *Protorosaurus speneri*; *Ezcurra, Scheyer & Butler, 2014*; *Legler & Schneider, 2008*) to Recent (*Crocodylus niloticus*).

**Unambiguous synapomorphies.** Postparietal absent as a separate ossification (84: 0 → 2); tabular absent (86: 0 → 1); posterior margin of quadrate continuously concave in lateral view (90: 0 → 1); intertuberal plate of parabasisphenoid absent (117: 0 → 1); retroarticular process anteroposteriorly long, extending considerably posterior to the glenoid fossa (162: 1 → 2); chevrons broaden gradually distally (223: 0 → 2).

**Lepidosauromorpha** *Gauthier, 1984*
**Definition.** *Sphenodon* and squamates and all saurians sharing a more recent common ancestor with them than they do with crocodiles and birds (*Gauthier, Estes & De Queiroz, 1988*).
**Temporal range.** Induan-early Olenekian (Early Triassic, *Paliguana whitei*; *Rubidge, 2005*; *Lucas, 2010*) to Recent (*Varanus niloticus*).
**Unambiguous synapomorphies.** Antorbital skull length versus total skull length between 0.35 and 0.37 (1: 0 → 1); anterior projection of the anterior process of the jugal up to or posterior to the level of mid-length of the orbit (37: 0 → 1); frontals fused to one another (54: 0 → 1); anterior process of the squamosal with a continuous contact along the posterior margin of the ventral process of the postorbital and contacting the jugal (64: 0 → 1); supratemporal absent (73: 1 → 2); quadratojugal absent or fused to quadrate (87: 1 → 0); broad palatine, forming the main component of the palate posterior to the choanae (98: 0 → 1); palatal dentition relatively large, similar size to marginal dentition (101: 0 → 1); pterygoid teeth present in three fields (T2, T3a and T3b) (102: 0 → 1); splenial absent (143: 0 → 1); absence of a posterodorsal process of the dentary (148: 1 → 0); presence of a posterocentral process of the dentary (149: 0 → 1); atlas centrum fused to axial intercentrum (183: 0 → 1); transverse processes of caudal vertebrae distinctly angled posterolaterally from base (221: 0 → 1); chevrons of mid-caudal vertebrae with anterior curvature (222: 0 → 1); ectepicondylar region of humerus with foramen (247: 1 → 0); anteroventral process of ilium draping across anterior surface of pubis (272: 0 → 1); thyroid fenestra present (273: 0 → 1); astragalus and calcaneum fused (289: 0 → 2); presence of medial plantar tubercle of metatarsal V (304: 0 → 1).

**Archosauromorpha** *Huene, 1946*
**Definition.** *Protorosaurus* and all other saurians that are related more closely to *Protorosaurus* than to Lepidosauria (*Dilkes, 1998*).
**Temporal range.** Wuchiapingian (late Permian, *Protorosaurus speneri*; *Ezcurra, Scheyer & Butler, 2014*; *Legler & Schneider, 2008*) to Recent (*Crocodylus niloticus*).
**Unambiguous synapomorphies.** Alveolar margin of maxilla concave in lateral view (25 0 → 1); supratemporal fossa of parietal expanded distinctly medially, resulting in a mediolaterally narrow parietal table (83: 0 → 1); anterior half of the surangular-angular suture anteroposteriorly concave ventrally in lateral view (158: 2 → 0); upturned retroarticular process (163: 0 → 1); diapophysis and parapophysis of anterior to mid-postaxial cervical vertebrae on different processes and nearly touching (188: 0 → 2); anterior margin of the neural spine of anterior and mid-postaxial cervical vertebrae anterodorsally inclined at an angle of more than 60 degrees from the horizontal plane (194: 0 → 1); long transverse

processes of mid-dorsal vertebrae (207: 0 → 1); presence of a shallow fossa lateral to the base of the neural spine of dorsal vertebrae (209: 0 → 1); coracoid moderately expanded posteriorly in lateral view (232: 2 → 1); proximal half of interclavicle subtriangular or diamond-shaped (237: 1 → 0); torsion between proximal and distal ends of humerus 35 degrees or less (241: 0 → 1); entepicondylar foramen of humerus absent (246: 0 → 1).

### Unnamed clade (*Prolacerta broomi* + *Jesairosaurus lehmani* + Dinocephalosauridae + Tanystropheidae + Allokotosauria + Rhynchosauria + *Teyujagua paradoxa* + Archosauriformes)

Unambiguous synapomorphies. Postnarial process of premaxilla well-developed, forming most of the ventral border of the external naris or excludes the maxilla from participation in the external naris but process does not contact prefrontal (9: 0 → 2); maxilla with abruptly ending dorsal apex and a concave posterior margin (20: 0 → 1); supraglenoid foramen of the scapula absent (231: 1 → 0); anterior margin of interclavicle with a median notch (236: 0 → 1); metatarsal V with a smooth curved proximal lateral margin (302: 1 → 0).

### Unnamed clade (*Jesairosaurus* + Dinocephalosauridae)

Unambiguous synapomorphies. Anterior process of the squamosal with a continuous contact along the posterior margin of the ventral process of the postorbital and contacting the jugal (64: 0 → 1); length of the postacetabular process of the ilium versus anteroposterior length of the acetabulum between 0.50 and 0.71 (267: 3 → 0); tibia longer than femur (285: 0 → 1).

### Dinocephalosauridae new clade

**Definition.** The most recent common ancestor of *Pectodens zhenyuensis* and *Dinocephalosaurus orientalis* and all of its descendants (node-based). A node-based definition of the new Dinocephalosauridae clade is preferred here because of the uncertain phylogenetic position of *Jesairosaurus lehmani*.

**Temporal range.** Anisian (Middle Triassic; *Sun et al., 2016*).

**Unambiguous synapomorphies.** Absence of a posterior process of the jugal (42: 0 → 1); glenoid fossa considerably ventrally displaced compared to the tooth row (161: 0 → 1); postaxial anterior or mid-cervical neural spine considerably shorter than the posterior articular surface of the centrum (178: 0/1 → 2); presence of between 11 and 13 cervical vertebrae (195: 1 → 2); anterior free-ending process of anterior cervical ribs present and long, extending anterior to the prezygapophyses of the corresponding vertebra when in articulation (200: 1 → 2).

### Tanystropheidae *Camp, 1945*

**Definition.** The most recent common ancestor of *Macrocnemus*, *Tanystropheus*, and *Langobardisaurus* and all of its descendants (*Dilkes, 1998*).

**Temporal range.** Induan-Olenekian (Early Triassic, *Elessaurus gondwanoccidens*; *De Oliveira et al., 2020*) to late Norian (late Late Triassic, *Sclerostropheus fossai*; *Rigo, Galli & Jadoul, 2009*; *Tackett & Tintori, 2019*).

**Unambiguous synapomorphies.** Dorsoventral height at the level of the anterior tip of the maxilla versus dorsoventral height at the level of the anterior border of the orbit between 0.20 and 0.27 (2: 1 → 0); prenarial process of premaxilla absent or incipient (7: 2 → 0); straight anterior part of the dorsal margin of the maxilla (21: 0 → 1); supratemporal absent (73: 1 → 2); distal margin of anterior and mid-postaxial cervical neural spines completely straight along anteroposterior length in lateral view (179: 0 → 1); anterior to mid-postaxial cervical neural spines with distal expansion restricted to the distal end of the neural spine (=spine table) (180: 0 → 2); neural spine of axis expanded anterodorsally in lateral view (186: 0 → 1); anterior margin of the neural spine of anterior and mid-postaxial cervical vertebrae anterodorsally inclined at an angle of less than 60 degrees from the horizontal plane (194: 1 → 2); height versus maximum anteroposterior length of anterior caudal neural spine between 0.42 and 0.83 (220: 1 → 0); presence of a large fenestra between scapula and coracoid (scapulocoracoidal fenestra) (227: 0 → 2); length of the longest metacarpal versus length of the longest metatarsal between 0.36 and 0.41 (258: 2/3 → 1); thyroid fenestra present (273: 0 → 1); length of metatarsal I versus metatarsal III between 0.54 and 0.63 (297: 1 → 2); length of metatarsal IV versus the proximodistal length of metatarsal V between 3.65 and 5.15 (305: 1 → 3).

*Macrocnemus bassanii + Macrocnemus fuyuanensis*
   Unambiguous synapomorphies. Pineal foramen of the parietal absent (77: 0 → 2); presence of teeth on the lateral ramus of the pterygoid (106: 0 → 1); presence of a medial plantar tubercle on metatarsal V (304: 0 → 1).

**Unnamed clade (*Macrocnemus obristi + Langobardisaurus pandolfii + Fuyuansaurus acutirostris + Ozimek volans + Elessaurus gondwanoccidens + Amotosaurus rotfeldensis + Augustaburiania vatagini + Raibliania vatagini + Sclerostropheus fossai + Tanytrachelos ahynis + AMNH FARB 7206 + Tanystropheus spp.*)**
   Unambiguous synapomorphies. There are no unambiguous synapomorphies that are shared by all MPTs for this node.

**Unnamed clade (*Langobardisaurus pandolfii + Fuyuansaurus acutirostris + Ozimek volans + Elessaurus gondwanoccidens + Amotosaurus rotfeldensis + Augustaburiania vatagini + Raibliania vatagini + Sclerostropheus fossai + Tanytrachelos ahynis* + AMNH FARB 7206 + *Tanystropheus* spp.)**
   Unambiguous synapomorphies. Length of metatarsal IV versus length of metatarsal III between 1.03 and 1.08 (299: 2 → 1); metatarsal V with an abruptly medially flexed proximal process (300: 2 → 1); phalanx V-1 is metatarsal-like, considerably longer than other non-ungual phalanges (306: 0 → 1).

**Unnamed clade (*Fuyuansaurus acutirostris + Ozimek volans + Elessaurus gondwanoccidens + Amotosaurus rotfeldensis + Augustaburiania vatagini + Raibliania vatagini + Sclerostropheus fossai + Tanytrachelos ahynis* + AMNH FARB 7206 + *Tanystropheus* spp.)**

Unambiguous synapomorphies. Concave anterior part of the dorsal margin of the maxilla (21: 1 → 2); not upturned retroarticular process (163: 1 → 0); epipophyses of cervical vertebrae overhanging the postzygapophysis posteriorly (191: 0 → 1); absence of interclavicle (234: 0 → 1); absence of posterior groove on astragalus (290: 0 → 1).

**Unnamed clade (*Tanytrachelos ahynis* + AMNH FARB 7206 + *Tanystropheus* spp.)**

Unambiguous synapomorphies. Glenoid fossa considerably ventrally displaced compared to the tooth row (161: 0 → 1); length of the postacetabular process of the ilium versus anteroposterior length of the acetabulum between 0.98 and 1.14 (267: 3 → 2).

**Unnamed clade (*Tanytrachelos ahynis* + AMNH FARB 7206)**

Unambiguous synapomorphies. Procoelous presacral vertebrae (181: 2 → 1); cervical ribs short, being less than two times the length of its respective vertebra, and shaft parallel to the neck (199: 2 → 1); preacetabular process of the ilium present and longer than two thirds of its height (264: 1 → 2).

**Tanystropheus spp. (part of *Meyer, 1847-1855*)**

Unambiguous synapomorphies. Postaxial anterior or mid-cervical neural spines are only present at the anterior and posterior ends of the vertebrae but are completely or virtually lost at their anteroposterior midpoints (178: 2 → 3); lengths of the fourth or fifth cervical centra versus the heights of their anterior articular surfaces between 14.58 and 15.58 or between 17.08 and 18.67 (187: 1 → 2/3); total length of the humerus versus the total length of the femur between 0.84 and 0.91 (248: 3/4 → 2).

**Crocopoda *Ezcurra, 2016***

**Definition.** All taxa more closely related to *Azendohsaurus madagaskarensis*, *Trilophosaurus buettneri*, *Rhynchosaurus articeps* and *Proterosuchus fergusi* than to *Protorosaurus speneri* or *Tanystropheus longobardicus* (*Ezcurra, 2016*).

**Temporal range.** Wuchiapingian (late Permian, *Eorasaurus olsoni*; *Ezcurra, Scheyer & Butler, 2014*) to Recent (*Crocodylus niloticus*).

**Unambiguous synapomorphies.** Alveolar margin of maxilla straight or convex in lateral view (25: 1 → 0/2); presence of a slightly elevated orbital rim formed by the margin of the jugal and/or postorbital (48: 0 → 1); roughly transverse orientation of the frontal-parietal contact in dorsal view (56: 1 → 0); posterolateral process of the parietal dorsoventrally deep, being plate-like in occipital view and subequal to the height of the supraoccipital (80: 0 → 1); height at the third alveolus of the dentary (or directly posterior to the tapering anterior end of the dentary in taxa with an anteriorly edentulous dentary) versus length of the alveolar margin (including edentulous anterior end if present) between 0.15 and 0.24 (144: 1 → 2); presence of hyposphene-hypantrum accessory intervertebral articulation in mid-posterior dorsal vertebrae (208: 0 → 1).

**Allokotosauria *Nesbitt et al., 2015***

**Definition.** The least-inclusive clade containing *Azendohsaurus madagaskarensis* and *Trilophosaurus buettneri* but not *Tanystropheus longobardicus*, *Proterosuchus fergusi*, *Protorosaurus speneri* or *Rhynchosaurus articeps* (*Nesbitt et al., 2015*).

**Temporal range.** Olenekian (Early Triassic, *Coelodontognathus donensis*; *Arkhangelskii & Sennikov, 2008*) to Revueltian (middle Norian, Late Triassic, *Trilophosaurus phasmalophos*; *Kligman et al., 2020*).

**Unambiguous synapomorphies.** Presence of rugose sculpturing on the lateral surface of the orbital margin of the prefrontal (46: 0 → 1); medial process of the squamosal long, forming entirely or almost entirely the posterior border of the supratemporal fenestra (70: 0 → 1); supratemporal absent (73: 1 → 2); presence of a posteriorly hooked dorsal end of the quadrate in lateral view (91: 0 → 1); vomerine teeth relatively large, similar in size to the marginal dentition (97: 0 → 1); palatine teeth relatively large, similar in size to marginal dentition (101: 0 → 1); teeth on the ventral surface of the anterior ramus of the pterygoid relatively large, similar in size to marginal dentition (105: 0 → 1); dentary ventrally curved or deflected at its anterior end (145: 0 → 2); retroarticular process anteroposteriorly short, being poorly developed posterior to the glenoid fossa (162: 2 → 1); lateral (=fibular) condyle of the femur projecting distally distinctly beyond medial (=tibial) condyle (282: 1 → 0); well-developed ventral tubercles of the pedal unguals, extended ventral to the articular portion of the ungual (307: 0 → 1).

### Unnamed clade (*Trilophosaurus buettneri* + *Azendohsaurus madagaskarensis*)

Unambiguous synapomorphies. Posterior end of the horizontal process of the maxilla distinctly ventrally deflected from the main axis of the alveolar margin (24: 0 → 1); postfrontal-frontal suture distinctly posteromedially inclined by a medial process of the postfrontal, resulting in posteriorly strongly narrowed frontal (57: 0 → 1); opisthotic and exoccipital fully fused (127: 0 → 1); paroccipital process of the opisthotic extends laterally or slightly posterolaterally (130: 1 → 0); multiple teeth of the marginal dentition with distinctly mesiodistally expanded crowns above the root (170: 0 → 1); second sacral rib does not bifurcate distally (216: 1 → 0); minimum anteroposterior length of the scapula less than a quarter of its proximodistal length (230: 1 → 2); presence of an olecranon process of the ulna (249: 0 → 1); presence of a lateral tuber on the proximal part of the ulna (250: 0 → 1); length of the postacetabular process of the ilium versus anteroposterior length of the acetabulum between 0.88 and 0.91 (267: 2 → 1); proximal articular surface of the femur well-ossified, being flat or convex (279: 1 → 0); attachment of the caudifemoralis musculature on the posterior surface of the femur crest-like and with intertrochanteric fossa (=internal trochanter), and not convergent with proximal end (280: 0 → 1).

### Rhynchosauria *Osborn, 1903*

**Definition.** All taxa more closely related to *Rhynchosaurus articeps* than to *Trilophosaurus buettneri*, *Prolacerta broomi* or *Crocodylus niloticus* (*Ezcurra, 2016*).

**Temporal range.** Induan-early Olenekian (Early Triassic, *Noteosuchus colletti*; *Rubidge, 2005*) to early Norian (Late Triassic, *Teyumbaita sulcognathus*; *Langer et al., 2007*; *Montefeltro, Langer & Schultz, 2010*).

**Unambiguous synapomorphies.** Lateral surface of the nasal meets entire dorsoventral height of medial surface of supra-alveolar portion of maxilla (34: 0 → 1); dorsal surface of frontals covered by shallow or deep pits across surface and/or low ridges (44: 0 → 1);

depression with deep pits on the dorsal surface of the postfrontal (59: 0 → 1); supratemporal present as broad element (73: 1 → 0); parietals fused with loss of suture (74: 0 → 1); fossa on the opisthotic immediately lateral to the foramen magnum (132: 0 → 1); presence of multiple Zahnreihen in maxilla and dentary (164: 0 → 1); crown base of the maxillary teeth circular in shape (174: 1 → 0); main axis of the postacetabular process of the ilium posterodorsally orientated in lateral or medial view (268: 1 → 0).

**Unnamed clade (*Howesia browni* + *Eohyosaurus wolvaardti*)**
Unambiguous synapomorphies. Presence of an anguli oris crest (27: 0 → 1); supratemporal fossa of the parietal well-exposed in dorsal view and mainly dorsally or dorsolaterally facing (82: 1 → 0); supratemporal fossa of parietal expanded distinctly medially and only separated from counterpart by a sagittal crest running along the midline of the parietal (83: 1 → 2).

**Unnamed clade (*Teyujagua paradoxa* + Archosauriformes)**
Unambiguous synapomorphies. Jugal bulges ventrolaterally at the point where its three processes meet (39: 0 → 1); ventral process of the postorbital ends much higher than the ventral border of the orbit (61: 1 → 0); supratemporal fossa restricted to the lateral edge of the parietal, resulting in a broad, flat parietal table (83: 1 → 0); posttemporal fenestra absent or developed as a foramen or very narrow slit (136: 0 → 2); presence of an external mandibular fenestra (152: 0 → 1); serrations distinctly present on the distal margin of maxillary/dentary tooth crowns and usually apically restricted; low or absent on the mesial margin (172: 0 → 1).

**Archosauriformes *Gauthier, Kluge & Rowe, 1988***
**Definition.** The least inclusive clade containing *Crocodylus niloticus* and *Proterosuchus fergusi* (*Nesbitt, 2011*).
**Temporal range.** Changhsingian (latest Permian, *Archosaurus rossicus*; *Sennikov & Golubev, 2012*) to Recent (*Crocodylus niloticus*).
**Unambiguous synapomorphies.** Rostrum dorsoventrally taller than transversely broad at the level of the anterior border of the orbit (3: 0 → 1); presence of an antorbital fenestra (22: 0 → 1); lacrimal contacts nasal but does not reach naris (35: 2 → 1); pineal foramen of the parietal absent (77: 1 → 2); postparietals sheet-like and not much narrower than the supraoccipital or small and splint-like (84: 2 → 0/1); tooth bearing portion of the dentary dorsally curved for all or most of its anteroposterior length (145: 0 → 1); presence of a posterocentral process of the dentary (149: 0 → 1); diapophysis and parapophysis of anterior to mid-postaxial cervical vertebrae on different processes and well-separated (188: 2 → 1).

**Proterosuchus spp. *Broom, 1903***
Unambiguous synapomorphies. Strongly downturned main body of the premaxilla (5: 1 → 2); length of the posterior process of the jugal versus the height of its base between 5.29 and 5.84 (43: 1/2 → 3); posterior process of the postorbital extends close to or beyond

the level of the posterior margin of the supratemporal fenestrae (60: 0 → 1); presence of teeth on the lateral ramus of the pterygoid (106: 0 → 1).

**Unnamed clade (*Erythrosuchus africanus* + *Euparkeria capensis*)**

Unambiguous synapomorphies. Dorsoventral height at the level of the anterior tip of the maxilla versus dorsoventral height at the level of the anterior border of the orbit between 0.56 and 0.78 (2: 1 → 2); the anterior process of the jugal is dorsoventrally expanded and partially covers the lateral surface of the posterior process of the maxilla (38: 0 → 1); supratemporal absent (73: 1 → 2); anterior process of the quadratojugal distinctly present, in which the lower temporal bar is complete, but the process terminates well posterior to the base of the posterior process of the jugal (88: 0/1 → 2); paroccipital process of the opisthotic extends laterally or slightly posterolaterally (130: 1 → 0); lateral shelf of the surangular present as a laterally or ventrolaterally projecting shelf with a lateral edge (155: 1 → 2); serrations distinctly present on both margins of maxillary/dentary tooth crowns (172: 1 → 2); mid-dorsal ribs with a dichocephalous proximal end (214: 1 → 0); second sacral rib does not bifurcate distally (216: 1 → 0); scapulacoracoid with a distinct notch present on the anterior margin at the level of the suture between both bones (227: 0 → 1); minimum anteroposterior length of the scapula less than a quarter of its proximodistal length (230: 1 → 2); coracoid unexpanded posteriorly in lateral view (232: 1 → 0); approximately straight-sided shape of the preacetabular process of the ilium with a distinct angle between the anterior and dorsal margins (265: 0 → 1); dorsalmost margin of the acetabulum on the ilium is unsculptured (271: 1 → 0); presence of a pubic apron (275: 0 → 1); pedal centrale absent as a separate ossification (291: 0 → 1); absence of both distal tarsals 1 and 2 (292: 0 → 2); absence of a perforating foramen between astragalus and calcaneum (295: 0 → 1); absence of a concave gap separating the dorsal prominence from the proximo-medial surface in metatarsal V (301: 1 → 0).

## DISCUSSION

The resolution of the SCTs of analyses 3 and 4, which employ ratio characters and treat relevant characters as ordered, is distinctly higher than those of analyses 1 and 2 and are calculated from fewer MPTs (Figs. 33–36). The topologies of the four analyses mostly correspond with each other, although several noteworthy differences occur. These relevant discrepancies between the analyses are discussed below for each corresponding OTU or clade. The results of our analyses broadly correspond to those of previous investigations of non-archosauriform archosauromorph interrelationships (e.g., *Butler et al., 2015*; *Ezcurra, 2016*; *Ezcurra, Montefeltro & Butler, 2016*; *Nesbitt et al., 2015*; *Pinheiro, Simão-Oliveira & Butler, 2019*; *Pritchard et al., 2015*; *Pritchard & Sues, 2019*). Archosauromorpha is comprised of the OTUs that have widely been accepted as members of this clade in the SCTs of all four analyses. Despite a larger sample size of "protorosaurs" compared to previous analyses, particularly through the inclusion of recently described and relatively completely known Chinese "protorosaurs", as well as several tanystropheid taxa that were previously not widely considered for phylogenetic analyses (e.g., *Sclerostropheus fossai*, *Ozimek volans*, *Raiblliania calligarisi*), all SCTs of our analyses still find "Protorosauria" to

be paraphyletic *Ezcurra, 2016* (Figs. 33–36), corroborating the results of recent analyses (e.g., *Ezcurra, 2016*; *Nesbitt et al., 2015*; *Pritchard et al., 2015*; *Pritchard & Sues, 2019*). Our analysis also corresponds to previous analyses in finding a monophyletic Rhynchosauria, Allokotosauria, and Archosauriformes, as well as in recovering *Teyujagua paradoxa* as the closest related taxon to Archosauriformes among our sampled OTUs. However, our results contradict with previous studies in the position of *Prolacerta broomi*, which is consistently found to be considerably more distantly related to Archosauriformes than previously considered. Furthermore, we recognize a new non-archosauriform archosaurmorph clade, which in all SCTs includes at least the Chinese taxa *Pectodens zhenyuensis* and *Dinocephalosaurus orientalis*. Below, the results are discussed in detail for each relevant taxon and clade.

## Tree resolution at the base of Archosauromorpha and the influence of ratio and ordered characters

Although some minor differences are present, the SCT topologies for the non-"protorosaurian" OTUs, namely all non-archosauromorphs and all previously recognized crocopods except for *Prolacerta broomi* (rhynchosaurs, allokotosaurs, *Teyujagua paradoxa*, and archosauriforms) do not exhibit consistent differences between the SCTs of analyses 3 and 4 and those of analyses 1 and 2 (Figs. 33A–36A). This indicates that their relative positions are quite stable for our data matrix, which is also supported by relatively high branch support values for Lepidosauromorpha, Archosauromorpha, and the relationships between and within rhynchosaurs, allokotosaurs, *Teyujagua paradoxa*, and Archosauriformes. The main difference between the results of analysis 1 and 2 compared to 3 and 4 is found in the resolution at the base of Archosauromorpha with the taxa that have previously been considered as "protorosaurs". In analyses 3 and 4 Tanystropheidae and Dinocephalosauridae are recovered as monophyletic clades (Figs. 35A–36A), whereas in analyses 1 and 2 these clades are collapsed and a large polytomy is formed by their OTUs, as well as *Protorosaurus speneri*, *Jesairosaurus lehmani*, and in analysis 1 *Czatkowiella harae* (Figs. 33A–34A). However, RSCT 1 of analysis 1 recovers Dinocephalosauridae, and RSCT 2 and 3 recover both Dinocephalosauridae and Tanystropheidae (Figs. 33B–33D), albeit in somewhat different compositions relative to analysis 3 and 4 as is discussed below. RSCTs 2 and 3 of analysis 2 also recover Tanystropheidae (Figs. 34C–34D), but a Dinocephalosauridae clade is not recovered since both taxa comprising this clade, *Pectodens zhenyuensis* and *Dinocephalosaurus orientalis*, were identified as unstable OTUs by the iter PCR function for this analysis. The broadly similar topologies between the SCTs of analyses 3 and 4 and the RSCTs of analyses 1 and 2 suggest that the addition of ratio characters and the ordering of characters has a positive effect on successfully resolving the phylogenetic relationships among non-archosauriform archosauromorphs. However, branch support values are low for all nodes within both Tanystropheidae and Dinocephalosauridae, which is likely attributable to several OTUs with large amounts of missing data within these clades. It is noteworthy to mention that the characters that were treated as ordered in analyses 3 and 4 were also formulated with the intention to be treated as such (including many
characters incorporated from). Therefore, future analyses that would decide against the use of ordered characters should reconsider the construction of some of these characters. For instance, character 84 could be split into one character treating the absence or presence of the postparietal and another describing its shape, in which the latter character is scored as inapplicable for OTUs in which the former character is scored as absent (*Brazeau, 2011*).

## The phylogenetic position of *Protorosaurus speneri*

*Protorosaurus speneri* has been widely considered as the sister taxon to all other archosauromorphs, although it has been recovered in a less inclusive clade with *Czatkowiella harae* or *Aenigmastropheus parringtoni* at the base of Archosauromorpha in the analyses of *Borsuk-Białynicka & Evans (2009b)* and *Ezcurra, Scheyer & Butler (2014)*, respectively. In *Ezcurra (2016)*, *Protorosaurus speneri* was recovered as the sister taxon of all archosauromorphs except for the enigmatic *Aenigmastropheus parringtoni*. *Aenigmastropheus parringtoni* is known from very fragmented remains and was not included in our analysis. *Protorosaurus speneri* was recovered as the sister taxon to all other archosauromorphs in the SCT of analysis 4 (Fig. 36A), whereas it was recovered in a polytomy at the base of Archosauromorpha in the SCTs of analyses 1 and 2 (Figs. 33A–34A). In analysis 1, *Protorosaurus speneri* also obtains the position as sister taxon to all other archosauromorphs in all RSCTs (Figs. 33B–33D). In analysis 2 the relationships of *Protorosaurus speneri* are only resolved after the *a posteriori* exclusion of *Jesairosaurus lehmani* in RSCT 3 (Fig. 34D). Therein, *Protorosaurus speneri* is found as the sister taxon to all other archosauromorphs as in the RSCTs of analysis 1 and the SCT of analysis 4. The position of *Protorosaurus speneri* only deviates in analysis 3, since in this analysis *Protorosaurus speneri* is consistently found as the sister taxon to all archosauromorphs except for *Jesairosaurus lehmani* (Fig. 35A). Our analyses agree with other studies in the position of *Protorosaurus speneri* as an early archosauromorph that is more distantly related to archosauriforms than most, if not all, other non-archosauriform archosauromorphs are, including the tanystropheids, dinocephalosaurids, allokotosaurs, and rhynchosaurs.

## The phylogenetic position of *Jesairosaurus lehmani*

*Jesairosaurus lehmani* was originally considered as a "protorosaur" that is closely related to Tanystropheidae (*Jalil, 1997*). More recently, *Ezcurra (2016)* recovered *Jesairosaurus lehmani* as the sister taxon to Tanystropheidae. The position of *Jesairosaurus lehmani* is not stable in our analyses. In the SCT of analysis 1 *Jesairosaurus lehmani* is found in a massive polytomy at the base of Archosauromorpha (Fig. 33A). The position of *Jesairosaurus lehmani* within archosauromorphs is resolved after the *a posteriori* exclusion of *Macrocnemus obristi* among Archosauromorpha and in all RSCTs of analysis 1 *Jesairosaurus lehmani* is recovered as the sister taxon to all archosauromorphs except *Protorosaurus speneri* (Figs. 33B–33D). *Jesairosaurus lehmani* was also found as part of a polytomy at the base of Archosauromorpha in the SCT of analysis 2 and identified as one of the unstable OTUs by the iter PCR function (Fig. 34). In most MPTs of analysis 2

*Jesairosaurus lehmani* is found as an early diverging archosauromorph or as an early diverging member of the clade consisting of the OTUs that form Tanystropheidae and Dinocephalosauridae in the other analyses. In all MPTs of analysis 3 *Jesairosaurus lehmani* is found as the sister taxon to all remaining archosauromorphs (Fig. 35A) and in all MPTs of analysis 4 *Jesairosaurus lehmani* forms the sister taxon to Dinocephalosauridae (Fig. 36A). The clade formed by *Jesairosaurus lehmani* and the dinocephalosaurids is found in a polytomy with *Prolacerta broomi*, Tanystropheidae, and Crocopoda. Thus, the position of *Jesairosaurus lehmani* as a non-crocopodan archosauromorph is corroborated, but its position among non-archosauriform Archosauromorpha remains contentious based on the results of our analyses.

## The phylogenetic position of *Czatkowiella harae*

Our data matrix is the first to incorporate *Czatkowiella harae* since the first description of this taxon by *Borsuk-Białynicka & Evans (2009b)*, in which it was found as the sister taxon to *Protorosaurus speneri* in a clade that formed the sister group to all other archosauromorphs. *Czatkowiella harae* was only included in analyses 1 and 3 in our study, since it was considered as one of the problematic taxa that were pruned *a priori* for analyses 2 and 4 together with the nomina dubia *Tanystropheus "conspicuus"* and *"Tanystropheus antiquus"*, as outlined above. Analyses 1 and 3 both recover *Czatkowiella harae* within Archosauromorpha (Figs. 33 and 35). In analysis 1 *Czatkowiella harae* and *Ozimek volans* were recovered in a clade in RSCTs 2 and 3 after the exclusion *a posteriori* of *Macrocnemus obristi* and *Elessaurus gondwanoccidens* among Archosauromorpha and this clade formed the sister group to a clade composed of *Fuyuansaurus acutirostris*, Tanystropheidae, and Crocopoda (Fig. 33C and 33D). *Czatkowiella harae* and *Ozimek volans* also form a clade in all MPTs of analysis 3, but this clade is recovered as the sister clade to *Fuyuansaurus acutirostris* and Dinocephalosauridae in this analysis (Fig. 35A). This suggests that at least some of the material referred to *Czatkowiella harae* can be attributed to Archosauromorpha, but that the taxon cannot be confidently referred to any of the less inclusive archosauromorph clades. The inclusion of *Czatkowiella harae* appears to have a distinct effect on the outcome of the results, since the sister taxon of *Czatkowiella harae*, *Ozimek volans*, is consistently recovered within Tanystropheidae in the analyses excluding *Czatkowiella harae a priori*. Therefore, as long as it cannot be corroborated that the fragmented remains of *Czatkowiella harae* can be attributed to a single taxon unambiguously, the inclusion of this taxon should be considered carefully, since its inclusion has the potential to influence tree topology among non-archosauriform archosauromorphs.

## Dinocephalosauridae

*Dinocephalosaurus orientalis* has been included in four previous phylogenetic analyses. In the analysis of *Rieppel, Li & Fraser (2008)*, it formed a polytomy with drepanosaurids, tanystropheids, and *Jesairosaurus lehmani*. In the analysis of *Liu et al. (2017)*, which is derived from the same character matrices as used by *Rieppel, Li & Fraser (2008)*, *Dinocephalosaurus orientalis* was recovered within Tanystropheidae as the sister taxon to

all other included tanystropheids. *Dinocephalosaurus orientalis* was also included in the phylogenetic analysis of *De Oliveira et al. (2020)*, which was based on a modification of the data matrix of *Pritchard et al. (2018)*. In this analysis *Dinocephalosaurus orientalis* was recovered in a clade with *Jesairosaurus lehmani* that represented the sister clade to all other archosauromorphs. However, the overall resolution of this analysis was poor and both *Dinocephalosaurus orientalis* and *Jesairosaurus lehmani* were pruned *a priori* for the final analysis presented by *De Oliveira et al. (2020)*. Finally, *Dinocephalosaurus orientalis* was also included in the data matrix of *Ezcurra & Butler (2018)*, which was constructed with the aim to investigate the morphological disparity of the middle Permian to early Carnian archosauromorphs rather than to resolve their phylogenetic relationships. In this analysis *Dinocephalosaurus orientalis* was found as closely related to Tanystropheidae, forming a polytomy with Tanystropheidae and the poorly known "protorosaur" *Trachelosaurus fischeri* in the second and least inclusive RSCT of that analysis. The phylogenetic relationships of *Pectodens zhenyuensis* have not been tested previously by a dedicated data matrix, but this taxon was also included in the matrix of *Ezcurra & Butler (2018)* and in the SCT and both RSCTs of this analysis it was found within Tanystropheidae in a large polytomy consisting of most included tanystropheid taxa and a clade formed by the three included *Macrocnemus* taxa.

In the MPTs of analysis 2 *Dinocephalosaurus orientalis* and *Pectodens zhenyuensis* are either found as closely related taxa in a clade at the base of Tanystropheidae or more deeply nested within Tanystropheidae as successive sister taxa to a clade composed of the *Tanystropheus* OTUs, as well as *Raibliania calligarisi*. The latter topology could potentially be attributed to the shared presence of aquatic adaptations in *Dinocephalosaurus orientalis* and *Tanystropheus* spp., which these genera have been considered to have acquired independently (e.g., *Li, Rieppel & LaBarbera, 2004*). In the SCT of analysis 2 both taxa are found in a massive polytomy at the base of Archosauromorpha (Fig. 34A). After the *a posteriori* exclusion of *Macrocnemus obristi* among Archosauromorpha, both taxa are recovered in a large polytomic clade in RSCT 1 with taxa that are generally considered as tanystropheids (Fig. 35B). Due to their unstable position in the MPTs, both *Dinocephalosaurus orientalis* and *Pectodens zhenyuensis* were omitted *a posteriori* by the iter PCR function for RSCTs 2 and 3.

In analysis 1 both *Dinocephalosaurus orientalis* and *Pectodens zhenyuensis* are part of a massive polytomy at the base of Archosauromorpha in the SCT (Fig. 33A). However, after the exclusion of the unstable OTU *Macrocnemus obristi*, both taxa are recovered in a monophyletic clade in RSCT 1, as well as the subsequent RSCTs 2 and 3 (Figs. 33B–33D). This monophyletic clade is composed of a polytomy of both taxa and *"Tanystropheus antiquus"* and *Sclerostropheus fossai*. In RSCT 3 *Pectodens zhenyuensis* is pruned *a posteriori*, and the relationships within the clade are resolved with *Dinocephalosaurus orientalis* forming the sister taxon to a clade composed of *"Tanystropheus antiquus"* and *Sclerostropheus fossai*. In both analyses 3 and 4 *Dinocephalosaurus orientalis* and *Pectodens zhenyuensis* form a monophyletic clade in all MPTs (Figs. 35A–36A), which in the case of analysis 3 also includes *"Tanystropheus antiquus"*. In both analysis 3 and 4 *Sclerostropheus fossai* is found well-nested within Tanystropheidae. It forms the sister taxon to

*Tanystropheus* spp. in RSCT 3 (Fig. 35D) and in analysis 4 it is recovered as the sister taxon to *Raibliania calligarisi* in the RSCT (Fig. 36B).

In summary, *Dinocephalosaurus orientalis* and *Pectodens zhenyuensis* form a monophyletic clade in the majority of all MPTs of all four analyses, with some MPTs also including *"Tanystropheus antiquus"* and *Sclerostropheus fossai* in this clade. However, the taxonomic status of *"Tanystropheus antiquus"* is currently unclear and its current phylogenetic placement should be considered with caution. *Sclerostropheus fossai* is exclusively known from a single, partial cervical column, and was found within Dinocephalosauridae in the RSCTs of analysis 1 (Figs. 33B–33D), but this taxon could also be referrable to Tanystropheidae based on the results of analyses 3 and 4 (Figs. 35 and 36). The presence of a new clade consisting of at least *Dinocephalosaurus orientalis* and *Pectodens zhenyuensis* among non-archosauriform archosauromorphs is well-supported based on our data matrix. Furthermore, the presence of more *Dinocephalosaurus*-like taxa has been alluded to through the description of IVPP V22788, an embryonic specimen that is very similar to *Dinocephalosaurus orientalis* but differs in several aspects, most notably a lower number of cervical vertebrae (Li, Rieppel & Fraser, 2017). Due to the very early ontogenetic stage of this specimen, it has not been referred to a separate taxon and it has not been included in our analyses, since very early ontogenetic features would have likely introduced biases into the analyses. Nevertheless, the clade formed by *Dinocephalosaurus orientalis* and *Pectodens zhenyuensis*, combined with the existence of at least one more *Dinocephalosaurus*-like taxon, merits the erection of the new higher-level taxon Dinocephalosauridae to define this clade.

Dinocephalosauridae as recovered by the RSCTs of analysis 1 forms the sister clade to a clade composing Tanystropheidae, Crocopoda, *Elessaurus gondwanoccidens*, *Ozimek volans*, *Czatkowiella harae*, and *Fuyuansaurus acutirostris* (Figs. 33B–33D). The dinocephalosaurid clade of analysis 3 is part of a larger clade formed with *Fuyuansaurus acutirostris*, *Ozimek volans*, and *Czatkowiella harae*, which forms the sister group to Tanystropheidae (Fig. 35A). The sister taxon to the dinocephalosaurid clade of analysis 4 is *Jesairosaurus lehmani* (Fig. 36A). Together these taxa form a clade in a polytomy that includes *Prolacerta broomi*, Tanystropheidae, and Crocopoda. Thus, Dinocephalosauridae can be considered as a separate clade among non-crocopodan archosauromorphs. More taxa referrable to this clade are likely to be discovered in China and possibly in other areas in the future, and their inclusion in phylogenetic analyses will aid in determining more confidently the position of Dinocephalosauridae among non-archosauriform archosauromorphs.

## The composition and interrelationships of Tanystropheidae

The monophyly of Tanystropheidae is widely supported by previous phylogenetic analyses (e.g., Benton & Allen, 1997; Dilkes, 1998; Evans, 1988; Ezcurra, 2016; Jalil, 1997; Pritchard et al., 2015; Rieppel, Fraser & Nosotti, 2003, but see Simões et al., 2018 for a notable exception) and this is corroborated by our data matrix, because a monophyletic Tanystropheidae is recovered in all analyses after the *a posteriori* exclusion of unstable OTUs. There has been less consensus on the position of tanystropheids among

non-archosauriform archosauromorphs, as well as the referral of several enigmatic taxa to Tanystropheidae. Several recent analyses recovered tanystropheids as the sister taxon to crocopods (e.g., *Ezcurra, 2016* and subsequent modifications of this matrix; *Nesbitt et al., 2015*). However, other analyses, which did not find a monophyletic Crocopoda, recovered tanystropheids as being more closely related to archosauriforms than are either rhynchosaurs (*Pritchard et al., 2015*) or allokotosaurs (*Pritchard & Sues, 2019*). Tanystropheidae are consistently found as more closely related to archosauriforms than *Protorosaurus speneri* in all MPTs of all four analyses and as more distantly related to archosauriforms than *Teyujagua paradoxa*, rhynchosaurs, and allokotosaurs are. The position of tanystropheids relative to *Prolacerta broomi*, *Jesairosaurus lehmani*, and dinocephalosaurs differs between the analyses and in the most stable analysis, analysis 4, Tanystropheidae forms a polytomy with *Prolacerta broomi*, Crocopoda, and a clade composed of *Jesairosaurus lehmani* and Dinocephalosauridae (Fig. 36).

*Fuyuansaurus acutirostris* was described as a "protorosaur" of uncertain phylogenetic affinities, since it shares features with known tanystropheids such as the presence of a long neck composed of elongate cervical vertebrae and accompanying ribs, but also lacks a thyroid fenestra between the pubis and ischium, which is considered a typical tanystropheid feature (*Fraser, Rieppel & Li, 2013*). *Fuyuansaurus acutirostris* was recovered in a large polytomy within Tanystropheidae by *Ezcurra & Butler (2018)*, which represents the only previous phylogenetic analysis that included this taxon. The conflicting morphology relative to (other) tanystropheids as suggested by *Fraser, Rieppel & Li (2013)* is reflected in our analyses. When the unstable OTUs *Macrocnemus obristi* and *Elessaurus gondwanoccidens* are excluded *a posteriori*, *Fuyuansaurus acutirostris* is found as the sister taxon to a clade composed of Tanystropheidae and Crocopoda in analysis 1 (Figs. 33C–33D). In analysis 2 *Fuyuansaurus acutirostris* is positioned in a large polytomy consisting of tanystropheid and dinocephalosaurid OTUs in RSCT 1 (Fig. 34B), but it is identified as an unstable taxon by the iter PCR function and excluded *a posteriori* in RSCTs 2 and 3. In all MPTs of analysis 3 *Fuyuansaurus acutirostris* forms the sister taxon to Dinocephalosauridae (Fig. 35A), whereas in the most stable analysis, analysis 4, *Fuyuansaurus* is quite deeply nested within Tanystropheidae (Fig. 36A). Therefore, the affinities of *Fuyuansaurus acutirostris* remain somewhat equivocal, and it can only very tentatively be referred to Tanystropheidae based on the currently available morphological information.

*Ozimek volans* differs from known tanystropheids in its extremely gracile and elongate appendicular elements and the morphology of its pectoral girdle and it was therefore not identified as a tanystropheid in the original description by *Dzik & Sulej (2016)*, but rather as a "protorosaur" closely related to *Sharovipteryx mirabilis*. However, the recent phylogenetic analysis by *Pritchard & Sues (2019)*, which included many archosauromorphs as well as non-saurian diapsids, recovered *Ozimek volans* as the sister taxon to *Langobardisaurus pandolfii* and *Tanytrachelos ahynis* deeply nested within Tanystropheidae. The position of *Ozimek volans* is inconsistent in our analyses. In the SCT of analysis 1 it is found in the massive polytomy at the base of Archosauromorpha, but after the *a posteriori* pruning of the unstable OTUs *Macrocnemus obristi* and *Elessaurus*

*gondwanoccidens* among archosauromorphs it is found in a clade with *Czatkowiella harae* in RSCTs 2 and 3 (Fig. 33). This clade forms the sister group to a clade composed of *Fuyuansaurus acutirostris*, Tanystropheidae, and Crocopoda. *Ozimek volans* is found in a similar large polytomy in the SCT of analysis 2, but it is recovered deeply nested in Tanystropheidae in a polytomy with *Amotosaurus rotfeldensis* and a clade composed of *Tanystropheus* spp. and *Raibliania calligarisi* in RSCTs 2 and 3 after the *a posteriori* pruning of *Macrocnemus obristi*, *Sclerostropheus fossai*, *Elessaurus gondwanoccidens*, *Pectodens zhenyuensis*, *Fuyuansaurus acutirostris*, and *Dinocephalosaurus orientalis* among the archosauromorph OTUs (Fig. 34). In all MPTs of analysis 3 *Ozimek volans* is again found in a clade with *Czatkowiella harae*. This clade forms the sister group to a clade composed of *Fuyuansaurus acutirostris* and Dinocephalosauridae (Fig. 35A). In all MPTs of analysis 4 *Ozimek volans* is also found deeply nested within Tanystropheidae, and it is recovered as the sister taxon to a clade composed of *Raibliania calligarisi* and *Sclerostropheus fossai* after the *a posteriori* exclusion of *Elessaurus gondwanoccidens* among the tanystropheids in the RSCT (Fig. 36). Thus, the inclusion of *Czatkowiella harae* seems to have a large effect on the position of *Ozimek volans*, since these OTUs form a clade in most MPTs in both analyses in which the former is included. However, only a single common unambiguous synapomorphy defines the *Czatkowiella harae*—*Ozimek volans* clade in analysis 3 (the only analysis to find this clade in all MPTs): a ratio of the length versus height in posterior dorsal vertebrae between 2.16 and 2.20. Regardless of the inclusion of ratio characters and the ordering of characters, the clade formed by *Czatkowiella harae* and *Ozimek volans* is found within Archosauromorpha as quite distantly related to Archosauriformes and outside Tanystropheidae (Figs. 33 and 35). Conversely, when the problematic OTU *Czatkowiella harae* is excluded *a priori* from the analyses *Ozimek volans* is relatively confidently recovered as a tanystropheid in both analyses 2 and 4 (Figs. 34 and 36). Therefore, the position of *Ozimek volans*, and by extension possibly the position of *Sharovipteryx mirabilis* and other putative sharovipterygids (*Dzik & Sulej, 2016*; *Pritchard & Sues, 2019*) among non-archosauriform archosauromorphs remains uncertain. Additional morphological information on *Ozimek volans*, including detailed comparisons to tanystropheids, other archosauromorphs, and Triassic diapsids such as drepanosauromorphs will likely aid in a more reliable phylogenetic interpretation of this taxon. This would be particularly valuable given the peculiar morphology of *Ozimek volans*, because the inclusion of a putative glider within Tanystropheidae would increase their known ecomorphological diversity considerably.

*Macrocnemus* and *Tanystropheus* represent the best-known tanystropheid genera. The postcrania of both *Tanystropheus hydroides* and *Tanystropheus longobardicus* possess well-known and easily recognizable characters that are considered derived compared to *Macrocnemus* spp., such as the presence of 13 extremely elongated cervical vertebrae and the presence of heterotopic bones in approximately 50% of specimens preserving the proximal caudal region (*Nosotti, 2007*; *Wild, 1973*). Recently the cranial morphology of both *Tanystropheus hydroides* and *Macrocnemus bassanii* were revised with the aid of high-resolution micro-computed tomography, highlighting a large cranial disparity between the two (*Miedema et al., 2020*; *Spiekman et al., 2020b*). The cranial morphology of

*Macrocnemus bassanii* shares many similarities with non-tanystropheid archosauromorphs such as *Prolacerta broomi* and *Protorosaurus speneri*, particularly in the temporal region of the skull (*Miedema et al., 2020*), which suggests that some of these features might be plesiomorphic to Tanystropheidae. In contrast, the skull of *Tanystropheus hydroides* exhibits clear specializations to an aquatic lifestyle (*Spiekman et al., 2020a*) and possesses many cranial characters unique among tanystropheids (e.g., the configuration of the palate, a dorsoventrally flattened rostrum, the presence of a posteriorly directed hook on the dorsal end of the quadrate, and the presence of a laterosphenoid; *Spiekman et al., 2020b*). This study is the first to incorporate the new cranial information for these taxa in a quantitative phylogenetic analysis. Our results consistently find *Macrocnemus bassanii* and *Macrocnemus fuyuanensis* in a clade that forms the sister group to all other tanystropheids when problematic OTUs are excluded (Figs. 33–36). The poorly known *Macrocnemus obristi* was identified as one of the unstable OTUs by the iter PCR function in analyses 1, 2, and 3. In analysis 4 it was recovered as the sister taxon to all remaining tanystropheids except *Macrocnemus bassanii* and *Macrocnemus fuyuanensis*. *Macrocnemus obristi* was found to differ from the other two *Macrocnemus* OTUs in the lack of a sigmoidal curvature of the femur (character 286), the ratio of the length of the pes versus the length of the tibia and fibula (character 294), and the ratio of the length of metatarsal I versus that of metatarsal III (character 297). The absence of morphological information on the skull, cervical vertebrae, or pectoral girdle for *Macrocnemus obristi* (*Fraser & Furrer, 2013*), which contain many diagnostic features among tanystropheids, might have contributed to its unstable position in the MPTs of analyses 1, 2, and 3, and could explain why it was not recovered as a direct sister taxon to the other *Macrocnemus* OTUs in the MPTs of the four analyses. Therefore, we consider there to be insufficient support to assign *Macrocnemus obristi* to a separate genus despite its aberrant position in our results. In contrast to the *Macrocnemus* OTUs, *Tanystropheus hydroides* and *Tanystropheus longobardicus*, the best-known *Tanystropheus* species, are consistently found deeply nested within Tanystropheidae in all MPTs of analysis 4 and in the RSCTs of the other three analyses, suggesting a derived position of the genus within Tanystropheidae.

The position of other tanystropheids is much less stable in our analyses. *Raibliania calligarisi* is found as closely related to *Tanystropheus* spp. in the RSCTs of analyses 1 and 3 and in all MPTs of analysis 2 (Figs. 33–35), reflecting the close morphological similarity of this taxon to *Tanystropheus longobardicus*, but it is recovered as quite distantly related to the *Tanystropheus* OTUs in the RSCT of analysis 4 (Fig. 36B). *Elessaurus gondwanoccidens*, which is only known from a single specimen comprising a hind limb and partial pelvis and vertebral column, was identified as an unstable OTU by the iter PCR function in all analyses, and in all SCTs is either found within Tanystropheidae or in a massive polytomy at the base of Archosauromorpha (Figs. 33–36). *Langobardisaurus pandolfii* and *Tanytrachelos ahynis* were found as sister taxa within Tanystropheidae in the analyses of *Pritchard & Nesbitt (2017)* and *Pritchard & Sues (2019)*, and a close relationship between these taxa was also recovered in all MPTs of analyses 1 and 2 (Figs. 33 and 34). *Tanytrachelos ahynis* was consistently found within Tanystropheidae in analysis

3, but its position within the clade is inconsistent in the MPTs and it was therefore pruned *a posteriori* by the iter PCR function (Fig. 35). In all MPTs of analysis 4, *Tanytrachelos ahynis* forms a clade with AMNH FARB 7206 that forms the sister group to a *Tanystropheus* spp. clade, whereas *Langobardisaurus pandolfii* was found as the sister taxon to all tanystropheids except *Macrocnemus* spp. (Fig. 36A). In RSCTs 2 and 3 of analysis 3 *Langobardisaurus pandolfii* forms a trichotomy with AMNH FARB 7206 and a clade composed of *Tanystropheus* spp., *Raibliania calligarisi*, and *Sclerostropheus fossai* (excluding the last two OTUs in RSCT 3) after the *a posteriori* exclusion of *Macrocnemus obristi*, *Elessaurus gondwanoccidens*, and *Tanytrachelos ahynis* (Figs. 35C and 35D). AMNH FARB 7206 was previously tentatively referred to *Tanytrachelos ahynis* by *Pritchard et al. (2015)* but was treated as a separate OTU here (see *Overview of "protorosaur" taxa* section above). AMNH FARB 7206 was only found as the direct sister taxon to *Tanytrachelos ahynis* in SCT of analysis 4 (Fig. 36). In the SCT of analysis 1 and all RSCTs of analysis 2 AMNH FARB 7206 was found as the sister taxon to a *Langobardisaurus pandolfii*—*Tanytrachelos ahynis* clade (Figs. 33 and 34). In RSCTs 2 and 3 of analysis 3 AMNH FARB 7206 forms a trichotomy with *Langobardisaurus pandolfii* and a clade composed of *Tanystropheus* spp., *Raibliania calligarisi*, and *Sclerostropheus fossai* (excluding the last two OTUs in RSCT 3) (Figs. 35C and 35D). Therefore, the results of our analysis are ambiguous when it comes to the referral of AMNH FARB 7206 to *Tanytrachelos ahynis* as was previously proposed (*Pritchard et al., 2015*). A detailed study of AMNH FARB 7206 and other specimens from the Lockatong Formation are required to determine whether this material represents a separate taxon to *Tanytrachelos ahynis*.

## The phylogenetic position of *Prolacerta broomi* and the composition of Crocopoda

Crocopoda is a recently erected clade that is defined as all archosauromorph taxa that are more closely related to *Trilophosaurus buettneri*, *Azendohsaurus madagaskarensis*, *Rhynchosaurus articeps*, and *Proterosuchus fergusi* than to *Protorosaurus speneri* and *Tanystropheus longobardicus* (*Ezcurra, 2016*, but see *Pritchard & Sues, 2019*, *Simões et al., 2018*, and *Spiekman et al., 2020a*, which found Crocopoda to be polyphyletic). *Prolacerta broomi* was previously considered to be very closely related to early archosauriforms (particulary the Early Triassic proterosuchids) and has been treated as such in discussions on character trait evolution (e.g., *Ezcurra & Butler, 2015a*; *Modesto & Sues, 2004*; *Pinheiro et al., 2016*; *Pritchard & Sues, 2019*). Congruently, phylogenetic analyses that found a monophyletic Crocopoda always recovered *Prolacerta broomi* within this clade (e.g., *Butler et al., 2019*; *Ezcurra & Butler, 2018*; *Ezcurra et al., 2017*, *2019*; *Maidment et al., 2020*; *Nesbitt et al., 2017a*, *2015*; *Pritchard et al., 2018*; *Pritchard & Nesbitt, 2017*; *Pritchard & Sues, 2019*; *Scheyer et al., 2020a*; *Sengupta, Ezcurra & Bandyopadhyay, 2017*; *Spiekman, 2018*; *Stocker et al., 2017*). In all MPTs of analyses 1 and 2 of our data matrix *Prolacerta broomi* is recovered within Crocopoda as the sister taxon to a clade composed of rhynchosaurs, allokotosaurs, *Teyujagua paradoxa*, and Archosauriformes (Figs. 33 and 34). In the SCTs of both analyses 3 and 4 *Prolacerta broomi* is found outside Crocopoda (Figs. 35 and 36). In the SCT of analysis 3 *Prolacerta broomi* forms the sister taxon to all

archosauromorphs except *Jesairosaurus lehmani* and *Protorosaurus speneri*, and *Prolacerta broomi* forms a polytomy with a clade composed of *Jesairosaurus lehmani* and Dinocephalosauridae, Tanystropheidae, and Crocopoda in the SCT of analysis 4. Our results therefore consistently find *Prolacerta broomi* to be considerably more distantly related to Archosauriformes than previously considered. This reflects the strong similarity in cranial morphology that has been observed between *Prolacerta broomi* and *Macrocnemus bassanii* (*Miedema et al., 2020*), indicating that both taxa share features that are possibly plesiomorphic to early archosauromorphs.

*Teyujagua paradoxa* exhibits a cranial morphology that in several ways is intermediate between non-archosauriform archosauromorphs and archosauriforms (*Pinheiro et al., 2016*; *Pinheiro, Simão-Oliveira & Butler, 2019*). *Teyujagua paradoxa* is found in a polytomy with rhynchosaurs, allokotosaurs, and archosauriforms in the SCT of analysis 2 (Fig. 34A), whereas all MPTs of the other three analyses consistently finds this OTU as the sister taxon to Archosauriformes (Figs. 33A, 35A and 36A). The internal nodes of the crocopodan clades Allokotosauria, Rhynchosauria, and Archosauriformes have the same composition in all four SCTs and possess relatively high support values (Figs. 33–36), which is in congruence with results of previous studies. However, the exact interrelationships between these clades differ slightly between the analyses. In all MPTs of analyses 1 and 3 rhynchosaurs are found to be more closely related to archosauriforms than allokotosaurs are (Figs. 33A and 35A). However, in the SCT of analysis 2 a polytomy is formed by *Teyujagua paradoxa*, Rhynchosauria, Allokotosauria, and Archosauriformes and in the SCT of analysis 4 a trichotomy is formed by Rhynchosauria, Allokotosauria, and a clade formed by *Teyujagua paradoxa* and Archosauriformes (Figs. 34A and 36A). Allokotosaurs were found to be closely related to Archosauriformes by *Nesbitt et al. (2015)*. However, other phylogenetic analyses (e.g., *Ezcurra, 2016* and subsequent modifications of that matrix; *Pritchard & Nesbitt, 2017*; *Pritchard & Sues, 2019*) found rhynchosaurs to be more closely related to archosauriforms. Our results reflect this ambiguity and are inconclusive regarding the relative position of Allokotosauria and Rhynchosauria among non-archosauriform Crocopoda.

## Macroevolutionary implications and prospectus

The results of our analyses, which include the largest "protorosaur" sample to date, reveal that non-archosauriform archosauromorphs are more diverse than previously considered, as is highlighted by the recognition of a new clade. Dinocephalosauridae includes at least two marine taxa with extensive aquatic adaptations and extremely elongated necks (*Li, Rieppel & Fraser, 2017*; *Rieppel, Li & Fraser, 2008*) and represents a remarkable convergence to plesiosaurs (Sauropterygia), a group of marine reptiles that was highly successful between the Late Triassic and Late Cretaceous. Much is currently unknown about the aquatic origins of *Dinocephalosaurus orientalis* and the temporal range of the Dinocephalosauridae. The skeleton of the dinocephalosaurid *Pectodens zhenyuensis* is poorly ossified, which could represent an aquatic adaptation but is more likely attributable to the early ontogenetic state of the only known specimen, since no other clear aquatic adaptations are present in this taxon (*Li et al., 2017*). The tanystropheids also include at

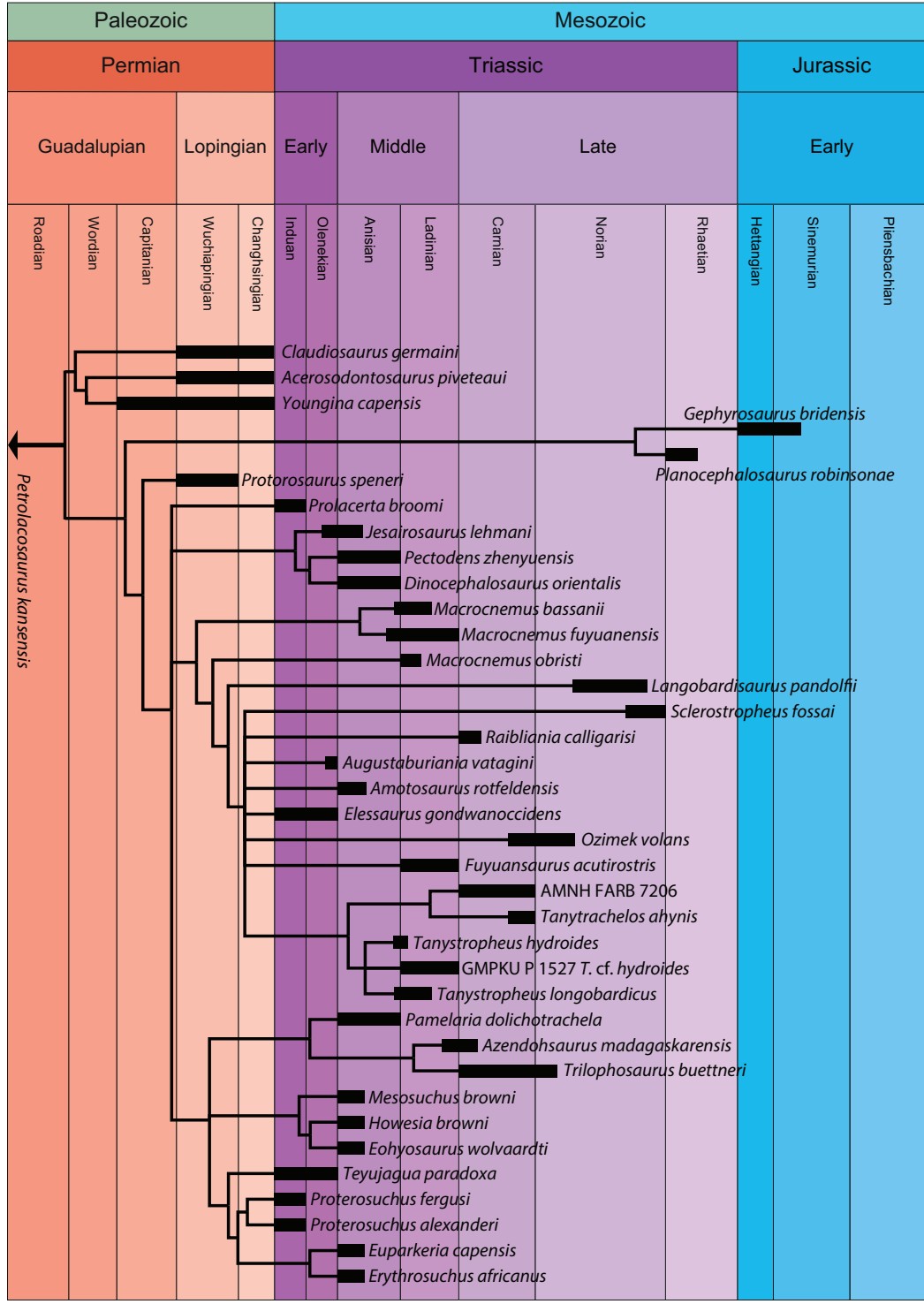

**Figure 37 Time-calibrated phylogenetic tree based on the relationships recovered in analysis 4.** The topology of the SCT is figured for Sauria, whereas the topology of the RSCT (excluding *a posteriori* the early Permian *Orovenator mayorum*) is used to indicate the relationships between the late Permian non-saurian diapsids. The black boxes indicate the possible temporal range of each OTU based on the available stratigraphic information (e.g., *Pectodens zhenyuensis* is known only from Member II of the Guanling Formation, which cannot be further restricted than being of Anisian age, and therefore the possible temporal range of *Pectodens zhenyuensis* covers the complete Anisian). The non-saurian diapsid

**Figure 37** (continued)
taxa *Petrolacosaurus kansensis* and *Orovenator mayorum* from the Carboniferous and early Permian, respectively, are not indicated since they are considerably older than the other taxa of this study. Their phylogenetic position is indicated in Fig. 36A. The timescale is based on the International Chronostratigraphic Chart of the International Commission on Stratigraphy (accessed May 2020).

least two genera, *Tanystropheus* and *Tanytrachelos*, that both exhibit considerable neck elongation and aquatic adaptations (*Olsen, 1979*; *Spiekman et al., 2020a*). *Tanystropheus hydroides*, *Tanystropheus longobardicus*, *Tanystropheus "conspicuus"*, and *Tanystropheus "haasi"* are all known from Middle Triassic marine deposits (*Spiekman & Scheyer, 2019*), but cervical vertebrae with strong similarities to those of *Tanystropheus* spp. have also been discovered in Middle Triassic fluvial sediments, indicating that certain *Tanystropheus*-like taxa also inhabited freshwater environments (*Formoso et al., 2019*). The Late Triassic *Tanytrachelos ahynis* is exclusively known from lacustrine sediments (*Casey, Fraser & Kowalewski, 2007*; *Fraser et al., 1996*). Recently it was shown that two species of *Tanystropheus* with a large size discrepancy, *Tanystropheus longobardicus* and *Tanystropheus hydroides*, exhibited niche partitioning within the marine habitat of the Besano Formation at Monte San Giorgio, since both species had evolved to exploit different food sources (*Spiekman et al., 2020a*). This remarkable diversity of aquatic forms, originating independently in multiple lineages, suggests that non-archosauriform archosauromorphs represented an important component of aquatic environments. Marine taxa are currently only known from the Middle Triassic of Europe and southern China, whereas only a single freshwater form (*Tanytrachelos ahynis*) has been formally described so far. Future research efforts are likely to reveal a substantial increase in the known diversity and temporal extent of aquatic non-archosauriform archosauromorphs.

The presence of tanystropheids in both South America (*De Oliveira et al., 2018*, *2020*) and Russia (*Sennikov, 2011*) during the Olenekian indicates that the clade already had achieved a wide, likely near-cosmopolitan, distribution before the end of the Early Triassic. Temporally, tanystropheids represent one of the most successful non-archosaurian archosauromorph lineages, extending from at least the Early Triassic until the late Late Triassic (*Sclerostropheus fossai*) (Fig. 37). Tanystropheids and dinocephalosaurids had almost certainly diverged from other known archosauromorph lineages by the late Permian and non-archosauriform archosauromorphs reached high ecomorphological disparity relatively soon after the Permo-Triassic mass extinction, as is indicated by the occurrence of herbivorous rhynchosaurs and allokotosaurs in the Early Triassic (*Arkhangelskii & Sennikov, 2008*; *Ezcurra, Montefeltro & Butler, 2016*) and the appearance of fully marine dinocelphalosaurids by the first half of the Middle Triassic (*Li, Rieppel & Fraser, 2017*; *Rieppel, Li & Fraser, 2008*). Consequently, our results provide additional support to the findings of previous phylogenetic and macroevolutionary studies (e.g., *Ezcurra, 2016*; *Ezcurra & Butler, 2018*; *Foth et al., 2016*) that have revealed that non-archosauriform archosauromorphs formed a major component of terrestrial ecosystems during the Triassic and that their palaeoecological diversity likely exceeded that

of the non-archosaurian archosauriforms. Our results further indicate that the diversity of aquatic non-archosauriform archosauromorphs is likely considerably higher than previous appreciated. The inclusion of the putative glider *Ozimek volans* among non-archosauriform archosauromorphs, and its possible referral to Tanystropheidae, provides further support for the high ecomorphological disparity of non-archosauriform Archosauromorpha. However, additional detailed comparisons of *Ozimek volans* and possibly other sharovipterygids to tanystropheids and other archosauromorphs are required to test these findings more rigorously.

The lack of resolution and the topological inconsistency in the interrelationships of Tanystropheidae in our analyses is attributable to the large amounts of missing data in most of the known tanystropheids. Many taxa are known from either a few isolated remains (*Tanystropheus "conspicuus"*, *Augustaburiania vatagini*) or a single, partial postcranial specimen (*Sclerostropheus fossai*, *Elessaurus gondwanoccidens*, *Macrocnemus obristi*, *Raibliania calligarisi*). Largely complete and articulated skeletons, including skulls, are known for several genera (e.g., *Tanystropheus*, *Macrocnemus*, *Tanytrachelos*, and *Langobardisaurus*) but their morphology is exceedingly hard to infer due to the poor preservation and the taphonomic flattening that has affected most tanystropheid fossils. However, the recent studies of flattened specimens of *Macrocnemus bassanii* and *Tanystropheus hydroides* using high-resolution synchrotron microtomographic scans (*Miedema et al., 2020*; *Spiekman et al., 2020a*, *2020b*) have revealed their cranial morphology in high detail, revealing a large cranial disparity between these taxa. The remarkable dentition of *Langobardisaurus pandolfii*, characterized by an edentulous anterior end of the snout and posterior to that tricuspid teeth and large crushing teeth in the posterior part of the jaw further indicates the high cranial diversity present among the tanystropheids (*Renesto & Dalla Vecchia, 2000*; *Saller, Renesto & Dalla Vecchia, 2013*). Therefore, an increased insight into the cranial morphology of other tanystropheid taxa, which are currently very poorly understood, will surely allow us to resolve tanystropheid interrelationships more confidently and will contribute to our understanding of their ecomorphological diversity. In addition to providing a detailed analysis of the interrelationships of former "protorosaurs", the data matrix constructed for this study is intended to provide a useful resource for further phylogenetic investigations on tanystropheids, dinocephalosaurids and other non-archosauriform archosaurmorph groups based on future findings, and to provide a phylogenetic framework for additional macroevolutionary or palaeobiographical studies.

## CONCLUSION

We provide a detailed overview of all known non-archosauriform archosauromorphs previously considered as "protorosaurs". The results of our phylogenetic analyses corroborate the polyphyly of "Protorosauria" as established by previous studies (e.g., *Dilkes, 1998*; *Ezcurra, 2016*; *Pritchard et al., 2015*) and affirm that the historical usage of "Protorosauria" as a clade that includes tanystropheids, dinocephalosaurids, and *Prolacerta broomi* should be abandoned. The use of both ratio and ordered characters has a positive result on tree resolution for our sample. The Chinese taxa *Pectodens zhenyuensis*

and *Dinocephalosaurus orientalis* form a newly erected clade, Dinocephalosauridae. *Jesairosaurus lehmani* and *Fuyuansaurus acutirostris* are non-crocopodan archosauromorphs that are closely related to tanystropheids and dinocephalosaurids, but their exact position among these groups is inconsistent in our analyses. The interrelationships within Tanystropheidae are poorly resolved, which can be attributed to the poorly known cranial morphology of most tanystropheid taxa. *Prolacerta broomi* is recovered as considerably more distantly related to Archosauriformes than previously considered.

## INSTITUTIONAL ABBREVIATIONS

| | |
|---|---|
| **AMNH** | American Museum of Natural History, New York, New York, USA |
| **AUP** | University of Aberdeen, Palaeontology collection, Aberdeen, UK |
| **BP** | Evolutionary Studies Institute, University of Witwatersrand, Johannesburg, South Africa |
| **BSPG** | Bayerische Staatssammlung für Paläontologie und Geologie, Munich, Germany |
| **FMNH** | Field Museum of Natural History, Chicago, Illinois, USA |
| **GMPKU** | Geological Museum of Peking University, Beijing, China |
| **IGWuG** | Institut für Geologische Wissenschaften und Geiseltalmuseum, Martin-Luther-Universität Halle-Wittenberg, Halle, Germany |
| **IVPP** | Institute of Vertebrate Paleontology and Paleoanthropology, Beijing, China |
| **KUVP** | Kansas University Museum of Natural History, Lawrence, Kansas, USA |
| **LPV** | Chengdu Center of the China Geological Survey, Chengdu, Sichuan, China |
| **MCSN** | Museo Cantonale di Scienze Naturali di Lugano, Lugano, Switzerland |
| **MCSNB** | Museo Civico di Scienze Naturali "E. Caffi" Bergamo, Bergamo, Italy |
| **MFSN** | Museo Friulano di Scienze Naturali, Udine, Italy |
| **MGUWr** | Geological Museum, Institute of Geological Sciences, University of Wrocław, Wrocław, Poland |
| **MSNM** | Museo di Storia Naturale, Milan, Italy |
| **NMS** | National Museums Scotland, Edinburgh, UK |
| **NHMW** | Naturhistorisches Museum Wien, Vienna, Austria |
| **NMK** | Naturkundemuseum im Ottoneum, Kassel, Germany |
| **OMNH** | Sam Noble Oklahoma Museum of Natural History, Norman, Oklahoma, USA |
| **P** | Palaeontological Collection of the Department of Geology and Palaeontology, University of Innsbruck, Innsbruck, Austria |
| **PIMUZ** | Paläontologisches Institut und Museum der Universität Zürich, Zurich, Switzerland |
| **PIN** | Paleontological Institute of the Russian Academy of Sciences, Moscow, Russia |
| **RC** | Rubidge Collection, Wellwood, Graaff-Reinet, South Africa |
| **RCSHC** | Royal College of Surgeons, Hunterian Collection, London, UK |

| | |
|---|---|
| **SAM-PK** | Iziko South African Museum, Cape Town, South Africa |
| **SMNS** | Staatliches Museum für Naturkunde Stuttgart, Stuttgart, Germany |
| **SMNK** | Staatliches Museum für Naturkunde Karlsruhe, Karlsruhe, Germany |
| **TMM** | Texas Memorial Museum, Austin, Texas, USA |
| **TWCMS** | Sunderland Museum and Winter Gardens, Tyne & Wear Archives & Museums, Sunderland, UK |
| **UA** | University of Antananarivo, Antananarivo, Madagascar |
| **UCMP** | University of California Museum of Paleontology, Berkeley, California, USA |
| **UFSM** | Universidade Federal de Santa Maria, Santa Maria, Brazil |
| **UMMP** | University of Michigan Museum of Paleontology, Ann Arbor, Michigan, USA |
| **UMZC** | University Museum of Zoology, Cambridge, UK |
| **UNIPAMPA** | Universidade Federal do Pampa, São Gabriel, Brazil |
| **UWBM** | Burke Museum of Natural History and Culture, University of Washington, Seattle, Washington, USA |
| **VMNH** | Virginia Museum of Natural History, Martinsville, Virginia, USA |
| **WMsN** | LWL-Museum für Naturkunde, Westfälisches Landesmuseum mit Planetarium, Münster, Germany |
| **YPM** | Yale Peabody Museum, New Haven, Connecticut, USA |
| **ZAR** | Zarzaitine Collection, Muséum National d'Histoire Naturelle, Paris, France |
| **ZMNH** | Zhejiang Museum of Natural History, Hangzhou, China |
| **ZPAL** | Institute of Paleobiology, Polish Academy of Sciences, Warsaw, Poland |

## ACKNOWLEDGEMENTS

We kindly thank Mark Norell and Carl Mehling (AMNH), Kevin Padian and Pat Holroyd (UCMP), Christian Sidor and Meredith Rivin (UWBM), Claire Browning and Roger Smith (SAM-PK), Anna Paganoni (MCSNB), Bernhard Zipfel, Sifelani Jirah, and Jonah Choiniere (BP), Rudolf Stockar (MCSN), Cristiano Dal Sasso and Stefania Nosotti (MSNM), Cristina Lombardo (MPUM), Nour-Eddine Jalil (ZAR), Christian Klug and Beat Scheffold (PIMUZ), Alexander Hastings and Christina Byrd (VMNH), Rainer Schoch (SMNS), Eberhard "Dino" Frey (SMNK), Lothar Schöllmann (WMsN), Cornelia Kurz (NMK), Giuseppe Muscio, Luca Simonetto, and Fabia Dalla Vecchia (MFSN), Chun Li (IVPP), Da-Yong Jiang (GMPKU), and Olivier Rieppel and William Simpson (FMNH), for access to specimens under their care. Martín Ezcurra, Thodoris Argyriou, Gabriel Aguirre Fernandez, Dylan Bastiaans, Feiko Miedema, and Olivier Rieppel are thanked for discussions. Martín Ezcurra and Sterling Nesbitt kindly provided photographs of several specimens of *Azendohsaurus madagaskarensis*, *Pamelaria dolichotrachela*, and *Proterosuchus alexanderi*. TNT is made freely available by the Willi Hennig Society. Finally, we would like to thank academic editor Claudia Marsicano and reviewers Martín Ezcurra and Felipe Pinheiro for their constructive recommendations that improved the quality of this manuscript.

### Funding

This study is part of the Swiss National Science Foundation project granted to Torsten Scheyer (no. 2'5321-162775). The Doris and Samuel P. Welles Fund provided financial support to Stephan Spiekman for the collection visit to the Museum of Paleontology in Berkeley. The funders had no role in study design, data collection and analysis, decision to publish, or preparation of the manuscript.

### Grant Disclosures

The following grant information was disclosed by the authors:
Swiss National Science Foundation: 2'5321-162775.
The Doris and Samuel P. Welles Fund.

### Competing Interests

The authors declare that they have no competing interests.

### Author Contributions

- Stephan N.F. Spiekman conceived and designed the experiments, performed the experiments, analyzed the data, prepared figures and/or tables, authored or reviewed drafts of the paper, and approved the final draft.
- Nicholas C. Fraser conceived and designed the experiments, authored or reviewed drafts of the paper, and approved the final draft.
- Torsten M. Scheyer conceived and designed the experiments, authored or reviewed drafts of the paper, and approved the final draft.

### Data Availability

  Raw data are available in the Supplemental Files.

### Supplemental Information

Supplemental information for this article can be found online at http://dx.doi.org/10.7717/peerj.11143#supplemental-information.

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
