# Peer review of "A new phylogenetic hypothesis of Tanystropheidae (Diapsida, Archosauromorpha) and other “protorosaurs”, and its implications for the early evolution of stem archosaurs"

_PeerJ, doi:10.7717/peerj.11143_

## Round 0.1 · original submission · Minor Revisions

Dear Dr. Spiekman,

Your Ms # 51657 entitled "A new phylogenetic hypothesis of Tanystropheidae (Diapsida, Archosauromorpha) and other “protorosaurs”, and its implications for the early evolution of stem archosaurs " co-authored with N. Fraser & T. Scheyer has been reviewed by two reviewers. After analyzing their comments, I consider that the Ms needs Minor Revisions before being considered for publication in PeerJ.

Both reviewers concur that the manuscript is a very welcome contribution to the discussion of the phylogenetic relationships of early archosauromorphs, and both consider that the Ms needs minor modifications.

The reviewers also included an annotated version of the Ms with general observations and several comments concerning the description and/or interpretation of many characters that should take into consideration.

I also agree with Reviewer # 1 (M. Ezcurra) that you should expand the Conclusions section in order to make it more robust and informative.

Thank you for submitting your Ms to PeerJ and I look forward to receiving your revision.

Sincerely,
Claudia Marsicano

·

Basic reporting

The manuscript of Spiekman and colleagues provides a detailed and comprehensive revision of the systematics of “protorosaur” archosauromorphs and the phylogenetic relationships of this group. This is a highly welcomed contribution because the phylogenetic relationships of tanystropheids have been poorly explored in recent analyses and several taxa haven’t been included in quantitative analyses or have been included only in the context of morphological disparity analyses. The introduction provides a detailed description of the problematic and the goals of the manuscript, the material and methods are well explained, and the results are described in detailed. The manuscript includes several anatomical figures that are crucial to understand new character formulations or modifications from previous data sets. As a result, I consider that this manuscript should be published after minor modifications.

Finally, I want to congratulate the authors for this well-built and robust phylogenetic analysis.

Yours sincerely,
Martin Ezcurra

Experimental design

The experimental design of the manuscript is clear and well conducted. I have minor comments about it that the authors should address before submitting the revised version of the manuscript (e.g. exclusion of taxa a priori based only on their high amount of missing data, calculation of retention and consistency indices).

Validity of the findings

The findings are well described and discussed. The main conclusion are well supported, but the authors should comment on the non-monophyly of groups based on the most parsimonious trees and not based on the topology of the strict consensus tree.

Additional comments

Before submitting the revised version of the manuscript, the authors should go through the edited version of the manuscript that I provide together with this letter and take a look to the more than 350 comments and modifications that I suggested. In addition, the authors should address the following minor to moderate comments:

- This sounds like you selected a priori which taxa should be pruned from the final analysis based only on the amount of missing data. Problematic taxa (i.e. OTUs with highly variable placements among MPTs) are a consequence of lack of phylogenetic information for a low-resolution placement. Thus, sometimes with a high amount of missing data (e.g. >95%) have stable phylogenetic positions. So, I strongly recommend to run a first analysis with all the OTUs (as you did) and after that use the iterPCR protocol to detect and prune terminals a posteriori.
On the other hand, if you have evidence to consider OTUs as problematic from a taxonomic point of view or because of problematic interpretations (e.g. Czatkowiella), I agree with their exclusion a priori from the analysis.

- You should mention that this character seems to be interdependent for your taxonomic sample, but the scorings of characters 20 and 76 in Ezcurra (2016: PeerJ) (and more recent iterations of this matrix) show that it is not the case. For example, rhynchosaurids have extremely short rostra but with a proportionally long nasal when compared with the frontal. You should mention this here in order to prevent future authors to delete character 76 from the data matrix of Ezcurra (2016).

- Discretization of ratios: it is not clear how you did that. In Ezcurra (2016), I conducted a cluster analysis to find different discretized states. I found this method very handy and useful.

- Character 17: I find difficult to find the difference between character state 17-0 and 17-1. In Prolacerta, the postnarial process also overlaps the maxilla laterally. Can you explain better and figure more clearly the difference between both character states.

- Character 23: it is very difficult to observe the presence of the fossa in the photograph of Dinocephalosaurus orientalis. Can you provide a drawing or a clearer photograph of that feature?

- I have a query about the calculation of the consistency and retention indices. If you have used the script STATS.RUN of TNT, this code calculates the values including all the terminals included in the dataset, without excluding the terminals excluded a priori before the analysis. You have excluded taxa a priori in some of your analyses, so, if it is the case, I strongly suggest you to use a script that I have modified that excludes these terminals also from the calculation of the indices. You can email me asking for this script, I think that my identity is quite clear.

- Page 116: you are providing a definition for an unnamed clade. Why? I agree to keep this clade unnamed until further work in done on this clade and the presence of non-tanystropheid taxa more closely related to tanystropheids than to other archosauromorphs is robustly supported.

- Page 118: why are you providing a new phylogenetic definition of Tanystropheidae. I consider that it is not necessary because the definition of Dilkes (1998: ZJLS) is well built. In addition, you are providing here a stem-based definition contrasting with the original spirit of the node-based definition of Dilkes (1998).

- Page 121: Unambiguous synapomorphies of Allokotosauria. The presence of three apomorphic character states related to the large palatal teeth of the vomer, palatine and pterygoid, respectively, sounds that they are interdependent for this part of the tree. You showed convincingly that in tanystropheids the presence of large vomerine teeth does not co-occur with the presence of large teeth in the palatine and pterygoid. So, I suggest to merge these three characters into two characters. One describing the presence/absence of large palatal teeth (as I did in 2016) and the other describing the distribution of such teeth across the palate (e.g. only in vomer, in palatine and pterygoid).

- End of text: I strongly suggest the authors to include some paragraphs in the Conclusion or as a separate section discussing where research efforts should be focused to improve tanystropheid taxonomy and phylogenetic relationships. I think that it is important for future workers and at the moment you are the most skilled people to provide such statements.

- The authors should provide in the supplementary files (NEXUS and TNT files) the code for character ordination. It is more time consuming to write the line for character ordination than to deactivate character ordination with a couple of clicks or a single command in TNT.

·

Basic reporting

Although early archosauromorphs historically known as the “Protorosauria” or “prolacertiforms” have been subject to several previous phylogenetic studies, the relationships of many taxa remained elusive, as former assessments suffered from low OTU representativeness or were especifically designed to resolve the phylogeny of more crownward clades. As such, the work of Spiekman et al. is most welcome, as it represents the state-of-the-art knowledge on early archosauromorph phylogeny, providing a new comprehensive dataset that icludes most taxa that were once considered as ‘Protorosauria’ or related to them. The manuscript is written in professional English, and I was only able to detect minor mistakes in this regard (but I’m not a native English speaker myself). A complete, fully referenced, background on early archosauromorph relationships is provided, and the MS is accompanied by high-quality images illustrating new morphological characters. The raw data is provided as nexus file, what will help future workers on this particular subject.

Experimental design

Spiekman et al. assembled a new datast based on characters from previous works and several new ones. Many old characters were modified to better fit a broad non-archosauriform sample, whereas others were merged or standardized following relevant criteria. This improved the quality of raw data, contributing to the reliabity of the outcomes. The experimental design and methodologies are robust and were sufficiently described.

Validity of the findings

Spiekman et al. present several trees, resulting from different phylogenetic parameters. More conservative hypotheses (analyses 3 and 4) are well-resolved and represent sound interpretations of early archosauromorph relationships. The main conclusions are derived strictly from the methodology empoyed and obtained results. The phylogenetic hypothesis sustained by the authors is useful in clarifying the relationships of several taxa, and the monophyly of important clades (such as the Tanystropheidae) is accessed with grater confidence than in previous literature. Some apparently aberrant results (such as the position of Prolacerta) could be better discussed by the authors, as it strongly disagrees with recently published hypotheses.

Additional comments

I have several minor issues, all of them are detailed in the attached pdf file. I would especially advise the authors to tone down their suggestion to abandon "Protorosauria". Not because they are wrong in their claim, but because this was already well-estabilished by several previous workers, whereas the authors sometimes seem to suggest that this is a direct outcome of their study (e.g. p. 123, lines 3393-3395).

Please pay special attention to the annotated pdf file I provide. All my concerns are highlighted in it!

Congratulations for the excellent work.

---

## Round 0.2 · accepted · Accept

Dear Dr. Spiekman,

I am pleased to inform you that your manuscript MS #51657 entitle "A new phylogenetic hypothesis of Tanystropheidae (Diapsida, Archosauromorpha) and other “protorosaurs”, and its implications for the early evolution of stem archosaurs” is now accepted for publication in PeerJ.

Thank you again for considering PeerJ and we look forward to your future contributions to the Journal.

sincerely,

Claudia Marsicano